

# The Cosmological Bootstrap:
# Spinning correlators from symmetries and factorization

Daniel Baumann[1,2], Carlos Duaso Pueyo[1], Austin Joyce[1,3],
Hayden Lee[4] and Guilherme L. Pimentel[5,1]

**1** Institute of Physics, University of Amsterdam, Amsterdam, 1098 XH, The Netherlands
**2** Department of Physics, National Taiwan University, Taipei 10617, Taiwan
**3** Department of Physics, Columbia University, New York, NY 10027, USA
**4** Department of Physics, Harvard University, Cambridge, MA 02138, USA
**5** Lorentz Institute for Theoretical Physics, Leiden, 2333 CA, The Netherlands

## Abstract

We extend the cosmological bootstrap to correlators involving massless spinning particles, focusing on spin-1 and spin-2. In de Sitter space, these correlators are constrained both by symmetries and by locality. In particular, the de Sitter isometries become conformal symmetries on the future boundary of the spacetime, which are reflected in a set of Ward identities that the boundary correlators must satisfy. We solve these Ward identities by acting with weight-shifting operators on scalar seed solutions. Using this weight-shifting approach, we derive three- and four-point correlators of massless spin-1 and spin-2 fields with conformally coupled scalars. Four-point functions arising from tree-level exchange are singular in particular kinematic configurations, and the coefficients of these singularities satisfy certain factorization properties. We show that in many cases these factorization limits fix the structure of the correlators uniquely, without having to solve the conformal Ward identities. The additional constraint of locality for massless spinning particles manifests itself as current conservation on the boundary. We find that the four-point functions only satisfy current conservation if the $s$, $t$, and $u$-channels are related to each other, leading to nontrivial constraints on the couplings between the conserved currents and other operators in the theory. For spin-1 currents this implies charge conservation, while for spin-2 currents we recover the equivalence principle from a purely boundary perspective. For multiple spin-1 fields, we recover the structure of Yang–Mills theory. Finally, we apply our methods to slow-roll inflation and derive a few phenomenologically relevant scalar-tensor three-point functions.

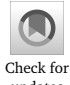

# 1 Introduction

Long-range forces determine the essential features of the macroscopic world. The large-scale structure of the universe is shaped by the force of gravity, while the electromagnetic force plays a fundamental role on a terrestrial scale. In quantum field theory, long-range forces are mediated by massless bosons, and the allowed forces are highly constrained by locality and unitarity [1]. In particular, the most salient properties of electromagnetism and gravity emerge from demanding consistency of scattering amplitudes involving massless particles of spin one and two [2–4]. On the other hand, massless particles with spin greater than two cannot interact consistently, ruling out the possibility of additional long-range forces.

Massless fields also play an important role during inflation [5–8] because their quantum fluctuations are amplified by the inflationary expansion, providing the seeds for structure formation in the late universe. Two massless modes are present in every inflationary model: a scalar mode (the Goldstone boson of broken time translations [9,10]) and a tensor mode (the graviton [11]). The former is the source of primordial density fluctuations, while the latter has not been observed yet, but is a primary target of future cosmological observations [12]. An important open problem is the systematic classification of inflationary scalar and tensor correlators, including the effects of new massive particles. Such a classification would provide the conceptual foundation for the discipline of "cosmological collider physics" [13–39], and facilitate a deeper understanding of theoretical constraints on cosmological correlators.

In this paper, we begin the systematic study and classification of spinning correlators in cosmological spacetimes. Due to the inherent difficulties in computing these objects with standard Lagrangian methods (see e.g. [40]), our current understanding is limited to the simplest cases [41,42]. This mirrors the limitations of the standard approach to computing scattering amplitudes of spinning particles. In that case, explicit computations using Feynman diagrams can be immensely complicated. Fortunately, this complexity is not reflected in the final answers, which are often remarkably simple [43,44]—in fact, the amplitudes for massless spinning particles are typically *simpler* than their scalar counterparts. The striking simplicity has motivated the modern "amplitudes bootstrap," in which the structure of scattering amplitudes is determined not by complex computations, but by much simpler and more fundamental consistency requirements [45,46]. It stands to reason that a similar approach will be fruitful in the cosmological context and, given the difficulties encountered in the direct computations of spinning correlators, the bootstrap approach is now a necessity and not just a luxury.

In [47], the bootstrap philosophy was applied to the study of inflationary scalar correlators (see also [48–54] for related work). Rather than tracking the inflationary time evolution explicitly, the late-time correlations were determined by consistency requirements alone. Concretely, the correlations arising from weakly interacting particles during inflation were constrained by the isometries of the inflationary spacetime [22,42,55], which act as conformal symmetries on the future boundary of the approximate de Sitter spacetime. In order to be consistent with these symmetries, the correlators must obey a set of conformal Ward identities, which are differential equations that dictate how the strength of correlations changes when the external momenta are varied [47,56]. Consistent inflationary processes correspond to solutions of these differential equations with the correct singularities [47]. Specifically, for Bunch–Davies initial conditions, the correlators should have no singularities in the so-called "folded limit," where two (or more) momenta become collinear, while in certain "factorization limits" the correlators must split into products of lower-point correlators (and/or lower-point scattering amplitudes) [51].

The goal of the present work is to extend the bootstrap approach to spinning correlators. Much as in flat space, there are both complications and simplifications associated with the introduction of spin. First, the conformal Ward identities for spinning fields are considerably more complicated than those for scalar fields, which naturally makes them much harder to solve directly. Second, correlation functions involving massless spinning fields obey additional consistency requirements. In particular, the operators dual to massless fields are conserved currents and must satisfy Ward–Takahashi (WT) identities associated to this current conservation. This implies that the structure of spinning correlators is more rigid and therefore more likely to be completely fixed by theoretical consistency, suggesting that the bootstrap approach should be particularly powerful.

In order to construct correlation functions involving massless spinning fields, we employ two complementary approaches:

- First, we use so-called "weight-shifting operators" to generate spinning correlators from

known scalar seed functions. The relevant weight-shifting operators were introduced for conformal field theories in [57, 58] and first applied in the cosmological context in [48]. Given a solution to the scalar conformal Ward identities, acting with a weight-shifting operator generates a solution to the spinning conformal Ward identities. The weight-shifting procedure therefore provides an efficient and algorithmic way to produce kinematically satisfactory spinning correlators with the right quantum numbers (spin and scaling dimension).

- Second, we will exploit our knowledge of the singularity structure of cosmological correlators to glue together more complicated correlators from simpler building blocks. For example, every correlator has a singularity when the total energy of the external fields vanishes, and the coefficient of this singularity is the flat-space scattering amplitude for the same process [42, 59]. Moreover, correlators arising from tree-level exchange have additional singularities when the sum of the energies entering a subgraph adds up to zero, and the coefficients of these singularities must satisfy certain factorization properties. As we will show, imposing that the correlators have only physical singularities with the correct residues is a powerful constraint and in many cases fixes the answer uniquely.

Using these two methods, we will provide a large amount of new theoretical data. In particular, we compute three- and four-point functions involving conserved spin-1 and spin-2 operators. At four points, the solutions to the conformal Ward identities are constructed separately for the $s$, $t$, and $u$-channels. We then show that the full correlator only satisfies the WT identities if the different channels are related to each other, leading to nontrivial constraints on the couplings between conserved currents and other operators in the theory. For spin-1 and spin-2 currents, this implies charge conservation and the equivalence principle, respectively, allowing us to re-discover these bulk facts from a purely boundary perspective. These constraints also have a deep relation to the singularity structure of cosmological correlators and we will show that the same conclusions can be reached by demanding consistency of the total energy singularity.

**Outline** The plan of the paper is as follows: In Section 2, we introduce our main objects of study, namely boundary correlators in de Sitter space. We describe the symmetries that these correlators must satisfy, derive the corresponding conformal Ward identities, and discuss the expected singularities of their solutions. In Section 3, we outline our strategy for computing conformal correlators with spin. We introduce the relevant weight-shifting operators that allow us to obtain complicated spinning correlators from much simpler scalar seed correlators. We explain that correlators involving conserved currents must satisfy additional Ward–Takahashi identities. We introduce spinor helicity variables that efficiently capture the polarization structure of the correlators and allow for a simple way to impose the WT identities. In Section 4, we use the weight-shifting formalism to derive three-point functions involving massless spin-1 and spin-2 operators. In Section 5, we extend our treatment to four-point functions. We show that the WT identities can only be satisfied if multiple channels are added and if the couplings in each channel are related to each other. In Section 6, we derive the same four-point correlators by imposing the correct singularities, without having to solve the conformal Ward identities explicitly. We show that the different channels must be added with specific normalizations in order for the total energy singularity to have a Lorentz-invariant residue. In Section 7, we comment on applications to inflation, providing simple derivations of a few mixed tensor-scalar three-point functions. Our conclusions are presented in Section 8.

A number of appendices contain additional technical details and review material: In Appendix A, we review basic elements of representation theory in de Sitter space. In Appendix B, we derive the Ward–Takahashi identities used in Sections 4 and 5. In Appendix C, we describe

the spinor helicity formalism, both in flat space and adapted to de Sitter space. In Appendix D, we derive the action of the special conformal generator on correlators in spinor helicity variables. In Appendix E, we present polarization tensors and polarization sums for spin-1 and spin-2 fields. In Appendix F, we cite results for the Compton scattering of spin-1 and spin-2 fields. In Appendix G, we provide an alternative derivation of the correlators associated to Compton scattering. Finally, in Appendix H, we list the most important variables used in this work.

**Reading guide**  Given the length of the paper, we provide a short reading guide: Section 2 contains mostly standard review material that can be skipped by experts, although we suggest skimming it to get familiar with our notation. The conceptual ideas of this work are presented in Section 3. Reading this section hopefully also provides a roadmap for the rest of the paper. Sections 4 and 5 are mostly a technical application of the weight-shifting procedure. This produces a lot of important theoretical data, but the sections can probably be skipped or skimmed on a first reading. Section 6 introduces a new way to construct complicated correlators from knowledge of their singularities. We feel that this method is only the beginning of a novel perspective on the problem of cosmological correlators and hope that some of our readers will be inspired to develop it further. Readers interested in summaries of the main results of Sections 5 and 6 can find them in §5.4 and §6.4. Section 7 should be read by readers interested in the application of these tools to inflationary correlators. Finally, the appendices are a mix of review material (added for the benefit of students and newcomers to the field) and computational details (added for the benefit of readers who would like to reproduce and/or extend our computations).

**Notation and conventions**  Throughout the paper, we use natural units, $\hbar = c \equiv 1$. Our metric signature is $(-+++)$. We use Greek letters for spacetime indices, $\mu = 0, 1, 2, 3$, and Latin letters for spatial indices, $i = 1, 2, 3$. Spatial vectors are denoted by $\vec{x}$ and their components by $x^i$. The corresponding three-momenta are $\vec{k}$. The magnitude of vectors is defined as $k = |\vec{k}|$ and unit vectors are written as $\hat{k} = \vec{k}/k$. We use Latin letters from the beginning of the alphabet to label the momenta of the different legs of a correlation function, i.e. $\vec{k}_a$ is the momentum of the $a$-th leg. The sum of two momenta $k_a$ and $k_b$ is often written as $k_{ab} \equiv k_a + k_b$.

Correlation functions in momentum space take the form

$$\langle O_{\vec{k}_1} \cdots O_{\vec{k}_n} \rangle = (2\pi)^3 \delta^3(\vec{k}_1 + \cdots + \vec{k}_n) \langle O_{\vec{k}_1} \cdots O_{\vec{k}_n} \rangle' . \tag{1}$$

To avoid notational clutter, we will usually drop the primes on the "stripped correlators." We will typically also drop the momentum labels on the operators $O_{\vec{k}_n}$ and let the order of appearance inside correlation functions indicate their momentum dependence, e.g. $\langle JOO \rangle \equiv \langle J_{\vec{k}_1} O_{\vec{k}_2} O_{\vec{k}_3} \rangle$. Our notation for the fields living in the bulk spacetime and their dual boundary operators is summarized in the following table:

| Dimension | Spin | Bulk | Boundary |
|:---:|:---:|:---:|:---:|
| $\Delta$ | $\ell$ | $\sigma_\Delta^{(\ell)}$ | $O_\Delta^{(\ell)}$ |
| 2 | 0 | $\varphi$ | $\varphi$ |
| 3 | 0 | $\phi$ | $\phi$ |
| 2 | 1 | $A_\mu$ | $J_i$ |
| 3 | 2 | $\gamma_{\mu\nu}$ | $T_{ij}$ |

To study spinning fields economically, we work with index-free notation: given a symmetric, spin-$\ell$ tensor operator, $O_{i_1 \cdots i_\ell}$, we introduce auxiliary null vectors $z^i$, and write

$$O_\Delta^{(\ell)} = z^{i_1} \cdots z^{i_\ell} \, O_{i_1 \cdots i_\ell} \, . \tag{2}$$

To extract the traceless part of the original tensor, the auxiliary vectors can be removed by acting with the differential operator [60]

$$D_z^i = \left( \frac{1}{2} + \vec{z} \cdot \frac{\partial}{\partial \vec{z}} \right) \frac{\partial}{\partial z_i} - \frac{1}{2} z^i \frac{\partial^2}{\partial \vec{z} \cdot \partial \vec{z}} \, . \tag{3}$$

We will often evaluate correlation functions for explicit choices of the external polarizations. We denote the polarization vectors by $\xi_i^\pm$, where $\pm$ labels the helicity of the external state. Polarization tensors for spin-2 operators are defined as $\xi_{ij}^\pm \equiv \xi_i^\pm \xi_j^\pm$. We often use the condensed notation $J^\pm \equiv \xi_i^\pm J^i$ and $T^\pm \equiv \xi_{ij}^\pm T^{ij}$. We will sometimes use a spinor helicity representation for the polarizations, which is reviewed briefly in §3.4, and more comprehensively in Appendix C.

Finally, we will use the following conventions for flat-space scattering amplitudes. All four-momenta $p_a^\mu$ are ingoing. Polarization vectors will be denoted by $\epsilon_a^\mu$. The Mandelstam variables are $S \equiv -(p_1 + p_2)^2$, $T \equiv -(p_1 + p_4)^2$ and $U \equiv -(p_1 + p_3)^2$. We capitalize the Mandelstam variables to avoid confusion with $s \equiv |\vec{k}_1 + \vec{k}_2|$, $t \equiv |\vec{k}_1 + \vec{k}_4|$ and $u \equiv |\vec{k}_1 + \vec{k}_3|$, which we employ for the exchange momenta in cosmological correlators.

## 2 De Sitter Correlators: Back to the Future

The fundamental observables in cosmology are correlation functions. If inflation is correct, then these correlations were created in a quasi-de Sitter spacetime

$$ds^2 = \frac{1}{H^2 \eta^2} \left( -d\eta^2 + d\vec{x}^2 \right), \tag{4}$$

where $H$ is the nearly constant Hubble parameter and $\eta$ is conformal time. The correlations imprinted on the future boundary of this spacetime (located at $\eta = 0$; see Fig. 1) both capture the dynamics during inflation and provide the initial conditions that evolve into the late-time structures that we see today. The correlations arising from weakly interacting particles are highly constrained by the isometries of the spacetime, which lead to conformal Ward identities satisfied by the boundary correlators (see Fig. 1). Beyond these kinematic constraints, locality and unitarity of the bulk time evolution place additional restrictions on the structure of all consistent correlations. In this section, we first review the kinematic consequences of the de Sitter symmetries for the boundary correlators of spinning fields, before discussing how bulk locality dictates the singularity structure of these correlators.

### 2.1 Boundary Correlators

Consider a set of bulk fields, $\sigma(\vec{x}, \eta)$, propagating in an approximate de Sitter spacetime. These fields include both matter fields (such as the inflaton $\phi$) and metric fluctuations (such as the graviton $\gamma_{\mu\nu}$), and we are interested in their spatial correlations. At sufficiently late times, all modes have crossed the horizon and only massless degrees of freedom survive, taking on time-independent spatial profiles, $\sigma(\vec{x})$. All information about these frozen fluctuations can be described by the so-called "wavefunction of the universe." This wavefunction is defined as the overlap between the vacuum state and a given state of the late-time fluctuations,

$$\Psi[\sigma(\vec{x})] \equiv \langle \sigma(\vec{x}) | 0 \rangle = \int \mathcal{D}\sigma \, e^{iS[\sigma(\vec{x})]} \approx e^{iS_{cl}[\sigma(\vec{x})]}, \tag{5}$$

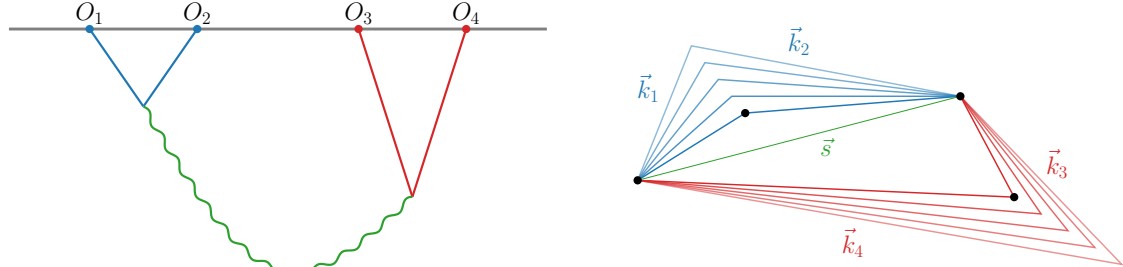

Figure 1: Correlations measured at the end of inflation are created during a period of quasi-de Sitter expansion in the early universe. These correlations capture information about the production and decay of massive particles during inflation (*left*). For weakly interacting particles, the boundary correlators are constrained by the isometries of the spacetime, which act as conformal transformations on the boundary. These constraints take the form of "conformal Ward identities," which are differential equations that dictate how the strength of the boundary correlations changes when the external momenta are varied (*right*). This momentum dependence encodes features of the inflationary time evolution.

where the final equality is the saddle-point approximation to the path integral that only holds for tree-level processes. As long as the fluctuations are small, the wavefunction has a perturbative expansion. It is useful to write this expansion in momentum space, the natural habitat of cosmological correlations.[1] The wavefunction then reads

$$\Psi[\sigma(\vec{k})] \simeq \exp\left(-\sum_{n=2}^{\infty} \frac{1}{n!} \int \frac{d^3 k_1 \cdots d^3 k_n}{(2\pi)^{3n}} \Psi_n(\vec{k}_N) \sigma_{\vec{k}_1} \cdots \sigma_{\vec{k}_n}\right), \tag{6}$$

where the kernels $\Psi_n$ are called *wavefunction coefficients* and $\vec{k}_N \equiv \{\vec{k}_1, \ldots, \vec{k}_n\}$ denotes the set of all momenta. Translation invariance implies that the wavefunction coefficients take the form

$$\Psi_n = (2\pi)^3 \delta^3(\vec{k}_1 + \cdots + \vec{k}_n) \langle O_{\vec{k}_1} \cdots O_{\vec{k}_n}\rangle', \tag{7}$$

where the prime on the expectation value indicates that the momentum-conserving delta function has been removed. In the following, we will often drop the primes, with the understanding that the delta function has been extracted. We see that the wavefunction coefficients can be interpreted as correlation functions[2] of operators $O$ dual to the bulk fields $\sigma$ [41].

The wavefunction describes the probability amplitude for observing a given set of perturbations and therefore encodes the late-time correlation functions:

$$\langle \sigma_{\vec{k}_1} \cdots \sigma_{\vec{k}_n}\rangle = \frac{\int \mathcal{D}\sigma \, \sigma_{\vec{k}_1} \cdots \sigma_{\vec{k}_n} |\Psi[\sigma]|^2}{\int \mathcal{D}\sigma \, |\Psi[\sigma]|^2} . \tag{8}$$

In perturbation theory, the relation between the wavefunction coefficients and the corresponding bulk in-in correlators can be made completely explicit. We simply expand the exponential

---

[1]There has been a flurry of activity in studying conformal field theories in momentum space, with applications ranging from cosmology to the conformal bootstrap in Euclidean and Lorentzian signatures [56, 61–71].

[2]For this reason, we will often abuse terminology and refer to the wavefunction coefficients as "correlators," although they should not be confused with the correlators of the bulk fields $\sigma(\vec{x}, \eta)$.

in (6) and perform the resulting Gaussian integrals in (8):

$$\langle \sigma_{\vec{k}_1} \sigma_{\vec{k}_2} \rangle = \frac{1}{2 \operatorname{Re}\langle O_{\vec{k}_1} O_{\vec{k}_2} \rangle}, \tag{9}$$

$$\langle \sigma_{\vec{k}_1} \sigma_{\vec{k}_2} \sigma_{\vec{k}_3} \rangle = -\frac{\operatorname{Re}\langle O_{\vec{k}_1} O_{\vec{k}_2} O_{\vec{k}_3} \rangle}{4 \prod_{n=1}^{3} \operatorname{Re}\langle O_{\vec{k}_n} O_{-\vec{k}_n} \rangle}, \tag{10}$$

$$\langle \sigma_{\vec{k}_1} \sigma_{\vec{k}_2} \sigma_{\vec{k}_3} \sigma_{\vec{k}_4} \rangle = \frac{\langle O^4 \rangle_{\mathrm{d}} - \langle O^4 \rangle_{\mathrm{c}}}{8 \prod_{n=1}^{4} \operatorname{Re}\langle O_{\vec{k}_n} O_{-\vec{k}_n} \rangle}. \tag{11}$$

The connected and disconnected contributions of the four-point function are

$$\langle O^4 \rangle_{\mathrm{c}} = \operatorname{Re}\langle O_{\vec{k}_1} O_{\vec{k}_2} O_{\vec{k}_3} O_{\vec{k}_4} \rangle, \tag{12}$$

$$\langle O^4 \rangle_{\mathrm{d}} = \frac{\operatorname{Re}\langle O_{\vec{k}_1} O_{\vec{k}_2} O_{-\vec{s}} \rangle \operatorname{Re}\langle O_{\vec{s}} O_{\vec{k}_3} O_{\vec{k}_4} \rangle}{2 \operatorname{Re}\langle O_{-\vec{s}} O_{\vec{s}} \rangle} + \mathrm{perms}, \tag{13}$$

where $\vec{s} \equiv \vec{k}_1 + \vec{k}_2$ and the permutations are over the external momenta. Although the above formulas were written for scalar fields, they generalize straightforwardly to spinning bulk fields.

## 2.2 Symmetries and Ward Identities

The dynamics of fields propagating in de Sitter space are constrained by the symmetries of the spacetime. These kinematic requirements have important consequences for the correlations generated during inflation—in particular, the late-time wavefunction is strongly constrained by the de Sitter symmetry [41,42,55,72–76]. These correlations reside on the future boundary, where the de Sitter isometries act like the conformal group. Consequently, the wavefunction coefficients satisfy the same kinematic constraints as the correlators of a conformal field theory (CFT).[3]

To make these symmetry constraints explicit, we examine the late-time action of the de Sitter isometries. The algebra of de Sitter isometries is generated by the following Killing vectors

$$
\begin{aligned}
P_i &= \partial_i, & D &= -\eta \partial_\eta - x^i \partial_i, \\
J_{ij} &= x_i \partial_j - x_j \partial_i, & K_i &= 2x_i \eta \partial_\eta + \left(2x^j x_i + (\eta^2 - x^2)\delta_i^j\right)\partial_j.
\end{aligned}
\tag{14}
$$

The isometries generated by $P_i$ and $J_{ij}$ are the translational and rotational symmetries of the flat spatial slices $\mathbb{R}^3$. The other transformations are maybe less familiar: $D$ generates a dilatation symmetry that rescales space and time equally. The last three transformations, $K_i$, mix the spatial and time coordinates in a rather complicated way. We will call them special conformal transformations (SCTs), anticipating their action at late times.

To see how the conformal group emerges, we consider the late-time evolution of a spin-$\ell$ field $\sigma^{(\ell)}$ of mass $m_\sigma^2$. Solving its equation of motion in the limit $\eta \to 0$, one finds

$$\sigma^{(\ell)}(\vec{x}, \eta \to 0) = \sigma_+^{(\ell)}(\vec{x})\eta^{\Delta_+ - \ell} + \sigma_-^{(\ell)}(\vec{x})\eta^{\Delta_- - \ell}, \tag{15}$$

where $\sigma_\pm^{(\ell)}(\vec{x})$ is the spatial field profile on the future boundary and we have employed index-free notation as in (2). The scaling dimensions in (15) are fixed in terms of the field's mass

---

[3]Note that the wavefunction is *not* the generating functional of a reflection-positive Euclidean conformal field theory. None of our considerations will rely on this distinction, nor do we require or assume the existence of any precise dS/CFT correspondence.

through the relation[4]

$$\Delta_\pm = \begin{cases} \dfrac{3}{2} \pm \sqrt{\dfrac{9}{4} - \dfrac{m_\sigma^2}{H^2}} & \text{(scalars)}, \\[2em] \dfrac{3}{2} \pm \sqrt{\left(\ell - \dfrac{1}{2}\right)^2 - \dfrac{m_\sigma^2}{H^2}} & \text{(spinning fields)}. \end{cases} \qquad (16)$$

Acting with (14) on (15), we obtain the following transformations for the boundary values of the field

$$P_i \sigma_\pm^{(\ell)} = \partial_i \sigma_\pm^{(\ell)}, \qquad (17)$$

$$J_{ij} \sigma_\pm^{(\ell)} = \left(x_i \partial_j - x_j \partial_i + z_i \partial_{z_j} - z_j \partial_{z_i}\right) \sigma_\pm^{(\ell)}, \qquad (18)$$

$$D \sigma_\pm^{(\ell)} = -\left(\Delta_\pm + \vec{x} \cdot \partial_{\vec{x}}\right) \sigma_\pm^{(\ell)}, \qquad (19)$$

$$K_i \sigma_\pm^{(\ell)} = \left(2x_i \Delta_\pm + \left(2x^j x_i - x^2 \delta_i^j\right) \partial_j - 2(\vec{x} \cdot \vec{z}) \partial_{z_i} + 2z_i \vec{x} \cdot \partial_{\vec{z}}\right) \sigma_\pm^{(\ell)}. \qquad (20)$$

These are precisely the transformation rules of a conformal primary operator with scaling dimension (weight) $\Delta_\pm$. We therefore see that, at late times, fields in de Sitter space have two characteristic fall-offs, the coefficients of which transform like conformal primaries.

Next, we want to understand how these symmetry transformations act on the wavefunction. For light fields,[5] the $\Delta_-$ fall-off dominates at late times, and we can identify the coefficient of this fall-off with the spatial field profile that appears in the wavefunction, $\sigma^{(\ell)}(\vec{x}) \equiv \sigma_-^{(\ell)}(\vec{x})$. The action of the de Sitter isometries on the late-time wavefunction is then easy to characterize —the field profiles just shift infinitesimally by (17)–(20). Equivalently, we can interpret the symmetry transformations as acting on the wavefunction coefficients, $\Psi_n$, as opposed to the field profiles $\sigma^{(\ell)}$, by integrating the derivatives by parts. Doing this explicitly, one finds that the de Sitter symmetries act also as conformal transformations on the dual operators $O^{(\ell)}$ in (7), but with weight $\Delta_+ = 3 - \Delta_-$.

In the following, we will write the boundary operators as $O_\Delta^{(\ell)}$, with $\Delta \equiv \Delta_+$. We will be particularly interested in scalar operators with $\Delta = 2$, which we will denote by $\varphi$. The corresponding bulk field $\varphi$ has mass $m^2 = 2H^2$ and is conformally coupled. For applications to inflation, we care about massless bulk fields $\phi$. The corresponding boundary operator has $\Delta = 3$ and will be written as $\phi$. In the case of spinning operators, we will primarily be interested in conserved currents. The conserved spin-1 current $J_i$ has dimension $\Delta = 2$ and is dual to a massless vector in the bulk, $A_\mu$, while the conserved spin-2 current, $T_{ij}$, has dimension $\Delta = 3$ and is dual to the bulk graviton $\gamma_{\mu\nu}$.

In Fourier space, the action of the conformal generators on the boundary operators $O_\Delta^{(\ell)}$ becomes

$$D O_\Delta^{(\ell)} = \left((3 - \Delta) + k^i \partial_{k^i}\right) O_\Delta^{(\ell)}, \qquad (21)$$

$$K_i O_\Delta^{(\ell)} = \left(2(\Delta - 3)\partial_{k^i} + k_i \partial_{k^j}\partial_{k^j} - 2k^j \partial_{k^j}\partial_{k^i} - 2z_j \partial_{k_j}\partial_{z_i} + 2z_i \partial_{k_j}\partial_{z_j}\right) O_\Delta^{(\ell)}. \qquad (22)$$

---

[4]The pair of conformal weights dual to a given bulk field are related by $\Delta_\pm = 3 - \Delta_\mp$. Operators of the same spin whose weights obey this relation are so-called "shadows" of each other, and belong to equivalent conformal representations. Operators can be mapped to their shadows by means of the shadow transform, which consists of convolving an operator with the two-point function of its shadow. See, for example, Appendix A of [77] for details.

[5]By "light fields" we mean fields belonging to the complementary or discrete series of representations of the de Sitter group. Fields in the principal series scale at late times with an admixture of $\Delta_\pm$ fall-offs. See Appendix A for a review of de Sitter representation theory.

This implies that the wavefunction coefficients must satisfy the *conformal Ward identities* [6]

$$0 = \left[ -3 + \sum_{a=1}^{n} D_a \right] \langle O_1 \cdots O_a \cdots O_n \rangle' ,$$

$$0 = \sum_{a=1}^{n} K_a^i \langle O_1 \cdots O_a \cdots O_n \rangle' ,$$

(23)

where we have introduced the shorthand $O_a \equiv O_{\Delta_a}^{(\ell_a)}(\vec{k}_a)$ and temporarily restored the primes on the correlators for clarity. These Ward identities determine how the strength of the correlations changes as the external momenta $\vec{k}_a$ are varied. Some solutions to the conformal Ward identities have been obtained for $n = 3$ in [30, 42, 55, 56, 61, 78–80] and for $n = 4$ in [47, 48, 62, 64].

Our goal is to solve the system of differential equations in (23) for spinning operators, subject to some conditions on the possible singularities that can appear in the solutions. In principle, we could try to solve these equations directly. However, the proliferation of tensor structures very quickly makes this intractable. Fortunately, there is a simpler and more elegant approach. We will exploit the fact that boundary correlation functions are conformally invariant and import tools from the CFT literature to study the correlators of spinning fields (see e.g. [57, 58]). In particular, we will employ weight-shifting operators [48] to generate spinning solutions to the differential equations (23). The virtue of the weight-shifting approach is that it allows us to transmute scalar de Sitter correlators into spinning correlation functions by acting with a simple set of differential operators, and therefore bypass the difficulties of solving the relevant equations directly. Using these techniques, we are able to construct a large number of spinning correlators, from which we can abstract general principles. We will outline this procedure in §3.1 and then apply it in Sections 4 and 5.

## 2.3 Physical Singularities

A key insight of the cosmological bootstrap is the fact that solutions of the Ward identities (23) can be classified by their singularities [47]. This provides a powerful organizing principle to understand the manifestations of bulk physics in purely boundary terms. Much like in the case of the flat-space $S$-matrix, only certain singularities characteristic of bulk processes are allowed in cosmological correlation functions and locality/unitarity constrains correlators to behave in universal ways in the vicinity of these singularities. As we will see, in many cases these constraints are strong enough to uniquely fix tree-level correlation functions. In this section, we describe the singularity structure of cosmological correlators, deferring a more complete discussion of how the behavior near these singularities can be used to re-construct the full correlator to Section 6 (see also [51, 52, 54]).

It is a remarkable fact that many cosmological correlators, $\Psi_n$, have a singularity[7] when the sum of the "energies" (absolute values of the momenta) vanishes, $E \equiv \sum |\vec{k}_a| \to 0$, and the coefficient of this singularity is the flat-space scattering amplitude for the same process, $A_n$ [42, 59]. This singularity is the cosmological avatar of the energy-conserving delta function for flat-space scattering amplitudes. In this precise sense, we can think of correlators as defor-

---

[6]The extra $-3$ in the dilatation Ward identity comes from the action of the dilatation operator on the momentum-conserving delta function that appears as a consequence of translation invariance. This additional contribution appears because we have defined "primed" momentum-space correlation functions, where we have removed the momentum-conserving delta function. See e.g. [42] for details.

[7]For particles with spin, there are some helicity configurations for which the correlators are nonzero, but the flat-space scattering amplitude vanishes—e.g. this is the case for Yang–Mills and Einstein gravity correlators with equal helicities. These correlators do *not* have total energy singularities for these choices of helicities.

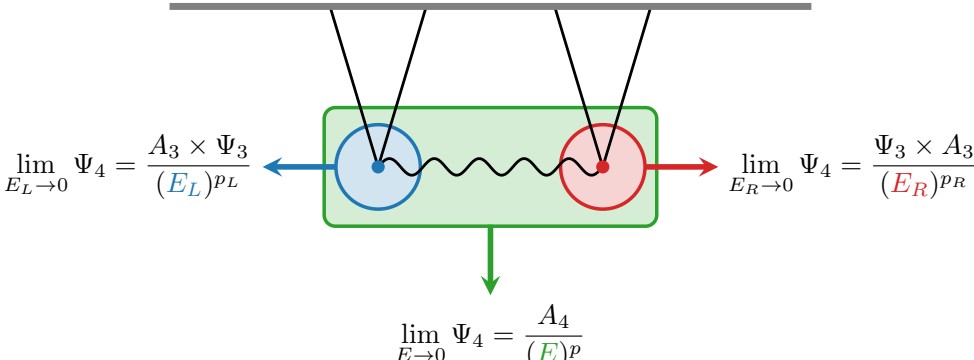

Figure 2: Illustration of the different singularities of four-point correlators in the *s*-channel. All correlators have a total energy singularity, while correlators arising from the exchange of a particle also have partial energy singularities when the energy at an interaction vertex is conserved. Requiring these singularities to have the correct residues is a powerful constraint on the structure of the correlators and in many cases fixes them completely.

mations of scattering amplitudes, which suggests that all the remarkable structures discovered in scattering amplitudes should have extensions to cosmological correlators.

For correlators arising from *contact interactions* in the bulk these "total energy singularities" are the only singularities. The derivative expansion of the bulk effective theory maps to a series of poles at $E = 0$ in the boundary correlators, with the order of the pole determined by the number of derivatives in the corresponding bulk interaction.

Correlators arising from *exchange interactions* in the bulk will have additional singularities when the sum of the energies entering a subgraph adds up to zero. We will call these "partial energy singularities." For example, for the *s*-channel diagram shown in Fig. 2, the four-point correlator, $\Psi_4$, has singularities when $E_L \equiv k_{12} + s \to 0$ or $E_R \equiv k_{34} + s \to 0$, where we have defined $k_{nm} \equiv |\vec{k}_n| + |\vec{k}_m|$ and $s \equiv |\vec{k}_1 + \vec{k}_2|$. At these singularities, the function $\Psi_4$ factorizes into a product of a three-point amplitude, $A_3$, and a three-point correlator, $\Psi_3$ (see Fig. 2). This is the analog of the factorization of scattering amplitudes when an intermediate particle goes on-shell. Imposing these factorization limits is a powerful constraint on the structure of the correlators.

Taken together, the total energy and partial energy singularities are the *only* singularities of consistent cosmological correlators. Even this is a nontrivial constraint, as generic bulk initial conditions would lead to singularities in the so-called "folded limit," when two (or more) momenta become collinear. For example, the correlator corresponding to the *s*-channel diagram shown in Fig. 2 should be regular for $k_{12} \to s$ and $k_{34} \to s$. Demanding the absence of such folded singularities therefore places an important constraint on the solutions to the Ward identities in (23). From the bulk perspective, we can think of this condition as imposing the adiabatic (Bunch–Davies) vacuum as an initial condition.[8]

The conformal Ward identities are second-order differential equations and therefore require two boundary conditions. One boundary condition is provided by demanding the absence of folded singularities. A second boundary condition is needed to fix the overall normalization of the solution.[9] There are several ways in which this condition can be chosen, but

---

[8] Note that this is a constraint on the initial *quantum state*: folded singularities are generically produced dynamically by classical evolution, and in that context can be thought of as signatures of the on-shell production or decay of particles [81].

[9] More specifically, there is a boundary condition for which the four-point correlator factorizes into a product of a three-point scattering amplitude and a "deformed" three-point correlator. We provide details later in the paper (see

perhaps the most natural is to impose that one of the partial energy singularities is normalized correctly. This then fixes the solution completely [47].

For massless spinning fields and scalars of integer conformal dimension, constraints on the allowed singularities and their residues are often strong enough to fix the answers completely, without having to solve the Ward identities in (23) explicitly. This is reminiscent of the situation in flat space, where interactions of massless particles are so strongly constrained that Lorentz invariance, locality, and unitarity uniquely fix the long-distance behavior of four-point scattering amplitudes in terms of three-point data only. In Section 6, we will explore this as a powerful alternative to the weight-shifting approach.

# 3 A Foray into Conformal Correlators with Spin

Solving the kinematic Ward identities in (23) is rather difficult, since they are a set of coupled partial differential equations in the momentum variables. For operators with spin, the polarization information carried by the external operators provides further complications. The direct approach to solving these equations can be carried out to some extent for spinning operators at three points [42, 55, 56] and for scalar operators at four points [47, 64, 82], but it quickly becomes intractable for spinning operators at four points and beyond. Fortunately, there is a more elegant approach that utilizes tools developed in the study of conformal field theory. By introducing a set of conformally-invariant *weight-shifting operators* [57, 58], new solutions to the conformal Ward identities can be generated given an initial seed solution. Due to the nature of the weight-shifting procedure, these new solutions will have different quantum numbers $\Delta$ and $\ell$. This allows us to economically build spinning solutions from known scalar solutions. The weight-shifting approach was first applied in the cosmological context in [48] to generate spin-exchange solutions for scalar correlators. Here, we will show that the formalism also provides a dramatic simplification for spinning correlators.

Our particular focus is on correlation functions involving spinning operators associated to conserved currents. These correlators must satisfy current conservation, which manifests itself as additional *Ward–Takahashi identities* that relate the longitudinal parts of the correlation functions to lower-point correlators. Our goal therefore is to simultaneously solve the Ward identities of both conformal symmetry and current conservation. Once this is done, we are free to add identically conserved correlation functions with the correct quantum numbers; these combined with the solution to the Ward–Takahashi identity parametrize the most general correlator.

## 3.1 Kinematics: Weight-Shifting Operators

We begin with a brief review of the relevant weight-shifting technology. We will focus on a subset of weight-shifting operators that will be of most use for our purposes. For a more detailed discussion, we refer the reader to our companion paper [48], as well as [57, 58].

Weight-shifting operators are most naturally constructed in the embedding space formalism, which introduces redundant variables to create an enlarged space in which conformal transformations act linearly [48, 57, 60, 83]. The physical space is then a particular projection of this higher-dimensional space. The embedding space approach makes it simpler to find differential operators that act on operators in correlation functions and change their representation weights. Once these weight-shifting operators have been identified in embedding

---

§6.1), but it is important to emphasize that there are two natural choices for the "deformed" three-point function—one gives the coefficient of the wavefunction of the universe, while the other computes the cosmological correlator directly. In [47, 48], we computed cosmological correlators, while in this paper, our focus is on the wavefunction coefficients instead.

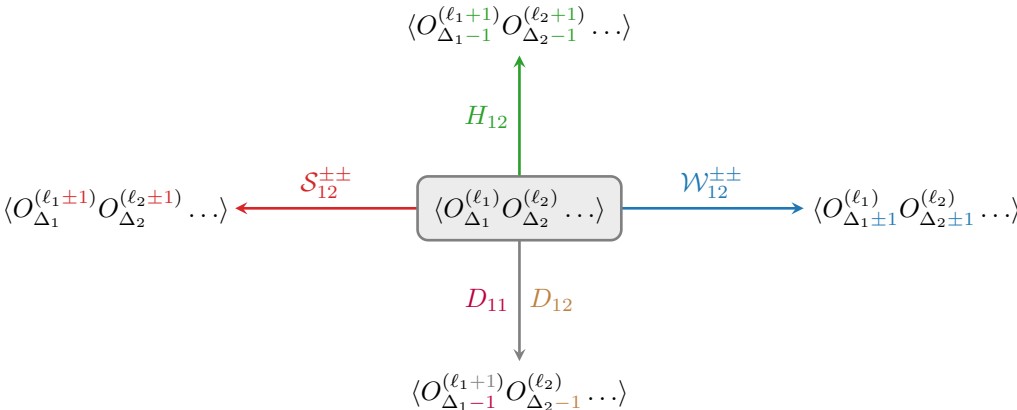

Figure 3: Schematic illustration of the different weight-shifting operators used in this paper.

space, they can straightforwardly be projected to the physical space and Fourier transformed. The details of this procedure can be found in [48], and here we merely quote the results.[10] A summary of the relevant weight-shifting operators and their effects is given in Fig. 3.

An important feature of the weight-shifting operators is that they are bi-local—they naturally act on a pair of operators in a correlation function. The simplest weight-shifting operator lowers the conformal scaling dimension $\Delta$ by one unit at each point it acts on. In Fourier space, this operator has the form

$$\mathcal{W}_{12}^{--} = \frac{1}{2}\vec{K}_{12}\cdot\vec{K}_{12}, \tag{24}$$

where $K_{12}^i \equiv \partial_{k_1^i} - \partial_{k_2^i}$ and, for concreteness, we have chosen the operator to act at points 1 and 2. The operator that raises the scaling dimension by one unit at each point, $\mathcal{W}_{12}^{++}$, is much more complicated. Acting on scalar operators, however, it reduces to the following manageable form

$$\mathcal{W}_{12}^{++} = \frac{1}{2}(k_1 k_2)^2 K_{12}^2 - (3-2\Delta_1)(3-2\Delta_2)\vec{k}_1\cdot\vec{k}_2 \\ + \left(k_2^2(3-2\Delta_1)\big(2-\Delta_1+\vec{k}_1\cdot\vec{K}_{12}\big) + (1\leftrightarrow 2)\right). \tag{25}$$

The version of this operator acting on operators with spin, and its (lengthy) explicit expression, can be found in [48].

Weight-shifting operators can also be used to change the spin of the operators they act on. For example, the following operator raises the spin by one unit at both the points 1 and 2:

$$\mathcal{S}_{12}^{++} = (\ell_1 + \Delta_1 - 1)(\ell_2 + \Delta_2 - 1)\vec{z}_1\cdot\vec{z}_2 - \frac{1}{2}(\vec{z}_1\cdot\vec{k}_1)(\vec{z}_2\cdot\vec{k}_2)K_{12}^2 \\ + \left[(\ell_1 + \Delta_1 - 1)(\vec{k}_2\cdot\vec{z}_2)(\vec{z}_1\cdot\vec{K}_{12}) + (1\leftrightarrow 2)\right]. \tag{26}$$

Some weight-shifting operators simultaneously change the spin and conformal weight of the operators they act on. For example, the following operator both lowers the weight by one unit and raises the spin by one unit at points 1 and 2:

$$H_{12} = 2\,(\vec{z}_1\cdot\vec{K}_{12})(\vec{z}_2\cdot\vec{K}_{12}) - (\vec{z}_1\cdot\vec{z}_2)K_{12}^2. \tag{27}$$

---

[10]There is, in principle, an infinite number of different weight-shifting operators, coming from the possible finite-dimensional representations of the conformal group. We restrict our attention to a particular set that is most useful for the purposes of this work, but other weight-shifting operators may be useful in other contexts.

This operator provides a useful alternative to the spin-raising operator $\mathcal{S}_{12}^{++}$, that will be especially convenient for the construction of identically conserved correlators (see §3.3).

So far, all the operators that we have introduced act in the same way at both points, but this is not required. In fact, there are many circumstances where we will want to act differently on the operators in a correlation function. There are two weight-shifting operators that we will find useful to do this:

$$D_{12} = (\Delta_1 + \ell_1 - 1)\vec{z}_1 \cdot \vec{K}_{12} - \frac{1}{2}(\vec{z}_1 \cdot \vec{k}_1)K_{12}^2, \tag{28}$$

$$D_{11} = \left(\Delta_2 - 3 + \vec{k}_2 \cdot \vec{K}_{12}\right)\vec{z}_1 \cdot \vec{K}_{12} - \frac{\vec{z}_1 \cdot \vec{k}_2}{2} K_{12}^2 - (\vec{z}_2 \cdot \vec{K}_{12})\vec{z}_1 \cdot \partial_{\vec{z}_2} + (\vec{z}_1 \cdot \vec{z}_2)\partial_{\vec{z}_2} \cdot \vec{K}_{12}. \tag{29}$$

The first operator, $D_{12}$, raises the spin at point 1 by one unit, while lowering the weight at point 2 by one unit. The second operator, $D_{11}$, lowers the weight by one unit and raises the spin by one unit at the same point (in this case point 1).

Taken together, these weight-shifting operators allow us to generate a large number of correlation functions of spinning operators starting from a relatively small number of "seed" correlation functions. We will provide explicit examples in Sections 4 and 5.

## 3.2 Locality: Ward–Takahashi Identities

The weight-shifting operators described in §3.1 provide us with an efficient and systematic way to construct a large number of solutions to the conformal Ward identities in (23). However, this is not sufficient for spin-$\ell$ operators with the special conformal weights

$$\Delta = t + 2, \tag{30}$$

where $t$ is a positive integer called the *depth*. These currents obey $\partial_{i_1} \cdots \partial_{i_{\ell-t}} J^{i_1 \cdots i_\ell} = 0$ [84,85], which leads to Ward–Takahashi (WT) identities for the correlation functions involving such conserved currents. Derivations of these identities can be found in Appendix B. In this paper, we will only be concerned with the case $t = \ell - 1$, corresponding to single conservation. Particularly important examples are conserved spin-1 currents, which have $\Delta = 2$, and the spin-2 stress tensor, which has $\Delta = 3$. The corresponding WT identities will play a crucial role in the derivation of correlation functions involving these operators.

Consider a correlator involving a conserved spin-1 current, $J_i$, and a set of generic operators, $O_a$, with $a = 2, \ldots, n$. Although classically $\partial_i J^i = 0$, inside of correlation functions this only holds for separated points, and the divergence of the correlator must satisfy the following WT identity

$$\frac{\partial}{\partial x_1^i} \langle J^i(\vec{x}_1)O_2(\vec{x}_2)\cdots O_n(\vec{x}_n)\rangle = -\sum_{a=2}^{n} \delta(\vec{x}_1 - \vec{x}_a)\langle O_2(\vec{x}_2)\cdots \delta O_a(\vec{x}_a)\cdots O_n(\vec{x}_n)\rangle, \tag{31}$$

where $\delta O_a$ stands for the action of the conserved charge associated to $J_i$ on the operator $O_a$. In the case of interest, the operators $O_a$ transform in a linear representation of the symmetry generated by $J_i$. The simplest case is an Abelian current where the operators are charged under a U(1) symmetry, so that $\delta O_a = -ie_a O_a$, where $e_a$ are the charges associated with the operators $O_a$. These charges are part of the data that defines the theory, just like the operator dimensions and spins. In Fourier space, the identity (31) becomes[11]

$$k_1^i \langle J_{\vec{k}_1}^i O_{\vec{k}_2} \cdots O_{\vec{k}_n}\rangle = -\sum_{a=2}^{n} e_a \langle O_{\vec{k}_2} \cdots O_{\vec{k}_a + \vec{k}_1} \cdots O_{\vec{k}_n}\rangle. \tag{32}$$

---

[11]More generally, we could have a multiplet of spin-1 currents, $J_i^A$, in which case the operators in the theory (including the currents themselves) can transform in representations of some non-Abelian group, leading to a WT identity of the form (B.9).

We see that this places a nontrivial constraint on the longitudinal component of the correlator, completely fixing it in terms of lower-point functions.

The spin-2 stress tensor is also conserved, $\partial_i T^{ij} = 0$, which leads to a similar identity for operators in the theory. In Fourier space, the WT identity for the stress tensor reads

$$z_1^i k_1^j \langle T_{\vec{k}_1}^{ij} O_{\vec{k}_2} \cdots O_{\vec{k}_n} \rangle = -\sum_{a=2}^{n} \kappa_a (\vec{z}_1 \cdot \vec{k}_a) \langle O_{\vec{k}_2} \cdots O_{\vec{k}_a + \vec{k}_1} \cdots O_{\vec{k}_n} \rangle, \tag{33}$$

where $\kappa_a$ is the coupling between the stress tensor and the operators in the theory. We will often suppress any flavor structure of the scalar operators, but allow for the possibility that they have different couplings to the stress tensor. Ultimately, we will find that in fact these couplings are required to have the same strength—because of the equivalence principle—but we allow them to be arbitrary at this point in order to see this constraint arise explicitly. Finally, the WT identities for correlators with multiple currents are slightly more complicated and will be introduced as needed.

### 3.3 Identically Conserved Correlators

After finding a "particular solution" to the "inhomogeneous" Ward–Takahashi identity, we are still free to add to it any identically conserved correlators. Moreover, in many cases of interest the right-hand side of the WT identity vanishes, in which case the relevant correlation functions must be identically conserved. Such correlation functions are somewhat simpler to construct. In particular, we can simplify the search for the relevant structures by acting on generic structures involving spinning operators with conformally-*invariant* differential operators that act as projectors onto the conserved structures.[12]

We are primarily interested in conserved currents of spins 1 and 2, so we will now describe the projectors in those cases explicitly.

- **Spin-1:** The simplest example is

$$J_i = \epsilon_{ijk} \partial_k B_j \equiv (\mathsf{P}_1)_{ij} B_j, \tag{34}$$

  which turns general spin-1 operators, $B_i$, into operators, $J_i$, that are identically conserved. In order for this to be consistent with conformal invariance, the operators $J_i$ must have dimension $\Delta = 2$. Since the projection operator in (34) increases the weight by one unit, the seed operators $B_i$ must therefore have $\Delta = 1$. To apply the projection operator in (34) to an operator in the index-free form (2), we must first extract the tensor operator using (3):

$$\mathsf{P}_a^{(1)} \equiv z_a^i (\mathsf{P}_1)_{ij} D_{z_a}^j. \tag{35}$$

  Working with such scalar projection operators reduces index proliferation.

  Since these projectors explicitly involve epsilon symbols, it would seem that they violate parity. However, this is not the case if we restrict our attention to the transverse components of correlation functions. This is done by taking the auxiliary vector, $z^i$, to be a polarization vector, $\xi^i$. Given a current with momentum $k$, polarization vectors are eigenvectors of the two-form $*k$, which implies $\xi_i^{\pm} = \pm \epsilon_{ijl} \xi_j^{\pm} k_l / k$. Contracting the polarization vector $\xi^i$ with the projection operator in (34) then leads to

$$\xi^i (\mathsf{P}_1)_{ij} = k \xi^i, \tag{36}$$

---

[12]We thank Petr Kravchuk and David Simmons-Duffin for discussions of this topic.

where the factor of $k$ effectively performs the shadow transform from $\Delta = 1$ to $\Delta = 2$. This means that it is extremely simple to implement the projection operator on the transverse components of correlators: we just have to contract the correlation functions of $\Delta = 1$ currents with a polarization vector $\xi^i$ and multiply by the magnitude of the momentum to get the (transverse part of) correlation functions involving conserved currents.

- **Spin-2:** Similarly, we can generate a conserved spin-2 operator through the following projection

$$T_{ij} = k^2 \left( \epsilon_{inm} k_m \pi_{jl} + \epsilon_{jnm} k_m \pi_{il} \right) B_{ln} \equiv (\mathsf{P}_2)_{ij,ln} B_{ln}, \tag{37}$$

where we have introduced

$$\pi_{ij}(\hat{k}) \equiv \delta_{ij} - \hat{k}_i \hat{k}_j. \tag{38}$$

The projection operator $(\mathsf{P}_2)_{ij,ln}$ in (37) is traceless in the $ij$ and $ln$ indices and transverse on all indices. It maps a general traceless tensor, $B_{ij}$, to a conserved traceless tensor with $\Delta' = \Delta + 3$. Since we have to land on $\Delta' = 3$, this operator should be applied to $\Delta = 0$ spin-2 operators. It is again useful to introduce a scalar projector

$$\mathsf{P}_a^{(2)} \equiv z_a^i z_a^j (\mathsf{P}_2)_{ij,lm} D_{z_a}^l D_{z_a}^m. \tag{39}$$

As before, the action of the projector is dramatically simpler for polarization vectors. In particular, we have

$$\xi^i \xi^j (\mathsf{P}_2)_{ij,lm} = k^3 \xi_l \xi_m. \tag{40}$$

This means that we can obtain the transverse part of a conserved correlator by simply contracting the correlation function of $\Delta = 0$ spin-2 currents with polarization vectors and multiplying by $k^3$.

Using this approach, it is relatively straightforward to construct identically conserved correlation functions. In many cases of interest these are the only possible contributions, but in other cases we must add contributions whose longitudinal pieces saturate the WT identities.

In this paper, we focus on solutions to the Ward identities corresponding to tree-level bulk processes, but the general strategy is also applicable at loop level. Notice that for correlation functions involving only conserved currents, the three-point function is totally fixed, which implies that all loops can do is possibly re-scale the inhomogeneous solution to the WT identity and shift the correlator by identically conserved pieces (at least at four points). This indicates that there is some universal piece—already present at tree level—that characterizes correlation functions of conserved currents, even accounting for loops.

## 3.4 Cosmological Spinor Helicity Variables

A disadvantage of the treatment described above is that we must keep track of the longitudinal polarizations (i.e. terms proportional to $\vec{z}_a \cdot \vec{k}_a$) in order to check the WT identities. As we include more and more spinning external operators, this will quickly become rather cumbersome. Since these pieces do not contribute to the correlators with transverse and traceless polarization states that we are interested in, it would be preferable to have a way to check the WT identity knowing only these transverse parts of correlation functions. It turns out that this is indeed possible, if we first write the correlators in spinor helicity variables. In this section, we will give a brief introduction to the spinor helicity formalism and then show that it provides a convenient way to impose the WT identities directly on correlators for states with definite helicities.

**Spinor helicity variables**

Spinning correlators are simplest in variables which manifest the helicity transformations of external operators. This is accomplished by rewriting momentum vectors in terms of spinor representations of the group of spatial rotations. Much as in flat space, it is convenient to complexify momenta by decomposing them into spinor representations of $\mathrm{SL}(2,\mathbb{C})$ (which is the complexification of the three-dimensional rotation group). Concretely, we convert momenta to helicity variables via the relation

$$\lambda_\alpha \bar{\lambda}^\beta = k_i (\sigma^i)_\alpha^\beta + k\, \mathbb{1}_\alpha^\beta\,, \tag{41}$$

where $(\sigma^i)_\alpha^\beta$ are the usual Pauli matrices and $k$ is the magnitude of the momentum $k^i$. Given a set of spinors, we can recover the original momentum vector via the inverse relation

$$k^i = \frac{1}{2}(\sigma^i)_\beta^\alpha \lambda_\alpha \bar{\lambda}^\beta\,. \tag{42}$$

For complex momenta, the variables $\lambda$ and $\bar{\lambda}$ can be thought of as independent—they will be related by a reality condition if we want to specialize to real momenta.

An important feature of (41) is that it does not uniquely assign the spinors $\lambda$ and $\bar{\lambda}$ to a particular spatial momentum vector. Instead, the transformation

$$\begin{aligned} \lambda_\alpha &\mapsto r\lambda_\alpha\,,\\ \bar{\lambda}^\alpha &\mapsto r^{-1}\bar{\lambda}^\alpha\,, \end{aligned} \tag{43}$$

leaves the three-momentum $k_i$ invariant. The eigenvalue under this $\mathrm{U}(1) \subset \mathrm{SL}(2,\mathbb{C})$ transformation is the helicity.

Correlators involving massless operators are rotationally invariant, so the spinor indices must be contracted in some way. There is a natural $\mathrm{SL}(2,\mathbb{C})$-invariant pairing between spinors given by $\epsilon^{\alpha\beta}$, and we denote the corresponding products with angle brackets,

$$\langle ab \rangle = \epsilon^{\alpha\beta} \lambda_\alpha^a \lambda_\beta^b\,, \tag{44}$$

$$\langle a\bar{b} \rangle = \epsilon^{\alpha\beta} \lambda_\alpha^a \bar{\lambda}_\beta^b\,. \tag{45}$$

Throughout this paper, we will always define these brackets as contractions of spinors with a *raised* epsilon symbol.[13]

In a cosmological background there is a distinguished time direction, which makes these spinor variables substantially different from their flat-space counterparts. For example, we can extract the energy associated to a pair of spinors by considering the mixed bracket

$$\langle \lambda\bar{\lambda} \rangle = -2k\,. \tag{46}$$

Similarly, we no longer have energy conservation in the situations of interest, but momentum is still conserved, which implies that the sum of spinors is proportional to the total energy[14]

$$\sum_{a=1}^n \lambda_\alpha^a \bar{\lambda}_\beta^a = E\,\epsilon_{\alpha\beta}\,, \tag{47}$$

---

[13]Additionally, it is often necessary to raise and lower spinor indices. We adopt the convention that indices are raised and lowered by contracting with the first index of the epsilon symbol, e.g. $\bar{\lambda}_\alpha = \epsilon_{\beta\alpha}\bar{\lambda}^\beta$.

[14]The placement of the spinor indices is important here. The analogous identity with raised indices would have a $-E$ on the right-hand side.

where $E \equiv \sum_a k_a$. A further important difference from the usual spinor helicity variables in flat space (see §C.1) is that polarization vectors can be defined without the need for an auxiliary spinor

$$\xi^+_{\alpha\beta} = \frac{\bar\lambda_\alpha \bar\lambda_\beta}{2k} \quad \text{and} \quad \xi^-_{\alpha\beta} = \frac{\lambda_\alpha \lambda_\beta}{2k}. \tag{48}$$

Looking at (48), it is clear that the transformation of the spinors under the rescaling (43) is what carries the helicity information.

We have presented an intrinsically boundary construction of the relevant spinor-helicity variables. See Appendix C for an explanation of the relation between this construction and four-dimensional spinor-helicity variables.

**Conformal generators and WT identities**

Besides providing an efficient way to describe the polarizations of the external states, the spinor helicity formalism also gives a simple way to check the WT identities. For this purpose, it is useful to consider the special conformal generator in spinor variables [86]

$$\widetilde{K}^i = 2(\sigma^i)^\beta_\alpha \frac{\partial^2}{\partial \lambda_\alpha \partial \bar\lambda^\beta}. \tag{49}$$

This differential operator acts on operators in different ways, depending on their quantum numbers [55]. For example, its action on $\Delta = 2$ scalars, $\varphi$, is

$$\widetilde{K}^i \varphi = -K^i \varphi, \tag{50}$$

where $K^i$ is the usual special conformal generator (22). Acting on conserved spin-1 and spin-2 currents, we instead get (see Appendix D)[15]

$$\widetilde{K}^i J^\pm = \left(-\xi^j_\pm K^i + 2\xi^i_\pm \frac{k^j}{k^2}\right) J_j, \tag{53}$$

$$\widetilde{K}^i \left(\frac{T^\pm}{k}\right) = \left(-\frac{1}{k}\xi^{(j}_\pm \xi^{l)}_\pm K^i + 12\xi^i_\pm \frac{\xi^{(j}_\pm k^{l)}}{k^3}\right) T_{jl}, \tag{54}$$

where $J^\pm \equiv \xi^\pm_j J^j$ and $T^\pm \equiv \xi^\pm_j \xi^\pm_l T^{jl}$ are the currents in the helicity basis. These formulas show that $\widetilde{K}$ not only acts on conserved currents like the conformal generator, but also contains a piece proportional to the divergence of the current. The operator $\widetilde{K}$ therefore reconstructs the longitudinal components of correlation functions purely from the corresponding correlators with definite helicities. This means that we can drop the longitudinal parts of correlation functions without losing any information.

The differential operator (49) is practically very useful for solving the WT identities. By utilizing weight-shifting operators, we are constructing correlators that are annihilated by the special conformal generator $K^i$. The action of $\widetilde{K}^i$ on these correlators therefore simply generates their longitudinal parts, which we then demand to satisfy the WT identity. Schematically, the action of $\widetilde{K}^i$ on a correlation function involving a conserved current then is

$$\widetilde{K}^i \langle J^\pm_1 O_2 \cdots O_n \rangle \sim \xi^i_\pm k_1 \cdot \langle J_1 O_2 \cdots O_n \rangle. \tag{55}$$

---

[15] Acting on tensors with uncontracted indices, the conformal generator takes the form

$$K^i O_{j_1 \cdots j_\ell} = \left[2(\Delta - 3)\partial_{k^i} + k^i \partial_{k^m}\partial_{k^m} - 2k^m \partial_{k^m}\partial_{k^i} - 2\partial_{k^m}\Sigma_{im}\right] O_{j_1 \cdots j_\ell}, \tag{51}$$

where

$$\Sigma_{im} O_{j_1 \cdots j_\ell} = \ell \left(O_{i(j_1 \cdots} \delta_{j_\ell)m} - O_{m(j_1 \cdots} \delta_{j_\ell)m}\right) \tag{52}$$

is the action of the generator of rotations in the vector representation.

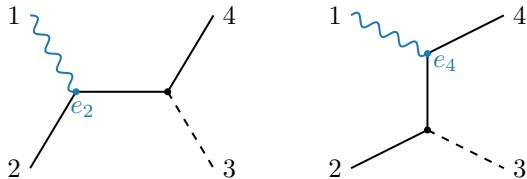

Figure 4: Feynman diagrams of the $s$- and $t$-channel contributions to photon-induced pion production.

Demanding the right-hand side of this equation to be consistent with the WT identity (32) is a nontrivial constraint on the form of the correlation function. In particular, at four points this constraint relates particle exchange in different channels.

### 3.5 Consistency Requires Multiple Channels

Starting at four points, there is an interesting complication to the general strategy sketched above. From the bulk perspective, four-point functions can arise from the exchange of particles in various channels, leading to different possible kinematic structures for the boundary correlators. We will see that the conformal Ward identities and the Ward–Takahashi identities can only be solved simultaneously if the different channels are related to each other, leading to nontrivial constraints on the couplings between conserved currents and other operators in the theory. Before we describe these constraints in the cosmological context, it is useful to review how these consistency constraints can arise in flat space.

**A flat-space example**

The scattering of massless particles in flat space is highly constrained, leading to a very small list of consistent interactions. In fact, demanding consistency of the four-particle $S$-matrix leads to powerful constraints on the space of viable quantum field theories [2–4, 87]. We will give a very simple illustration of these restrictions by showing how gauge invariance of the $S$-matrix requires both a combination of multiple channels and charge conservation.

Consider the following process in the $s$-channel: a photon (particle 1) is absorbed by a charged scalar (particle 2) which then decays into two scalar particles (particles 3 and 4); see Fig. 4 for an illustration. For simplicity, we will take particle 3 to be un-charged. In the Standard Model, such a scattering process describes neutral pion photo-production from charged pions. The $s$-channel contribution to the scattering amplitude is

$$A_s = e_2 \frac{\epsilon_1 \cdot p_2}{S}, \tag{56}$$

where $S \equiv -(p_1 + p_2)^2$ is the Mandelstam invariant, $e_2$ is the coupling of the photon to particle 2 (i.e. it is the charge of this particle), and we have set the coupling between the scalar particles to unity. For simplicity, we have taken all particles to be massless. This amplitude is not gauge-invariant: substituting $\epsilon_1 \mapsto p_1$, the amplitude does not vanish, but instead becomes $A_s \to -\frac{1}{2} e_2 \neq 0$. To rectify this problem, we must add the $t$-channel contribution:

$$A_t = e_4 \frac{\epsilon_1 \cdot p_4}{T}, \tag{57}$$

where $T \equiv -(p_1 + p_4)^2$ and $e_4$ is the charge of particle 4. The total amplitude will be gauge-invariant only if $e_2 = -e_4 \equiv e$, meaning that the charge is conserved.[16] We then have

$$A_{s+t} = e \left( \frac{\epsilon_1 \cdot p_2}{S} - \frac{\epsilon_1 \cdot p_4}{T} \right), \tag{58}$$

---

[16]Recall that all momenta are defined as incoming momenta.

which indeed vanishes upon the substitution $\epsilon_1 \mapsto p_1$. We see that demanding gauge invariance of the amplitude has forced us to have both $s$- and $t$-channel contributions.

The fact that the individual channels are not gauge-invariant tells us that splitting them in this way is somewhat arbitrary—exchanges in a given channel do not have any independent physical meaning. It is therefore desirable to phrase things in a slightly more on-shell language. This requires working in terms of spinor helicity variables. We will see that there is an essential tension between locality and the correct factorization of amplitudes when intermediate particles go on shell.

Consider the same scattering process where the photon has negative helicity ($1^-$). The form of the amplitude in the $s$-channel is fixed by the correct factorization on the pole at $S = 0$. In four-dimensional spinor helicity variables (see Appendix C), we get[17]

$$A_s = e_2 \frac{\langle 12 \rangle \langle 1I_s \rangle}{\langle 2I_s \rangle} \times \frac{1}{S} \times 1 = +e_2 \frac{\langle 12 \rangle [24] \langle 41 \rangle}{ST}, \tag{60}$$

where $I_s \equiv p_1 + p_2$ and $T = \langle 14 \rangle [14]$. The fact that the residue of the $s$-channel pole has a pole at $T = 0$ means that we also have to consider factorization in the $t$-channel to get a consistent amplitude. An amplitude that factorizes correctly in the $t$-channel is

$$A_t = e_4 \frac{\langle 14 \rangle \langle 1I_t \rangle}{\langle 4I_t \rangle} \times \frac{1}{T} \times 1 = -e_4 \frac{\langle 12 \rangle [24] \langle 41 \rangle}{ST}, \tag{61}$$

where $I_t \equiv p_1 + p_4$ and $S = \langle 12 \rangle [12]$. The goal then is to find an amplitude at general kinematics which factorizes correctly in both channels. It is clear that this is only possible if $e_2 = -e_4 \equiv e$, in which case the total amplitude is

$$A_{s+t} = e \frac{\langle 12 \rangle [24] \langle 41 \rangle}{ST}. \tag{62}$$

This amplitude has the correct residues on both the $s$- and $t$-channel poles. Interestingly, demanding consistency of the two factorization channels has fixed the amplitude completely.

**One channel is not enough**

Consistency constraints on correlation functions run parallel to those of scattering amplitudes. Individual channels satisfy the conformal Ward identities, but a sum of exchanges in multiple channels is needed to satisfy the Ward–Takahashi identities. The latter play a role analogous to the requirement of gauge invariance of the $S$-matrix. In the case of the S-matrix, the individual Feynman diagrams generate Lorentz-covariant tensors. After contracting the answers with polarization vectors, the resulting objects are, in general, *not* Lorentz invariant. This is because, despite appearances, polarization vectors do not transform as Lorentz vectors. The requirement of gauge invariance then becomes equivalent to imposing Lorentz invariance of the full $S$-matrix. Likewise, in cosmology, the correlators corresponding to individual exchange channels are de Sitter covariant, being solutions of the conformal Ward identities. However, if their contractions with polarization vectors fail to obey the WT identity, it means that the results for the particular exchange channel by itself is not conformally invariant.

In practice, we implement the WT constraint by acting with the operator $\widetilde{K}$ on the four-point function of a conserved operator with exchange in a given channel. We will find that we must introduce exchanges in additional channels with correlated couplings to obtain consistent correlators. This will reproduce bulk facts like charge conservation and the equivalence principle from a purely boundary perspective.

---

[17]To obtain the second equality in (60), we performed the following spinor manipulations

$$\frac{\langle 1I_s \rangle}{\langle 2I_s \rangle} = \frac{\langle 1I_s \rangle}{\langle 2I_s \rangle} \cdot \frac{[I_s 4]}{[I_s 4]} = \frac{\langle 1|I_s|4]}{\langle 2|I_s|4]} = \frac{\langle 12 \rangle [24]}{\langle 21 \rangle [14]} = -\frac{[24]}{[14]} = \frac{[24] \langle 41 \rangle}{T}. \tag{59}$$

# 4 Three-Point Functions from Weight-Shifting

We now have all the technical machinery required to begin our study of correlation functions involving conserved operators. As a first step, we consider three-point functions. Although it has long been known that conformal invariance completely fixes the form of correlation functions for three local operators up to a finite number of coefficients [88,89], most of the classic results are phrased in position space, while cosmological applications require results in momentum space. We will see that the weight-shifting procedure allows us to easily generate these three-point functions involving conserved currents.[18] Our goal is not to be entirely exhaustive, but rather to illustrate our approach in a variety of examples.

## 4.1 Scalar Seed Correlators

Our strategy for obtaining general solutions to the conformal Ward identities in (23) is to relate them to known scalar solutions. It is therefore useful to first collect the relevant scalar seed correlators.

- The three-point function for generic scalar operators $O_a$ (of dimensions $\Delta_a$) is known in Fourier space in various forms. Its most economical representation is as an integral over Bessel-$K$ functions [56]

$$\langle O_1 O_2 O_3 \rangle = k_1^{\Delta_1 - \frac{3}{2}} k_2^{\Delta_2 - \frac{3}{2}} k_3^{\Delta_3 - \frac{3}{2}} \int_0^\infty dz\, z^{\frac{1}{2}} K_{\Delta_1 - \frac{3}{2}}(k_1 z) K_{\Delta_2 - \frac{3}{2}}(k_2 z) K_{\Delta_3 - \frac{3}{2}}(k_3 z), \quad (63)$$

where the overall normalization is not fixed by conformal symmetry. For general weights, the integral can also be written in terms of the Appell $F_4$ function [56,61], a two-variable generalized hypergeometric function. For weights $\Delta_a$ that lead to Bessel functions of half-integral order, the integral can be evaluated in terms of elementary functions (some examples of which are given below).

- The three-point function of $\Delta = 2$ scalars $\varphi$ is given by

$$\langle \varphi \varphi \varphi \rangle = \log(K/\mu), \quad (64)$$

where $K \equiv k_1 + k_2 + k_3$. This expression solves the Ward identities "anomalously." The scale variation of the logarithm does not vanish, but is instead a function that is analytic in the momenta (in this case it is just a constant), indicating that it is a contact term in position space. This correlation function therefore satisfies the conformal Ward identities at separated points, which is all that is required. Note that we can freely add an arbitrary constant to this correlation function by shifting the (arbitrary) scale $\mu$.[19]

- The three-point function of $\Delta = 3$ scalars $\phi$ is [96]

$$\langle \phi \phi \phi \rangle = \log(K/\mu) \sum_a k_a^3 - \sum_{a \neq b} k_a^2 k_b + k_1 k_2 k_3. \quad (65)$$

The term involving the logarithm again only solves the conformal Ward identities at separated points, and changes in $\mu$ correspond to the freedom to add the arbitrary local term, $\sum_a k_a^3$, which solves the Ward identities by itself.

---

[18]For previous work analyzing three-point functions in flat and curved space, and their constraints coming from Ward identities, see [89–95].

[19]From the $\Delta = 2$ three-point function, it is straightforward to obtain the three-point function of $\Delta = 1$ scalars: $\langle \widetilde{\varphi} \widetilde{\varphi} \widetilde{\varphi} \rangle = (k_1 k_2 k_3)^{-1}$, which we can think of as the shadow transform of the constant that can be added to the result (64); in Fourier space, this amounts to multiplying by $(k_1 k_2 k_3)^{-1}$. In this case, the shadow transform of the logarithm is not conformally invariant; it is invariant under special conformal transformations, but does not satisfy the dilation constraint (even anomalously).

- The three-point function of two $\Delta = 2$ scalars and a scalar of general dimension $\Delta$ is [22]:

$$\langle \varphi \varphi O \rangle = k_3^{\Delta-2} {}_2F_1\left[\begin{array}{cc} 2-\Delta, & \Delta-1 \\ & 1 \end{array} \middle| \frac{1-p}{2}\right],\qquad(66)$$

where $p \equiv (k_1 + k_2)/k_3$. This solution is valid for generic $\Delta$ in the principal series, but naive continuation to integer weight representations does not reproduce the logarithms or contact terms present in those correlation functions. In cases where the third operator also belongs to a special representation, the other results above should be used.

- Other mixed correlators can be obtained by acting with appropriate weight-shifting operators. For example, acting with the weight-lowering operator $\mathcal{W}_{12}^{--}$ on (65) gives

$$\begin{aligned} \langle \varphi \varphi \phi \rangle &= \mathcal{W}_{12}^{--} \langle \phi \phi \phi \rangle \\ &\propto (k_1 + k_2)\log(K/\tilde{\mu}) - K\,, \end{aligned}\qquad(67)$$

where we have shifted the scale $\mu$ as $\log(\tilde{\mu}/\mu) = -7/12$ in order to make the answer more symmetric. It can be checked straightforwardly that this correlation function is consistent with the result of an explicit bulk computation.

## 4.2 Correlators with Spin-1 Currents

As a first illustration of the spin-raising procedure, we consider three-point correlators involving conserved spin-1 currents.

### 4.2.1 ⟨JOO⟩

We begin with a correlator involving one spin-1 current and two general operators. This correlator must satisfy the following Ward–Takahashi identity

$$k_1^i \langle J_{\vec{k}_1}^i O_{\vec{k}_2} O_{\vec{k}_3} \rangle = -e_2 \langle O_{\vec{k}_2+\vec{k}_1} O_{\vec{k}_3} \rangle - e_3 \langle O_{\vec{k}_2} O_{\vec{k}_3+\vec{k}_1} \rangle\,,\qquad(68)$$

where spin and conformal weight labels have been suppressed.

We first consider the case where the two additional operators are scalars, which must have equal weights in order for any conformally-invariant structure to exist [97].[20] A candidate for the correlator $\langle JO_\Delta O_\Delta \rangle$ can be constructed by acting with $D_{11}$ on the three-point scalar correlator with one $\Delta = 3$ scalar and two scalars of general weight[21]

$$\langle JO_\Delta O_\Delta \rangle = D_{11} \langle \phi\, O_\Delta O_\Delta \rangle\,.\qquad(69)$$

The operator $D_{11}$ both raises the spin and lowers the weight of the first operator to the conserved value $\Delta_J = 2$. In this case, there is a nontrivial WT identity, given by (68), that we have to verify is satisfied. For generic scalars, the result is not easily expressed in terms of elementary functions. However, for special values of $\Delta$ the answer dramatically simplifies (and indeed can be written as a rational function). As an example, consider the correlator involving

---

[20]It is well-known that the two-point function of operators in a conformal field theory vanishes for unequal weights: $\langle O_\Delta O_{\Delta'} \rangle \propto \delta_{\Delta\Delta'}$. The right-hand side of (68) then vanishes, and the correlator $\langle JO_\Delta O_{\Delta'} \rangle$ can be constructed using the projection operators introduced in §3.3. However, it turns out that the projector actually annihilates any putative correlator, so the correlation function of a spin-1 conserved current with two scalar operators must vanish if the scalars have unequal weights.

[21]In order for this correlation function to be nonzero, the scalar operators must contain additional flavor indices and be antisymmetric under their exchange. Since our focus is on kinematic information, we suppress these labels.

two $\Delta = 2$ scalars; this can be obtained by acting with $D_{11}$ on (67). In that case, the expression is easy to evaluate, and we find

$$\langle J \varphi \varphi \rangle = D_{11} \langle \phi \varphi \varphi \rangle = \frac{2}{K} (\vec{k}_2 \cdot \vec{z}_1) + \frac{k_1 - k_2 + k_3}{k_1 K} (\vec{k}_1 \cdot \vec{z}_1), \tag{70}$$

where $J \equiv \vec{z}_1 \cdot \vec{J}_{\vec{k}_1}$. We must still check that this result is compatible with the WT identity (68). Letting $\vec{z}_1 \to \vec{k}_1$ in (70), we get

$$\vec{k}_1 \cdot \langle \vec{J}_{\vec{k}_1} \varphi_{\vec{k}_2} \varphi_{\vec{k}_3} \rangle = N_{J\varphi\varphi}(k_3 - k_2), \tag{71}$$

where we have introduced the amplitude of the three-point function, $N_{J\varphi\varphi}$. Using[22] $\langle \varphi \varphi \rangle = k$, we see that this is only consistent with (68) if

$$e_2 + e_3 = 0, \quad N_{J\varphi\varphi} = -e_2. \tag{72}$$

We have therefore discovered that the three-point function is only nonzero if the total charge is conserved, and found that the normalization of this three-point function is fixed by the charges. Of course, this is expected from the bulk point of view. When the kinetic term for the scalar field is written in covariant form, by coupling it to a gauge field, the cubic interactions are fixed by gauge invariance to be proportional to the charge. However, it is satisfying to reproduce these bulk facts purely from a boundary perspective.

An alternative way to check the WT identity is to act with the operator (49) on (70) in spinor helicity variables (see §3.4 and Appendix C). In terms of these variables, the correlator takes the form[23]

$$\langle J^- \varphi \varphi \rangle = N_{J\varphi\varphi} \frac{\langle 12 \rangle \langle \bar{2}1 \rangle}{2k_1} \frac{1}{K}, \tag{74}$$

where the result for positive helicity is related by parity (swapping barred and un-barred spinors). Acting with (49) on (74) leads to the expression

$$\sum_{a=1}^{3} \vec{b} \cdot \widetilde{K}_a \langle J_1^- \varphi_2 \varphi_3 \rangle = \frac{2\vec{b} \cdot \vec{\xi}_1^-}{k_1^2} (k_3 - k_2) N_{J\varphi\varphi}. \tag{75}$$

Using (53) and (50), we can also write the left-hand side as

$$\sum_{a=1}^{3} \vec{b} \cdot \widetilde{K}_a \langle J_1^- \varphi_2 \varphi_3 \rangle = \frac{2\vec{b} \cdot \vec{\xi}_1^-}{k_1^2} \vec{k}_1 \cdot \langle \vec{J}_{\vec{k}_1} \varphi_{\vec{k}_2} \varphi_{\vec{k}_3} \rangle = -\frac{2\vec{b} \cdot \vec{\xi}_1^-}{k_1^2} \left[ e_2 k_3 + e_3 k_2 \right], \tag{76}$$

where we have used the WT identity (68) and substituted $\langle \varphi \varphi \rangle = k$. We see that the results (75) and (76) are only consistent if (72) holds.

Given a correlation function that saturates the WT identity (70), we can add to it the most general identically conserved correlation function. However, it is relatively easy to check that the projector (35) annihilates all kinematically-allowed possibilities and the structure (71) is therefore unique. The result is also consistent with a bulk calculation of the correlator between a photon and a conformally coupled scalar in de Sitter, and matches the answer in [56].[24]

---

[22]The normalization of the two-point function $\langle \varphi \varphi \rangle$ can be changed by re-defining the operators. For simplicity, we have chosen $N_{\varphi^2} = 1$.

[23]Using the identity

$$\langle \bar{2}1 \rangle = (k_3 + k_2 - k_1) \frac{\langle 13 \rangle}{\langle 23 \rangle}, \tag{73}$$

we can write this correlator in variables that are well-defined in four dimensions. It can then be checked that the residue of the pole at $E = 0$ is the flat-space scattering amplitude for a photon and two massless scalars.

[24]In [56], the expression for a three-point function involving a conserved current and two $\Delta = 1$ scalars is given. This can be shadow transformed to give our $\Delta = 2$ result.

Finally, we let one of the operators have spin $\ell$, i.e. we wish to determine $\langle JOO^{(\ell)} \rangle$. In that case, the right-hand side of the WT identity vanishes, and the result (if it exists) must come from the projection operator introduced in §3.3. If the third operator has spin 1, we can write the correlator as

$$\langle J\varphi O^{(1)} \rangle = \mathsf{P}_1^{(1)} k_1 D_{11} k_2 D_{32} \langle \varphi \varphi O \rangle \,, \tag{77}$$

which has a compact expression in terms of the scalar three-point function. This three-point function comes from the bulk diagram with vertex $\varphi F_{\mu\nu} G^{\mu\nu}$, where $G_{\mu\nu}$ is the "field strength" of the massive spin-1 particle.

### 4.2.2 $\langle JJO \rangle$

Next, we consider the correlator of two conserved spin-1 currents and a generic (non-conserved) operator. Since the two-point function $\langle JO_\Delta^{(\ell)} \rangle$ necessarily vanishes, the Ward–Takahashi identity simply reads

$$k_1^i \, \langle J_{\vec{k}_1}^i J_{\vec{k}_2} O_{\vec{k}_3} \rangle = 0 \,, \tag{78}$$

where we have suppressed the spin and conformal weight labels. These correlation functions can therefore be constructed completely using projectors.

Consider first the case where the third operator is a scalar. By starting with the three-point correlator $\langle \varphi\varphi O_\Delta \rangle$ given in (66) and acting with the operator $H_{12}$, given in (27), we obtain a correlation function for two currents of spin-1 and weight $\Delta = 1$. This is the shadow dimension to a conserved current, so we can apply the projector (35) to obtain an identically conserved three-point function

$$\langle JJO_\Delta \rangle = \mathsf{P}_1^{(1)} \mathsf{P}_2^{(1)} H_{12} \langle \varphi\varphi O_\Delta \rangle \,. \tag{79}$$

The fact that this correlation function is identically conserved reflects the fact that it arises from a non-minimal coupling of the form $\sigma F_{\mu\nu}^2$ in the bulk, constructed from gauge-invariant objects.

As an analytically tractable example, we consider the special case where the scalar operator has $\Delta = 2$.[25] In this case, we find[26]

$$
\begin{aligned}
\langle JJ\varphi \rangle &= \mathsf{P}_1^{(1)} \mathsf{P}_2^{(1)} H_{12} \langle \varphi\varphi\varphi \rangle \\
&= \frac{(k_1 + k_2 - k_3)}{2K} (\vec{z}_1 \cdot \vec{z}_2) + \frac{1}{K^2} (\vec{k}_1 \cdot \vec{z}_2)(\vec{k}_2 \cdot \vec{z}_1) - \frac{(k_1^2 + k_2^2 - k_3^2)}{2k_1 k_2 K^2} (\vec{k}_1 \cdot \vec{z}_1)(\vec{k}_2 \cdot \vec{z}_2) \\
&\quad - \frac{k_2}{k_1 K^2} (\vec{k}_1 \cdot \vec{z}_1)(\vec{k}_1 \cdot \vec{z}_2) - \frac{k_1}{k_2 K^2} (\vec{k}_2 \cdot \vec{z}_1)(\vec{k}_2 \cdot \vec{z}_2) \,.
\end{aligned}
\tag{80}
$$

In spinor helicity variables, this becomes (see also Appendix B of [98])

$$\langle J^- J^- \varphi \rangle = \frac{\langle 12 \rangle^2}{4K^2} \,, \tag{81}$$

$$\langle J^- J^+ \varphi \rangle = 0 \,. \tag{82}$$

We see that, when the third particle is a scalar, choosing opposite helicities causes the correlator to vanish, by angular momentum conservation. We therefore have only one independent structure corresponding to the case of equal helicities. Were we to apply a weight-lowering operator and the projector again, we would obtain a result proportional to the same three-point function, thus not generating a new structure.

---

[25]The result for $\Delta = 3$ can also easily be generated by acting on $\langle \varphi\varphi\phi \rangle$.

[26]If we shadow transform the scalar operator to $\Delta = 1$, this matches the expression found by [56].

We can also consider the case where the third operator carries spin. In this case, it is straightforward to spin-up the scalar operator in (79), by using the operator $D_{33}$ in (29). Applying this operator $\ell$ times will generate the correlation function $\langle JJO^{(\ell)}_{\Delta-\ell}\rangle$. There are then two distinct cases: when $\ell$ is odd, the resulting structure $\langle JJO^{(\ell)}\rangle$ is antisymmetric in the currents. If the $J$'s are the same current, there would therefore be *zero* kinematically allowed correlators—this is the correlator version of the (generalized) Landau–Yang theorem [99,100], which states that a massive spin-1 particle cannot decay into two photons. When $\ell$ is even, there are *two* kinematically allowed structures, corresponding to equal and opposite helicities for the currents. Starting at spin 2, we obtain other structures by applying the weight-lowering operator plus projectors:

$$\langle JJO^{(\ell)}\rangle_B = \mathsf{P}_1^{(1)}\mathsf{P}_2^{(1)}\mathcal{W}_{12}^{--}\langle JJO^{(\ell)}\rangle_A. \tag{83}$$

As alluded to above, performing this procedure for $\ell = 0, 1$ would not produce a new structure, because the resulting correlator would be proportional to its seed.

### 4.2.3 $\langle JJJ\rangle$

Finally, we consider the three-point function of three conserved spin-1 currents. The novelty compared to the previous examples lies in the fact that there are now two structures—one that solves the Ward–Takahashi identity nontrivially and another that is identically conserved.

The WT identity for three currents is given by

$$\vec{k}_1 \cdot \langle J^A_{\vec{k}_1} J^B_{\vec{k}_2} J^C_{\vec{k}_3}\rangle = f^{ABD}\langle J^D_{\vec{k}_2+\vec{k}_1} J^C_{\vec{k}_3}\rangle - f^{ADC}\langle J^B_{\vec{k}_2} J^D_{\vec{k}_3+\vec{k}_1}\rangle. \tag{84}$$

For simplicity, we will restrict our attention to the case where the tensors $f^{ABC}$ are totally antisymmetric.[27] We therefore only have to impose the WT identity for one current, and it will then be satisfied for all of the currents by permutation symmetry.

There are several ways to generate a correlation function with the correct kinematics. One possibility is to start with the correlator $\langle J\varphi\varphi\rangle$ and raise the spin of the second and third operator using $\mathcal{S}_{23}^{++}$:

$$\langle JJJ\rangle = \mathcal{S}_{23}^{++}\langle J_1\varphi_2\varphi_3\rangle + \text{cyclic perms.}, \tag{85}$$

where we have suppressed the color indices and summed over cyclic permutations (which effectively antisymmetrizes the kinematic factor). We can then check that this correlator does indeed solve the WT identity (84) if we normalize it by $f^{ABC}$.

In order to simplify the later construction of four-point correlation functions (see Section 5), it is convenient to construct this correlation function in a more intricate way. Rather than starting with the correlation function $\langle J\varphi\varphi\rangle$, we instead consider $\langle\phi\phi J\rangle$ and act with the following operator

$$\begin{aligned}\langle JJJ\rangle &= (H_{12} + D_{11}D_{22} - 2D_{12}D_{21})\langle\phi\phi J\rangle \\ &\equiv \mathcal{D}_{12}^{(1)}\langle\phi\phi J\rangle.\end{aligned} \tag{86}$$

The specific linear combination of weight-shifting operators has been chosen in order to generate a correlation function that satisfies the WT identity.[28] The important feature of this

---

[27]There are no possible interactions between three spin-1 currents for symmetric $f^{ABC}$. In the context of QED, this goes by the name Furry's theorem [101]. There are kinematically satisfactory correlators if the tensors $f^{ABC}$ have mixed symmetry, but these structures are not consistent with conservation of all three vector operators, so we do not consider them here.

[28]We do not have an independent justification for this precise combination of weight-shifting paths, beyond the fact that it produces the correct structure that satisfies the WT identity. It would be very interesting to understand if this combination of operators has an independent interpretation.

approach is that it only requires acting on two of the operators. Both (85) and (86) yield the same correlation function, which in spinor variables takes the form

$$\langle J^- J^- J^- \rangle_{\text{YM}} = f^{ABC} \langle 12 \rangle \langle 23 \rangle \langle 31 \rangle \frac{1}{k_1 k_2 k_3}, \tag{87}$$

$$\langle J^- J^- J^+ \rangle_{\text{YM}} = f^{ABC} \langle 12 \rangle \langle 2\bar{3} \rangle \langle \bar{3}1 \rangle \frac{k_1 + k_2 - k_3}{k_1 k_2 k_3} \frac{1}{K}, \tag{88}$$

where we have introduced the color factor $f^{ABC}$ as the normalization (but suppressed the color indices on the left-hand side). It is straightforward to check using the operator (49) that this correlation function satisfies the WT identity [42]. From the bulk perspective, the correlator above is generated by the cubic Yang–Mills vertex $f_{ABC} \partial_\mu A^A_\nu A^{B\,\mu} A^{C\,\nu}$.[29]

Having found a structure that saturates the WT identity, we are free to add to it an identically conserved correlation function, constructed using the projector (35). We act on $\langle J\varphi\varphi \rangle$ with $H_{23}$ to generate the correlation function $\langle JBB \rangle$, which involves three spin-1 operators, one with $\Delta = 2$ and two with $\Delta = 1$, and then project onto the identically conserved structure:

$$\langle JJJ \rangle = \mathsf{P}^{(1)}_2 \mathsf{P}^{(1)}_3 H_{23} \langle J_1 \varphi_2 \varphi_3 \rangle. \tag{89}$$

It turns out that, despite the fact that we have only projected two of the operators onto their conserved structure, the resulting correlation function is identically conserved in all three arguments. In terms of spinor variables, we get

$$\langle J^- J^- J^- \rangle_{F^3} = f^{ABC} \langle 12 \rangle \langle 23 \rangle \langle 31 \rangle \frac{1}{K^3}, \tag{90}$$

$$\langle J^- J^- J^+ \rangle_{F^3} = 0. \tag{91}$$

From the bulk perspective, this identically conserved correlator is generated by the curvature coupling $f_{ABC} F^{A\,\mu}_\nu F^{B\,\nu}_\rho F^{C\,\rho}_\mu$. The most general three-point correlator for conserved spin-1 currents is a mixture of the two structures found above.

## 4.3 Correlators with Spin-2 Currents

Next, we perform a similar analysis for correlators involving conserved spin-2 currents, i.e. the stress tensor operator.

### 4.3.1 $\langle TOO \rangle$

The correlator with a single conserved spin-2 current is very similar to the spin-1 case. In this case, we must satisfy the stress tensor Ward–Takahashi identity

$$z^i_1 k^j_1 \langle T^{ij}_{\vec{k}_1} O_{\vec{k}_2} O_{\vec{k}_3} \rangle = -\kappa_2 (\vec{k}_2 \cdot \vec{z}_1) \langle O_{\vec{k}_2 + \vec{k}_1} O_{\vec{k}_3} \rangle - \kappa_3 (\vec{k}_3 \cdot \vec{z}_1) \langle O_{\vec{k}_2} O_{\vec{k}_3 + \vec{k}_1} \rangle, \tag{92}$$

where spin and conformal weight labels have been suppressed.

Let the two operators $O_2$ and $O_3$ be scalars. As in the spin-1, the correlator then vanishes unless the scalar operators have the same weight.[30] A candidate correlator can be generated by acting with the operators $D_{12}$ and $D_{13}$ on the three-point function of a $\Delta = 3$ scalar $\phi$ with two general weight scalars

$$\langle TO_\Delta O_\Delta \rangle = D_{12} D_{13} \langle \phi\, O_{\Delta+1} O_{\Delta+1} \rangle. \tag{93}$$

---

[29]Notice that the $---$ correlator does not have the usual singularity at $E = 0$. This is because the flat-space amplitude vanishes. In the cosmological context, the flat-space factor of $E$ cancels against the would-be $1/E$ pole to give something finite as $E \to 0$.

[30]The argument is the same as for the spin-1 case: the right-hand side of the WT identity vanishes, so the correlation function must be constructible using the projectors from §3.3, but the projector annihilates any possibility.

For generic weights, the result cannot be written in terms of elementary functions, but scalar operators with special weights will lead to simple expressions. For example, starting with the three-point function of $\Delta = 3$ scalars (65), we obtain the correlation function of the stress tensor with two $\Delta = 2$ scalars

$$
\begin{aligned}
\langle T\varphi\varphi \rangle &= D_{12}D_{13}\langle\phi\phi\phi\rangle \\
&= \frac{9(k_1 - k_2 + k_3)^2}{2k_1 K^2}(\vec{k}_1 \cdot \vec{z}_1)^2 + \frac{36(k_1 + k_3)}{K^2}(\vec{k}_1 \cdot \vec{z}_1)(\vec{k}_2 \cdot \vec{z}_1) \\
&\quad + \frac{18(2k_1 + k_2 + k_3)}{K^2}(\vec{k}_2 \cdot \vec{z}_1)^2 \,.
\end{aligned}
\tag{94}
$$

This reproduces the result presented in [56], after reintroducing the longitudinal parts of the correlator there. In spinor helicity variables, the result takes the form

$$
\langle T^-\varphi\varphi \rangle = N_{T\varphi\varphi} \frac{(K + k_1)}{k_1^2 K^2}\langle 12\rangle^2 \langle 1\bar{2}\rangle^2 \,,
\tag{95}
$$

where we have introduced the amplitude $N_{T\varphi\varphi}$. By applying the operator $\mathcal{W}_{23}^{++}$, we can also raise the weight of the scalars to obtain $\langle T\phi\phi \rangle$. This object is interesting because it is related in a simple way to the inflationary correlator $\langle\gamma\zeta\zeta\rangle$ (see Section 7).

The expression (95) is kinematically satisfactory, but we still have to verify that it satisfies the WT identity (92). This is most simply done in spinor helicity variables, where acting with the operator $\widetilde{K}$ leads to

$$
\sum_{a=1}^{3} \vec{b} \cdot \widetilde{K}_a \left( \frac{1}{k_1}\langle T_1^- \varphi_2\varphi_3 \rangle \right) = 96 N_{T\varphi\varphi} \frac{\vec{b} \cdot \vec{\xi}_1^-}{k_1^3}\left(\vec{k}_2 \cdot \vec{\xi}_1^- k_3 + \vec{k}_3 \cdot \vec{\xi}_1^- k_2 \right).
\tag{96}
$$

Using (54), we can also write the left-hand side as

$$
\sum_{a=1}^{3} \vec{b} \cdot \widetilde{K}_a \left( \frac{1}{k_1}\langle T_1^- \varphi_2\varphi_3 \rangle \right) = -\frac{12\,\vec{b} \cdot \vec{\xi}_1^-}{k_1^3}\left(\kappa_2 \vec{k}_2 \cdot \vec{\xi}_1^- k_3 + \kappa_3 \vec{k}_3 \cdot \vec{\xi}_1^- k_2 \right),
\tag{97}
$$

where we have used the WT identity (92) and substituted $\langle\varphi\varphi\rangle = k$. We see that the results (96) and (97) are only consistent if

$$
\kappa_2 = \kappa_3 \equiv \kappa \,, \qquad N_{T\varphi\varphi} = -\frac{1}{8}\kappa \,.
\tag{98}
$$

We see that the WT identity forces the couplings of the scalars to the stress tensor to be the same, and fixes the normalization of the three-point function $\langle T\varphi\varphi\rangle$ in terms of this coupling. Of course, both of these features are expected from the bulk perspective. The three-point correlator arises from a bulk action of the form

$$
S = \int \mathrm{d}^4x \sqrt{-g}\left(-\frac{1}{2}g^{\mu\nu}\partial_\mu\varphi\partial_\nu\varphi - H^2\varphi^2\right),
\tag{99}
$$

which makes it clear that the equality of the couplings $\kappa_2$ and $\kappa_3$ is a manifestation of the equivalence principle, and that the relation between the normalizations follows from diffeomorphism invariance, which fixes the $\gamma\varphi^2$ coupling relative to the $\varphi^2$ terms.[31]

---

[31]For a single bulk scalar, the equality of $\kappa_2$ and $\kappa_3$ follows from symmetry under exchange of these operators, but even if we allow some nontrivial flavor structure these coupling constants have to be the same. This latter statement is the essential output of the equivalence principle in this context.

### 4.3.2 $\langle TTO \rangle$

Next, we consider the case with two spin-2 currents. This is again quite similar to the spin-1 analysis, so we will not dwell on the details. The right-hand side of the WT identity (92) vanishes, so we can construct this correlation function using the projection operator (39). We obtain a kinematically satisfactory correlator by applying $H_{12}$ twice to (66) and then using the projector:

$$\langle TTO_\Delta \rangle = \mathsf{P}_1^{(2)} \mathsf{P}_2^{(2)} H_{12}^2 \langle \varphi \varphi O_\Delta \rangle . \tag{100}$$

For generic weight $\Delta$, this is a complicated expression, but it once again simplifies dramatically when the scalar has $\Delta = 2$. Even then the full answer in terms of auxiliary vectors is long and not particularly illuminating. However, the expression becomes much simpler in spinor helicity variables:

$$\langle T^- T^- \varphi \rangle = \frac{k_1 k_2}{K^4} \langle 12 \rangle^4 , \tag{101}$$

$$\langle T^- T^+ \varphi \rangle = 0 , \tag{102}$$

which agrees with results in the literature [98, 102]. This correlator is identically conserved, indicating that it arises from a non-minimal bulk curvature coupling, such as $\varphi W_{\mu\nu\rho\sigma}^2$, where $W_{\mu\nu\rho\sigma}$ is the Weyl tensor.

 We can also consider the case where the third operator has spin. For simplicity we restrict to even parity correlators and assume that there is a single conserved spin-2 current. If the spin of the third operator is odd, then there are no conformally invariant correlators [60]. However, if it has even spin, there are possible structures. If the third operator has spin-2, then there is a unique possible correlator. For spin $\geq 4$, there are two possible conformally invariant structures [60,87].[32] In both cases, the correlators are identically conserved, and therefore can be built by acting with the projector (39) on a correlator with the correct quantum numbers.

### 4.3.3 $\langle TTT \rangle$

As a last example, we consider the correlation function of three stress tensors. In this case, the Ward–Takahashi identity can be written as [42, 56, 80, 89][33]

$$
\begin{aligned}
\xi_1^i k_1^j \langle T_{\vec{k}_1}^{ij} T_{\vec{k}_2} T_{\vec{k}_3} \rangle = & -(\vec{\xi}_1 \cdot \vec{k}_2) \langle T_{\vec{k}_2+\vec{k}_1} T_{\vec{k}_3} \rangle + 2(\vec{\xi}_1 \cdot \vec{\xi}_2) k_2^i \xi_2^j \langle T_{\vec{k}_2+\vec{k}_1}^{ij} T_{\vec{k}_3} \rangle \\
& -(\vec{\xi}_1 \cdot \vec{k}_3) \langle T_{\vec{k}_2} T_{\vec{k}_3+\vec{k}_1} \rangle + 2(\vec{\xi}_1 \cdot \vec{\xi}_3) k_3^i \xi_3^j \langle T_{\vec{k}_2} T_{\vec{k}_3+\vec{k}_1}^{ij} \rangle \\
& +(\vec{k}_1 \cdot \vec{\xi}_2) \xi_1^i \xi_2^j \langle T_{\vec{k}_2+\vec{k}_1}^{ij} T_{\vec{k}_3} \rangle + (\vec{\xi}_1 \cdot \vec{\xi}_2) k_1^i \xi_2^j \langle T_{\vec{k}_2+\vec{k}_1}^{ij} T_{\vec{k}_3} \rangle \\
& +(\vec{k}_1 \cdot \vec{\xi}_3) \xi_1^i \xi_3^j \langle T_{\vec{k}_2} T_{\vec{k}_1+\vec{k}_3}^{ij} \rangle + (\vec{\xi}_1 \cdot \vec{\xi}_3) k_1^i \xi_3^j \langle T_{\vec{k}_2} T_{\vec{k}_1+\vec{k}_3}^{ij} \rangle .
\end{aligned} \tag{103}
$$

Using the explicit expression for the stress tensor two-point function, $\langle T_{\vec{k}} T_{-\vec{k}} \rangle = 2k^3 (\vec{\xi}_1 \cdot \vec{\xi}_2)^2$, this identity simplifies dramatically:

$$\xi_1^i k_1^j \langle T_{\vec{k}_1}^{ij} T_{\vec{k}_2} T_{\vec{k}_3} \rangle = 2(\vec{\xi}_2 \cdot \vec{\xi}_3) \left[ (\vec{\xi}_1 \cdot \vec{k}_2)(\vec{\xi}_2 \cdot \vec{\xi}_3) + \text{cyclic perms.} \right] \left( k_2^3 - k_3^3 \right). \tag{104}$$

---

 [32]This counting assumes unbroken parity. If parity is broken, then a third structure is possible for even spin, $\langle T^+ T^+ O^{(\ell)} \rangle \neq \langle T^- T^- O^{(\ell)} \rangle$. Moreover, in the case of broken parity, there is also a structure $\langle T^- T^+ O^{(\ell)} \rangle \neq 0$ for operators with odd spin $\ell \geq 5$. We thank Sasha Zhiboedov for a discussion on this.

 [33]There is an ambiguity in this identity, corresponding to the freedom to perform bulk field redefinitions. In principle, we are allowed to add an arbitrary multiple of the last two lines in (103). See Appendix B for details. This shifts the coefficient of the contact term (108). Our choice of WT identity fixes the stress tensor three-point function to be the one that arises from a bulk computation if the graviton fluctuation is written as $h_{ij} = (e^\gamma)_{ij}$ [42].

Intriguingly, the tensor structure that appears in the brackets is precisely the Yang–Mills three-point amplitude. Given this identity, our goal is the same as before: find a solution and then characterize the most general identically conserved correlation function.

As in the spin-1 case, several different weight-shifting paths are required to construct a solution to the WT identity. Using the three-point function of $\Delta = 3$ scalars as a seed, we can act with the following combinations of weight-shifting operators to generate correlation functions of $\Delta = 3$ spin-2 operators:

$$\langle TTT \rangle_A = \mathcal{S}_{31}^{++} \mathcal{S}_{23}^{++} \mathcal{S}_{12}^{++} \langle \phi \phi \phi \rangle, \tag{105}$$

$$\langle TTT \rangle_B = \left( \mathcal{S}_{23}^{++} \right)^2 \mathcal{W}_{23}^{++} D_{13} D_{12} \langle \phi \phi \phi \rangle + \text{perms.}, \tag{106}$$

$$\langle TTT \rangle_C = D_{13} D_{12} \left( \mathcal{S}_{23}^{++} \right)^2 \mathcal{W}_{23}^{++} \langle \phi \phi \phi \rangle + \text{perms.}, \tag{107}$$

where "perms." in the last two paths indicates that we should sum over symmetric permutations to symmetrize the correlator in the three operators. Note that there is also a contact term,

$$\langle TTT \rangle_{\text{loc}} = (k_1^3 + k_2^3 + k_3^3)(\vec{\xi}_1 \cdot \vec{\xi}_2)(\vec{\xi}_2 \cdot \vec{\xi}_3)(\vec{\xi}_3 \cdot \vec{\xi}_1), \tag{108}$$

that satisfies the conformal Ward identities. Although we are free to add an arbitrary amount of this term to the correlation function, the WT identity fixes the contribution from this piece.

The requirement that $\langle TTT \rangle$ satisfies the WT identity (103) fixes the precise linear combination of weight-shifting paths, leading to the result

$$\langle TTT \rangle_E = -\frac{1}{81} \langle TTT \rangle_A + \frac{7}{1458} \langle TTT \rangle_B + \frac{2}{3645} \langle TTT \rangle_C + \frac{8}{15} \langle TTT \rangle_{\text{loc}}. \tag{109}$$

The somewhat peculiar-looking coefficients appearing in this expression are a consequence of the (arbitrary) way that we have normalized the various weight-shifting operators. Writing the explicit correlation function in terms of spinor helicity variables, we obtain

$$\langle T^- T^- T^- \rangle_E = \langle 12 \rangle^2 \langle 23 \rangle^2 \langle 31 \rangle^2 \frac{\left( K^3 - K(k_1 k_2 + k_2 k_3 + k_3 k_1) - k_1 k_2 k_3 \right)}{256(k_1 k_2 k_3)^2}, \tag{110}$$

$$\langle T^- T^- T^+ \rangle_E = \langle 12 \rangle^2 \langle 2\bar{3} \rangle^2 \langle \bar{3}1 \rangle^2 \frac{(k_1 + k_2 - k_3)^2 \left( K^3 - K(k_1 k_2 + k_2 k_3 + k_3 k_1) - k_1 k_2 k_3 \right)}{256(k_1 k_2 k_3)^2 K^2}, \tag{111}$$

where the other helicity configurations can be obtained from these by swapping barred and un-barred spinors. This expression agrees with [42, 98], and from the bulk perspective arises from the cubic interaction in the Einstein–Hilbert term $\sqrt{-g}R$.[34]

Just like in the Yang–Mills case, in order to simplify later calculations, it is useful to consider a more complicated path to the same result. In this route, we take as a starting point the correlator of a spin-2 current with two $\Delta = 5$ scalar operators. It is then possible to write the Einstein–Hilbert three-point function by acting with a specific differential operator:

$$\langle TTT \rangle_E = \Big[ (126 H_{12} + 6 D_{22} D_{11} + 16 \mathcal{S}_{12}^{++} \mathcal{W}_{12}^{--}) D_{22} D_{11}$$
$$+ (15 D_{12} D_{21} + 5 \mathcal{S}_{12}^{++} \mathcal{W}_{12}^{--}) D_{12} D_{21} + 2 (\mathcal{S}_{12}^{++})^2 (\mathcal{W}_{12}^{--})^2 \Big] \langle O_5 O_5 T \rangle \tag{112}$$

$$\equiv \mathcal{D}_{12}^{(2)} \langle O_5 O_5 T \rangle, \tag{113}$$

---

[34]The −−+ correlator has an $E^{-2}$ pole, with residue the flat-space Einstein–Hilbert three-point scattering amplitude, as can be verified by using spinor identities to remove the barred spinors. The order of the pole is consistent with the bulk interaction being a two-derivative vertex. Note also that the −−− correlator has no such pole, which is consistent with the Einstein–Hilbert term not generating a scattering amplitude with these helicities.

which gives the same result as (109). For future reference, we have defined the differential operator $\mathcal{D}^{(2)}_{12}$ that implements this procedure. The key benefit of the operator $\mathcal{D}^{(2)}_{12}$ is that it acts only on the first two operators appearing in the correlator. This will be useful when we construct the four-point function with two external stress tensors in §5.3.2.

Having found a solution to the WT identity (103), we can add to it any identically conserved correlator. As before, it is relatively straightforward to obtain identically conserved shapes using the projectors. We can weight shift the $\Delta = 3$ scalar three-point function to a correlator of the form $\langle Thh \rangle$, which involves a $\Delta = 3$ spin-2 conserved current $T$, and two $\Delta = 0$ spin-2 operators $h$. Applying the projector (37) then gives

$$\langle TTT \rangle_{W^3} = \mathsf{P}^{(2)}_2 \mathsf{P}^{(2)}_3 \left( \mathcal{W}^{--}_{23} \right)^2 \left( \mathcal{S}^{++}_{23} \right)^2 D_{13} D_{12} \langle \phi \phi \phi \rangle \,. \tag{114}$$

In spinor helicity variables, this takes the form

$$\langle T^- T^- T^- \rangle_{W^3} = \langle 12 \rangle^2 \langle 23 \rangle^2 \langle 31 \rangle^2 \frac{k_1 k_2 k_3}{K^6} \,, \tag{115}$$

$$\langle T^- T^- T^+ \rangle_{W^3} = 0 \,, \tag{116}$$

which agrees with the result of [42]. From the bulk perspective, this correlation function arises from a higher-derivative cubic coupling, such as the Weyl tensor cubed, $\sqrt{-g}\, W^3_{\mu\nu\rho\sigma}.$[35]

# 5 Four-Point Functions from Weight-Shifting

We now turn to the four-point functions of spinning operators. This time we are not guaranteed to find solutions to both the conformal Ward identities and the Ward–Takahashi identities for conserved currents. This is both a complication and an opportunity: in flat space, locality and unitarity impose strong constraints on four-particle scattering amplitudes, which has helped to carve out the space of consistent theories [2, 4]. Our hope is to learn similar lessons in the cosmological setting.[36] In this paper, we will focus mainly on cosmological correlators for which a consistent four-particle amplitude is known to exist. The case of no-go results and exotic representations, unique to de Sitter space, will be considered elsewhere. We remind the reader that longitudinal terms do not contribute to the observables so here we present exclusively the transverse part of four-point functions.

## 5.1 Scalar Seed Correlators

Our strategy is essentially the same as it was at three points. We first utilize weight-shifting operators to generate candidate structures with the correct kinematics from known seed solutions, and then impose the Ward–Takahashi identities to further constrain the possibilities. We begin with a brief review of the required seed functions [47].

**Conformally coupled scalars**

An important seed is the four-point function of conformally coupled scalars arising from the exchange of a massive scalar, which we denote by $F \equiv \langle \varphi \varphi \varphi \varphi \rangle$. Invariance under rotations and translations implies that this correlation function a priori depends on six independent kinematic variables. However, since the correlator $F$ arises from the exchange of a scalar,

---

[35]As for the Einstein–Hilbert correlator, this correlation function has a total energy pole (in this case $E^{-6}$), with a residue that is the flat-space scattering amplitude. The order of the pole is consistent with the correlator coming from a six-derivative bulk vertex.

[36]There is a growing literature on higher-point correlation functions of spinning fields in curved space [103–115].

it is simpler and depends only on three variables. Using dilatation symmetry, one of these variables can be reduced to an overall factor. In the case of $s$-channel exchange, it makes sense to re-scale the correlator by the magnitude of the internal momentum, $s = |\vec{k}_1 + \vec{k}_2|$, so that the kinematically nontrivial information is given by a function of just two dimensionless variables:[37]

$$F = s^{-1}\hat{F}(w, v), \qquad \begin{aligned} w &= \frac{s}{k_1 + k_2}, \\ v &= \frac{s}{k_3 + k_4}, \end{aligned} \tag{117}$$

where the function $\hat{F}(w, v)$ is constrained by special conformal symmetry. Explicitly, the Ward identities in (23) reduce to [47]

$$(\Delta_w - \Delta_v)\hat{F} = 0, \tag{118}$$

where $\Delta_w \equiv w^2(1 - w^2)\partial_w^2 - 2w^3\partial_w$. The solutions to this equation can be classified by their singularity structures (see §2.3).

**Contact solutions**

The simplest solutions to (118) only have singularities at vanishing total energy and correspond to contact interactions in the bulk,

$$\hat{C}_0 = \frac{wv}{w + v} = \frac{s}{E} \quad \text{and} \quad \hat{C}_n = \Delta_w^n \hat{C}_0, \tag{119}$$

where $s^{-1}\hat{C}_0$ arises from a $\varphi^4$ interaction and the solutions $\hat{C}_n$ correspond to higher-derivative interactions. The most general contact solution is a linear combination of the solutions in (119). These are all of the contact solutions that arise from integrating out a scalar particle at tree level, reproducing the effective field theory expansion of the bulk theory.

**Exchange solutions**

Solutions corresponding to the tree-level exchange of a massive scalar are obtained by splitting the partial differential equation (118) into a pair of ordinary differential equations

$$\begin{aligned} \left[\Delta_w - (\Delta_\sigma - 1)(\Delta_\sigma - 2)\right]\hat{F} &= \hat{C}, \\ \left[\Delta_v - (\Delta_\sigma - 1)(\Delta_\sigma - 2)\right]\hat{F} &= \hat{C}, \end{aligned} \tag{120}$$

where $\hat{C}$ is one of the contact solutions in (119) and $\Delta_\sigma$ is the weight of the exchanged operator.[38] For generic weights, the solutions can be written in terms of generalized two-variable hypergeometric functions [47]. However, in the following, we will only need the solutions for the specific weights $\Delta_\sigma = 2$ and $3$, corresponding to the exchange of conformally coupled and massless scalars in the bulk, respectively. For $\hat{C} = \hat{C}_0$, the relevant solutions of (120) are [22, 47][39]

$$\hat{F}_{\Delta_\sigma = 2} = \frac{1}{2}\left[\text{Li}_2\left(\frac{k_{12} - s}{E}\right) + \text{Li}_2\left(\frac{k_{34} - s}{E}\right) + \log\left(\frac{k_{12} + s}{E}\right)\log\left(\frac{k_{34} + s}{E}\right)\right], \tag{121}$$

$$\hat{F}_{\Delta_\sigma = 3} = \left((1 - w^2)\partial_w - \frac{1}{w}\right)\left((1 - v^2)\partial_v - \frac{1}{v}\right)\hat{F}_{\Delta_\sigma = 2} + \frac{wv + 1}{w + v}, \tag{122}$$

---

[37]In [47], we used the dimensionless variables $u$ and $v$. In this paper, we instead use $w$ and $v$ to avoid confusion with the exchange momentum $u = |\vec{k}_1 + \vec{k}_3|$.

[38]The weight is related to the mass of the corresponding bulk field via $(\Delta_\sigma - 1)(\Delta_\sigma - 2) = 2 - m_\sigma^2/H^2$.

[39]Recall that, in this paper, $\hat{F}$ denotes a wavefunction coefficient, while, in [22, 47], we computed boundary correlators. As can be seen in (11), the two differ by a disconnected part, which is the homogeneous solution to the equations (120).

where $\mathrm{Li}_2(x)$ is the dilogarithm and $E = k_{12} + k_{34}$ is the total energy.

Given the scalar-exchange solutions generated in this fashion, we can get the corresponding spin-exchange solutions by acting with appropriate spin-raising operators [47,48]. These spin-exchange solutions, in general, depend on the full set of six kinematic variables at four points. It is convenient to parametrize the additional variables as

$$\alpha \equiv \frac{k_1 - k_2}{s}, \qquad \beta \equiv \frac{k_3 - k_4}{s}, \qquad \hat{\tau} \equiv \frac{\vec{\alpha} \cdot \vec{\beta}}{s^2}, \tag{123}$$

where $\vec{\alpha} \equiv \vec{k}_1 - \vec{k}_2$ and $\vec{\beta} \equiv \vec{k}_3 - \vec{k}_4$. Note that $\hat{\alpha} \neq \vec{\alpha}/|\vec{\alpha}|$. Written as a sum over helicity contributions, the solution for spin-$\ell$ exchange then takes the following form [47,48]

$$\hat{F}^{(\ell)} = \sum_{m=0}^{\ell} \Pi_{\ell,m}(\alpha, \beta, \hat{\tau}) \, \mathcal{D}_{wv}^{(\ell,m)} \hat{F}^{(0)}, \tag{124}$$

where we have defined the scalar-exchange solution as $\hat{F}^{(0)} \equiv \hat{F}(w, v)$. Explicit expressions for the differential operators $\mathcal{D}_{wv}^{(\ell,m)}$ and the polarization sums $\Pi_{\ell,m}$ can be found in [48]. In this paper, we will only need the following operators

$$\text{spin 1:} \quad \mathcal{D}_{wv}^{(1,1)} = (wv)^2 \partial_w \partial_v \equiv D_{wv}, \quad \mathcal{D}_{wv}^{(1,0)} = \Delta_w, \tag{125}$$

$$\text{spin 2:} \quad \mathcal{D}_{wv}^{(2,2)} = D_{wv}^2, \qquad \mathcal{D}_{wv}^{(2,1)} = D_{wv}(\Delta_w - 2), \quad \mathcal{D}_{wv}^{(2,0)} = \Delta_w(\Delta_w - 2). \tag{126}$$

The polarization sums for spins 1 and 2 are given in Appendix E.

We will also need spin-exchange solutions in the $t$- and $u$-channel. These solutions will have the same structure as the solutions in (124), with suitable generalizations of the polarization sums $\Pi_{\ell,m}^{(t)}$ and $\Pi_{\ell,m}^{(u)}$. These polarization sums are functions of the following variables

$$t\text{-channel:} \qquad \alpha_t \equiv \frac{k_1 - k_4}{t}, \qquad \beta_t \equiv \frac{k_2 - k_3}{t}, \qquad \hat{\tau}_t \equiv \frac{(\vec{k}_1 - \vec{k}_4) \cdot (\vec{k}_2 - \vec{k}_3)}{t^2}, \tag{127}$$

$$u\text{-channel:} \qquad \alpha_u \equiv \frac{k_1 - k_3}{u}, \qquad \beta_u \equiv \frac{k_2 - k_4}{u}, \qquad \hat{\tau}_u \equiv \frac{(\vec{k}_1 - \vec{k}_3) \cdot (\vec{k}_2 - \vec{k}_4)}{u^2}, \tag{128}$$

which are the natural permutations of the variables defined in (123).

## 5.2 Correlators with Spin-1 Currents

Many of the interesting features of spinning correlation functions are already present for correlators involving conserved spin-1 currents. Although this case is phenomenologically less relevant than the case of external spin-2 currents (which are dual to graviton fluctuations) it is computationally simpler, and therefore provides a useful setting in which the structure of spinning operators can be explored. As in Section 4, our goal is not to be completely comprehensive, but rather to present a set of examples which illustrate the most interesting physical phenomena.

### 5.2.1 ⟨*JOOO*⟩

We begin with the correlator of one conserved spin-1 current and three $\Delta = 2$ scalars (see Fig. 5).[40] The relevant Ward–Takahashi identity is

$$\vec{k}_1 \cdot \langle \vec{J}_{\vec{k}_1} \varphi_{\vec{k}_2} \varphi_{\vec{k}_3} \varphi_{\vec{k}_4} \rangle = -e_2 \langle \varphi_{\vec{k}_2 + \vec{k}_1} \varphi_{\vec{k}_3} \varphi_{\vec{k}_4} \rangle - e_3 \langle \varphi_{\vec{k}_2} \varphi_{\vec{k}_3 + \vec{k}_1} \varphi_{\vec{k}_4} \rangle - e_4 \langle \varphi_{\vec{k}_2} \varphi_{\vec{k}_3} \varphi_{\vec{k}_4 + \vec{k}_1} \rangle$$

$$= -N_{\varphi^3} \left[ e_2 \log\left( \frac{k_{34} + s}{\mu} \right) + e_3 \log\left( \frac{k_{23} + u}{\mu} \right) + e_4 \log\left( \frac{k_{24} + t}{\mu} \right) \right], \tag{129}$$

---

[40]We could also consider the analogous correlation function with scalar operators of general weights, but the corresponding expressions are substantially more unwieldy and do not add any additional insights.

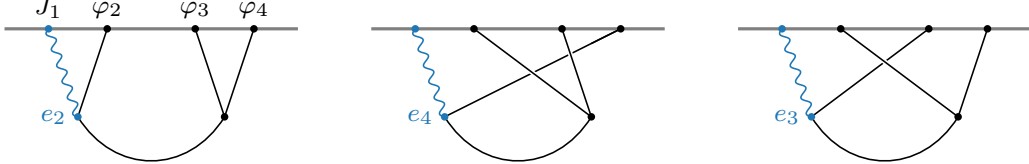

Figure 5: Diagrammatic representation of the $s$, $t$, and $u$-channel contributions to $\langle J\varphi\varphi\varphi\rangle$.

where $e_a$ are the charges of the operators $\varphi_a$ and we substituted $\langle\varphi\varphi\varphi\rangle = N_{\varphi^3}\log(K/\mu)$ in the second equality. Our strategy for deriving the correlator is to first determine $\langle J\varphi\varphi\varphi\rangle_s$ in the $s$-channel and then to demonstrate that multiple channels are needed to satisfy the WT identity (129).

It is straightforward to increase the spin of one of the operators in $\langle\varphi\varphi\varphi\varphi\rangle_s$ by acting with the operator $D_{12}$ defined in (28). Besides raising the spin of particle 1, this operator lowers the weight of particle 2. To undo the weight-lowering, and get back to a $\Delta = 2$ scalar, we multiply by $k_2$ (this corresponds to shadow transforming particle 2):

$$\langle J\varphi\varphi\varphi\rangle_s = k_2 D_{12}\big(F_{\Delta_\sigma=2}\big), \tag{130}$$

where the seed function $F_{\Delta_\sigma=2} = s^{-1}\hat{F}_{\Delta_\sigma=2}$ was defined in (121). The explicit answer for this correlator then is

$$\langle J\varphi\varphi\varphi\rangle_s = \frac{\vec{\xi}_1\cdot\vec{k}_2}{(k_{12}+s)(k_{12}-s)}\log\left(\frac{k_{34}+s}{E}\right), \tag{131}$$

where $J \equiv \vec{\xi}\cdot\vec{J}$. We have re-scaled the correlator to have a convenient overall normalization. We will later allow the normalization to be arbitrary and see that it is fixed by the WT identity, so there is no loss of generality. We have also kept only the transverse part of this correlation function by contracting it with a polarization vector. Here, and throughout this section, we are suppressing flavor indices on the operators $\varphi$. Since the kinematic structures in (131) are not symmetric under permutations of the external momenta, we require nontrivial flavor factors to restore Bose symmetry for the full correlator. To avoid notational clutter, we do not write these explicitly.

*One channel is not enough.*—Next, we will show that the $s$-channel contribution (131) on its own is not consistent with the WT identity. As we described in §3.4, it is easiest to check the WT identity by first writing the result in spinor helicity variables and then acting on it with the operator $\widetilde{K}$ defined in (49). Before doing this, we multiply the result (131) by a normalization factor $e_2 N_{J\varphi^3}$, where $N_{J\varphi^3}$ captures the overall normalization of the various channels, and the charge $e_2$ is the relative normalization of the $s$-channel. Acting with $\widetilde{K}$, we then get

$$\sum_{a=1}^{4}\vec{b}\cdot\widetilde{K}_a\big(e_2 N_{J\varphi^3}\langle J^-\varphi\varphi\varphi\rangle_s\big) = -e_2 N_{J\varphi^3}\frac{\vec{b}\cdot\vec{\xi}_1^-}{k_1^2}\left[\log\left(\frac{k_{34}+s}{E}\right)+\frac{k_1}{E}\right]. \tag{132}$$

Alternatively, we can use (53) to relate the action of $\widetilde{K}$ on the full correlator to the WT identity. This gives

$$\sum_{a=1}^{4}\vec{b}\cdot\widetilde{K}_a\langle J^-\varphi\varphi\varphi\rangle_s = \frac{2\vec{b}\cdot\vec{\xi}_1^-}{k_1^2}\vec{k}_1\cdot\langle J^-_{\vec{k}_1}\varphi_{\vec{k}_2}\varphi_{\vec{k}3}\varphi_{\vec{k}_4}\rangle$$

$$= -2N_{\varphi^3}\frac{\vec{b}\cdot\vec{\xi}_1^-}{k_1^2}\left[e_2\log\left(\frac{k_{34}+s}{\mu}\right)+e_3\log\left(\frac{k_{23}+u}{\mu}\right)+ \tag{133}\right.$$

$$e_4 \log\left(\frac{k_{24} + t}{\mu}\right)\Bigg],$$

where we have substituted (129) in the second equality. Consistency requires (132) and (133) to agree with each other. However, we see that the $1/E$ and $\log E$ terms in (132) cannot be matched with the right-hand side of (133), showing that the $s$-channel answer alone is not sufficient.

To achieve consistency with the WT identity, we must add the $t$- and/or $u$-channel contributions to the correlator. These contributions are simply permutations[41] of the $s$-channel answer in (131), which we combine as

$$\langle J^-\varphi\varphi\varphi\rangle_{s+t+u} \equiv N_{J\varphi^3}\Big[e_2 \langle J^-\varphi\varphi\varphi\rangle_s + e_4 \langle J^-\varphi\varphi\varphi\rangle_t + e_3 \langle J^-\varphi\varphi\varphi\rangle_u\Big]. \tag{134}$$

Acting on this sum of the channels with the operator $\widetilde{K}$, we get

$$\sum_{a=1}^{4} \vec{b}\cdot\widetilde{K}_a \langle J^-\varphi\varphi\varphi\rangle_{s+t+u} = -N_{J\varphi^3}\frac{\vec{b}\cdot\vec{\xi}_1^-}{k_1^2}\Bigg\{e_2 \log\left(\frac{k_{34}+s}{\mu}\right) + e_3 \log\left(\frac{k_{23}+u}{\mu}\right) \tag{135}$$
$$+ e_4 \log\left(\frac{k_{24}+t}{\mu}\right) + (e_2+e_3+e_4)\Big[\log\left(\frac{\mu}{E}\right) + \frac{k_1}{E}\Big]\Bigg\}.$$

This expression agrees with (133)—and hence is consistent with the WT identity—if $N_{J\varphi^3} = 2N_{\varphi^3}$ and all the charges add up to zero,

$$e_2 + e_3 + e_4 = 0. \tag{136}$$

At a minimum, this requires including two channels with opposite charges. Hence, we see that the WT identity requires the presence of multiple channels and that the couplings satisfy charge conservation.

*Identically conserved correlators.*—Having found a solution to the WT identity, we are still free to add to it any correlator with the correct quantum numbers as long as it is identically conserved. As explained in §3.3, such correlators can be constructed using projection operators. In particular, acting with the spin-raising operator $D_{11}$ on one of the solutions $F^{(\ell)}$ in (124) and then projecting it onto the identically conserved form, we obtain

$$\langle J\varphi\varphi\varphi\rangle_s = \mathsf{P}_1^{(1)} D_{11}\big(F^{(\ell)}\big), \tag{137}$$

where $\mathsf{P}_1^{(1)}$ was defined in (35). It is easy to check that the scalar-exchange solution $F^{(0)}$ is annihilated by the projection operator. This is the CFT version of the flat-space statement that the on-shell three-particle amplitude between a photon and two massive scalars vanishes unless the scalars have equal mass, in which case there is a unique amplitude. Correspondingly, for external $\Delta = 2$ scalars, the only scalar operator that can be exchanged must have $\Delta = 2$, giving the solution (131). Taking any of the spin-exchange solutions $F^{(\ell\neq 0)}$ as a seed, the more complicated kinematics allow for other nonzero possibilities. Finally, we can also let the seed in (137) be a contact solution to generate contact interactions between the spinning field and scalar operators. We have not attempted to systematically classify all possible contact solutions of this form, but it would be interesting to do so.

### 5.2.2 $\langle JOJO \rangle$

The next-simplest case involves two spin-1 currents (see Fig. 6). This is the analog of *Compton scattering*. For simplicity, we will restrict to the case where the two additional operators are $\Delta = 2$ scalars. There are two conceptually distinct situations, depending on whether we have only one kind of conserved current, or a multiplet of currents.

---

[41]In detail, to go from the $s$-channel to the $t$-channel, we permute the legs 2 and 4, which sends $s \mapsto t$. Similarly, permuting 2 and 3 gives the $u$-channel answer, with $s \mapsto u$.

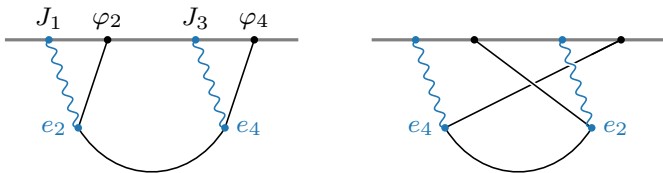

Figure 6: Illustration of the $s$- and $t$-channel contributions to the Abelian Compton correlator.

**Abelian Compton scattering**

We first consider the example of one type of conserved current, corresponding to Abelian Compton scattering. In this case, the WT identity reads

$$\vec{k}_1 \cdot \langle \vec{J}_{\vec{k}_1} \varphi_{\vec{k}_2} J_{\vec{k}_3} \varphi_{\vec{k}_4} \rangle = -e_2 \langle \varphi_{\vec{k}_2+\vec{k}_1} J_{\vec{k}_3} \varphi_{\vec{k}_4} \rangle - e_4 \langle \varphi_{\vec{k}_2} J_{\vec{k}_3} \varphi_{\vec{k}_4+\vec{k}_1} \rangle$$
$$= 2e^2 \left[ \frac{\vec{\xi}_3 \cdot \vec{k}_4}{(k_{34}+s)} + \frac{\vec{\xi}_3 \cdot \vec{k}_2}{(k_{23}+t)} \right], \tag{138}$$

where in the second line we have used (70) for the three-point function $\langle J\varphi\varphi \rangle$. We also used the constraints (72) derived at three points to set $e_2 = -e_4 = e$. Acting twice with the weight-shifting operators in (130) generates a solution with the right kinematic properties:[42]

$$\langle J\varphi J\varphi \rangle_s = k_4 D_{34} \, k_2 D_{12} \big( F_{\Delta_\sigma=2} \big). \tag{139}$$

The result of this weight-shifting procedure is

$$\langle J\varphi J\varphi \rangle_s = \frac{(\vec{\xi}_1 \cdot \vec{k}_2)(\vec{\xi}_3 \cdot \vec{k}_4)}{(k_{12}+s)(k_{34}+s)E}, \tag{140}$$

where we have re-scaled the correlator. The result in the $t$-channel is obtained straightforwardly from (140) through the permutation $2 \leftrightarrow 4$.

*One channel is not enough.*—As before, the $s$-channel correlator alone is not consistent with the WT identity. This time, however, adding just the $t$-channel is not sufficient. We add together the $s$- and $t$-channels as

$$\langle J\varphi J\varphi \rangle_{s+t} \equiv N_{J\varphi J\varphi} \big[ \langle J\varphi J\varphi \rangle_s + \langle J\varphi J\varphi \rangle_t \big], \tag{141}$$

where $N_{J\varphi J\varphi}$ is the overall normalization of the correlator. Acting with the operator $\widetilde{K}$ on this sum of channels, we get

$$\sum_{a=1}^4 \vec{b} \cdot \widetilde{K}_a \langle J^-\varphi J^+\varphi \rangle_{s+t} = -N_{J\varphi J\varphi} \frac{\vec{b} \cdot \vec{\xi}_1^-}{k_1^2} \left( \frac{\vec{\xi}_3^+ \cdot \vec{k}_4}{(k_{34}+s)} + \frac{\vec{\xi}_3^+ \cdot \vec{k}_2}{(k_{23}+t)} - \left( \frac{k_1}{E^2} + \frac{1}{E} \right) \vec{\xi}_3^+ \cdot \vec{k}_1 \right)$$
$$- \big( 3 \leftrightarrow 1, 2 \leftrightarrow 4 \big). \tag{142}$$

Alternatively, the action of $\widetilde{K}$ on the full correlator can be computed by using the WT identity (138):

$$\sum_{a=1}^4 \vec{b} \cdot \widetilde{K}_a \langle J^-\varphi J^+\varphi \rangle = 2e^2 \frac{\vec{b} \cdot \vec{\xi}_1^-}{k_1^2} \left( \frac{\vec{\xi}_3^+ \cdot \vec{k}_4}{(k_{34}+s)} + \frac{\vec{\xi}_3^+ \cdot \vec{k}_2}{(k_{23}+t)} \right) + \big( 3 \leftrightarrow 1, 2 \leftrightarrow 4 \big). \tag{143}$$

---

[42]A seemingly reasonable alternative weight-shifting procedure is to apply $\mathcal{S}_{13}^{++}$ directly to (121). Although this generates a correlation function with the correct quantum numbers, it turns out that this correlator does not have the right pole structure to satisfy the WT identity and hence does not correspond to any physical bulk process.

Notice that there are additional poles at $E = 0$ in (142) that aren't matched by (143). To cancel off these total energy poles, we must add the following contact solution

$$\langle J\varphi J\varphi\rangle_c = \mathcal{S}_{13}^{++} C_0 = e_c \frac{\vec{\xi}_1 \cdot \vec{\xi}_3}{E}. \tag{144}$$

Indeed, acting with $\widetilde{K}$ on (144), we get

$$\sum_{a=1}^{4} \vec{b} \cdot \widetilde{K}_a \langle J^- \varphi J^+ \varphi \rangle_c = 2e_c \frac{\vec{b} \cdot \vec{\xi}_1^-}{k_1^2}\left( \frac{k_1}{E^2} + \frac{1}{E} \right) \vec{\xi}_3^+ \cdot \vec{k}_1 + (1 \leftrightarrow 3). \tag{145}$$

This cancels the unwanted terms in (142) if $e_c = \frac{1}{2} N_{J\varphi J\varphi}$. We then see that we can make (142) and (143) consistent by setting $N_{J\varphi J\varphi} = -2e^2$. The full solution is

$$\langle J\varphi J\varphi\rangle_{s+t+c} = -2e^2 \left( \frac{(\vec{\xi}_1 \cdot \vec{k}_2)(\vec{\xi}_3 \cdot \vec{k}_4)}{(k_{12}+s)(k_{34}+s)} + \frac{(\vec{\xi}_1 \cdot \vec{k}_4)(\vec{\xi}_3 \cdot \vec{k}_2)}{(k_{14}+t)(k_{23}+t)} + \frac{\vec{\xi}_1 \cdot \vec{\xi}_3}{2} \right) \frac{1}{E}. \tag{146}$$

It is easy to verify that the coefficient of the $E \to 0$ singularity is indeed the amplitude for Compton scattering; cf. Appendix F.

**Non-Abelian Compton scattering**

Next, we consider a multiplet of conserved currents. The WT identity now reads

$$\vec{k}_1 \cdot \langle J_{\vec{k}_1}^A \varphi_{\vec{k}_2}^a J_{\vec{k}_3}^B \varphi_{\vec{k}_4}^b \rangle = -i(T^A)^a{}_c \langle \varphi_{\vec{k}_2+\vec{k}_1}^c J_{\vec{k}_3}^B \varphi_{\vec{k}_4}^b \rangle - i(T^A)^b{}_c \langle \varphi_{\vec{k}_2}^a J_{\vec{k}_3}^B \varphi_{\vec{k}_4+\vec{k}_1}^c \rangle$$
$$+ f^{ABC} \langle \varphi_{\vec{k}_2}^a J_{\vec{k}_3+\vec{k}_1}^C \varphi_{\vec{k}_4}^b \rangle, \tag{147}$$

where we have displayed the flavor structure explicitly. The currents are charged under the symmetries that they generate, with structure constants $f^{ABC}$, and $(T^A)_{ab}$ are the couplings between the currents and a multiplet of charged scalar operators. Note that the last term in (147) requires special care, because it involves a polarization vector contracted with a current at a *shifted* momentum, $J_{\vec{k}_3+\vec{k}_1}^C \equiv \vec{\xi}_3 \cdot \vec{J}_{\vec{k}_3+\vec{k}_1}^C$. To evaluate this expression, we write $J_{\vec{q}=\vec{k}_3+\vec{k}_1}^C$ as

$$J_{\vec{q}}^C \equiv \vec{\xi}_3 \cdot \vec{J}_{\vec{q}}^C = \xi_3^i \Big[ (\delta_{ij} - \hat{q}^i \hat{q}^j) + \hat{q}^i \hat{q}^j \Big] J_{\vec{q}}^j = \sum_{\pm} \vec{\xi}_3 \cdot \vec{\xi}_{\vec{q}}^{\pm} \vec{\xi}_{\vec{q}}^{\mp} \cdot \vec{J}_{\vec{q}}^C + \frac{\vec{\xi}_3 \cdot \vec{q}}{q^2} \vec{q} \cdot \vec{J}_{\vec{q}}. \tag{148}$$

The first term leads to the transverse part of $\langle \varphi J \varphi \rangle$, while the second term is constrained by the WT identity and is proportional to $\langle \varphi\varphi \rangle$.

From the bulk perspective, the correlators in the Abelian Compton example arise from the exchange of a scalar operator. In the non-Abelian case, it is also possible for a conserved current to be exchanged in the $u$-channel (see Fig. 7), and this contribution is crucial to satisfy the WT identity. One of the vertices involved in the correlator $\langle J\varphi J\varphi \rangle_u$ is the Yang–Mills three-point vertex, whose corresponding correlator was found in §4.2.3. In (86), this correlator was written in terms of a differential operator $\mathcal{D}_{12}^{(1)}$ acting on $\langle \phi\phi J \rangle$. This suggests that we can write $\langle J\varphi J\varphi \rangle_u$ as

$$\langle J\varphi J\varphi \rangle_u \overset{?}{=} \mathcal{D}_{13}^{(1)} \langle \phi\varphi\phi\varphi \rangle_{u,J}, \tag{149}$$

where the seed function $\langle \phi\varphi\phi\varphi \rangle_{u,J}$ involves the exchange of a conserved spin-1 current in the $u$-channel. This turns out not to be quite right, but the true answer is very closely related. As we will describe, the above weight-shifting procedure only generates the correct $u$-channel exchange solution after a suitable manipulation of the seed.

The required scalar seed correlator is obtained by first raising the spin of the exchanged field in (121) and then raising the weight of two of the external legs:

$$
\begin{aligned}
\langle \phi \varphi \phi \varphi \rangle_{u,J} &= \mathcal{W}_{13}^{++} \langle \varphi \varphi \varphi \varphi \rangle_{u,J} \\
&= \mathcal{W}_{13}^{++} \left( \Pi_{1,1}^{(u)} D_{wv} + \Pi_{1,0}^{(u)} \Delta_w \right) F_{\Delta_\sigma = 2} \,,
\end{aligned}
\tag{150}
$$

where $F_{\Delta_\sigma=2} = u^{-1} \hat{F}_{\Delta_\sigma=2}(w,v)$, with $w \equiv u/k_{13}$ and $v \equiv u/k_{24}$.[43] The polarization sums $\Pi_{1,1}^{(u)}$ and $\Pi_{1,0}^{(u)}$ are the obvious generalizations of the polarization sums in (124) to the $u$-channel (see also Appendix E). Evaluating (150) using (25) leads to the following expression

$$
\langle J \varphi J \varphi \rangle_u \stackrel{?}{=} \left[ (\vec{\xi}_1 \cdot \vec{\xi}_3) \left( \Pi_{1,1}^{(u)} D_{wv} + \Pi_{1,0}^{(u)} \Delta_w \right) + 2(\vec{\xi}_1 \circ \vec{\xi}_3) D_{wv} \right] (\Delta_w - 12) \Delta_w F_{\Delta_\sigma=2} \,, \tag{151}
$$

where we have defined

$$
\begin{aligned}
\vec{\xi}_1 \circ \vec{\xi}_3 &\equiv \frac{4}{u^2} \xi_{1i} \xi_{3j} k_2^{[i} k_4^{j]} = \frac{2}{u^2} \left( (\vec{\xi}_1 \cdot \vec{k}_2)(\vec{\xi}_3 \cdot \vec{k}_4) - (\vec{\xi}_1 \cdot \vec{k}_4)(\vec{\xi}_3 \cdot \vec{k}_2) \right) \\
&= \frac{2}{u^2} \left( (\vec{\xi}_1 \cdot \vec{k}_3)(\vec{\xi}_3 \cdot \vec{k}_2) - (\vec{\xi}_1 \cdot \vec{k}_2)(\vec{\xi}_3 \cdot \vec{k}_1) \right) .
\end{aligned}
\tag{152}
$$

In the second line of this equation, we have used momentum conservation to eliminate $\vec{k}_4$.

The result in (151) has the right quantum numbers, but does *not* have the correct poles expected for particle exchange in the $u$-channel. These singularities are present in the seed $F_{\Delta_\sigma=2}$, but are removed by the weight-shifting procedure. This follows from inspection of (120), which implies that

$$
(\Delta_w - 12) \Delta_w F_{\Delta_\sigma=2} = \Delta_w C - 12 C \,, \tag{153}
$$

so that (151) corresponds to a pure contact solution. However, the fix is simple: to recover the singularity structure associated with the particle exchange, we replace $(\Delta_w - 12)\Delta_w F_{\Delta_\sigma=2}$ by $F_{\Delta_\sigma=2}$ in (151). This leads to

$$
\langle J \varphi J \varphi \rangle_u = \frac{1}{(k_{13} + u)(k_{24} + u)E} \left[ \mathcal{P}_1^{(u)} \vec{\xi}_1 \cdot \vec{\xi}_3 + 2u^2 \vec{\xi}_1 \circ \vec{\xi}_3 \right], \tag{154}
$$

where we have defined the following combination of internal polarization sums (see §E.2)

$$
\mathcal{P}_1^{(u)} \equiv u^2 \Pi_{1,1}^{(u)} - (k_{13} + u)(k_{24} + u) \Pi_{1,0}^{(u)} . \tag{155}
$$

Notice that the coefficient of the $E \to 0$ singularity reproduces the $u$-channel Feynman diagram for non-Abelian Compton scattering in axial gauge; cf. Appendix F.[44] An alternative derivation of the final result (154) is given in Appendix G, which bypasses some of the complications discussed here.

*One channel is not enough.*—As before, the full correlator is only consistent with the Ward–Takahashi identity if all channels, and an appropriate contact solution, are added with correlated couplings. The relevant couplings are shown in Fig. 7, so that the sum of channels can be written as

$$
\begin{aligned}
\langle J^A \varphi^a J^B \varphi^b \rangle &\equiv N_{J^2 \varphi^2}^{(s)} (T^A T^B)_{ab} \langle J \varphi J \varphi \rangle_s + N_{J^2 \varphi^2}^{(t)} (T^B T^A)_{ab} \langle J \varphi J \varphi \rangle_t \\
&\quad + N_{J^2 \varphi^2}^{(u)} f^{ABC} T_{ab}^C \langle J \varphi J \varphi \rangle_u + N_{J^2 \varphi^2}^{(c)} (T^{(A} T^{B)})_{ab} \langle J \varphi J \varphi \rangle_c \,,
\end{aligned}
\tag{156}
$$

---

[43]We will use these definitions for the variables $w$ and $v$ when computing $u$-channel correlators, instead of the $s$-channel definitions given in (117). It should always be clear from the context whether we are dealing with the $s$- or $u$-channel variables.

[44]This did not have to be the case. Individual flat-space Feynman diagrams and individual correlator channels are not physically meaningful, so it is not required that they match. However, it turns out that our weight-shifting procedure generates objects that do indeed have total energy singularities that match the individual flat-space diagrams computed in axial gauge. See Section 6 for a more detailed discussion.

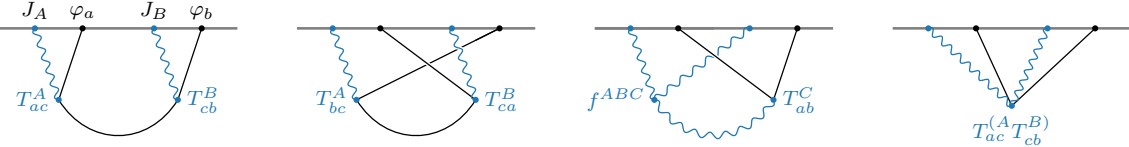

Figure 7: Diagrammatic representation of the contributions to the non-Abelian Compton correlator.

where we have written the flavor structure—which is carried by the couplings—explicitly, but allowed the relative normalizations of the channels to be arbitrary. The kinematic structures associated to each channel and to the contact term are given by (140), (144) and (154). Acting with the operator $\widetilde{K}$ on each of them, we find

$$\sum_a \vec{b} \cdot \widetilde{K} \langle J^- \varphi J^+ \varphi \rangle_s = -\frac{\vec{b} \cdot \vec{\xi}_1^-}{k_1^2} \vec{\xi}_3^+ \cdot \vec{k}_4 \left( \frac{1}{k_{34} + s} - \frac{k_1}{E^2} - \frac{1}{E} \right) + (1 \leftrightarrow 3), \tag{157}$$

$$\sum_a \vec{b} \cdot \widetilde{K} \langle J^- \varphi J^+ \varphi \rangle_c = +\frac{\vec{b} \cdot \vec{\xi}_1^-}{k_1^2} \vec{\xi}_3^+ \cdot \vec{k}_1 \left( \frac{k_1}{E^2} + \frac{1}{E} \right) + (1 \leftrightarrow 3), \tag{158}$$

$$\sum_a \vec{b} \cdot \widetilde{K} \langle J^- \varphi J^+ \varphi \rangle_u = -\frac{\vec{b} \cdot \vec{\xi}_1^-}{k_1^2} \left[ 2\vec{\xi}_3^+ \cdot \vec{k}_2 \left( \frac{1}{k_{24} + u} - \frac{k_1}{E^2} - \frac{1}{E} \right) \right.$$
$$\left. + \vec{\xi}_3^+ \cdot \vec{k}_1 \left( \frac{-k_2 + k_4 + u}{u(k_{24} + u)} - \frac{k_1}{E^2} - \frac{1}{E} \right) \right] + (1 \leftrightarrow 3), \tag{159}$$

where the $t$-channel contribution can be obtained from the $s$-channel by permuting $2 \leftrightarrow 4$. On the other hand, the WT identity (147), together with the three-point function

$$\langle J^A \varphi^a \varphi^b \rangle = -i T_{ab}^A \frac{2}{K} (\vec{k}_2 \cdot \vec{\xi}_1), \tag{160}$$

implies that the action of this operator on the full correlator (156) must be equal to

$$\sum_a \vec{b} \cdot \widetilde{K} \langle J_-^A \varphi^a J_+^B \varphi^b \rangle = -\frac{4\vec{b} \cdot \vec{\xi}_1^-}{k_1^2} \left[ (T^A T^B)_{ab} \frac{\vec{\xi}_3^+ \cdot \vec{k}_4}{k_{34} + s} + (T^B T^A)_{ab} \frac{\vec{\xi}_3^+ \cdot \vec{k}_2}{k_{23} + t} \right.$$
$$\left. + f^{ABC} T_{ab}^C \left( \frac{\vec{\xi}_3^+ \cdot \vec{k}_4}{k_{24} + u} + \frac{\vec{\xi}_3^+ \cdot \vec{k}_1 (k_2 - k_4 + u)}{2u(k_{24} + u)} \right) \right]. \tag{161}$$

Consistency with the WT identity then requires that the relative normalizations of the different channels are

$$N_{J^2 \varphi^2}^{(s)} = N_{J^2 \varphi^2}^{(t)} = 4, \quad N_{J^2 \varphi^2}^{(u)} = -1, \quad N_{J^2 \varphi^2}^{(c)} = 2, \tag{162}$$

and that the couplings satisfy

$$[T^A, T^B]_{ab} = f^{ABC} T_{ab}^C. \tag{163}$$

That is, the matter couplings $T_{ab}^A$ must transform in a representation of the Lie algebra.

*Identically conserved correlators.*—Having found a correlation function that satisfies the WT identity, we are still free to add to it any identically conserved correlators. For example, we can add an $s$-channel contribution of the form

$$\langle J \varphi J \varphi \rangle_s = P_1^{(1)} P_3^{(1)} H_{13}(F^{(\ell)})$$
$$= k_1 k_3 \left[ (\vec{\xi}_1 \cdot \vec{\xi}_3) K_{13}^2 - 2(\vec{\xi}_1 \cdot \vec{K}_{13})(\vec{\xi}_3 \cdot \vec{K}_{13}) \right] F^{(\ell)}, \tag{164}$$

where, in the second line, we have used the fact that the polarization vectors are eigenvectors of the projection operators. By feeding arbitrary exchange or contact solutions into this equation, we obtain identically conserved correlation functions.

### 5.3 Correlators with Spin-2 Currents

In the following, we repeat the above analysis for correlators involving spin-2 currents. The treatment is conceptually very similar to that of the previous section, so we will suppress some of the details.

#### 5.3.1 $\langle TOOO \rangle$

First, we consider the correlation function of one spin-2 current and three conformally coupled scalars. This correlator satisfies the following WT identity

$$
\begin{aligned}
k_1^i \langle T_{\vec{k}_1}^i \varphi_{\vec{k}_2} \varphi_{\vec{k}_3} \varphi_{\vec{k}_4} \rangle = {}&-\kappa_2(\vec{\xi}_1 \cdot \vec{k}_2)\langle \varphi_{\vec{k}_1+\vec{k}_2} \varphi_{\vec{k}_3} \varphi_{\vec{k}_4} \rangle - \kappa_3(\vec{\xi}_1 \cdot \vec{k}_3)\langle \varphi_{\vec{k}_2} \varphi_{\vec{k}_1+\vec{k}_3} \varphi_{\vec{k}_4} \rangle \\
&- \kappa_4(\vec{\xi}_1 \cdot \vec{k}_4)\langle \varphi_{\vec{k}_2} \varphi_{\vec{k}_3} \varphi_{\vec{k}_1+\vec{k}_4} \rangle ,
\end{aligned}
\tag{165}
$$

where $\kappa_a$ are the gravitational couplings of the operators $\varphi_a$, which we have allowed to be different. We see that this WT identity is similar to that for $\langle J\varphi\varphi \rangle$ in (129).

Using (121) as a seed function, an $s$-channel correlator with the correct quantum numbers is

$$
\langle T\varphi\varphi\varphi \rangle_s = k_2 D_{12}^2 \mathcal{W}_{12}^{++} \langle \varphi\varphi\varphi\varphi \rangle_s .
\tag{166}
$$

The result is a mix of the desired exchange solution and additional contact solutions. This can be diagnosed by taking the limit $E \to 0$. We expect this limit to be singular and the coefficient of the singularity to be the flat-space amplitude (see §6.1). Simple dimensional analysis then tells us that the correlator should diverge as $E^{-1}$ in this limit. The result in (166), however, diverges as $E^{-3}$, suggesting the presence of additional contact solutions. Subtracting the contact solutions associated to the $E^{-3}$ and $E^{-2}$ singularities,[45] and rescaling the answer, we obtain the following pure exchange contribution

$$
\langle T\varphi\varphi\varphi \rangle_s = (\vec{\xi}_1 \cdot \vec{k}_2)^2 \left[ \frac{k_{12}^2 + 2k_{12}k_1 - s^2}{(k_{12}+s)^2(k_{12}-s)^2} \log\left(\frac{k_{34}+s}{E}\right) + \frac{1}{(k_{12}+s)(k_{12}-s)}\frac{k_1}{E} \right].
\tag{169}
$$

The $t$- and $u$-channel answers are simple permutations of (169).

*One channel is not enough.*—We now want to use the expression (169) to solve the WT identity (165). As in the previous examples, it is not consistent to only include this $s$-channel contribution. Indeed, we will see that all the scalar operators involved in the correlator must couple to the spin-2 stress tensor, corresponding to scalar exchange in all three channels.

To see this, we normalize the correlator (169) with a factor of $\kappa_2 N_{T\varphi^3}$, where $N_{T\varphi^3}$ is the overall normalization of the final answer, and $\kappa_2$ is the coupling of the graviton to operator 2 (see Fig. 8), which parametrizes the normalization relative to the other channels. Acting with

---

[45]The required contact solutions can also be generated by weight shifting. The two contact solutions accounting for the $E^{-3}$ and $E^{-2}$ divergences are obtained from distinct seeds. One seed is the solution $\hat{C}_0$ of (119), corresponding to a bulk $\varphi^4$ interaction, and the other is the contact solution $\hat{C}_0$ obtained in Appendix D of [47], which is given by

$$
\hat{C}_0(w,v) \equiv \frac{1}{3}\left[ \left(\frac{1}{w^3} + \frac{1}{v^3}\right)\log\left(\frac{\mu}{E}\right) + \left(\frac{1}{w} + \frac{1}{v}\right)\frac{1}{wv} \right],
\tag{167}
$$

where $\mu$ is some momentum-independent mass scale. The correlator arising from a bulk $\phi^4$ interaction can then be written as

$$
\mathcal{C}_0 = s^3 O_{12} O_{34} \hat{C}_0 ,
\tag{168}
$$

where the differential operators appearing in this expression are defined as $O_{ab} \equiv 1 - \frac{k_a k_b}{k_{ab}}\partial_{k_{ab}}$.

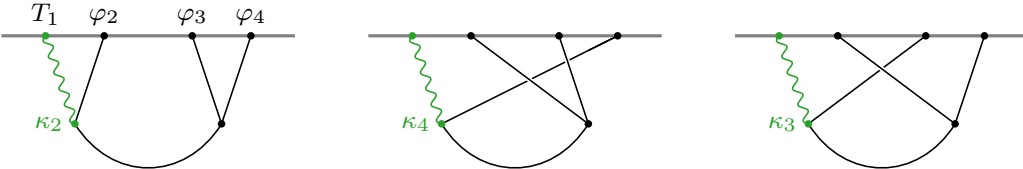

Figure 8: Diagrammatic representation of the $s$, $t$, and $u$-channel contributions to $\langle T\varphi\varphi\varphi\rangle$.

the operator $\widetilde{K}$, we get

$$\sum_a \vec{b} \cdot \widetilde{K}_a \left( \kappa_2 N_{T\varphi^3} k_1^{-1} \langle T^-_{\vec{k}_1} \varphi_{\vec{k}_2} \varphi_{\vec{k}_3} \varphi_{\vec{k}_4} \rangle_s \right)$$
$$= 6 N_{T\varphi^3} \frac{\vec{b} \cdot \vec{\xi}_1^-}{k_1^3} \left( \kappa_2 (\vec{\xi}_1^- \cdot \vec{k}_2) \left[ \frac{k_1^2}{3E^2} + \frac{k_1}{E} + \log\left( \frac{s + k_3 + k_4}{E} \right) \right] \right). \tag{170}$$

At the same time, the WT identity requires that the action of $\widetilde{K}$ on the full correlator gives

$$\sum_a \vec{b} \cdot \widetilde{K}_a \left( k_1^{-1} \langle T^-_{\vec{k}_1} \varphi_{\vec{k}_2} \varphi_{\vec{k}_3} \varphi_{\vec{k}_4} \rangle \right) = -12 N_{\varphi^3} \frac{\vec{b} \cdot \vec{\xi}_1^-}{k_1^3} \left( (\vec{\xi}_1^- \cdot \vec{k}_2) \log(s + k_3 + k_4) + 2 \text{ perms.} \right), \tag{171}$$

where $N_{\varphi^3}$ is the normalization of $\langle\varphi\varphi\varphi\rangle$. We see that there are terms with $E^{-2}$, $E^{-1}$, and $\log E$ singularities in (170) that should be absent. It is relatively straightforward to see how to rectify the situation. Each of these terms multiplies $\vec{\xi}_1^- \cdot \vec{k}_2$, so if we add the analogous $t$- and $u$-channel contributions with identical coupling constants, then these terms will be proportional to $\vec{\xi}_1^- \cdot (\vec{k}_2 + \vec{k}_3 + \vec{k}_4)$, which vanishes after using momentum conservation. More explicitly, we find that the sum of the three exchange channels,

$$\langle T\varphi\varphi\varphi\rangle_{s+t+u} \equiv N_{T\varphi^3} \left[ \kappa_2 \langle T\varphi\varphi\varphi\rangle_s + \kappa_4 \langle T\varphi\varphi\varphi\rangle_t + \kappa_3 \langle T\varphi\varphi\varphi\rangle_u \right], \tag{172}$$

solves the WT identity if the couplings satisfy

$$N_{T\varphi^3} = -2 N_{\varphi^3} \quad \text{and} \quad \kappa_2 = \kappa_3 = \kappa_4 \equiv \kappa. \tag{173}$$

We see that the normalization of the four-point function is fixed in terms of the three-point function, so that all channels have the same coupling to the stress tensor. This is, of course, a manifestation of the bulk equivalence principle.

*Identically conserved correlators.*—As before, we are still free to add any identically conserved correlator with the correct quantum numbers. One way to build such correlators is as

$$\langle T\varphi\varphi\varphi\rangle_s = P_1^{(2)} (D_{11})^2 \left( F^{(\ell)} \right), \tag{174}$$

where $P_1^{(2)}$ is the spin-2 projector (39). By acting with this differential operator on any of the spin-exchange solutions, or alternatively on a $\Delta = 2$ contact solution, we can construct a wide variety of identically conserved correlators involving a single stress tensor. It would be nice to determine whether this spans all possible contact solutions, by matching with the possible flat-space structures.

### 5.3.2 $\langle TOTO\rangle$

Finally, we consider the four-point correlator involving two spin-2 currents and two conformally coupled scalars. This is the gravitational analog of Compton scattering. The relevant



Figure 9: Diagrammatic representation of different contributions to the gravitational Compton correlator.

WT identity is (see Appendix B for a derivation)

$$
\begin{aligned}
k_1^i \langle T^i_{\vec{k}_1} \varphi_{\vec{k}_2} T_{\vec{k}_3} \varphi_{\vec{k}_4} \rangle = &- \kappa (\vec{\xi}_1 \cdot \vec{k}_2) \langle \varphi_{\vec{k}_2+\vec{k}_1} T_{\vec{k}_3} \varphi_{\vec{k}_4} \rangle - \kappa (\vec{\xi}_1 \cdot \vec{k}_4) \langle \varphi_{\vec{k}_2} T_{\vec{k}_3} \varphi_{\vec{k}_4+\vec{k}_1} \rangle \\
&+ 2\kappa^2 (\vec{\xi}_1 \cdot \vec{\xi}_3)(\vec{\xi}_3 \cdot \vec{k}_2) \langle \varphi_{\vec{k}_2+\vec{k}_3+\vec{k}_1} \varphi_{\vec{k}_4} \rangle \\
&+ 2\kappa^2 (\vec{\xi}_1 \cdot \vec{\xi}_3)(\vec{\xi}_3 \cdot \vec{k}_4) \langle \varphi_{\vec{k}_4+\vec{k}_3+\vec{k}_1} \varphi_{\vec{k}_2} \rangle \\
&- \kappa_g (\vec{\xi}_1 \cdot \vec{k}_3) \xi_3^i \xi_3^j \langle \varphi_{\vec{k}_2} T^{ij}_{\vec{k}_3+\vec{k}_1} \varphi_{\vec{k}_4} \rangle + 2\kappa_g (\vec{\xi}_1 \cdot \vec{\xi}_3) \xi_3^j k_3^i \langle \varphi_{\vec{k}_2} T^{ij}_{\vec{k}_3+\vec{k}_1} \varphi_{\vec{k}_4} \rangle \\
&+ \kappa_g (\vec{\xi}_3 \cdot \vec{k}_1) \xi_1^i \xi_3^j \langle \varphi_{\vec{k}_2} T^{ij}_{\vec{k}_3+\vec{k}_1} \varphi_{\vec{k}_4} \rangle + \kappa_g (\vec{\xi}_1 \cdot \vec{\xi}_3) \xi_3^i k_1^j \langle \varphi_{\vec{k}_2} T^{ij}_{\vec{k}_3+\vec{k}_1} \varphi_{\vec{k}_4} \rangle,
\end{aligned}
\tag{175}
$$

where we are using the shorthand notation $T_{\vec{k}_3} \equiv \xi_3^i \xi_3^j T^{ij}_{\vec{k}_3}$. We have used (173) to set the couplings of the scalar operators to the graviton to be equal, but have allowed for the possibility that the graviton self-coupling is normalized differently. The three-point functions appearing on the right-hand side can be obtained from (94). As in the case involving spin-1 currents, special care must be taken to include the longitudinal and trace components of the $\langle T \varphi \varphi \rangle$ correlators, both of which are fixed by WT identities; see (B.38) for the trace identity.

This gravitational Compton example is conceptually quite similar to the non-Abelian Compton correlator for spin-1 currents. There are two distinct types of exchanges (see Fig. 9), one where a scalar is exchanged (which can happen in either the $s$- or $t$-channel) and another where the exchanged field is the graviton (in the $u$-channel). The sum of all of these processes—along with a particular contact solution—is required to solve the WT identity (175), and we will consider each of them in turn.

*Scalar exchange.*—The contribution from scalar exchange (in the $s$-channel) is obtained by acting twice with the weight-shifting operator of (166) on the seed function (121):

$$
\langle T \varphi T \varphi \rangle_s \stackrel{?}{=} \left( k_4 D_{34}^2 \mathcal{W}_{34}^{++} \right) \left( k_2 D_{12}^2 \mathcal{W}_{12}^{++} \right) \langle \varphi \varphi \varphi \varphi \rangle_s .
\tag{176}
$$

Using the explicit expressions for the weight-shifting operators, this combination can be simplified into the following form

$$
\langle T \varphi T \varphi \rangle_s \stackrel{?}{=} (\vec{\xi}_1 \cdot \vec{k}_2)^2 (\vec{\xi}_3 \cdot \vec{k}_4)^2 Q_{12} Q_{34} (\Delta_w - 2)(\Delta_v - 2) F_{\Delta_\sigma = 2} ,
\tag{177}
$$

where we have defined the differential operator

$$
\begin{aligned}
Q_{12} &= \frac{w}{1-w^2} \Big[ (s + w\alpha)\Delta_w + 2(u\alpha + s(2-w^2)\partial_w \Big] \\
&= w^2 \big( 4s + 2w\alpha + w(s + w\alpha)\partial_w \big) \partial_w .
\end{aligned}
\tag{178}
$$

The operator $Q_{34}$ is the same as $Q_{12}$, with $w \to v, \alpha \to \beta$. Notice that the expression (177) contains pieces involving $\Delta_w$ and $\Delta_v$ acting on the $\Delta = 2$ exchange seed $F_{\Delta_\sigma = 2}$. From (120), we see that these operators collapse the exchange singularities and create a contact solution. There are two ways to deal with this: either we explicitly subtract off these contact solution contributions, or we can make the replacement $(\Delta_u - 2)(\Delta_v - 2) F_{\Delta_\sigma = 2} \to F_{\Delta_\sigma = 2}$ in (177), which then isolates the exchange contribution. We choose this latter, more economical, route. Acting with the operators $Q_{ab}$, the pure exchange contribution then is

$$\langle T\varphi T\varphi\rangle_s = 4(\vec{\xi}_1\cdot\vec{k}_2)^2(\vec{\xi}_3\cdot\vec{k}_4)^2\left[\frac{1}{(k_{12}+s)(k_{34}+s)}\left(\frac{2k_1k_3}{E^3}+\frac{k_{13}}{E^2}+\frac{1}{E}\right)\right.$$
$$\left.+\frac{1}{(k_{12}+s)^2(k_{34}+s)^2}\left(\frac{2sk_1k_3}{E^2}+\frac{2k_1k_3+(k_{12}+s)k_3+(k_{34}+s)k_1}{E}\right)\right],\tag{179}$$

where we have rescaled the correlator to have a convenient overall normalization. In §6.2, we will show how this rather complex result can also be obtained by imposing that the correlator has the right singularities. The $t$-channel result is a simple permutation of (179).

*Graviton exchange.*—The contribution from graviton exchange, $\langle T\varphi T\varphi\rangle_u$, presents the same problem as for $\langle J\varphi J\varphi\rangle_u$. The naive weight-shifting procedure, suggested by (113), in this case is

$$\langle T\varphi T\varphi\rangle_u \overset{?}{=} \mathcal{D}_{13}^{(2)}\langle O_5\varphi O_5\varphi\rangle_{u,T},\tag{180}$$

where $\langle O_5\varphi O_5\varphi\rangle_{u,T}$ corresponds to the graviton exchange contribution to the scalar seed. This seed correlator is obtained by raising the spin of the exchanged field of (122) by two units, which gives $\langle\varphi\varphi\varphi\varphi\rangle_{u,T}$, and then raising the conformal weight of two legs to $\Delta=5$:

$$\langle O_5\varphi O_5\varphi\rangle_{u,T} = \left(\mathcal{W}_{13}^{++}\right)^3\langle\varphi\varphi\varphi\varphi\rangle_{u,T},$$
$$= \left(\mathcal{W}_{13}^{++}\right)^3\left[\Pi_{2,2}^{(u)}D_{wv}^2+\Pi_{2,1}^{(u)}D_{wv}(\Delta_w-2)+\Pi_{2,0}^{(u)}\Delta_w(\Delta_w-2)\right]F_{\Delta_\sigma=3}.\tag{181}$$

Substituting this into (180), leads to

$$\langle T\varphi T\varphi\rangle_u \overset{?}{=} \mathcal{D}_{T\varphi T\varphi}(\Delta_w-72)(\Delta_w-42)(\Delta_w-12)(\Delta_w-2)\Delta_w F_{\Delta_\sigma=3},\tag{182}$$

where the differential operator $\mathcal{D}_{T\varphi T\varphi}$ is defined in (G.19). Just like in the analysis for $\langle J\varphi J\varphi\rangle_u$, the poles expected for particle exchange are absent in this solution—the operator $(\Delta_w-2)$ acting on the seed has eliminated them, cf. (120). The correct exchange solution is obtained by replacing $(\Delta_w-72)\cdots(\Delta_w-2)\Delta_w F_{\Delta_\sigma=3}$ with $F_{\Delta_\sigma=3}$. After some nontrivial algebra, the final result can be written as

$$\langle T\varphi T\varphi\rangle_u = \frac{1}{(k_{13}+u)(k_{24}+u)}\left(\frac{2k_1k_3}{E^3}+\frac{k_{13}}{E^2}\right)\mathcal{M}$$
$$+\frac{u}{(k_{13}+u)^2(k_{24}+u)^2}\left(\frac{2k_1k_3}{E^2}+\frac{k_{13}+u}{E}\right)\mathcal{N}+\frac{1}{E}\mathcal{L},\tag{183}$$

where we have defined the following polarization structures

$$\mathcal{N}\equiv\mathcal{Q}_2\frac{(\vec{\xi}_1\cdot\vec{\xi}_3)^2}{6}+u^2\mathcal{Q}_1(\vec{\xi}_1\cdot\vec{\xi}_3)(\vec{\xi}_1\circ\vec{\xi}_3)+u^4(\vec{\xi}_1\circ\vec{\xi}_3)^2,$$
$$\mathcal{M}\equiv\mathcal{P}_2\frac{(\vec{\xi}_1\cdot\vec{\xi}_3)^2}{6}+u^2\mathcal{P}_1(\vec{\xi}_1\cdot\vec{\xi}_3)(\vec{\xi}_1\circ\vec{\xi}_3)+u^4(\vec{\xi}_1\circ\vec{\xi}_3)^2,\tag{184}$$
$$\mathcal{L}\equiv\frac{\left(u^2-(k_1-k_3)^2\right)\left(u^2-3(k_2-k_4)^2\right)}{2u^2}\frac{(\vec{\xi}_1\cdot\vec{\xi}_3)^2}{6},$$

with the angular functions

$$\mathcal{Q}_2\equiv u^4\Pi_{2,2}^{(u)}-(k_{13}+u)^2(k_{24}+u)^2\,\Pi_{2,0}^{(u)},$$
$$\mathcal{Q}_1\equiv u^2\Pi_{1,1}^{(u)},$$
$$\mathcal{P}_2\equiv u^4\Pi_{2,2}^{(u)}-(k_{13}+u)(k_{24}+u)u^2\Pi_{2,1}^{(u)}+(k_{13}+u)^2(k_{24}+u)^2\,\Pi_{2,0}^{(u)},\tag{185}$$
$$\mathcal{P}_1\equiv u^2\Pi_{1,1}^{(u)}-(k_{13}+u)(k_{24}+u)\,\Pi_{1,0}^{(u)}.$$

The full answer has both a contribution with the exchange singularities required to reproduce the flat-space scattering amplitude—the first line of (183)—and a piece with higher-order partial energy singularities, which are sub-leading in the flat-space limit, but are required by conformal invariance.

*Contact solution.*—Given our experience with the spin-1 Compton correlator, it is natural to expect that an additional contract solution will be required in order to solve the WT identity (175). A candidate contact solution can be constructed by the following procedure: The scalar and spin-2 exchange contributions to the correlator have total energy singularities of order $E^3$ and lower, and we know that the $\langle T\varphi T\varphi \rangle$ correlator must have mass dimension one. This leaves three possible polarization structures compatible with both of these constraints. We can then construct these structures via weight-shifting and fix their relative coefficients be demanding consistency with the WT identity. The end result of this procedure is

$$\langle T\varphi T\varphi \rangle_c = (\mathcal{S}_{13}^{++})^2 \mathcal{W}_{13}^{++} C_0 + \frac{1}{2}\mathcal{S}_{13}^{++}(D_{12}D_{34} + D_{14}D_{32})\mathcal{C}_0\,, \tag{186}$$

where $C_0 = 1/E$, and $\mathcal{C}_0$ is the $\phi^4$ contact solution given by (168). Explicitly evaluating the action of the weight-shifting operators, and rescaling by an overall factor, we obtain

$$\begin{aligned}
\langle T\varphi T\varphi \rangle_c = \frac{1}{6}&\left( (k_{13}^2 - u^2)\left( \frac{2k_1 k_3}{E^3} + \frac{k_{13}}{E^2} + \frac{1}{E} \right) - 2k_1 k_3 \left( \frac{k_{13}}{E^2} + \frac{1}{E} \right)\right)(\vec{\xi}_1 \cdot \vec{\xi}_3)^2 \\
&+ 2\left( \frac{2k_1 k_3}{E^3} + \frac{k_{13}}{E^2} + \frac{1}{E} \right)\left[ (\vec{\xi}_1 \cdot \vec{k}_4)(\vec{\xi}_3 \cdot \vec{k}_2) + (\vec{\xi}_1 \cdot \vec{k}_2)(\vec{\xi}_3 \cdot \vec{k}_4) \right](\vec{\xi}_1 \cdot \vec{\xi}_3).
\end{aligned} \tag{187}$$

This correlator has only total energy singularities, which have a similar structure to those appearing in the exchange contributions.

In addition to the pieces that we have constructed, there are possible local terms that in principle could contribute to the correlator. However, we find that they are not needed to satisfy the WT identity (175), so we do not have to consider them here.

*One channel is not enough.*—To solve the Ward–Takahashi identity (175), we combine the exchange and contract contributions in the following way

$$\langle T\varphi T\varphi \rangle = N_{T^2\varphi^2}^{(s)}\langle T\varphi T\varphi \rangle_s + N_{T^2\varphi^2}^{(t)}\langle T\varphi T\varphi \rangle_t + N_{T^2\varphi^2}^{(u)}\langle T\varphi T\varphi \rangle_u + \kappa_c^2\langle T\varphi T\varphi \rangle_c\,, \tag{188}$$

where we have allowed for independent normalizations in the different channels. Going through the same procedure as before, consistency between the action of $\widetilde{K}$ and the WT identity imposes the following constraints:

$$N_{T^2\varphi^2}^{(s)} = N_{T^2\varphi^2}^{(t)} = -\kappa^2\,, \quad N_{T^2\varphi^2}^{(u)} = -\kappa\kappa_g \quad \text{and} \quad \kappa_g = \kappa_c = \kappa\,. \tag{189}$$

To obtain these relations, it is also necessary to use the WT identity (92) for $\langle TOO \rangle$, which relates the normalization of $\langle T\varphi\varphi \rangle$ to that of $\langle \varphi\varphi \rangle$ (which we have taken to be 1). We learn that the various channels have to be added together with precise relative normalizations and further that the self-coupling of graviton must be the same as its coupling to other operators. Additionally, the normalization of the contact contribution is completely fixed in terms of these couplings. The fact that all of these coupling are the same is another manifestation of the equivalence principle. This will be made more explicit in Section 6, where the normalizations of the various correlators are directly related to the bulk couplings in a more transparent way.

*Identically conserved correlators.*—As before, we are still free to add identically conserved correlators as homogeneous solutions to the WT identity. A possible such correlator is

$$\langle T\varphi T\varphi \rangle_s = \mathsf{P}_1^{(2)}\mathsf{P}_3^{(2)}(H_{13})^2\big(F^{(\ell)}\big)\,, \tag{190}$$

where $F^{(\ell)}$ can be any of the exchange or contact solutions that we have discussed.

## 5.4 Summary of Results

In this section, we used the weight-shifting formalism to derive selected four-point correlations between conserved currents (of spin 1 and 2) and conformally coupled scalars. Here, we briefly summarize our results.

- The result for the $s$-channel correlator $\langle J\varphi\varphi\varphi\rangle_s$ is given in (131), with the corresponding $t$- and $u$-channel answers related to this by permutation symmetry. In order to satisfy the WT identity, the different channels must be added with correlated coefficients, cf. (136). From the bulk perspective, this is understood as the requirement of charge conservation. Finally, we can add to the solution any correlator that is identically conserved. A large class of such correlators was presented in (137).

- The $s$-channel contribution to Abelian Compton scattering, $\langle J\varphi J\varphi\rangle_s$, is given in (140). This time consistency with current conservation requires both the $t$-channel and a contact solution. The complete Abelian Compton correlator can be found in (146). Non-Abelian Compton scattering receives an additional contribution from the exchange of the conserved current in the $u$-channel. The solution for $\langle J\varphi J\varphi\rangle_u$ is given in (154). In deriving this, we had to modify the naive seed function in the weight-shifting procedure in order to recover the expected poles characteristic of particle exchange. The full correlator is only consistent with the WT identity if all channels, and the contact solution, are added with correlated couplings. Specifically, the matter couplings must transform in a representation of the Lie algebra, cf. (163). We may add to the full correlator any amount of the identically conserved solutions in (164).

- The result for the $s$-channel correlator $\langle T\varphi\varphi\varphi\rangle_s$ is given in (169). Again, the $t$- and $u$-channels are related to this by permutation symmetry and must be added with the correct coefficients. This constraint provides a boundary derivation of the equivalence principle. The identically conserved correlators (174) may be added independently. From the bulk perspective, this corresponds to the freedom to add higher-derivative corrections to Einstein gravity.

- The result for gravitational Compton scattering in the $s$-channel, $\langle T\varphi T\varphi\rangle_s$, is (179), and the corresponding $t$-channel answer is related to this by permutation symmetry. To obtain this result, contact contributions had to be subtracted from the naive weight-shifting result. The $u$-channel correlator $\langle T\varphi T\varphi\rangle_u$ arises from graviton exchange and is given in (183). As in the corresponding spin-1 example, we had to modify the naive seed function in order to recover the expected poles characteristic of particle exchange. The full correlator also must have the contact solution (187). The sum of all contributions is only consistent if the couplings satisfy the equivalence principle. A class of higher-derivative corrections are captured by the identically conserved correlators in (190).

## 6 Four-Point Functions from Factorization

The analytic structure of tree-level scattering amplitudes is well understood [46]. Locality demands that the poles that arise when intermediate particles go on-shell are simple poles. On these poles, the amplitudes must factorize into products of lower-point amplitudes with positive coefficients. In fact, knowing all factorization channels, together with locality and Lorentz symmetry, is often sufficient to construct tree-level amplitudes from lower-point data alone. This feature of tree-level amplitudes has been formalized in the celebrated BCFW recursion relations [116].

By comparison, the singularity structure of cosmological correlation functions is much less understood. Nevertheless, new insights into the singularities of cosmological correlators have recently emerged, see e.g. [51–54, 59, 117]. In particular, cosmological correlators develop singularities when the sum of the energies entering a subgraph vanishes, and the coefficients of these singularities are related to the corresponding flat-space scattering amplitudes. In §6.1, we will establish these facts through perturbative calculations of wavefunction coefficients. We will then explain, in §6.2, how this provides an efficient way to determine the correlators derived in Section 5. The relevant correlators will be constructed separately for the $s$, $t$, and $u$-channels. Finally, we will show, in §6.3, that in many cases the full correlator is only consistent if the individual channels are related to each other.

## 6.1 Singularities of Cosmological Correlators

To gain some intuition for the possible singularities of cosmological correlators, it is helpful to first study a few explicit perturbative computations of the wavefunction of the universe. We will begin with a brief review of the Feynman rules for these computations (see e.g. [30, 118] for more details), and then use them to investigate the types of singularities that can arise. Much of the relevant physics can already be understood from simple examples in flat space. As a concrete model, we will consider a scalar field in flat space, with an arbitrary set of interactions

$$S = \int \mathrm{d}^4 x \left( -\frac{1}{2}(\partial \sigma)^2 - \frac{m^2}{2}\sigma^2 + \mathcal{L}_{\text{int}}[\sigma] \right). \tag{191}$$

Despite its simplicity, this model has a lot of structure [51, 52, 117]. After understanding the singularity structure of correlation functions in this simplified setting, we will describe the generalization to cosmological backgrounds.

### 6.1.1 The Perturbative Wavefunction

We are interested in the wavefunction $\Psi[\bar{\sigma}, t_\star] \equiv \langle \bar{\sigma}|0\rangle$, for the (spatial) field configuration $\sigma_{\vec{k}}(t_\star) \equiv \bar{\sigma}_{\vec{k}}$ at a late time $t_\star$. Without loss of generality, we will set $t_\star = 0$. The wavefunction can then be computed via the path integral

$$\Psi[\bar{\sigma}] = \int_{\substack{\sigma(0)=\bar{\sigma} \\ \sigma(-\infty)=0}} \mathcal{D}\sigma\, e^{iS[\sigma]}. \tag{192}$$

This path integral is a sum over field configurations with the indicated boundary conditions at $t = 0$ and $t = -\infty$ (where the standard $i\epsilon$ prescription is implied). The latter condition defines the vacuum as the initial state. Although computing the path integral exactly is difficult, in perturbation theory it can be written as a saddle-point approximation, $\Psi[\bar{\sigma}] \approx \exp(iS[\bar{\sigma}_{\text{cl}}])$, where $S$ is the on-shell action and $\bar{\sigma}_{\text{cl}}$ is the boundary value of the solution $\sigma_{\text{cl}}$ to the classical equations of motion:

$$\sigma_{\text{cl}}(t, \vec{k}) = \mathcal{K}(\vec{k}, t)\bar{\sigma}_{\vec{k}} + \int \mathrm{d}t'\, \mathcal{G}(\vec{k}; t, t') \frac{\delta S_{\text{int}}}{\delta \bar{\sigma}_{\vec{k}}(t')}. \tag{193}$$

The formal solution in (193) can be evaluated iteratively to any desired order in $\bar{\sigma}$. As in the corresponding computations in anti-de Sitter space, this expansion can be organized in Feynman–Witten diagrams. The essential objects involved in (193) are the bulk-to-boundary

propagator and the bulk-to-bulk propagator:

$$\mathcal{K}(\vec{k}, t) = e^{iE_k t}, \tag{194}$$

$$\mathcal{G}(\vec{k}; t, t') = \frac{1}{2E_k} \left( e^{iE_k(t'-t)}\theta(t-t') + e^{iE_k(t-t')}\theta(t'-t) - e^{iE_k(t+t')} \right), \tag{195}$$

where $E_k$ is the energy of a given momentum mode $\vec{k}$. The function $\mathcal{K}(\vec{k}, t)$ is a solution to the linearized equation of motion that oscillates with a positive frequency in the far past and goes to 1 at $t = 0$. The bulk-to-bulk propagator is a Green's function for the Klein–Gordon equation, $(\partial_t^2 + k^2)\mathcal{G}(k, t, t') = -i\delta(t - t')$. Note that the first two terms of (195) are identical to the Feynman propagator, while the third term arises because $\mathcal{G}(\vec{k}; 0, t') = \mathcal{G}(\vec{k}; t, 0) = 0$ in order for $\sigma_{\rm cl}$ to have the correct limit on the boundary.

The recipe for computing the wavefunction then is:

- draw all diagrams with a fixed number of lines ending on the boundary
- assign a vertex factor, $iV$, to each bulk interaction
- assign a bulk-to-bulk propagator, $\mathcal{G}$, to each internal line
- assign a bulk-to-boundary propagator, $\mathcal{K}$, to each external line
- integrate over the time insertions of all bulk vertices.

We will use these Feynman rules to examine the singularities that can arise in the simple model (191). The final result is fairly simple to state: correlation functions have singularities whenever the energy flowing into a subgraph adds up to zero. The residues of these singularities are determined in terms of scattering amplitudes and lower-point correlation functions.

### 6.1.2 Total Energy Singularity

A nearly universal signature of local physics in correlation functions is the presence of a total energy singularity [59]. To understand the appearance of this singularity, we note that the computation of correlation functions in perturbation theory parallels that of scattering amplitudes, except that the time integrals range from $-\infty$ to 0 rather than from $-\infty$ to $+\infty$. It is these time integrals that, for scattering amplitudes, result in a delta function enforcing total energy conservation. For correlators, the integrals instead generate singularities at $E = 0$.[46] Since the rest of the computation is the same, the coefficients of these singularities are the flat-space scattering amplitudes. In the following, we will make this intuition more precise.

**Flat-space correlators**

We begin with the flat-space example introduced in (191). It is simplest to see the relationship between wavefunction coefficients and scattering amplitudes for contact interactions. Using the Feynman rules, we can write each as

$$\Psi_n^{(c)}(\vec{k}_a, E_{\rm tot}) = iV(\vec{k}_a) \int_{-\infty}^{0} dt\, e^{iE_{\rm tot}t} = \frac{V(\vec{k}_a)}{E_{\rm tot}}, \tag{196}$$

$$iA_n^{(c)}(\vec{k}_a, E_{\rm tot}) = iV(\vec{k}_a) \int_{-\infty}^{\infty} dt\, e^{iE_{\rm tot}t} = iV(\vec{k}_a)\, 2\pi\delta(E_{\rm tot}), \tag{197}$$

---

[46]These singularities arise from the early-time part of the integrals where the boundary is infinitely far way, which explains why the coefficient is Lorentz invariant. In the cosmological context, the divergences are again localized near $t \to -\infty$ and the limit $E \to 0$ corresponds to a flat-space limit, yielding a Lorentz-invariant residue.

where $a$ labels the external lines and $E_{\text{tot}} \equiv \sum_a E_a$. We added the subscript on $E_{\text{tot}}$ for clarity. The temporal component of the four-vector $k_a^\mu$ is defined implicitly in terms of the spatial components via $E_a^2 = \vec{k}_a^2 + m^2$. Note that this is a somewhat non-standard presentation of scattering amplitudes, where we have Fourier-transformed with respect to the spatial coordinates, but have left the time dependence as an explicit integral. We see that the only difference between the two computations is the range of the time integral, so that the correlator indeed has a pole at $E_{\text{tot}} = 0$, whose residue is the scattering amplitude.

Next, we consider a diagram with internal lines. It is simplest to consider tree-level correlation functions, for which a powerful recursive formula will allow us to prove the existence of a total energy singularity by induction. Let us consider a wavefunction coefficient coming from a graph of the form

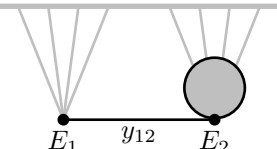

where the grey blob stands for an arbitrary completion of the graph.

In flat space, the bulk-to-boundary propagator is particularly simple, being a pure exponential factor. It therefore isn't necessary to track the individual external lines emanating from a vertex. Instead, the only physically important information is the total energy flowing out of a vertex to the boundary. We can therefore work with truncated graphs and keep track only of the energies associated to vertices—which are denoted $E_1$ and $E_2$ in the diagram above—and those attached to internal lines, like $y_{12}$ in the above example. The wavefunction coefficient corresponding to the graph above then has the following schematic form

$$\Psi_n = \int_{-\infty}^0 \mathrm{d}t_2 \, e^{iE_2 t_2} \; \bigcirc \; \int_{-\infty}^0 \mathrm{d}t_1 \, e^{iE_1 t_1} \mathcal{G}(y_{12}, t_1, t_2). \tag{198}$$

Using the frequency-space representation of the bulk-to-bulk propagator,

$$\mathcal{G}(y_{12}, t_1, t_2) = \int_{-\infty}^\infty \frac{\mathrm{d}\omega}{2\pi} \left( \frac{i e^{-i\omega(t_1 - t_2)}}{\omega^2 - y_{12}^2 + i\epsilon} - \frac{i e^{-i\omega(t_1 + t_2)}}{\omega^2 - y_{12}^2 + i\epsilon} \right), \tag{199}$$

we can perform the time integral over the vertex 1:

$$
\begin{aligned}
I &\equiv \int_{-\infty}^0 \mathrm{d}t_1 \, e^{iE_1 t_1} \mathcal{G}(y_{12}, t_1, t_2) \\
&= \int_{-\infty}^\infty \frac{\mathrm{d}\omega}{2\pi} \frac{1}{(E_1 - i\epsilon) - \omega} \left( \frac{e^{i\omega t_2}}{\omega^2 - y_{12}^2 + i\epsilon} - \frac{e^{-i\omega t_2}}{\omega^2 - y_{12}^2 + i\epsilon} \right) \\
&= \frac{-i}{y_{12}^2 - E_1^2} \left( e^{iE_1 t_2} - e^{i y_{12} t_2} \right).
\end{aligned}
\tag{200}
$$

From this, we see that the effect of the additional internal line is to shift the energy of the vertex 2. This implies the following recursion relation [51]

$$\underset{E_1 \quad y_{12} \quad E_2}{\bullet\!\!-\!\!\bullet\!\!\bigcirc} \;=\; \frac{-i}{y_{12}^2 - E_1^2} \left[ \underset{E_1 + E_2}{\bigcirc} - \underset{E_2 + y_{12}}{\bigcirc} \right]. \tag{201}$$

This recursive formula allows us to build tree diagrams by successively adding internal lines.

Using the above recursion relation, it is easy to prove by induction that all tree-level correlators have a pole at $E_{\text{tot}} = 0$, and that its coefficient is the flat-space scattering amplitude. Let us assume that the correlator represented by  has a pole at $E'_{\text{tot}} = E_2$, whose residue is the flat-space scattering amplitude. In the recursion relation (201), the first term on the right-hand side has the energy of this correlator shifted by $E_1$, so the pole at $E_2$ becomes a pole at $E_{\text{tot}} = E_1 + E_2$ for the correlator on the left-hand side. Next, we note that when $E_{\text{tot}} = 0$, the multiplicative factor in the recursion relation is exactly the Feynman propagator for the internal line we are attaching. If we then assume that the residue at the pole coming from the ⬭ part of the graph is the scattering amplitude with $r - 1$ internal lines then adding this Feynman propagator turns it into the scattering amplitude with $r$ internal lines. We have already proven this for the base case where $r = 1$, so the full answer follows.

In this section, we will restrict our attention to correlators of scalar operators. For spinning correlators, the $E \to 0$ limit of expressions with polarization vectors is somewhat subtle. Since we are working in terms of three-dimensional (equal-time) variables, the residue takes the form of a gauge-fixed Feynman diagram in axial gauge. This object is not physical, because it is not a Lorentz scalar. In §6.3, we will see that the presence of polarization vectors requires us to combine multiple channels in order to obtain the correct Lorentz-invariant scattering amplitude.

**De Sitter correlators**

The previous arguments apply to the computation of the flat-space wavefunction, but the presence of a total energy singularity is completely generic, holding also for the cosmological wavefunction in a de Sitter background. The essential reason for this universality is that the total energy singularity arises from the temporal integration region in the infinite past. As we approach this limit in de Sitter space, we are probing scales that are much smaller than the cosmological horizon and the corresponding momentum modes therefore behave in the same way as in flat space.

As before, it is simplest to first consider contact interactions. In this case, the integral representation of the wavefunction is

$$\Psi_n^{(c)}(\vec{k}_a, E_{\text{tot}}) = i \int_{-\infty}^{0} d\eta \, V(\vec{k}_a, \eta) \prod_{a=1}^{n} \mathcal{K}_{\Delta_a}(\vec{k}_a, \eta), \tag{202}$$

where $n$ is the number of external fields. This is similar to (196), except for two important differences. First, the vertex factors now depend on (conformal) time. Second, the bulk-to-boundary propagator is no longer the simple exponential (194), but is rather given by

$$\mathcal{K}_\Delta(\vec{k}, \eta) = -\frac{i\pi}{2^{\Delta - \frac{3}{2}} \Gamma[\Delta - \frac{3}{2}]} (-k\eta_\star)^{\Delta - \frac{3}{2}} \left(\frac{\eta}{\eta_\star}\right)^{\frac{3}{2}} H^{(2)}_{\Delta - \frac{3}{2}}(-k\eta), \tag{203}$$

where $H^{(2)}_\nu(x)$ is the Hankel function of the second kind. The propagator has been normalized so that it goes to 1 as $\eta \to \eta_\star \to 0$, where $\eta_\star$ is an infrared regulator. Since the energies of the external lines that are connected to a vertex no longer simply add, the computation of the de Sitter wavefunction is more complicated. However, in the infinite past, the form of the bulk-to-boundary propagator simplifies greatly:

$$\mathcal{K}_\Delta(\vec{k}, \eta) \underset{\eta \to -\infty}{\sim} e^{\frac{i\pi}{2}(\Delta + 1)} \frac{\pi^{\frac{1}{2}}}{2^{\Delta - 2}} \frac{(-k\eta_\star)^{\Delta - 3}}{\Gamma[\Delta - \frac{3}{2}]} \left(-ik\eta + \frac{(\Delta - 1)(\Delta - 2)}{2} + \cdots\right) e^{ik\eta}, \tag{204}$$

where the terms we have dropped are subleading in $(k\eta)^{-1}$. The early-time limit of the integral in (202) then is

$$\Psi_n^{(c)}(\vec{k}_a, E_{\text{tot}}) \simeq i\tilde{V}(\vec{k}_a) \prod_a k_a^{\Delta_a - 2} \int_{-\infty}^{0} d\eta \, \eta^{n+p-4} e^{iE_{\text{tot}}\eta}, \qquad (205)$$

where we have defined the vertex factor with factors of $\eta$ removed, $V = \eta^p \tilde{V}$. For scalar integrals, the parameter $p$ counts the number of derivatives in the vertex (which lead to inverse metric factors). This implies that the wavefunction has a singularity of the form[47]

$$\Psi_n^{(c)}(\vec{k}_a, E_{\text{tot}}) \xrightarrow{E_{\text{tot}} \to 0} \frac{\prod_a k_a^{\Delta_a - 2}}{E_{\text{tot}}^{n+p-3}} A_n^{(c)}(\vec{k}_a), \qquad (206)$$

where $A_n^{(c)}$ is the flat-space scattering amplitude corresponding to the same process. Notice that the universality of the bulk-to-boundary propagator at early times (204) allows us to determine the subleading singularities as $E_{\text{tot}} \to 0$. We will use this information later to reconstruct the wavefunction from singularities alone.

The situation with internal lines is similar—the presence of a total energy singularity follows almost immediately from the flat-space arguments combined with the expansion (204). The only additional ingredient needed to determine the order of the pole is the scaling of the bulk-to-bulk propagator in the far past. In the de Sitter case, this Green's function, for $\eta' < \eta$, takes the form

$$\mathcal{G}_\nu(\vec{k}, \eta, \eta') = \frac{\pi}{4}(\eta\eta')^{\frac{3}{2}} \left[ H_\nu^{(1)}(-k\eta) H_\nu^{(2)}(-k\eta') - \frac{H_\nu^{(1)}(-k\eta_\star)}{H_\nu^{(2)}(-k\eta_\star)} H_\nu^{(2)}(-k\eta) H_\nu^{(2)}(-k\eta') \right], \quad (207)$$

where $\eta_\star$ is a late-time regulator and $H_\nu^{(1,2)}(x)$, with $\nu \equiv \Delta - \frac{3}{2}$, are Hankel functions of the first and second kind. Notice that—exactly as in flat space—the bulk-to-bulk propagator has a non-time-ordered piece required by the Dirichlet boundary conditions. The general form of the propagator is somewhat complicated, but it simplifies substantially at early times to $\eta\eta'$ times the corresponding flat-space propagator. For an additional discussion, see [59].

**An example**

The presence of additional factors of conformal time in the interaction vertices and the propagators leads to a richer singularity structure in cosmological spacetimes than in flat space. As an illustration of this, we consider the total energy singularity of the correlator $\langle T\varphi T\varphi \rangle$. This example will be useful in §6.2.3. There are several possible bulk processes that contribute to this correlation function: a contact interaction, scalar exchange and graviton exchange.
*Contact solution.*—We first consider the contact contribution. This piece arises from bulk interactions with two gravitons and two scalars:

$$h_{\mu\nu}^2 \varphi^2, \quad h_{\mu\nu} h^{\nu\alpha} \partial_\alpha \varphi \partial^\mu \varphi, \quad h_{\mu\nu}^2 (\partial\varphi)^2. \qquad (208)$$

From the bulk perspective, it is natural to compute correlation functions in axial gauge, so that $h_{0\mu} = 0$. In this case, the first two interactions in (208) give similar contributions to the correlator, while the last interaction has two types of terms: those with time derivatives acting on the $\varphi$ lines, and those with only spatial derivatives. In order to isolate the leading total energy singularity, we focus on the interactions without time derivatives.

---

[47]We are dropping a $\Delta$-dependent phase that arises from the phases in (204). This phase will not affect any of our manipulations.

A generic contact interaction leads to a time integral of the following form

$$
\Psi^{(c)}_{T\varphi T\varphi} \xrightarrow{E\to 0} i\tilde{V}_c(\vec{k}_a) \int \mathrm{d}\eta\, e^{iE\eta} (-ik_1\eta + 1)(-ik_3\eta + 1)
$$
$$
= \tilde{V}_c(\vec{k}_a)\left(\frac{2k_1 k_3}{E^3} + \frac{k_{13}}{E^2} + \frac{1}{E}\right),
\tag{209}
$$

where we have dropped the subscript on $E_{\mathrm{tot}}$. The momentum-dependent vertex factor, $\tilde{V}_c(\vec{k}_a)$, contains the polarization information, but is not important for extracting the total energy scaling. We see that the leading total energy singularity scales as $E^{-3}$, as expected on dimensional grounds, and comes with a specific set of subleading poles. It is important to emphasize that these are not necessarily the *only* total energy singularities, but rather they are just the ones associated with the leading singularity. Indeed, the interaction with time derivatives acting on $\varphi$ will lead to contributions that scale as $E^{-2}$ and are not tied to the leading $E^{-3}$ singularity.

*Scalar exchange.*—Next, we discuss the possible exchange contributions. Depending on the type of exchange channel, either a scalar or a graviton propagates on the intermediate line, and the two cases are qualitatively different. We first consider the case of scalar exchange in the $s$-channel. In this case, the bulk-to-bulk propagator is (207), with $\nu = 1/2$—which is just $\eta\eta'$ times (195)—so that the relevant time integrals take the form

$$
\Psi^{(s)}_{T\varphi T\varphi} \simeq i\tilde{V}_s(\vec{k}_a) \int \mathrm{d}\eta\, \mathrm{d}\eta'\, e^{i(k_{12}\pm s)\eta} e^{i(k_{34}\mp s)\eta'} (-ik_1\eta + 1)(-ik_3\eta' + 1),
\tag{210}
$$

where $\tilde{V}_s$ is built from the flat-space Feynman rules and the signs in the exponentials depend on the precise time ordering. At very early times, the time ordering of the integrals is immaterial, so we can define a "center of time" coordinate $\bar{\eta} \equiv \eta + \eta'$ and a corresponding difference of times (which we set to zero at leading order). Performing the integral over $\bar{\eta}$, we get

$$
\Psi^{(s)}_{T\varphi T\varphi} \xrightarrow{E\to 0} i\tilde{V}_s(\vec{k}_a) \int \mathrm{d}\bar{\eta}\, e^{iE\bar{\eta}} (-ik_1\bar{\eta} + 1)(-ik_3\bar{\eta} + 1)
$$
$$
\propto \tilde{V}_s(\vec{k}_a)\left(\frac{2k_1 k_3}{E^3} + \frac{k_{13}}{E^2} + \frac{1}{E}\right).
\tag{211}
$$

As for the contact solution, this singularity structure is only that associated with the *leading* total energy singularity, and there can be additional poles in the full answer arising from the details of the nested time-ordered integrals. We have neglected these subtleties, so that this argument only tells us that we should expect the subleading singularities in (209) to multiply the same polarization structures as the $E^{-3}$ pole.

*Graviton exchange.*—The case of graviton exchange in the $u$-channel is more complicated. Since the operator being exchanged has $\Delta = 3$, the bulk-to-bulk propagator (207) contains subleading terms in the early-time limit. These terms complicate the estimation of the subleading total energy singularities. In particular, the approximation we made in the $s$-channel of integrating over a center of time coordinate is no longer reliable and we must do the bulk time integral exactly. However, our goal is not to do the full bulk computation, but rather to extract some information about the singularity structure to aid the later construction of the full answer (see §6.2). We therefore focus on the highest helicity components of the exchange. In particular, we can make the simplifying assumption that both the interaction vertices and the bulk-to-bulk propagator are transverse and traceless. Specifically, we combine the scalar-scalar-graviton vertex

$$
V^{ij}_{\gamma\varphi\varphi} = \frac{1}{2\eta^2}\beta^i_u \beta^j_u,
\tag{212}
$$

with the graviton three-point vertex

$$V_{\gamma\gamma\gamma}^{kl} = \frac{1}{\eta^2}(\vec{\xi}_1 \cdot \vec{\xi}_3)^2 \alpha_u^k \alpha_u^l + \cdots, \tag{213}$$

where we have restricted both vertices to their transverse-traceless components, and have only kept track of the highest helicity component of the graviton self-interaction.

The bulk time integral of interest connects these vertices with the transverse-traceless part of the graviton bulk-to-bulk propagator, which consists of $(\Pi_{2,2})_{kl}^{ij}$ multiplied by (207), with $\nu = 3/2$. Putting all of these pieces together, the time integral of interest takes the form

$$\Psi_{T\varphi T\varphi}^{(u)} \simeq \left(u^4 \Pi_{2,2}^{(u)}(\vec{\xi}_1 \cdot \vec{\xi}_3)^2 + \cdots\right) \int \frac{d\eta d\eta'}{\eta^2}(1 - ik_1\eta)(1 - ik_3\eta)e^{ik_{12}\eta} \mathcal{G}_{\frac{3}{2}}(\vec{u}, \eta, \eta') e^{ik_{34}\eta'}, \tag{214}$$

where we have written explicitly only the terms proportional to the helicity-2 polarization sum $\Pi_{2,2}^{(u)}$. Doing this integral, we find the terms multiplying this leading polarization sum[48]

$$\Psi_{T\varphi T\varphi}^{(u)} \supset u^4 \Pi_{2,2}^{(u)}(\vec{\xi}_1 \cdot \vec{\xi}_3)^2 \left[ \frac{1}{(k_{13}+u)(k_{24}+u)}\left(\frac{2k_1k_3}{E^3} + \frac{k_{13}}{E^2}\right) \right.$$
$$\left. + \frac{u}{(k_{13}+u)^2(k_{24}+u)^2}\left(\frac{2k_1k_3}{E^2} + \frac{k_{13}+u}{E}\right) \right]. \tag{215}$$

The total energy singularities naturally separate into two distinct pieces shown in the first and second line. We see that the leading $E^{-3}$ singularity is now coupled to a $E^{-2}$ singularity, but no $E^{-1}$ singularity. We will see later in (295) that the final answer indeed splits in this way. Notice that in (215) we have incomplete information about the final $u$-channel correlator, but this is enough to understand how the leading total energy pole appears, along with the specific subleading singularities required by conformal invariance. In the following, we will see how to combine this information with information about partial energy singularities to reconstruct the full answer.

### 6.1.3 Partial Energy Singularities

In addition to the total energy singularity highlighted in the previous section, the wavefunction also has "partial energy" singularities when the total energy of a subgraph vanishes. These partial energy singularities are signatures of particle exchange—on these singularities the wavefunction factorizes and can be written in terms of lower-point objects. This provides a consistency constraint on the structure of correlation functions that is similar to the factorization of the tree-level $S$-matrix when an intermediate particle goes on-shell.

**Flat-space correlators**

It is helpful to first consider the partial energy singularities of the flat-space wavefunction. The simplest type of partial energy singularity (and indeed the only type we will need in the following) occurs when the energy of a subgraph connected by a single internal line to rest of the diagram is conserved. On this singularity, the wavefunction factorizes

$$\xrightarrow{E_1 + y_{12} \to 0} \quad \frac{A_{(1)} \times \widetilde{\Psi}_{(2)}}{E_1 + y_{12}}, \tag{216}$$

---

[48]This requires integrating over both time orderings of the bulk-to-bulk propagator.

where $A_{(1)}$ is the flat-space amplitude associated to the left subgraph (including the internal line) and $\widetilde{\Psi}_{(2)}$ is a "shifted wavefunction" defined by

$$\widetilde{\Psi}_{(2)}(E_2, y_{12}) \equiv \frac{1}{2y_{12}}\Big(\Psi_{(2)}(E_2 - y_{12}) - \Psi_{(2)}(E_2 + y_{12})\Big). \tag{217}$$

This factorization phenomenon can be understood intuitively by examining the bulk-to-bulk propagator. Recall that the physical reason for divergences in the wavefunction is that the integration limits in the various time integrals are running off to $-\infty$, and these integrals are unsuppressed when some subset of energies are conserved. In this case, the divergence at $E_1 + y_{12} = 0$ is coming from sending $t_1 \to -\infty$. In this limit, the bulk-to-bulk propagator takes the form

$$\mathcal{G}(y_{12}; t_1 \to -\infty, t_2) = \frac{e^{iy_{12}t_1}}{2y_{12}}\Big(e^{-iy_{12}t_2} - e^{iy_{12}t_2}\Big). \tag{218}$$

The theta function has removed one of the terms because $t_1 < t_2$ in this regime. It is then easy to see that the effect of the bulk-to-bulk propagator is to shift the energy of the vertex 2 by $\pm y_{12}$, leading to the shifted wavefunction (217). On the other side of the diagram, the total energy flowing into the vertex 1 vanishes, so the residue of this singularity is the corresponding scattering amplitude, by the argument in the previous section.

**De Sitter correlators**

Much like for the total energy singularity, the essential behavior of the partial energy singularities is the same in cosmological backgrounds as in flat space. When the energy of a subgraph adds up to zero, the wavefunction diverges and the coefficient is a product of a scattering amplitude and a shifted wavefunction coefficient. We restrict our attention to cases where the singularities are poles or logarithms, but see [54] for some examples with more general singularity structure.

In what follows, we will be interested exclusively in the singularities of the four-point function in de Sitter space, so we list the properties of these partial energy singularities explicitly. The four-point function has two possible partial energy singularities, when the energies in either the left vertex or the right vertex add up to zero.

Consider tree-level exchange in the $s$-channel. When $E_L \equiv k_1 + k_2 + s \to 0$, the four-point function factorizes as

$$\xrightarrow{\;E_L = k_{12} + s \to 0\;} \frac{\widetilde{A}_{(L)} \times \widetilde{\Psi}_{(R)}}{E_L^{p_L}}, \tag{219}$$

where the parameter $p_L$ controlling the order of the pole can be fixed by dimensional analysis. Associated with the left vertex is the "dressed" three-point scattering amplitude

$$\widetilde{A}_{(L)} \equiv k_1^{\Delta_1 - 2} k_2^{\Delta_2 - 2} s^{\Delta_I - 2} A_{(L)}, \tag{220}$$

where the factors of $k^\Delta$ appear because of the expansion of the mode functions near $\eta \to -\infty$; cf. (204). Taking the $\eta' \to -\infty$ limit in (207), we find that associated with the right vertex is the following "shifted wavefunction"

$$\widetilde{\Psi}_{(R)}(s, k_3, k_4) \equiv \frac{(-1)^{\Delta_I}}{2s^{2\Delta_I - 3}}\Big(\Psi_{(R)}(-s, k_3, k_4) - \Psi_{(R)}(s, k_3, k_4)\Big), \tag{221}$$

where the overall factor of energy is fixed by the dimension $\Delta_I$ of the exchanged field.

By exchange symmetry, the four-point function of course also has a singularity when $E_R \equiv k_3 + k_4 + s \to 0$, where it factorizes into a shifted correlator on the left and a scattering amplitude on the right. All of the exchange correlators we have computed via weight-shifting have these partial energy singularities, $E_{L,R} \to 0$, and factorize in this characteristic way in their vicinity. In the next section, we will invert the logic and try to reconstruct the full correlators from their known singularity structure.

## 6.2 Correlators from Consistent Factorization

We would like to determine to what degree the four-point functions of interest are constrained by consistent factorization in the limits $E_{L,R} \to 0$. The challenge is to find solutions at general kinematics that reduce to these limits. As we will see, naive extrapolations away from the factorization channels often lead to spurious folded singularities. Requiring these to be absent then leads to the physically expected singularity at $E \to 0$, and in some cases is stringent enough to reconstruct the full correlator uniquely. In other cases, we will need to impose additional constraints to fix subleading poles.

We will use the following notation for the partial energies in the $s$, $t$, and $u$-channels:

$$s\text{-channel:} \quad E_L \equiv k_{12} + s\,, \quad E_R \equiv k_{34} + s\,, \tag{222}$$

$$t\text{-channel:} \quad E_L^{(t)} \equiv k_{14} + t\,, \quad E_R^{(t)} \equiv k_{23} + t\,, \tag{223}$$

$$u\text{-channel:} \quad E_L^{(u)} \equiv k_{13} + u\,, \quad E_R^{(u)} \equiv k_{24} + u\,, \tag{224}$$

i.e. quantities without the superscript are defined in the $s$-channel. Throughout this subsection, we will construct the kinematic parts of correlators from factorization. That is, we set all of the three-point couplings to unity. One advantage of the factorization approach is that the overall normalization of correlators comes out correct, and it will only be necessary to constrain the relative normalizations. We will re-introduce these coupling constants in §6.3 when we discuss how the sum of all channels is constrained by the total energy singularity.

### 6.2.1 A Few Instructive Examples

To illustrate the basic logic, we begin with a few simple examples. After that, we will show how this approach can be applied to the correlators studied in Section 5.

**Massless scalar in flat space** Consider the four-point function of a massless scalar field $\varphi$ in flat space. In the limits $E_{L,R} \to 0$, the $s$-channel must factorize into a product of three-point amplitude and a (shifted) three-point correlator.[49] For a $\varphi^3$ interaction, the three-point wavefunction is $\Psi_{\varphi\varphi\varphi} = 1/K$, so that the relevant building blocks are

$$\widetilde{\Psi}_{\varphi\varphi\varphi}(k_1, k_2, s) = \frac{1}{(k_{12} + s)(k_{12} - s)}\,,$$
$$A_{\varphi\varphi\varphi}(k_1, k_2, s) = 1\,. \tag{225}$$

Using these expressions, the four-point function on each of the poles is[50]

$$\langle \varphi\varphi\varphi\varphi \rangle_s \xrightarrow{E_L \to 0} \frac{1}{E_L} A_{\varphi\varphi\varphi} \cdot \widetilde{\Psi}_{\varphi\varphi\varphi} = \frac{1}{E_L} \frac{1}{E_R(k_{34} - s)}\,, \tag{226}$$

$$\langle \varphi\varphi\varphi\varphi \rangle_s \xrightarrow{E_R \to 0} \frac{1}{E_R} \widetilde{\Psi}_{\varphi\varphi\varphi} \cdot A_{\varphi\varphi\varphi} = \frac{1}{E_R} \frac{1}{E_L(k_{12} - s)}\,. \tag{227}$$

---

[49]See footnote 2.
[50]Notice that we use $\Psi_{OOOO}$ and $\langle OOOO \rangle$ interchangeably, cf. (7).

The goal is to find an object at general kinematics that reproduces these two factorization limits. This is a constrained problem because the residue of the singularity at $E_L = 0$ has a singularity as $E_R \to 0$, and vice versa, so we cannot just add (226) and (227) together and treat the result for general momenta. Notice, however, that the factors of $(k_{12} - s)$ and $(k_{34} - s)$ can both be interpreted as the total energy $E = k_{12} + k_{34}$ evaluated in the limits $E_R \to 0$ and $E_L \to 0$. Consequently, there is a unique object that factorizes correctly in both limits:

$$\langle \varphi \varphi \varphi \varphi \rangle_s = \frac{1}{E E_L E_R} \, . \tag{228}$$

We see that the constraint of consistent factorization has forced us to introduce an additional physical singularity at $E = 0$. As expected, the residue of this singularity is the flat-space scattering amplitude:

$$\langle \varphi \varphi \varphi \varphi \rangle_s \xrightarrow{E \to 0} \frac{1}{E} A^{(s)}_{\varphi \varphi \varphi \varphi} \, , \tag{229}$$

where we used that $E_L E_R = -S$ in the limit $E \to 0$.

**Photon exchange**   A slightly more nontrivial example is the four-point function of conformally coupled scalars arising from the exchange of a massless vector (e.g. the photon). Since the interactions are conformally invariant, this example is the same in flat space and de Sitter space. The relevant three-point data are

$$\widetilde{\Psi}_{\varphi \varphi J}(k_1, k_2, s) = \frac{\vec{\alpha} \cdot \vec{\xi}_s}{(k_{12} + s)(k_{12} - s)} \, , \tag{230}$$

$$A_{\varphi \varphi J}(k_1, k_2, s) = \vec{\alpha} \cdot \vec{\xi}_s \, , \tag{231}$$

where $\vec{\alpha} \equiv \vec{k}_1 - \vec{k}_2$ and $\vec{\xi}_s$ is the polarization vector associated with the exchanged photon. In the factorization limits, the four-point function must become

$$\langle \varphi \varphi \varphi \varphi \rangle_{s,J} \xrightarrow{E_L \to 0} \frac{A_{\varphi \varphi J} \otimes \widetilde{\Psi}_{J \varphi \varphi}}{E_L} = \frac{1}{E_L} \frac{s^2 \Pi_{1,1}}{E_R (k_{34} - s)} \, , \tag{232}$$

$$\langle \varphi \varphi \varphi \varphi \rangle_{s,J} \xrightarrow{E_R \to 0} \frac{\widetilde{\Psi}_{\varphi \varphi J} \otimes A_{J \varphi \varphi}}{E_R} = \frac{1}{E_R} \frac{s^2 \Pi_{1,1}}{E_L (k_{12} - s)} \, , \tag{233}$$

where the symbol $\otimes$ denotes a sum over helicities, which in practice we implement by contracting with the transverse-traceless projector $\pi_{ij}$, and we have defined the polarization sum $s^2 \Pi_{1,1} \equiv \alpha^i \pi_{ij} \beta^j$, with $\vec{\beta} \equiv \vec{k}_3 - \vec{k}_4$. As in the previous example, consistent factorization requires us to replace $k_{12} - s$ and $k_{34} - s$ by the total energy $E$. This generates a singularity at $E = 0$. The residue of this singularity has the correct $1/S$ scaling, but does not have the correct angular dependence. Indeed, in the limit $E \to 0$, we expect

$$\langle \varphi \varphi \varphi \varphi \rangle_{s,J} \xrightarrow{E \to 0} \frac{1}{E} A^{(s)}_{\varphi \varphi \varphi \varphi} = \frac{1}{E} P_1 \left( 1 + \frac{2U}{S} \right) \, . \tag{234}$$

A solution that satisfies all of the above limits is

$$\langle \varphi \varphi \varphi \varphi \rangle_{s,J} = \frac{\mathcal{P}_1}{E E_L E_R} \, , \tag{235}$$

where we have introduced

$$\mathcal{P}_1 \equiv s^2 \Pi_{1,1} - E_L E_R \Pi_{1,0} \xrightarrow{E \to 0} -S \, P_1 \left( 1 + \frac{2U}{S} \right) \, , \tag{236}$$

with $\Pi_{1,0} \equiv \hat{\alpha}\hat{\beta}$. Notice that the longitudinal piece, proportional to $\Pi_{1,0}$, is not fixed by the factorization limits (232) and (233), but is required in order for the flat-space limit (234) to have the correct angular structure.

**Graviton exchange**  Another illustrative example is the four-point function of conformally coupled scalars arising from the exchange of a massless spin-2 particle (i.e. the graviton). In this case, the relevant three-point data are

$$\widetilde{\Psi}_{\varphi\varphi T}(k_1, k_2, s) = \frac{\left(\vec{\alpha} \cdot \vec{\xi}_s\right)^2}{(k_{12} + s)^2 (k_{12} - s)^2}, \tag{237}$$

$$\widetilde{A}_{\varphi\varphi T}(k_1, k_2, s) = \frac{s}{2} \left(\vec{\alpha} \cdot \vec{\xi}_s\right)^2. \tag{238}$$

Notice that we have included an additional factor of $s$ in the scattering amplitude relative to the usual flat-space expression. This accounts for the fact that the stress tensor has conformal dimension $\Delta = 3$; cf. (220). The four-point function then factorizes as

$$\langle \varphi\varphi\varphi\varphi \rangle_{s,T} \xrightarrow{E_L \to 0} \frac{\widetilde{A}_{\varphi\varphi T} \otimes \widetilde{\Psi}_{T\varphi\varphi}}{E_L^2} = \frac{1}{3} \frac{s^5}{E_L^2 E_R^2 (k_{34} - s)^2} \Pi_{2,2}, \tag{239}$$

$$\langle \varphi\varphi\varphi\varphi \rangle_{s,T} \xrightarrow{E_R \to 0} \frac{\widetilde{\Psi}_{\varphi\varphi T} \otimes \widetilde{A}_{T\varphi\varphi}}{E_R^2} = \frac{1}{3} \frac{s^5}{E_R^2 E_L^2 (k_{12} - s)^2} \Pi_{2,2}, \tag{240}$$

where the orders of the poles are fixed by the scaling symmetry. We have summed over internal helicities, using the spin-2 polarization tensor $(\Pi_{2,2})^{ij}{}_{lm}$ (see Appendix E), and defined the scalar polarization sum

$$\frac{2s^4}{3} \Pi_{2,2} = \alpha_i \alpha_j (\Pi_{2,2})^{ij}{}_{lm} \beta^l \beta^m. \tag{241}$$

We want to find an expression at general kinematics that reduces to these expressions around each of the singularities. A function that correctly reproduces the factorization limits (239) and (240) is

$$\langle \varphi\varphi\varphi\varphi \rangle_{s,T} \stackrel{?}{=} \frac{1}{3} \frac{s^5}{E^2 E_L^2 E_R^2} \Pi_{2,2}. \tag{242}$$

However, this expression does not have the correct scaling and angular dependence in the limit $E \to 0$. The expected flat-space limit is

$$\langle \varphi\varphi\varphi\varphi \rangle_{s,T} \xrightarrow{E \to 0} \frac{1}{E^3} A^{(s)}_{\varphi\varphi\varphi\varphi} = -\frac{1}{E^3} \frac{S}{3} P_2 \left(1 + \frac{2U}{S}\right). \tag{243}$$

A function that also incorporates this limit is

$$\langle \varphi\varphi\varphi\varphi \rangle_{s,T} = \frac{1}{3} \frac{s\mathcal{Q}_2}{E^2 E_L^2 E_R^2} + \frac{1}{3} \frac{\mathcal{P}_2}{E^3 E_L E_R}, \tag{244}$$

where we have introduced the functions

$$\mathcal{Q}_2 \equiv s^4 \Pi_{2,2} - E_L^2 E_R^2 \Pi_{2,0}, \tag{245}$$

$$\mathcal{P}_2 \equiv s^4 \Pi_{2,2} - E_L E_R s^2 \Pi_{2,1} + E_L^2 E_R^2 \Pi_{2,0}. \tag{246}$$

In the flat-space limit, the combination $\mathcal{P}_2$ reproduces the expected angular structure of the amplitude:

$$\mathcal{P}_2 \xrightarrow{E \to 0} S^2 P_2 \left(1 + \frac{2U}{S}\right). \tag{247}$$

The second term in $\mathcal{Q}_2$ is not fixed by the factorization limit, but is needed in order for the answer to be conformally invariant.

### 6.2.2  Correlators with Spin-1 Currents

After these warmup examples, we now apply the factorization argument to the correlators studied in Section 5. We begin with the correlators involving spin-1 currents. We will find that all poles are of sufficiently low order, so that the answers are completely fixed by the factorization limits and the total energy singularity.

**Single photon correlator**

As a first example, we consider the correlator involving one photon and three conformally coupled scalars, $\langle J\varphi\varphi\varphi\rangle$. Let us first consider this correlation function in flat space. In the factorization limit, we will need the following wavefunction coefficient and three-point amplitude

$$\widetilde{\Psi}_{J\varphi\varphi}(k_1, k_2, s) = \frac{2\,\vec{\xi}_1 \cdot \vec{k}_2}{(k_{12} + s)(k_{12} - s)}, \tag{248}$$

$$A_{J\varphi\varphi}(k_1, k_2, s) = 2\,\vec{\xi}_1 \cdot \vec{k}_2, \tag{249}$$

together with the scalar three-point quantities (225). The two factorization limits of the $s$-channel contribution to $\langle J\varphi\varphi\varphi\rangle$ then are

$$\langle J\varphi\varphi\varphi\rangle_s \xrightarrow{\;E_L \to 0\;} \frac{1}{E_L} A_{J\varphi\varphi} \cdot \widetilde{\Psi}_{\varphi\varphi\varphi} = \frac{2\,\vec{\xi}_1 \cdot \vec{k}_2}{E_L E_R(k_{34} - s)}, \tag{250}$$

$$\langle J\varphi\varphi\varphi\rangle_s \xrightarrow{\;E_R \to 0\;} \frac{1}{E_R} \widetilde{\Psi}_{J\varphi\varphi} \cdot A_{\varphi\varphi\varphi} = \frac{2\,\vec{\xi}_1 \cdot \vec{k}_2}{E_R E_L(k_{12} - s)}. \tag{251}$$

A function with factorizes correctly on both poles is

$$\langle J\varphi\varphi\varphi\rangle_s = \frac{2\,\vec{\xi}_1 \cdot \vec{k}_2}{E E_L E_R}. \tag{252}$$

We again see the appearance of a total energy singularity, whose coefficient is the $s$-channel contribution to the relevant scattering process. However, due to the presence of the polarization vector, the residue this time is *not* Lorentz-invariant, and therefore the $s$-channel contribution by itself is not consistent. To obtain a consistent correlator, we have to add the $t$-channel and impose charge conservation. We will explore this more fully in §6.3.

It is straightforward to repeat the exercise in de Sitter space. The relevant wavefunction coefficients and three-point amplitudes are the same as in (225) and (249), except for

$$\widetilde{\Psi}_{\varphi\varphi\varphi}(s, k_3, k_4) = -\frac{1}{2s} \log\left(\frac{k_{34} + s}{k_{34} - s}\right). \tag{253}$$

The two factorization limits of the $s$-channel contribution then are

$$\langle J\varphi\varphi\varphi\rangle_s \xrightarrow{\;E_L \to 0\;} \frac{1}{E_L} A_{J\varphi\varphi} \cdot \widetilde{\Psi}_{\varphi\varphi\varphi} = -\vec{\xi}_1 \cdot \vec{k}_2 \frac{1}{E_L} \frac{1}{s} \log\left(\frac{E_R}{k_{34} - s}\right), \tag{254}$$

$$\langle J\varphi\varphi\varphi\rangle_s \xrightarrow{\;E_R \to 0\;} \log(E_R/\mu)\, \widetilde{\Psi}_{J\varphi\varphi} \cdot A_{\varphi\varphi\varphi} = 2\,\vec{\xi}_1 \cdot \vec{k}_2 \frac{\log(E_R/\mu)}{E_L(k_{12} - s)}, \tag{255}$$

where the fact that the singularity at $E_R = 0$ is logarithmic follows from dimensional analysis. As before, the goal is to find an object at general kinematics that reproduces the two factorization channels (254) and (255). An object with the correct factorization limits is

$$\langle J\varphi\varphi\varphi\rangle_s \overset{?}{=} \frac{2\,\vec{\xi}_1 \cdot \vec{k}_2}{E_L(k_{12} - s)} \log\left(\frac{E_R}{k_{34} - s}\right). \tag{256}$$

However, although this function factorizes correctly for $E_L, E_R \to 0$, it has an additional folded singularity as $k_{34} \to s$, whose residue does not have any physical interpretation. This unwanted feature is avoided if we send $k_{34} - s \to E$ in the ansatz (256). A function with the correct factorization limits, and without folded singularities, therefore is[51]

$$\langle J\varphi\varphi\varphi\rangle_s = \frac{2\,\vec{\xi}_1 \cdot \vec{k}_2}{E_L(k_{12}-s)} \log\left(\frac{E_R}{E}\right), \tag{257}$$

which is the same as the result (131) obtained by weight-shifting. Note that the apparent singularity at $k_{12} = s$ is cancelled by the $\log(E_R)$ factor in the numerator. Again, the solution has a singularity at $E = 0$ describing the flat-space limit. As before, this limit of the $s$-channel contribution is not Lorentz-invariant and demands the addition of the $t$-channel.

**Compton correlator**

Next, we consider the correlator $\langle J\varphi J\varphi\rangle$. The $s$-channel contribution has the following factorization limits:

$$\langle J\varphi J\varphi\rangle_s \xrightarrow{E_L \to 0} \frac{1}{E_L} A_{J\varphi\varphi} \cdot \widetilde{\Psi}_{\varphi J\varphi} = (\vec{\xi}_1 \cdot \vec{k}_2)(\vec{\xi}_3 \cdot \vec{k}_4) \frac{4}{E_L E_R(k_{34}-s)}, \tag{258}$$

$$\langle J\varphi J\varphi\rangle_s \xrightarrow{E_R \to 0} \frac{1}{E_R} \widetilde{\Psi}_{J\varphi\varphi} \cdot A_{\varphi J\varphi} = (\vec{\xi}_1 \cdot \vec{k}_2)(\vec{\xi}_3 \cdot \vec{k}_4) \frac{4}{E_R E_L(k_{12}-s)}, \tag{259}$$

where the relevant wavefunction coefficients and three-point amplitudes were given in (249). An expression that factorizes correctly on both poles is

$$\langle J\varphi J\varphi\rangle_s = (\vec{\xi}_1 \cdot \vec{k}_2)(\vec{\xi}_3 \cdot \vec{k}_4) \frac{4}{EE_L E_R}, \tag{260}$$

which matches the weight-shifting result (140). As with the single-current correlators, this $s$-channel Compton correlator is not consistent by itself. As we will describe in §6.3, we must add the $t$-channel permutation of this correlator, as well as a contact contribution, in order for the limit $E \to 0$ to be Lorentz-invariant.

Allowing for non-Abelian interactions, it becomes possible to exchange a vector operator in the $u$-channel. To obtain this from factorization, we require the following additional three-point data:

$$A_{JJJ}(k_1, k_3, u) = (\vec{\alpha}_u \cdot \vec{\xi}_u)\vec{\xi}_1 \cdot \vec{\xi}_3 + 2(\vec{k}_3 \cdot \vec{\xi}_1)(\vec{\xi}_3 \cdot \vec{\xi}_u) - 2(\vec{k}_1 \cdot \vec{\xi}_3)(\vec{\xi}_u \cdot \vec{\xi}_1), \tag{261}$$

$$\widetilde{\Psi}_{JJJ}(k_1, k_3, u) = \frac{1}{(k_{13}+u)(k_{13}-u)} A_{JJJ}, \tag{262}$$

where $\vec{\alpha}_u = \vec{k}_1 - \vec{k}_3$ and $\vec{\xi}_u$ is the polarization vector of the exchanged field in the $u$-channel. Using these expressions, together with permutations of (230) and (231), the factorization

---

[51]It is interesting that we have to explicitly forbid the presence of folded singularities in de Sitter space, but not in flat space. This is a manifestation of the fact that there are many de Sitter-invariant vacua, so we must impose an additional condition to uniquely select the Bunch–Davies state. In contrast, there is a unique vacuum in flat space, so folded singularities are automatically absent.

limits in the $u$-channel become

$$\langle J\varphi J\varphi\rangle_u \xrightarrow{E_L^{(u)}\to 0} \frac{1}{E_L^{(u)}} A_{JJJ}\otimes\widetilde{\Psi}_{J\varphi\varphi} = \frac{u^2\Pi_{1,1}^{(u)}\vec{\xi}_1\cdot\vec{\xi}_3 + 2u^2\vec{\xi}_1\circ\vec{\xi}_3}{E_L^{(u)}E_R^{(u)}(k_{24}-u)}, \tag{263}$$

$$\langle J\varphi J\varphi\rangle_u \xrightarrow{E_R^{(u)}\to 0} \frac{1}{E_R^{(u)}} \widetilde{\Psi}_{JJJ}\otimes A_{J\varphi\varphi} = \frac{u^2\Pi_{1,1}^{(u)}\vec{\xi}_1\cdot\vec{\xi}_3 + 2u^2\vec{\xi}_1\circ\vec{\xi}_3}{E_R^{(u)}E_L^{(u)}(k_{13}-u)}, \tag{264}$$

where we have used the shorthand notation introduced in (152):

$$\vec{\xi}_1\circ\vec{\xi}_3 = \frac{2}{u^2}\Big[(\vec{\xi}_1\cdot\vec{k}_2)(\vec{\xi}_3\cdot\vec{k}_4)-(\vec{\xi}_1\cdot\vec{k}_4)(\vec{\xi}_3\cdot\vec{k}_2)\Big]. \tag{265}$$

Again, a naive extrapolation of (263) and (264) would have folded singularities at $k_{13}=u$ and $k_{24}=u$. As before, these are avoided by sending $k_{13}-u\to E$ and $k_{24}-u\to E$. An expression that factorizes correctly, and has no folded singularities, therefore is

$$\langle J\varphi J\varphi\rangle_u = \frac{1}{EE_L^{(u)}E_R^{(u)}}\Big[\mathcal{P}_1^{(u)}\vec{\xi}_1\cdot\vec{\xi}_3 + 2u^2\vec{\xi}_1\circ\vec{\xi}_3\Big], \tag{266}$$

where $\mathcal{P}_1^{(u)}$ is the same function as in (236) (adapted to the $u$-channel):

$$\mathcal{P}_1^{(u)} = u^2\Pi_{1,1}^{(u)} - E_L^{(u)}E_R^{(u)}\Pi_{1,0}^{(u)} \xrightarrow{E\to 0} -U\left(1+\frac{2S}{U}\right). \tag{267}$$

The result in (266) is the same as the result (154) obtained by weight-shifting. As before, the longitudinal component of the polarization sum was introduced in (267), so that the $E\to 0$ singularity has the correct angular structure. Consistency of the flat-space limit $E\to 0$, furthermore, requires the addition of the contact solution (144).

**Yang–Mills correlator**

As a final example involving spin-1 currents, we show how to construct the Yang–Mills four-point function $\langle JJJJ\rangle$ from its factorization singularities. It is at this point that we begin to reap the full rewards of the factorization approach. From §5.2.2 and §G.1, it is clear that constructing even the non-Abelian Compton correlator by weight-shifting is a rather intricate task. The Yang–Mills correlator is even more complicated, and attempting to go beyond four points using these techniques would seem to be infeasible. We therefore need a simpler approach. It turns out that factorization and gluing provide such an approach and make the construction of this correlator remarkably straightforward.

The relevant building blocks are the amplitude and shifted correlator in (261) and (262). Using these, we obtain the following factorization limits of the $s$-channel correlator

$$\langle JJJJ\rangle_s \xrightarrow{E_L\to 0} \frac{1}{E_L} A_{JJJ}\otimes\widetilde{\Psi}_{JJJ} = \frac{Z_{JJJJ}^{(s)}}{E_L E_R(k_{12}-s)}, \tag{268}$$

$$\langle JJJJ\rangle_s \xrightarrow{E_R\to 0} \frac{1}{E_R} \widetilde{\Psi}_{JJJ}\otimes A_{JJJ} = \frac{Z_{JJJJ}^{(s)}}{E_R E_L(k_{34}-s)}, \tag{269}$$

where we have defined

$$Z_{JJJJ}^{(s)} = (\vec{\xi}_1\cdot\vec{\xi}_2)(\vec{\xi}_3\cdot\vec{\xi}_4)s^2\Pi_{1,1} + 2s^2(\vec{\xi}_1\cdot\vec{\xi}_2)(\vec{\xi}_3\circ\vec{\xi}_4) + 2s^2(\vec{\xi}_3\cdot\vec{\xi}_4)(\vec{\xi}_1\circ\vec{\xi}_2) + \tilde{Z}^{(s)}, \tag{270}$$

$$\tilde{Z}^{(s)} = 4\big[(\vec{\xi}_1\cdot\vec{k}_2)\vec{\xi}_2-(\vec{\xi}_2\cdot\vec{k}_1)\vec{\xi}_1\big]\cdot\big[(\vec{\xi}_3\cdot\vec{k}_4)\vec{\xi}_4-(\vec{\xi}_4\cdot\vec{k}_3)\vec{\xi}_3\big]. \tag{271}$$

As in the previous examples, we must include the longitudinal part of the polarization sum to get the correct limit as $E \to 0$; in other words, in (270) we make the replacement

$$s^2 \Pi_{1,1} \to \mathcal{P}_1 \equiv s^2 \Pi_{1,1} - E_L E_R \Pi_{1,0}. \tag{272}$$

Since the numerator factor is the same on both poles, a correlator that factorizes correctly on all of the $s$-channel singularities is then given by

$$\langle JJJJ \rangle_s = \frac{Z^{(s)}_{JJJJ}}{E E_L E_R}, \tag{273}$$

which agrees with the direct computation in [112]. The $t$- and $u$-channel contributions are simply permutations of the $s$-channel result.

### 6.2.3 Correlators with Spin-2 Currents

Next, we consider correlators involving the stress tensor and build them from their singularities. The singularities will generally have higher powers of energy compared to the spin-1 counterparts, rendering a full reconstruction more challenging. Nonetheless, we show below that almost the entire structure of the four-point correlators in each channel can be fixed from those singularities (and the absence of further unphysical singularities).

**Single graviton correlator**

To construct the correlator $\langle T \varphi \varphi \varphi \rangle$, we require various building blocks: the scalar three-point amplitude (which is just a constant), the shifted wavefunction (253), as well as

$$\widetilde{\Psi}_{T\varphi\varphi}(k_1, k_2, s) = 2(\vec{\xi}_1 \cdot \vec{k}_2)^2 \frac{k_{12}^2 + 2k_{12}k_1 - s^2}{(k_{12} + s)^2 (k_{12} - s)^2}, \tag{274}$$

and the scattering amplitude (238). Notice that (274) is different from (237), because it is shifted with respect to one of the scalar legs as opposed to the leg associated with the spin-2 current. Putting these pieces together, we obtain the following factorization limits

$$\langle T \varphi \varphi \varphi \rangle_s \xrightarrow{E_L \to 0} \frac{1}{E_L^2} \widetilde{A}_{T\varphi\varphi} \cdot \widetilde{\Psi}_{\varphi\varphi\varphi} = -(\vec{\xi}_1 \cdot \vec{k}_2)^2 \frac{k_1}{E_L^2} \frac{1}{s} \log\left(\frac{E_R}{k_{34} - s}\right), \tag{275}$$

$$\langle T \varphi \varphi \varphi \rangle_s \xrightarrow{E_R \to 0} \log(E_R/\mu) \widetilde{\Psi}_{T\varphi\varphi} \cdot A_{\varphi\varphi\varphi} = 2(\vec{\xi}_1 \cdot \vec{k}_2)^2 \frac{k_{12}^2 + 2k_{12}k_1 - s^2}{E_L^2 (k_{12} - s)^2} \log(E_R/\mu), \tag{276}$$

where the orders of the left and right singularities are fixed by dimensional analysis.

As before, the goal is to find an expression that has both of these factorization singularities, but no additional unphysical singularities in folded configurations. We avoid the folded singularity at $k_{34} = s$ by interpreting the factor of $(k_{34} - s)$ in (275) as $E$. A naive extrapolation away from (276) would then give

$$\langle T \varphi \varphi \varphi \rangle_s \overset{?}{=} 2(\vec{\xi}_1 \cdot \vec{k}_2)^2 \frac{k_{12}^2 + 2k_{12}k_1 - s^2}{E_L^2 (k_{12} - s)^2} \log\left(\frac{E_R}{E}\right) \tag{277}$$

$$\xrightarrow{k_{12} \to s} (\vec{\xi}_1 \cdot \vec{k}_2)^2 \frac{k_1}{s E} \frac{1}{(k_{12} - s)}. \tag{278}$$

The expression in (277) correctly reproduces both factorization channels, but as shown in (278), it still has a folded singularity at $k_{12} = s$. Note that this cannot be cured by interpreting the factor of $(k_{12} - s)^2$ as $E^2$, because the expression would not reproduce the left factorization limit (275) correctly. In order to cancel this singularity, we must instead add an extra term to (277), with the result

$$\langle T\varphi\varphi\varphi\rangle_s = 2(\vec{\xi}_1 \cdot \vec{k}_2)^2 \left[ \frac{k_{12}^2 + 2k_{12}k_1 - s^2}{E_L^2(k_{12}-s)^2} \log\left(\frac{E_R}{E}\right) + \frac{k_1}{EE_L(k_{12}-s)} \right]. \qquad (279)$$

This is indeed consistent with the previous result (169) obtained by weight-shifting. Note that for $E_L \to 0$, the second term is subleading and the first term becomes (275). The correlator (279) is therefore the unique function that factorizes correctly and has no folded singularities.

Requiring the absence of folded singularities has achieved two things: First, it completely fixed the correlator in a nontrivial way. Second, the extra term that we were forced to add in (279) contains the leading singularity in the limit $E \to 0$:

$$\langle T\varphi\varphi\varphi\rangle_s \xrightarrow{E\to 0} -\frac{k_1}{E}\frac{2(\vec{\xi}_1 \cdot \vec{k}_2)^2}{S} = \frac{k_1}{E}A_{T\varphi\varphi\varphi}^{(s)}. \qquad (280)$$

As expected, the coefficient of this total energy singularity is precisely the flat-space scattering amplitude, but it arises in an interesting way in this example.

**Gravitational Compton correlator**

Next, we consider the correlator $\langle T\varphi T\varphi\rangle$ corresponding to gravitational Compton scattering (see Fig. 9). This correlator splits into two distinct contributions: $s$- and $t$-channels from scalar exchange and a $u$-channel from graviton exchange. We will now show how each of these contributions can be derived from consistent factorization.

*Scalar exchange.*—For the $s$-channel contribution, the relevant factorization limits are

$$\langle T\varphi T\varphi\rangle_s \xrightarrow{E_L\to 0} \frac{1}{E_L^2}\widetilde{A}_{T\varphi\varphi} \otimes \widetilde{\Psi}_{\varphi T\varphi}, \qquad (281)$$

$$\langle T\varphi T\varphi\rangle_s \xrightarrow{E_R\to 0} \frac{1}{E_R^2}\widetilde{\Psi}_{T\varphi\varphi} \otimes \widetilde{A}_{\varphi T\varphi}, \qquad (282)$$

where the orders of the singularities are determined by dimensional analysis. Substituting the relevant three-point scattering amplitude and shifted correlator given in (238) and (274), we find

$$\langle T\varphi T\varphi\rangle_s \xrightarrow{E_L\to 0} 4(\vec{\xi}_1 \cdot \vec{k}_2)^2(\vec{\xi}_3 \cdot \vec{k}_4)^2 \frac{E(E_R+k_3)k_1 + E_R k_1 k_3}{E_L^2 E_R^2(k_{34}-s)^2}, \qquad (283)$$

$$\langle T\varphi T\varphi\rangle_s \xrightarrow{E_R\to 0} 4(\vec{\xi}_1 \cdot \vec{k}_2)^2(\vec{\xi}_3 \cdot \vec{k}_4)^2 \frac{E(E_L+k_1)k_3 + E_L k_1 k_3}{E_R^2 E_L^2(k_{12}-s)^2}. \qquad (284)$$

As before, we want to find a correlator that factorizes correctly on both of these poles and is devoid of spurious singularities. A natural guess, consistent with the permutation symmetry $\{E_L, k_3\} \longleftrightarrow \{E_R, k_1\}$, is

$$\langle T\varphi T\varphi\rangle_s \overset{?}{=} (\vec{\xi}_1 \cdot \vec{k}_2)^2(\vec{\xi}_3 \cdot \vec{k}_4)^2 \frac{4}{E_L^2 E_R^2}\left(\frac{2sk_1k_3}{E^2} + \frac{2k_1k_3 + E_L k_3 + E_R k_1}{E}\right). \qquad (285)$$

This is not yet the full solution, however, since we can have subleading poles in $E_L$ and $E_R$, that are not fixed by the factorization limits. These additional terms are constrained by the $E \to 0$ limit and conformal invariance. Let us compare this to the weight-shifting result (179):

$$\langle T\varphi T\varphi\rangle_s = (\vec{\xi}_1 \cdot \vec{k}_2)^2 (\vec{\xi}_3 \cdot \vec{k}_4)^2 \left[ \frac{4}{E_L^2 E_R^2}\left( \frac{2sk_1k_3}{E^2} + \frac{2k_1k_3 + E_Lk_3 + E_Rk_1}{E} \right) \right.$$
$$\left. + \frac{4}{E_L E_R}\left( \frac{2k_1k_3}{E^3} + \frac{k_{13}}{E^2} + \frac{1}{E} \right) \right]. \tag{286}$$

The terms in the first line (in blue) are fixed by the factorization limits, while the first term in the second line (in green) is determined by the limit $E \to 0$. The last two terms have subleading $E^{-2}$ and $E^{-1}$ poles (in red) whose coefficients, in principle, are unfixed. Their presence is required by conformal symmetry. Note that the pole structure in the second line is the same as that predicted in (211). In particular, we saw from the bulk perspective that these specific subleading total energy singularities must come along with the leading $E^{-3}$ pole due to symmetry. It would be interesting to understand this better without any reference to the bulk computation. Finally, the $t$-channel result is simply a permutation of the $s$-channel answer (286).

*Graviton exchange.*—Next, we consider the $u$-channel. To apply the factorization limits, we need the following three-point data

$$\widetilde{A}_{TTT}(k_1,k_3,u) = \frac{k_1 k_3 u}{2}\left[ (\vec{a}_u \cdot \vec{\xi}_u)\vec{\xi}_1 \cdot \vec{\xi}_3 + 2(\vec{k}_3 \cdot \vec{\xi}_1)(\vec{\xi}_3 \cdot \vec{\xi}_u) - 2(\vec{k}_1 \cdot \vec{\xi}_3)(\vec{\xi}_u \cdot \vec{\xi}_1) \right]^2, \tag{287}$$

$$\widetilde{A}_{T\varphi\varphi}(u,k_2,k_4) = \frac{u}{2}\left( \vec{\xi}_u \cdot \vec{\beta}_u \right)^2, \tag{288}$$

$$\widetilde{\Psi}_{TTT}(k_1,k_3,u) = -\frac{u^2 - k_{13}^2 - 2k_1 k_3}{(k_{13}+u)^2(k_{13}-u)^2}A_{TTT}, \tag{289}$$

$$\widetilde{\Psi}_{T\varphi\varphi}(u,k_2,k_4) = \frac{2A_{T\varphi\varphi}}{(k_{24}+u)^2(k_{24}-u)^2}. \tag{290}$$

In the factorization limits, we then require

$$\langle T\varphi T\varphi\rangle_u \xrightarrow{E_L^{(u)}\to 0} \frac{1}{(E_L^{(u)})^2}\widetilde{A}_{TTT} \otimes \widetilde{\Psi}_{T\varphi\varphi} = \frac{2uk_1k_3}{(E_L^{(u)})^2(E_R^{(u)})^2(k_{24}-u)^2}Z_{T\varphi T\varphi}^{(u)}, \tag{291}$$

$$\langle T\varphi T\varphi\rangle_u \xrightarrow{E_R^{(u)}\to 0} \frac{1}{(E_R^{(u)})^2}\widetilde{\Psi}_{TTT} \otimes \widetilde{A}_{T\varphi\varphi} = \frac{2uk_1k_3 + uEE_L^{(u)}}{(E_R^{(u)})^2(E_L^{(u)})^2(k_{13}-u)^2}Z_{T\varphi T\varphi}^{(u)}, \tag{292}$$

where we have defined the following polarization structure

$$Z_{T\varphi T\varphi}^{(u)} \equiv u^4 \Pi_{2,2}^{(u)}\frac{(\vec{\xi}_1 \cdot \vec{\xi}_3)^2}{6} + u^4 \Pi_{1,1}^{(u)}(\vec{\xi}_1 \cdot \vec{\xi}_3)(\vec{\xi}_1 \circ \vec{\xi}_3) + u^4(\vec{\xi}_1 \circ \vec{\xi}_3)^2, \tag{293}$$

and where $\xi_1 \circ \xi_3$ is defined as in (152). A function that factorizes correctly in both channels is

$$\langle T\varphi T\varphi\rangle_u \stackrel{?}{=} \frac{u}{(E_L^{(u)})^2(E_R^{(u)})^2}\left( \frac{2k_1k_3}{E^2} + \frac{E_L^{(u)}}{E} \right)Z_{T\varphi T\varphi}^{(u)}. \tag{294}$$

Notice, however, that this does not yet have the correct total energy singularity which should scale as $E^{-3}$. Indeed, the complete solution (183) takes the following form:

$$\langle T\varphi T\varphi\rangle_u = \frac{u}{(E_L^{(u)}E_R^{(u)})^2}\left( \frac{2k_1k_3}{E^2} + \frac{E_L^{(u)}}{E} \right)\mathcal{N} + \frac{1}{E_L^{(u)}E_R^{(u)}}\left( \frac{2k_1k_3}{E^3} + \frac{k_{13}}{E^2} \right)\mathcal{M} + \frac{1}{E}\mathcal{L}, \tag{295}$$

where the polarization structures $\mathcal{N}$, $\mathcal{M}$ and $\mathcal{L}$ were defined in (184). A large part of this answer is fixed by the expected poles: the first term (in blue) is fixed by the left and right singularities. The function $\mathcal{Q}_2^{(u)} = u^4 \Pi_{2,2}^{(u)} - (E_L^{(u)})^2 (E_R^{(u)})^2 \Pi_{2,0}^{(u)}$ inside of $\mathcal{N}$ contains a subleading pole that is required by conformal symmetry. The leading $E^{-3}$ pole (in green) is fixed by the total energy singularity, while the subleading poles (in red) are a consequence of conformal symmetry. Notice that the $E^{-2}$ pole is precisely the one predicted in (215). As for the $s$-channel answer, we have some understanding of why these precise singularities arise by appealing to the bulk dynamics, but it would be more satisfying to understand why these precise combinations come together from a symmetry perspective. A natural expectation is that these combinations of total energy singularities have particularly simple conformal transformation properties, and understanding their structure is likely to be helpful in constructing more complicated correlation functions. Finally, consistency with the flat-space limit requires us to add the contact solution (187).

**Four-point graviton correlator**

Given the three-point function that arises from Einstein gravity, it is straightforward to write down the factorization limits of $\langle TTTT \rangle$. As in the case of $\langle T\varphi T\varphi \rangle$, these limits will allow for unfixed subleading poles in both $E_{L,R}$ and $E$. It would be relatively straightforward to parametrize these subleading poles and fix their coefficients using conformal symmetry. The real challenge is to determine the contact solution. For $\langle T\varphi T\varphi \rangle$, we were able to derive the contact solution through weight-shifting and cross-check it against an explicit bulk calculation. For $\langle TTTT \rangle$, this does not seem feasible—even in flat space, the explicit computation of the contact contribution to the four-graviton scattering amplitude is extremely complicated. We therefore leave the derivation of $\langle TTTT \rangle$ as an interesting challenge for the future (see [103, 114, 119, 120] for earlier works).

## 6.3 One Channel Is Not Enough

In Section 5, we showed that the different channels of spinning correlators cannot be treated independently. In particular, we proved that the relevant Ward–Takahashi identities can only be satisfied if all channels are added with correlated couplings. This is a reflection of the fact that the individual channels aren't gauge-invariant and only their sum is physical. We will now provide a more on-shell perspective on the same problem. Specifically, we will show that the total energy singularities of the correlators are only consistent if the channels are added with the correct couplings.

### 6.3.1 $\langle JOOO \rangle$: Charge Conservation

We begin with the correlator of one spin-1 current and three conformally coupled scalars, $\langle J\varphi\varphi\varphi \rangle$. It is instructive to write the $s$-channel result (257) in spinor helicity variables

$$\langle J^- \varphi\varphi\varphi \rangle_s = e_2 \frac{\langle 12 \rangle \langle \bar{2}1 \rangle}{2k_1} \frac{1}{S} \log(E/E_R), \tag{296}$$

where have re-introduced the normalization of the correlator $e_2$ (see Fig. 5), and defined $S \equiv (k_{12} + s)(k_{12} - s)$.[52] The presence of the pole at $2k_1 = \langle \bar{1}1 \rangle \to 0$ suggests that this $s$-channel contribution by itself is inconsistent. In particular, the flat-space limit, $E \to 0$, is *not*

---

[52]Note that this definition of $S$ is not the same as $S_{34} \equiv (k_{34} + s)(k_{34} - s)$ at general energy values, but they coincide when $E \to 0$. In this section, when working away from $E = 0$, we always mean by $S, T, U$ the objects built out of the energy variables associated to the "left" vertex.

Lorentz-invariant. This is an on-shell diagnostic of the non-gauge-invariance of the $s$-channel correlator discussed in §5.2.1.

To isolate the problem, we write the mixed bracket in (296) as

$$\langle\bar{2}1\rangle = \frac{\langle\bar{2}4\rangle}{\langle\bar{1}4\rangle}\langle\bar{1}1\rangle + \frac{\langle\bar{1}\bar{2}\rangle}{\langle\bar{1}4\rangle}\langle\bar{4}1\rangle. \tag{297}$$

Substituting this into (296), we find

$$\langle J^-\varphi\varphi\varphi\rangle_s = -e_2\left(\frac{\langle12\rangle\langle\bar{2}4\rangle\langle41\rangle}{ST} - \frac{\langle14\rangle\langle\bar{4}1\rangle}{2k_1}\frac{1}{T}\right)\log(E/E_R), \tag{298}$$

where $T \equiv \langle14\rangle\langle\bar{1}4\rangle = (k_{14}+t)(k_{14}-t)$. We see that the Lorentz-violating part of the answer has a pole at $T = 0$. This suggests that Lorentz invariance can be restored by adding an appropriate contribution from the $t$-channel, which is

$$\langle J^-\varphi\varphi\varphi\rangle_t = e_4\frac{\langle14\rangle\langle\bar{4}1\rangle}{2k_1}\frac{1}{T}\log\left(E/E_R^{(t)}\right). \tag{299}$$

Indeed, combining the two channels and taking the limit $E \to 0$, we obtain

$$\frac{1}{\log E}\langle J^-\varphi\varphi\varphi\rangle_{s+t} \xrightarrow{E\to0} -e_2\frac{\langle12\rangle\langle\bar{2}4\rangle\langle41\rangle}{ST} + (e_2+e_4)\frac{\langle14\rangle\langle\bar{4}1\rangle}{2k_1}\frac{1}{T}. \tag{300}$$

The non-Lorentz-invariant pieces in both channels therefore cancel if we impose *charge conservation*, $e_2 = -e_4$.[53] Moreover, the remaining Lorentz-invariant term is precisely the flat-space scattering amplitude.

There is a simpler way to diagnose the same Lorentz non-invariance of the flat-space limit, which will prove to be useful in more complicated examples. We imagine taking the limits $E \to 0$ and $k_1 \to 0$ simultaneously. In a Lorentz-invariant theory, this limit has to be regular. Since $k_1$ is an energy, a divergence as it goes to zero would select a preferred Lorentz frame. We reach $2k_1 = \langle\bar{1}1\rangle \to 0$ by setting 1 parallel to $\bar{1}$, so that $\langle12\rangle\langle\bar{2}1\rangle \to -S$ and $\langle14\rangle\langle\bar{4}1\rangle \to -T$. In this limit, the sum of $s$- and $t$-channels is given by

$$\frac{1}{\log E}\langle J^-\varphi\varphi\varphi\rangle_{s+t} \xrightarrow{E,k_1\to0} -\frac{e_2+e_4}{2k_1}. \tag{301}$$

We see that the pole at $k_1 = 0$ is only absent when $e_2 = -e_4$.

There is some interesting physics to this perspective: demanding the absence of a $k_1 = 0$ singularity has required the presence of the $t$-channel. This is essentially a manifestation of *radiation-reaction*.[54] In the $s$-channel, the $k_1$-singularity is a signal of instantaneous propagation (clearly in violation of Lorentz invariance). This singularity can only be removed by allowing the exchanged particle to also propagate in the $t$-channel. Hence, given a charged source that can emit photons via an $s$-channel process (radiation), this source will experience a self-force from the $t$-channel process (reaction).

The $s$- and $t$-channel answers can be combined into a single expression as

$$\langle J^-\varphi\varphi\varphi\rangle_{s+t} = \frac{\langle12\rangle\langle\bar{2}4\rangle\langle41\rangle}{2ST}\log\left(\frac{E^2}{E_R^{(s)}E_R^{(t)}}\right) + \frac{\langle\bar{2}1\rangle\langle\bar{1}4\rangle + \langle\bar{4}1\rangle\langle\bar{1}\bar{2}\rangle}{2\langle\bar{1}1\rangle\langle\bar{1}\bar{2}\rangle\langle\bar{1}4\rangle}\log\left(\frac{E_R^{(s)}}{E_R^{(t)}}\right). \tag{302}$$

---

[53]Note that—exactly as in §5.2.1—we are only *required* to add either the $t$- or $u$-channel exchange correlator to the $s$-channel result in order for everything to be consistent. For the sake of simplicity, we have chosen this minimal route. (We could have instead added the $u$-channel by permuting $4 \leftrightarrow 3$ in (297).) However, it is completely acceptable to also allow particle exchange in the $u$-channel. Consistency of the total energy singularity would then require total charge conservation, as in (136).

[54]We thank Nima Arkani-Hamed for a discussion of this viewpoint.

This way of writing the correlator manifestly has the correct total energy singularity (to which only the first term contributes). However, this presentation obscures the partial energy singularities, with the two terms now combining to give the correct coefficients. Furthermore, the second term is regular in the limit $2k_1 = \langle \bar{1}1 \rangle \to 0$, but not manifestly so. The would-be singularity is cancelled by the vanishing of the log in the limit. This difficulty with making all properties of the correlator simultaneously manifest is strongly reminiscent of flat-space scattering amplitudes written in terms of spinor helicity variables, and is highly suggestive that there should be a more illuminating way of writing this object.

### 6.3.2 $\langle TOOO \rangle$: Equivalence Principle I

A similar analysis applies to the correlator of one stress tensor and three conformally coupled scalars, $\langle T\varphi\varphi\varphi \rangle$. The relevant $s$-channel correlator (279) is

$$\langle T^- \varphi\varphi\varphi \rangle_s = \frac{\kappa_2}{2} \left( \frac{\langle 12 \rangle \langle \bar{2}1 \rangle}{2k_1} \right)^2 \left[ \frac{1}{S} \frac{k_1}{E} + \frac{2S + 4k_{12}k_1}{S^2} \log\left( \frac{E_R}{E} \right) \right], \tag{303}$$

where the first term in the bracket dominates in the limit $E \to 0$, and the normalization is fixed in terms of the coupling $\kappa_2$ (see Fig. 8). Using Schouten identities, this limit can be written as

$$\frac{E}{k_1} \langle T^- \varphi\varphi\varphi \rangle_s \xrightarrow{E \to 0} \frac{\kappa_2}{2} \left( \frac{\langle 12 \rangle^2 \langle 14 \rangle \langle \bar{2}4 \rangle \langle 13 \rangle \langle \bar{2}3 \rangle}{STU} + A + B + (T+U)C \right), \tag{304}$$

where we have defined the following quantities:

$$A \equiv \frac{\langle 12 \rangle \langle 13 \rangle \langle 14 \rangle \langle \bar{2}4 \rangle}{TU} \frac{\langle \bar{3}1 \rangle}{2k_1}, \tag{305}$$

$$B \equiv \frac{\langle 12 \rangle \langle 13 \rangle \langle 14 \rangle \langle \bar{2}3 \rangle}{TU} \frac{\langle \bar{4}1 \rangle}{2k_1}, \tag{306}$$

$$C \equiv -\frac{\langle 14 \rangle \langle 13 \rangle}{TU} \frac{\langle \bar{3}1 \rangle \langle \bar{4}1 \rangle}{4k_1^2}. \tag{307}$$

Again, we have additional non-Lorentz-invariant terms that need to be cancelled by adding the other channels. The flat-space limits of the $t$- and $u$-channel contributions are

$$\frac{E}{k_1} \langle T^- \varphi\varphi\varphi \rangle_t \xrightarrow{E \to 0} -\frac{\kappa_4}{2} \left[ \frac{\langle 14 \rangle \langle 13 \rangle \langle 12 \rangle \langle \bar{2}3 \rangle}{TU} \frac{\langle \bar{4}1 \rangle}{2k_1} - \frac{\langle 14 \rangle \langle 13 \rangle}{U} \frac{\langle \bar{3}1 \rangle \langle \bar{4}1 \rangle}{4k_1^2} \right], \tag{308}$$

$$\frac{E}{k_1} \langle T^- \varphi\varphi\varphi \rangle_u \xrightarrow{E \to 0} -\frac{\kappa_3}{2} \left[ \frac{\langle 13 \rangle \langle 14 \rangle \langle 12 \rangle \langle \bar{2}4 \rangle}{TU} \frac{\langle \bar{3}1 \rangle}{2k_1} - \frac{\langle 13 \rangle \langle 14 \rangle}{T} \frac{\langle \bar{4}1 \rangle \langle \bar{3}1 \rangle}{4k_1^2} \right], \tag{309}$$

where we have re-written mixed spinor brackets in a convenient way. Combining all channels, we then get

$$\frac{E}{k_1} \langle T^- \varphi\varphi\varphi \rangle_{s+t+u} \xrightarrow{E \to 0} \frac{\kappa_2}{2} \frac{\langle 12 \rangle^2 \langle 14 \rangle \langle \bar{2}4 \rangle \langle 13 \rangle \langle \bar{2}3 \rangle}{STU}$$

$$- \frac{(\kappa_3 - \kappa_2)}{2} A - \frac{(\kappa_4 - \kappa_2)}{2} B - \left[ \frac{(\kappa_3 - \kappa_2)}{2} T + \frac{(\kappa_4 - \kappa_2)}{2} U \right] C. \tag{310}$$

We find that all Lorentz-violating pieces cancel only if all the coupling strengths are equal, $\kappa_2 = \kappa_3 = \kappa_4 \equiv \kappa$. This is of course nothing but the *equivalence principle*, and the remaining term correctly reproduces the flat-space amplitude.

It is also instructive to reproduce the same result via the shortcut of taking the limit $E, k_1 \to 0$ simultaneously. This gives

$$\frac{E}{k_1} \langle T^- \varphi \varphi \varphi \rangle_{s+t+u} \xrightarrow{E, k_1 \to 0} \frac{1}{8k_1^2} \left( \kappa_2 S + \kappa_4 T + \kappa_3 U \right), \tag{311}$$

so that the $k_1 = 0$ pole is only absent if $\kappa_2 = \kappa_3 = \kappa_4 \equiv \kappa$ (using the fact that $S + T + U = 0$).

### 6.3.3 $\langle JOJO \rangle$: Yang–Mills Theory

Next, let us consider the correlator $\langle J \varphi J \varphi \rangle$, with multiple non-Abelian gauge fields (see Fig. 7). Focusing on mixed helicities, the results for the different exchange solutions (260) and (266), as well as the contact solution (144), can be written as

$$\langle J_A^- \varphi_a J_B^+ \varphi_b \rangle_s = (T^A T^B)_{ab} \frac{1}{4EE_L E_R} \frac{\langle 12 \rangle \langle \bar{2}1 \rangle \langle \bar{3}4 \rangle \langle 4\bar{3} \rangle}{k_1 k_3}, \tag{312}$$

$$\langle J_A^- \varphi_a J_B^+ \varphi_b \rangle_t = (T^B T^A)_{ab} \frac{1}{4EE_L^{(t)} E_R^{(t)}} \frac{\langle 14 \rangle \langle \bar{4}1 \rangle \langle \bar{3}2 \rangle \langle 2\bar{3} \rangle}{k_1 k_3}, \tag{313}$$

$$\langle J_A^- \varphi_a J_B^+ \varphi_b \rangle_u = -f^{ABC} T_{ab}^C \frac{\mathcal{P}_1^{(u)} \langle \bar{3}1 \rangle^2 + 2\langle 12 \rangle \langle \bar{2}1 \rangle \langle 1\bar{3} \rangle \langle \bar{3}1 \rangle - 2\langle 13 \rangle \langle \bar{3}1 \rangle \langle 2\bar{3} \rangle \langle \bar{3}2 \rangle}{8EE_L^{(u)} E_R^{(u)} k_1 k_3}, \tag{314}$$

$$\langle J_A^- \varphi_a J_B^+ \varphi_b \rangle_c = -\left( (T^A T^B)_{ab} + (T^B T^A)_{ab} \right) \frac{1}{E} \frac{\langle \bar{3}1 \rangle^2}{8k_1 k_3}, \tag{315}$$

where we have normalized the various contributions according to the three-point couplings in Fig. 7. Again, we want to ensure that the flat-space limit, $E \to 0$, is Lorentz invariant. This time the undesired potential singularities arise when $k_1 \to 0$ and $k_3 \to 0$. To isolate the behavior in the vicinity of these locations, we can simultaneously take the limit $E, k_1, k_3 \to 0$. The relevant correlators greatly simplify to give

$$E \langle J_A^- \varphi_a J_B^+ \varphi_b \rangle_s \xrightarrow{E, k_1, k_3 \to 0} -(T^A T^B)_{ab} \frac{S}{4k_1 k_3}, \tag{316}$$

$$E \langle J_A^- \varphi_a J_B^+ \varphi_b \rangle_t \xrightarrow{\hspace{1.5cm}} -(T^B T^A)_{ab} \frac{T}{4k_1 k_3}, \tag{317}$$

$$E \langle J_A^- \varphi_a J_B^+ \varphi_b \rangle_u \xrightarrow{\hspace{1.5cm}} -f^{ABC} T_{ab}^C \frac{T - S}{8k_1 k_3}, \tag{318}$$

$$E \langle J_A^- \varphi_a J_B^+ \varphi_b \rangle_c \xrightarrow{\hspace{1.5cm}} -\left( (T^A T^B)_{ab} + (T^B T^A)_{ab} \right) \frac{U}{8k_1 k_3}. \tag{319}$$

Adding up the various channels, we see that the only way to get the coefficient of $(k_1 k_3)^{-1}$ to vanish is if the coupling matrices satisfy

$$[T^A, T^B]_{ab} = f^{ABC} T_{ab}^C. \tag{320}$$

That is, the couplings must transform in a representation of the *Lie algebra*.

### 6.3.4 $\langle TOTO \rangle$: Equivalence Principle II

The different contributions to gravitational Compton scattering were computed in §5.3.2 and §6.2.3. It is straightforward to write the results in spinor helicity variables and see that, as for

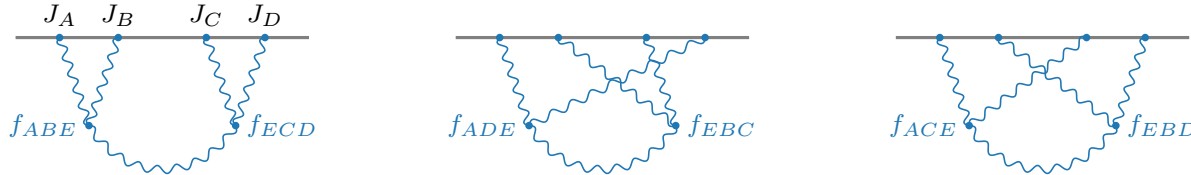

Figure 10: Illustration of the couplings involved in the $s$, $t$ and $u$-channel contributions to the four-point function of non-Abelian vector fields.

$\langle J^-\varphi J^+\varphi\rangle$, there are Lorentz-violating poles at $k_1 = 0$ and $k_3 = 0$. Indeed, taking the limit $E, k_1, k_3 \to 0$, of (286), (295) and (187), we find

$$E^3\langle T^-\varphi T^+\varphi\rangle_s \xrightarrow{E,k_1,k_3\to 0} -\kappa^2 \frac{S^3}{32k_1k_3}, \tag{321}$$

$$E^3\langle T^-\varphi T^+\varphi\rangle_t \xrightarrow{\hspace{2cm}} -\kappa^2 \frac{T^3}{32k_1k_3}, \tag{322}$$

$$E^3\langle T^-\varphi T^+\varphi\rangle_u \xrightarrow{\hspace{2cm}} -\kappa\kappa_g \frac{U(6S^2 + 6SU + U^2)}{192k_1k_3}, \tag{323}$$

$$E^3\langle T^-\varphi T^+\varphi\rangle_c \xrightarrow{\hspace{2cm}} \kappa_c^2 \frac{U(12ST - 5U^2)}{192k_1k_3}, \tag{324}$$

where we have included the couplings shown in Fig. 9. When the couplings are taken to be equal, the sum of channels greatly simplifies

$$E^3\langle T^-\varphi T^+\varphi\rangle_{s+t+u+c} \xrightarrow{E,k_1,k_3\to 0} -\kappa^2 \frac{S(T+U)(S+T+U)}{32k_1k_3} = 0. \tag{325}$$

Hence, we see that the Lorentz-violating poles cancel if we take

$$\kappa_g = \kappa_c = \kappa, \tag{326}$$

which is, in fact, the only way to make the limit regular. This relation is, of course, the *equivalence principle*, which now also holds for the graviton self-couplings.

### 6.3.5 $\langle JJJJ\rangle$: Jacobi Identity

In §6.2.2, the Yang–Mills correlator, $\langle JJJJ\rangle$, was derived by factorization. The contributions from the individual exchange channels are given by (273) and its permutations. Though we do not pursue it further here, it should be possible to see along the lines of §6.3.3 that the total energy singularity requires us to add together the individual channels in a particular way. Moreover, a contact contribution should be required in order for the final answer to be consistent.

Going through this in detail, we should find that it is only possible to construct a correlation function that has all the correct factorization singularities, along with the correct total energy pole, if the Yang–Mills coupling constants satisfy the Jacobi identity (see Fig. 10)

$$\sum_E f_{ABE}f_{CDE} + f_{BCE}f_{ADE} + f_{CAE}f_{BDE} = 0. \tag{327}$$

We strongly suspect, however, that there is a more elegant way to construct the full correlator directly, without the intermediate sum over different exchange channels, at least for some helicity configurations. This intuition comes from the remarkably concise expressions that exist for Yang–Mills scattering amplitudes, for example the Parke–Taylor formula [43].

### 6.3.6 $\langle TTTT \rangle$: Equivalence Principle III

If we had been able to compute the different contributions to the four-graviton correlator, $\langle TTTT \rangle$, we could now use them to derive further constraints on the graviton self-couplings, as in §6.3.4. We would have to find that the exchange contributions alone are not consistent and require the presence of the contact solution (see Fig. 11). The quartic coupling of the contact contribution must be related to the cubic couplings appearing in the exchange solutions, which is another manifestation of the equivalence principle for the graviton self-interactions.

It will be illuminating to understand these factorization/consistency requirements for graviton correlation functions in more detail. We have not fully explored pure graviton correlators partially because we anticipate—much as for the Yang–Mills case—that there is a deeper structure to be uncovered from which the decomposition into exchange channels can be recovered as an output. Finding such a formulation remains an important goal for the future.

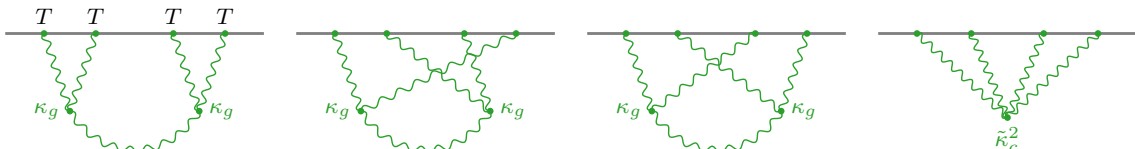

Figure 11: Illustration of the couplings involved in the exchange and contact contributions to the four-point function of the stress tensor.

### 6.3.7 A No-Go Example

One might get the impression that it is always possible to add sufficient channels to cancel all of the unwanted Lorentz-violating singularities. We therefore now give a simple flat-space example that shows this not to be the case.

Consider a massless scalar with a $\varphi^3$ interaction coupled to a massless spin-$\ell$ particle. On-shell there is a unique coupling between the scalar and the spin-$\ell$ particle. The corresponding four-point function in the $s$-channel is

$$\langle J_{(\ell)}\varphi\varphi\varphi \rangle_s = g_2 \frac{(\vec{\xi}_1 \cdot \vec{k}_2)^\ell}{E(k_{12}+s)(k_{34}+s)}. \tag{328}$$

It is easy to check that this four-point function is consistent with factorization. In spinor helicity variables, we obtain

$$\langle J_{(\ell)}^- \varphi\varphi\varphi \rangle_s = g_2 \left( \frac{\langle 12 \rangle \langle \bar{2}1 \rangle}{2k_1} \right)^\ell \frac{1}{E(k_{12}+s)(k_{34}+s)}. \tag{329}$$

The $s$-channel contribution clearly has an unwanted Lorentz-violating singularity as $E, k_1 \to 0$. We can reach this singularity by setting 1 parallel to $\bar{1}$, so that $\langle 12 \rangle \langle \bar{2}1 \rangle = \langle 12 \rangle \langle \bar{2}\bar{1} \rangle = -S$ and hence

$$\langle J_{(\ell)}^- \varphi\varphi\varphi \rangle_s \xrightarrow{E,k_1 \to 0} \frac{(-1)^{\ell-1}}{(2k_1)^\ell E} g_2 S^{\ell-1}. \tag{330}$$

Adding the $t$- and $u$-channels, we find

$$\langle J_{(\ell)}^- \varphi\varphi\varphi \rangle_{s+t+u} \xrightarrow{E,k_1 \to 0} \frac{(-1)^{\ell-1}}{(2k_1)^\ell E} \left( g_2 S^{\ell-1} + g_4 T^{\ell-1} + g_3 U^{\ell-1} \right). \tag{331}$$

Whether the singularity at $k_1 = 0$ can be removed depends on the spin $\ell$. For $\ell = 1$ and $\ell = 2$, this is captured by our discussion above. For $\ell \geq 3$, on the other hand, there is no way to

make the parenthesis in (331) vanish—there is simply no identically vanishing invariant that can be constructed from a sum of powers of Mandelstam variables. This reproduces from the correlator perspective the well-known statement that a particle with spin $\ell \geq 3$ cannot couple consistently to scalar matter in flat space.

## 6.4   Summary of Results

In this section, we presented an alternative approach to determine the four-point correlations between conserved currents and conformally coupled scalars, based on their singularity structure. In particular, we showed that most correlators are completely fixed by demanding $i$) the correct factorization on partial energy singularities, $ii$) the correct coefficient of the total energy singularity, and $iii$) the absence of any folded singularities. Sometimes, a subset of these conditions was sufficient to obtain the answer. In a few cases, we had to use conformal symmetry to fix the contributions from subleading poles. As in Section 5, we constructed the correlators separately in the $s$, $t$, and $u$-channels, and then used Lorentz-invariance in the flat-space limit to connect the different channels.

In the following, we briefly summarize our results:

- We began with a few warmup examples.

    - The correlator of a massless scalar in flat-space, $\langle \varphi\varphi\varphi\varphi \rangle$, is completely fixed by factorization alone. The answer for the $s$-channel is given in (228).

    - The four-point function of conformally coupled scalars arising from the exchange of a massless vector, $\langle \varphi\varphi\varphi\varphi \rangle_J$, has two parts: the helicity-one part is fixed by factorization alone, while the helicity-zero part is determined by the total energy singularity. The final answer for the $s$-channel is (235).

    - Similarly, the scalar correlator arising from the exchange of a graviton, $\langle \varphi\varphi\varphi\varphi \rangle_T$, is constrained by a combination of factorization and the total energy singularity. This almost fixes the answer uniquely. One subleading pole in the factorization limit is not constrained, and we must appeal to conformal symmetry to fix it. The final result is given in (244).

- We then determined the correlator $\langle J\varphi\varphi\varphi \rangle$, both in flat space and in de Sitter space. The flat-space result (252) is completely fixed by factorization alone. This is very similar to the analysis for $\langle \varphi\varphi\varphi\varphi \rangle$. To obtain the de Sitter result (257), we must impose the absence of an unwanted folded singularity. This automatically leads to the required total energy singularity with the flat-space amplitude as its coefficient.

- The correlator for Abelian Compton scattering, $\langle J\varphi J\varphi \rangle$, is completely fixed by the factorization limits. The answer for the $s$-channel is given in (260), with the $t$-channel related to this by a simple permutation. For non-Abelian Compton scattering, we have an additional $u$-channel contribution arising from the exchange of the vector field. This contribution is constrained by demanding the correct factorization and the absence of any folded singularities. A longitudinal component has to be added to give the correct limit for vanishing total energy. This is very similar to the analysis for $\langle \varphi\varphi\varphi\varphi \rangle_J$. The final answer is given in (266).

- A nice example of the power of the approach advocated in this section is the Yang–Mills correlator $\langle JJJJ \rangle$. It would be hard to determine this correlator by the weight-shifting method of Section 5. Instead, factorization fixes the answer almost completely. A subleading longitudinal piece has to be added to get the correct total energy singularity, as in all examples of vector exchange. The final result is (273).

- Another interesting example is the correlator $\langle T\varphi\varphi\varphi\rangle$. In that case, demanding the correct factorization and the absence of any folded singularities completely fixes the answer (279), and the correct total energy singularity is simply an output.

- The limits of the factorization method—at least in the way that we are currently implementing it—are exposed by the graviton Compton correlator, $\langle T\varphi T\varphi\rangle$. The $s$-channel result (286) is still completely fixed by a combination of factorization, total energy singularity and the absence of spurious folded singularities. Much of the $u$-channel result (295) is also fixed by these requirements, except for two subleading poles. First, we have to add a longitudinal piece to the answer in the factorization limit. This is the same term that had to be added to the naive answer for $\langle\varphi\varphi\varphi\varphi\rangle_T$. Second, the flat-space limit allows for an additional $E^{-1}$ pole. We don't have a simple way of fixing this pole, except appealing to conformal symmetry. Finally, the full correlator must have an additional contact solution. The leading $E^{-3}$ singularity of this solution is fixed by the flat-space limit. Additional $E^{-2}$ and $E^{-1}$ singularities, as well as a finite term, must be fixed by conformal symmetry.

- Given the challenges arising already for $\langle T\varphi T\varphi\rangle$, it is clear that imposing the above restrictions on the allowed singularities will leave a significant part of the four-graviton correlator $\langle TTTT\rangle$ undetermined. Bootstrapping the answer for $\langle TTTT\rangle$ therefore remains an important open problem (see [103, 114, 119, 120] for related work).

- In §6.3, we showed that multiple channels, with correlated coefficients, have to be added in order for the complete correlator to be Lorentz-invariant in the flat-space limit. Away from the flat-space limit, this is related to the conformal invariance of the full correlator, while the results of the individual channels are only covariant.

    - For $\langle J\varphi\varphi\varphi\rangle$, consistency of the full correlator implies charge conservation, while, for $\langle T\varphi\varphi\varphi\rangle$, it leads to the requirement that all matter couplings must satisfy the equivalence principle.

    - Consistency of $\langle J\varphi J\varphi\rangle$ requires the addition of a contact solution. Moreover, all matter couplings must transform in the representation of a Lie algebra. Similarly, consistency of $\langle T\varphi T\varphi\rangle$ requires that the gravitational self-couplings must be equal to the couplings to matter.

    - Consistency of $\langle JJJJ\rangle$ should require the self-couplings of the non-Abelian vector field (i.e. the structure constants $f_{ABC}$) to satisfy the Jacobi identity (327), and it would be interesting to show this explicitly. Combined with the constraints on the matter couplings from $\langle J\varphi J\varphi\rangle$ this then fixes the structure of Yang–Mills theory.

    - Although we weren't able to analyze $\langle TTTT\rangle$ explicitly, it is clear what to expect. Consistency requires the addition of a quartic contact interaction, whose coupling is equal to the square of the cubic couplings of the gravitons. This amounts to a nonlinear completion of Einstein gravity from the bootstrap perspective.

# 7 Applications to Inflation

So far, we have studied correlators of fields on a fixed de Sitter background and took all external scalars to be conformally coupled. For applications to inflation, however, we must consider massless scalars (playing the role of the inflaton field) and break some of the de Sitter symmetries. In slow-roll inflation, the symmetry breaking can be treated perturbatively, and the inflationary correlators are still constrained by the approximate conformal symmetry. In this

section, we will present a few simple cases of mixed scalar-tensor non-Gaussianity, but it should be clear that the machinery developed in this paper is more broadly applicable.

The inflationary scalar fluctuations are typically parametrized by the comoving curvature perturbation $\zeta$, which in single-field slow-roll inflation can be written as

$$\zeta = -\frac{H}{\dot{\phi}} \delta\phi \,, \tag{332}$$

where $\delta\phi$ are the inflaton fluctuations in spatially flat gauge and $\dot{\phi}$ controls the small deviation from a pure de Sitter background, with $\epsilon \equiv \frac{1}{2}\dot{\phi}^2/(M_{\text{pl}}^2 H^2) \ll 1$. To leading order in the slow-roll approximation, cosmological correlators can be computed first in terms of a (nearly) massless scalar field in a de Sitter background and are then converted to correlators for $\zeta$ using (332).

## 7.1 $\langle \gamma\zeta\zeta \rangle$

We begin with the mixed tensor-scalar-scalar bispectrum $\langle \gamma\zeta\zeta \rangle$. This was first computed in [41] and recently re-derived by solving the conformal Ward identity [55]. Here, we show that the weight-shifting approach provides a very simple path to the answer.

In terms of the dual wavefunction coefficients, the tensor-scalar-scalar correlator $\langle \gamma\,\delta\phi\,\delta\phi \rangle$ can be written as

$$\langle \gamma\,\delta\phi\,\delta\phi \rangle = -\frac{1}{4} \frac{\text{Re}\langle T\phi\phi \rangle}{\text{Re}\langle TT \rangle (\text{Re}\langle\phi\phi\rangle)^2} \,. \tag{333}$$

The relevant three-point correlation function $\langle T\phi\phi \rangle$ between one stress tensor and two massless scalars is related by a weight-raising operator to the result for conformally coupled scalars:

$$\langle T\phi\phi \rangle = \mathcal{W}_{23}^{++} \langle T\varphi\varphi \rangle \,, \tag{334}$$

where $\langle T\varphi\varphi \rangle$ is given by (95). Applying the weight-shifting operator (25), we find that the transverse part of the correlation function is

$$\langle T\phi\phi \rangle = c_{T\phi\phi}\left( K - \frac{k_1 k_2 + k_1 k_3 + k_2 k_3}{K} - \frac{k_1 k_2 k_3}{K^2} \right)(\vec{\xi}_1 \cdot \vec{k}_2)(\vec{\xi}_1 \cdot \vec{k}_3) \,, \tag{335}$$

where the normalization $c_{T\phi\phi}$ is *not* arbitrary, but fixed in terms of the size of the scalar two-point function $\langle \phi\phi \rangle$. Using

$$\langle \phi\phi \rangle = \frac{1}{M_{\text{pl}}^2}\langle TT \rangle = \frac{1}{H^2}k^3 \,, \tag{336}$$

we find that the stress tensor Ward–Takahashi identity requires that $c_{T\phi\phi} = -2/H^2$. Putting everything together, and converting from $\delta\phi$ to $\zeta$ using (332), we get

$$\langle \gamma\zeta\zeta \rangle = \frac{H^4}{4M_{\text{Pl}}^4 \epsilon} \frac{1}{(k_1 k_2 k_3)^3}\left( -K + \frac{k_1 k_2 + k_1 k_3 + k_2 k_3}{K} + \frac{k_1 k_2 k_3}{K^2} \right)(\vec{\xi}_1 \cdot \vec{k}_2)(\vec{\xi}_1 \cdot \vec{k}_3), \tag{337}$$

which agrees precisely with the result in [41].

## 7.2 $\langle \gamma\gamma\zeta \rangle$

Next, we compute the tensor-tensor-scalar correlator $\langle \gamma\gamma\zeta \rangle$. We first derive this correlator in standard slow-roll inflation (assuming Einstein gravity) and then consider higher-curvature corrections.

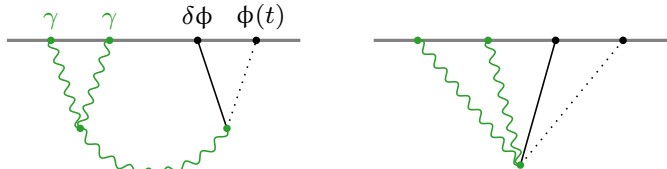

Figure 12: Illustration of the inflationary tensor-tensor-scalar bispectrum arising from the soft limit of a de Sitter trispectrum.

**Einstein gravity**

To determine the three-point function $\langle \gamma\gamma\zeta \rangle$ in slow-roll inflation, we must first compute the de Sitter four-point function of two gravitons and two massless scalars, or its dual wavefunction coefficient $\langle TT\phi\phi \rangle$. To relate this to the three-point function in an inflationary background, we perturb the conformal weight of the external scalars, $\Delta = 3 - \epsilon$ (where $\epsilon$ is the slow-roll parameter), and take the soft limit $k_4 \to 0$ (see Fig. 12). The later corresponds to evaluating one of the inflaton fields on its time-dependent background $\phi(t)$. The inflationary bispectrum $\langle \gamma\gamma\zeta \rangle$ can then be written as

$$\langle \gamma\gamma\zeta \rangle = -\frac{H}{\dot\phi}\langle \gamma\gamma\,\delta\phi \rangle = \frac{1}{4}\frac{H}{\dot\phi}\lim_{k_4\to 0}\frac{\mathrm{Re}\langle TT\phi\phi_{\Delta_4=3-\epsilon} \rangle}{(\mathrm{Re}\langle TT \rangle)^2\mathrm{Re}\langle \phi\phi \rangle}, \tag{338}$$

where we have used (332) to convert $\delta\phi$ to $\zeta$.

Our first task therefore is to compute the correlator $\langle TT\phi\phi \rangle$ in de Sitter space. We present the full correlator in §G.3, but only part of the answer is needed for the inflationary bispectrum. In particular, the scalar exchange and contact parts vanish when any of the scalar legs is taken to be soft, so they don't contribute to the inflationary bispectrum. We can therefore focus on the graviton exchange contribution.

For $\langle TT\phi\phi \rangle$, graviton exchange arises in the $s$-channel, while for $\langle T\varphi T\varphi \rangle$ in §5.3.2 it was in the $u$-channel. In principle, the graviton exchange involves contributions from all helicities, but only its longitudinal part has a non-vanishing soft limit and hence affects the inflationary bispectrum. This longitudinal piece is (see §G.3)

$$\langle TT\phi\phi \rangle_{s,L} = \left(\vec\xi_1 \cdot \vec\xi_2\right)^2\langle \phi\phi\phi\phi \rangle_{s,T}, \tag{339}$$

where $\langle \phi\phi\phi\phi \rangle_{s,T}$ is the four-point function of massless scalars exchanging a graviton in the $s$-channel. This can be written as

$$\langle \phi\phi\phi\phi \rangle_{s,T} = \mathcal{W}_{12}^{++}\mathcal{W}_{34}^{++}\left[\Pi_{2,0}(\Delta_w - 2)\langle \varphi\varphi\varphi\varphi \rangle_s\right] + \langle \phi\phi\phi\phi \rangle_c + \cdots, \tag{340}$$

where the weight-raising operators $\mathcal{W}_{ab}^{++}$ are defined in (25) and the relevant seed function $\langle \varphi\varphi\varphi\varphi \rangle_s$ is given in (122). A contact solution $\langle \phi\phi\phi\phi \rangle_c$ was added in (340) to remove the leading total energy singularity of the term proportional to $\Pi_{2,0}$, reducing the dominant scaling from $E^{-5}$ to $E^{-3}$. It is given by

$$\langle \phi\phi\phi\phi \rangle_c = \left(2\mathcal{W}_{12}^{++}\mathcal{W}_{34}^{++} - 3\mathcal{W}_{13}^{++}\mathcal{W}_{24}^{++} - 3\mathcal{W}_{14}^{++}\mathcal{W}_{23}^{++}\right)\frac{1}{E}. \tag{341}$$

The ellipses in (340) denote terms proportional to $\Pi_{2,2}$ and $\Pi_{2,1}$ that vanish in the soft limit and therefore don't contribute to the inflationary bispectrum.

As can be seen from (25), the weight-raising operator $\mathcal{W}_{34}^{++}$ depends on the perturbed scaling dimension $\Delta_4 = 3 - \epsilon$. Expanding (339) to linear order in the slow-roll parameter $\epsilon$,

we find

$$\lim_{k_4 \to 0} \langle TT\phi\phi_{\Delta_4 = 3-\epsilon} \rangle = \epsilon \left( \vec{\xi}_1 \cdot \vec{\xi}_2 \right)^2 \Big[ k_3^2 \mathcal{W}_{12}^{++} \Pi_{2,0} (\Delta_w - 2) \langle \varphi\varphi\varphi\varphi \rangle_s$$

$$+ \left( 2k_3^2 \mathcal{W}_{12}^{++} - 3k_2^2 \mathcal{W}_{13}^{++} - 3k_1^2 \mathcal{W}_{23}^{++} \right) \frac{1}{E} \Big]_{\vec{k}_4 \to 0} . \tag{342}$$

Substituting this into (338), we get

$$\langle \gamma\gamma\zeta \rangle = \frac{H^4}{8M_{\text{pl}}^4} \frac{(\vec{\xi}_1 \cdot \vec{\xi}_2)^2}{(k_1 k_2 k_3)^3} \left[ \frac{4k_1^2 k_2^2}{K} + \frac{1}{2} k_3 (k_1^2 + k_2^2) - k_1^3 - k_2^3 + \frac{k_3^3}{2} \right], \tag{343}$$

which matches the result in [41] up to a local term.[55]

**Higher-derivative correction**

In §4.3.2, we defined the three-point function between two stress tensors and a generic scalar operator,

$$\langle TT O_\Delta \rangle = \mathsf{P}_1^{(2)} \mathsf{P}_2^{(2)} H_{12}^2 \langle \varphi\varphi O_\Delta \rangle, \tag{344}$$

where $\mathsf{P}_a^{(2)}$ are the projection operators defined in (39) and $H_{12}$ is the weight-shifting operator introduced in (27). For $\Delta = 3$, this allows us to compute a higher-derivative correction to the inflationary tensor-tensor-scalar correlator

$$\langle \gamma\gamma\zeta \rangle = -\frac{H}{\dot{\phi}} \langle \gamma\gamma\, \delta\phi \rangle = -\frac{1}{4} \frac{H}{\dot{\phi}} \frac{\text{Re}\langle TT\phi \rangle}{(\text{Re}\langle TT \rangle)^2 \, \text{Re}\langle \phi\phi \rangle} . \tag{345}$$

The bulk interaction that gives rise to this correlator is $\phi W^2$, where $W_{\mu\nu\rho\sigma}$ is the Weyl tensor. Using (67) in (344), we find

$$\langle TT\phi \rangle = \frac{c_{TT\phi}}{K^4} \left[ f_1 \big( \vec{\xi}_1 \cdot \vec{\xi}_2 \big)^2 + 4 f_2 \big( \vec{\xi}_1 \cdot \vec{\xi}_2 \big) \big( \vec{\xi}_1 \cdot \vec{k}_2 \big) \big( \vec{\xi}_2 \cdot \vec{k}_1 \big) + 4 f_3 \big( \vec{\xi}_1 \cdot \vec{k}_2 \big)^2 \big( \vec{\xi}_2 \cdot \vec{k}_1 \big)^2 \right], \tag{346}$$

where

$$f_1 \equiv K^2 (K - 2k_3) \Big[ K^{(4)} + 3k_3 K^{(3)} + k_1 k_2 \big( K^{(2)} - 7k_3^2 \big) - 3k_3^3 K^{(1)} \Big], \tag{347}$$

$$f_2 \equiv -3K^{(5)} + 4K K^{(4)} + 3k_3^2 K^{(3)} + 3k_3 \big( 4k_1 k_2 - k_3^2 \big) K^{(2)} + k_1^2 k_2^2 \big( 7K + 9k_3 \big), \tag{348}$$

$$f_3 \equiv -3 \big( K^{(3)} + 2k_3^3 \big) + 4K \big( K^{(2)} + 2k_3^2 \big) + 12 k_1 k_2 k_3, \tag{349}$$

with $K^{(n)} \equiv \sum_a k_a^n - 2k_3^n$. The result simplifies considerably in spinor helicity variables,

$$\langle T^- T^- \phi \rangle = c_{TT\phi} \frac{k_1 k_2 (K + 3k_3)}{K^4} \langle 12 \rangle^4 . \tag{350}$$

Substituting (346) or (350) into (345) gives the result for $\langle \gamma\gamma\zeta \rangle$.

# 8  Conclusions and Outlook

The study of spinning cosmological correlators is still nascent, and we are just beginning to discover its organizing principles. Computing correlators of spinning fields directly is rather

---

[55]Notice that the expression we found has vanishing soft limit. The missing local term can be fixed by the inflationary consistency condition for the squeezed limit [41].

complicated, so only a handful of these calculations have been done. Similar challenges were encountered for flat-space scattering amplitudes, where the usual perturbative approaches are also prohibitively difficult for particles with spin. In the case of scattering amplitudes, these obstacles have been overcome through the use of modern on-shell techniques. Fundamental principles like locality, unitarity, and causality are granted primacy, and the final observable scattering amplitudes are determined by theoretical consistency. Aside from providing concrete computational results, this bootstrap approach has led to new conceptual insights, revealing hidden symmetries and mathematical structures that are completely invisible at the level of Lagrangians and Feynman diagrams [121].

In this paper, we applied the bootstrap philosophy to spinning correlators in de Sitter space. Instead of tracking the detailed time evolution of the bulk physics, we have focused directly on the final boundary correlators and derived them from consistency conditions alone. In particular, we used that all correlators must satisfy the conformal Ward identities, and that the correlation functions involving conserved currents must obey the Ward–Takahashi identities. The challenge is to satisfy these identities simultaneously, subject to constraints on the singularity structure allowed by local bulk physics.

We developed two complementary approaches to construct consistent spinning correlators:

- First, we showed that solutions to the conformal Ward identities for spinning correlators can be obtained by acting with weight-shifting operators on simple scalar seed correlators [57,58]. Using this weight-shifting approach, we derived many three- and four-point functions involving spinning fields, focusing on the phenomenologically most relevant cases of massless spin-1 and spin-2 fields. Imposing the WT identities on these correlators led to interesting constraints on the couplings of the fields, reproducing charge conservation and the equivalence principle from a purely boundary perspective.

- We then derived the same four-point correlators by imposing consistent factorization, obviating the need to solve the conformal Ward identities directly. This approach exploited the fact that all correlators have singularities when the sum of the energies entering a subgraph vanishes. At these loci, the correlators must factorize, with coefficients related to the corresponding flat-space scattering amplitudes. We showed that in many cases this singularity structure is sufficient to completely fix the correlator. Furthermore, demanding the coefficient of the total energy singularity to be Lorentz invariant gives the same constraints on the couplings as those obtained from the WT identities.

Our work suggests a number of concrete directions that are worthy of further exploration:

- We have concentrated on correlators of fields with integer spin. The weight-shifting approach, however, can also be used to generate correlators involving fermionic fields [58]. As an interesting application, one could derive correlators with massless gravitinos. It is known that, in flat space, the gravitational couplings of massless spin-3/2 particles must be supersymmetric, leading to a bootstrap derivation of supergravity [2,4]. In de Sitter space, on the other hand, supersymmetry must be broken, and it would be illuminating to understand how this symmetry breaking manifests itself in the cosmological correlators.

- We have focused mostly on correlators arising from theories that are known to be consistent in flat space. It would be very interesting to explore more uncharted territory and use theoretical consistency to map out the broader landscape of allowed field theories in de Sitter space. Clear targets are to understand (broken) supersymmetry and the viability of Vasiliev-like theories [77,122] from this bootstrap perspective. Beyond this, it would also be interesting to constrain the interactions of partially massless fields. These

are exotic representations unique to de Sitter space (see Appendix A), for which there are some no-go results [123–127]. The bootstrap approach should help shed some light on these theories.

- We have shown that consistent factorization in many examples completely fixes the exchange contributions to the correlators. However, in some of the cases, certain subleading poles cannot be determined solely from the factorization singularities, and instead have to be fixed by imposing conformal symmetry. This is the analogue of using Lorentz symmetry to determine the structure of flat-space scattering amplitudes away from factorization limits. While it is easy to construct Lorentz-invariant amplitudes (e.g. by writing the factorization limits in terms of Mandelstam variables), we do not yet have a simple way to enforce conformal symmetry of the correlators away from their singularities. Developing a method to connect conformally-invariant building blocks will be essential in order to apply the factorization method to more complicated examples. A concrete challenge would be to construct the four-point graviton correlator, which is sufficiently complicated as to provide a useful stress-test for more sophisticated techniques.

- The standard approach to computing scattering amplitudes in gauge theories is complicated because the calculation is broken up into non-gauge-invariant pieces. Each Feynman diagram contains redundant information that does not contribute to the final, gauge-invariant answer. The modern bootstrap approaches avoid these complications by focusing directly on the physical degrees of freedom, at the cost of manifest locality. In the cosmological context, we do not yet have such a purely on-shell formulation. In particular, the initial outputs of both of our approaches—weight shifting and factorization—are exchange correlators in particular channels. We then found that consistency requires us to combine the various channels in a precise way. It would be more satisfying to find an approach that constructs this final answer directly, without the intermediate decomposition into channels. One promising avenue would be to solve the conformal Ward identity and the WT identity simultaneously by suitably combining them into a single differential equation. Another approach worth exploring is to search for cosmological analogues of BCFW-like recursion relations [116], which would systematize the decomposition into channels (see [113, 128, 129] for related work in AdS).

The modern amplitudes revolution was catalyzed by the discovery of the Parke–Taylor formula [43], a remarkably compact and universal formula describing gluon scattering. Initially discovered by brute force, it is now easily derived using modern recursion methods [46]. The elegance of the Parke–Taylor result was the first indication of a hidden simplicity in scattering amplitudes and has since sparked many fascinating developments. The study of cosmological correlators has not yet had its Parke–Taylor moment, but the fact that scattering amplitudes live inside cosmological correlators [42, 59] strongly suggests that similar structures exist. Exposing this hidden simplicity remains an important goal of the cosmological bootstrap.

# Acknowledgements

We are grateful to Ana Achúcarro, Paolo Benincasa, Matteo Biagetti, Heng-Yu Chen, Wei-Ming Chen, Frederik Denef, Garrett Goon, Daniel Green, Aaron Hillman, Kurt Hinterbichler, Yu-tin Huang, Lam Hui, Hiroshi Isono, Sadra Jazayeri, Petr Kravchuk, Arthur Lipstein, Paul McFadden, Scott Melville, Alberto Nicolis, Toshifumi Noumi, Rachel Rosen, Luca Santoni, David Simmons-Duffin, John Stout, Zimo Sun, Mark Trodden, Dong-Gang Wang, Zhong-Zhi Xianyu, Sasha Zhiboedov, and Siyi Zhou for helpful discussions. A special thank you to Nima Arkani-Hamed for collaboration, many inspiring discussions and encouragement. DB is grateful to

Yu-tin Huang and the Center for Theoretical Physics at the National Taiwan University for their hospitality during his sabbatical. DB and CDP thank the National Taiwan University for its hospitality during the workshop "Cosmology and Conformal Field Theory." DB thanks the participants of the Simons Symposium "Amplitudes Meet Cosmology" for many helpful discussions. DB, AJ, and GP thank Harvard University for hospitality while this work was in progress. GP thanks Toshifumi Noumi and the cosmology group of Kobe University for their warm hospitality and for many lively discussions. DB and CDP are supported by a VIDI grant of the Netherlands Organisation for Scientific Research (NWO) that is funded by the Dutch Ministry of Education, Culture and Science (OCW). HL is supported by DOE grant DE-SC0019018. GP is supported by the European Union's Horizon 2020 research and innovation programme under the Marie-Sklodowska Curie grant agreement number 751778. The work of DB, AJ, and GP is part of the Delta-ITP consortium.

# A    De Sitter Representations

This appendix contains a brief review of representations of the de Sitter group. Further details can be found in [130–134].

## A.1    De Sitter Algebra

The de Sitter algebra, so(4, 1), is generated by the (anti-Hermitian) generator $J_{AB}$, with commutation relations

$$[J_{AB}, J_{CD}] = \eta_{AC} J_{BD} - \eta_{BC} J_{AD} + \eta_{BD} J_{AC} - \eta_{AD} J_{BC} , \tag{A.1}$$

where $\eta_{AB} \equiv \mathrm{diag}\left(\delta_{ij}, 1, -1\right)$, with $i, j \in \{1, 2, 3\}$. The relation between these generators and those in (14) is

$$\begin{aligned} D &= J_{45} , \\ P_i &= J_{4i} - J_{5i} , \\ K_i &= J_{4i} + J_{5i} . \end{aligned} \tag{A.2}$$

The fact that the $J_{AB}$ are anti-Hermitian implies that these generators are anti-Hermitian as well:

$$D^{\dagger} = -D , \quad P_i^{\dagger} = -P_i , \quad K_i^{\dagger} = -K_i . \tag{A.3}$$

The goal is to enumerate representations that are unitary with respect to an inner product consistent with this notion of conjugation.[56] Such representations have been classified in various places [130–134], they are naturally labeled by their quadratic and quartic Casimir eigenvalues [131]

$$\mathcal{C}_2 = \frac{1}{2} J_{AB} J^{AB} = \Delta(3 - \Delta) - \ell(\ell + 1) , \tag{A.4}$$

$$\mathcal{C}_4 = W_A W^A = -\ell(\ell + 1)(\Delta - 2)(\Delta - 1) , \tag{A.5}$$

where $W_A \equiv \frac{1}{8} \epsilon_{ABCDE} J^{BC} J^{DE}$ is the de Sitter analogue of the Pauli–Lubanski (pseudo)vector. Unitary representations are therefore labeled by $[\Delta, \ell]$, where $\ell$ is the spin of the corresponding bulk field and $\Delta$ is related to the mass through the relation (16).

---

[56]This reality condition differs from the standard one imposed in the study of Euclidean CFT, for example in [135]. The reason for the difference is that in those cases the interest is in studying representations in Euclidean signature that are the analytic continuation of unitary representations in Lorentzian signature, corresponding to a different reality condition on the complexified algebra. Here, we are interested in representations that are unitary with respect to the reality condition (A.3).

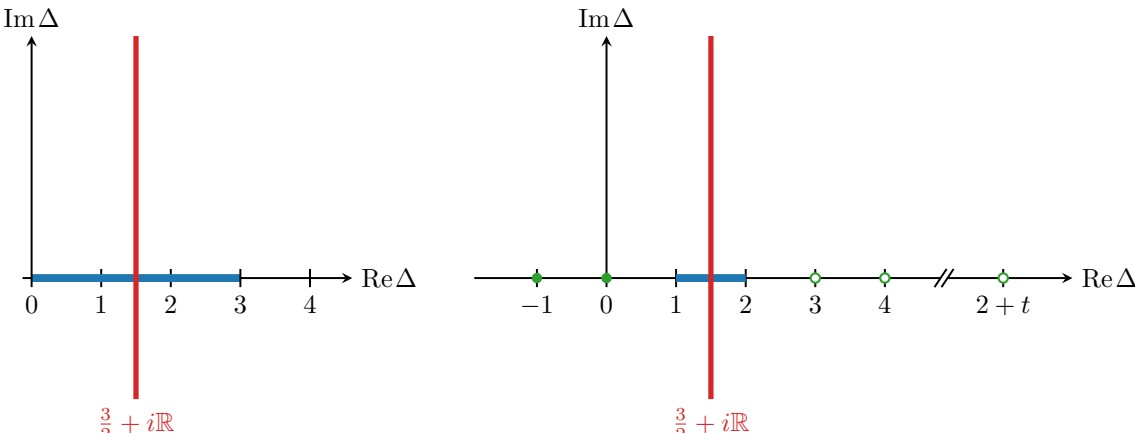

Figure 13: Illustration of scalar (*left*) and spin-$\ell$ (*right*) representations of SO(4, 1) in the complex $\Delta$ plane. The red lines corresponds to the principal series, while the blue lines mark the complementary series. The green circles for the spin-$\ell$ case denote the representations of the discrete series. Open circles are the shadows of the filled ones.

## A.2 Unitary Representations

The unitary representations of the de Sitter algebra are qualitatively different for scalars and spinning fields, so we must treat them separately. We summarize the results in Fig. 13.

**Scalar fields** We begin by considering scalar representations $\ell = 0$. In this case the quartic Casimir vanishes. In order for the representation to be unitary, at minimum the quadratic Casimir eigenvalue must be real, which places some restrictions on the possible values of $\Delta$. There are then additional constraints imposed by requiring a positive-definite inner product. The unitary representations split into two families:

- **Principal series:** Representations in this family have conformal dimensions $\Delta = \frac{3}{2} + i\mu$, with $\mu \in \mathbb{R}$. Using the relation (16), we see that these representations correspond to heavy fields in de Sitter with $m^2 \geq \frac{9}{4}H^2$.

- **Complementary series:** Representations in this series have conformal dimensions in the range $0 < \Delta < 3$. This corresponds to light fields with $0 < m^2 < \frac{9}{4}H^2$.

The most important scalar representations for our purposes lie in the complementary series. The late-time wavefunction coefficients of a conformally coupled scalar in dS$_4$ are captured by correlation functions of $\Delta = 2$ scalar operators, while massless scalars correspond to $\Delta = 3$[57].

**Spin-$\ell$ fields** Fields with spin have a slightly different classification. Along with the principal and complementary series of representations, there is an additional family of unitary representations, whose weights take on discrete values.

- **Principal series:** Spinning representations in the principal series have conformal dimensions satisfying $\Delta = \frac{3}{2} + i\mu$, with $\mu \in \mathbb{R}$. The corresponding bulk representations can be inferred from (16) and are given by heavy fields with $m^2 \geq (\ell - \frac{1}{2})^2 H^2$.

---

[57]Strictly speaking, these representations have an additional shift symmetry, and are better referred to as members of the discrete series of representations, see e.g. [136].

- **Complementary series:** Fields in this family have conformal dimensions in the range $1 < \Delta < 2$. Note that this is bounded away from $\Delta = 0$. These representations correspond to fields with masses in the range $\ell(\ell-1)H^2 < m^2 < (\ell - \frac{1}{2})^2 H^2$. A qualitative difference from the scalar case is that spinning representations with $\ell \geq 2$ have a lower bound on their mass in order to remain unitary. This lower bound is typically called the *Higuchi bound* [137].

- **Discrete series:** For spinning fields there is an additional set of unitary representations. Fields in this series have conformal dimensions with the discrete set of values $\Delta = 2 + t$, or with the shadow weight $\Delta = 1 - t$, where the parameter $t$ is an integer $t \in \{0, 1, \cdots, \ell - 1\}$. Correspondingly bulk fields have discrete mass values

$$\frac{m^2}{H^2} = \ell(\ell-1) - (t+1)t\,. \tag{A.6}$$

The representation with $t = \ell - 1$ is a massless field, other values of $t$ correspond to partially massless fields [138–140].[58] Correspondingly, the parameter $t$ is typically called the depth of partial masslessness. The $t = 0$ points, which coincide with the endpoints of the complementary series, are sometimes called the *exceptional series*.

In this paper, we are primarily interested in correlation functions of external fields that lie in the complementary and discrete series—so-called "light" fields in de Sitter. This is because correlation functions of operators at these special weights are particularly simple.

# B  Ward–Takahashi Identities

In this appendix, we describe the derivation of the Ward–Takahashi identities used in Sections 4 and 5. Additional discussions can be found in [42, 56, 80, 89, 141]. Our starting point is the late-time wavefunctional $\Psi[\gamma_{ij}, A_i^B, \sigma^a]$. This plays the role of a generating functional for a three-dimensional quantum field theory, where the sources are the late-time profiles and the correlators are the wavefunction coefficients. Specifically, the relation between the wavefunction and the correlators of interest is given by the following one-point functions in the presence of sources

$$\langle O^b(\vec{x}) \rangle = \frac{1}{\sqrt{\gamma(\vec{x})}} \frac{\delta \Psi}{\delta \sigma^b(\vec{x})}\,, \tag{B.1}$$

$$\langle J_i^B(\vec{x}) \rangle = \frac{1}{\sqrt{\gamma(\vec{x})}} \frac{\delta \Psi}{\delta A_i^B(\vec{x})}\,, \tag{B.2}$$

$$\langle T^{ij}(\vec{x}) \rangle = \frac{2}{\sqrt{\gamma(\vec{x})}} \frac{\delta \Psi}{\delta \gamma_{ij}(\vec{x})}\,. \tag{B.3}$$

Invariance under gauge transformations and diffeomorphisms in the bulk translate into identities satisfied by these wavefunction coefficients. These identities are precisely the Ward–Takahashi identities of interest.

In this appendix, we will derive the WT identities by using our knowledge of bulk gauge transformations. However, it is an important conceptual point that these WT identities also have a purely boundary interpretation, where they are part of the *definition* of conserved current operators. The WT identities could then also be derived from a more axiomatic point

---

[58]The obvious generalization of these facts to $d$ dimensions is wrong, (partially) massless fields are not in the discrete series in general but rather lie in the exceptional series [134]. We thank Frederik Denef and Zimo Sun for a discussion of this point.

of view, using the fact that currents are conserved only up to contact terms (see e.g. [135]). A useful aspect of this more boundary-centric point of view is the notion that the action of the stress tensor on other operators $O_a$ can have an associated "charge," which we call $\kappa_a$. Eventually, all of these couplings are of course required to be the same due to the equivalence principle, but we allow these normalizations to float in the main text in order to see this constraint come out explicitly.

## B.1 Spin-1 Identities

We first consider the WT identities satisfied by correlation functions involving spin-1 currents. Under a gauge transformation the late-time field profiles change as follows

$$\delta A_i^B = \partial_i \Lambda^B - i f^{BCD} A_i^C \Lambda^D \,, \tag{B.4}$$

$$\delta \sigma^a = \Lambda_A (T^A)_{ab} \, \sigma^b \,. \tag{B.5}$$

Demanding that the wavefunction is invariant under (small) gauge transformations[59] of these sources implies the identity

$$\delta_\Lambda \Psi = \int \mathrm{d}^3 x \left[ \left( \partial_i \Lambda^B - i f^{BCD} A_i^C \Lambda^D \right) \frac{\delta}{\delta A_i^B} + \Lambda_A (T^A)_{ab} \, \sigma^b \frac{\delta}{\delta \sigma^a} \right] \Psi = 0 \,. \tag{B.6}$$

Using the relations (B.1) and (B.2), we can write this as

$$\partial^i \langle J_i^A(\vec{x}) \rangle - i f^{ABC} A_i^B(\vec{x}) \langle J_i^C(\vec{x}) \rangle + (T^A)_{ab} \, \sigma^b(\vec{x}) \langle O^a(\vec{x}) \rangle = 0 \,. \tag{B.7}$$

The WT identities involving currents can be obtained from this formula by repeated differentiation with respect to $\sigma$ and $A_i$, after which we set the sources to zero. The presence of sources in some of the terms is the reason that these identities relate correlation functions with different numbers of fields.

As an example, consider the identity for correlators involving a single current and arbitrarily many scalar operators $\langle J_1 O_2 \cdots O_n \rangle$. The corresponding WT identity can be obtained from (B.7) by differentiating $n$ times with respect to the $\sigma_a$ source:

$$\partial^i \langle J_i^A(\vec{x}_1) O^{b_2}(\vec{x}_2) \cdots O^{b_n}(\vec{x}_n) \rangle = -\sum_{a=2}^n \delta(\vec{x}_1 - \vec{x}_a)(T^A)_{b_a c} \langle O^{b_2}(\vec{x}_2) \cdots O^c(\vec{x}_a) \cdots O^{b_n}(\vec{x}_n) \rangle \,. \tag{B.8}$$

Transforming to Fourier space, the delta function on the right-hand side shifts the momentum argument of the $a$-th operator, so that

$$\vec{k}_1 \cdot \langle \vec{J}_{\vec{k}_1}^A O_{\vec{k}_2}^{b_2} \cdots O_{\vec{k}_n}^{b_n} \rangle = -\sum_{a=2}^n i(T^A)_{b_a c} \langle O_{\vec{k}_2}^{b_2} \cdots O_{\vec{k}_a + \vec{k}_1}^c \cdots O_{\vec{k}_n}^{b_n} \rangle \,. \tag{B.9}$$

Other identities involving currents can be obtained in a similar way by taking functional derivatives of (B.7).

## B.2 Spin-2 Identities

Next, we consider the WT identities associated to spin-2 currents. In this case, there are two different types of WT identities, associated to either current conservation or the vanishing of the trace of the current. We will consider each of these in turn.

---

[59]In contrast, the wavefunction is typically not invariant under *large* gauge transformations. See, e.g., [142].

**Current conservation**

The action of bulk diffeomorphisms implies that the sources transform as

$$\delta\gamma_{ij} = -2\nabla_{(i}\xi_{j)}, \tag{B.10}$$

$$\delta A_i^B = -\xi^j\nabla_j A_i^B - \nabla_i\xi^j A_j^B, \tag{B.11}$$

$$\delta\sigma^a = -\xi^i\nabla_i\sigma^a, \tag{B.12}$$

where $\nabla$ is the covariant derivative associated to $\gamma_{ij}$. Varying the wavefunction, we obtain the identity

$$\delta_\xi\Psi = \int \mathrm{d}^3x\left[-2\nabla_{(i}\xi_{j)}\frac{\delta}{\delta\gamma_{ij}} - \left(\xi_j\nabla^j A_i^B + \nabla_i\xi_j A^{jB}\right)\frac{\delta}{\delta A_i^B} - \xi_i\nabla^i\sigma^a\frac{\delta}{\delta\sigma^a}\right]\Psi = 0. \tag{B.13}$$

Using the relations (B.1)–(B.3), this can be written as

$$\nabla_i\langle T^{ij}(\vec{x})\rangle - \nabla^j A^{iB}\langle J_i^B(\vec{x})\rangle + \nabla^i\left(A^{jB}\langle J_i^B(\vec{x})\rangle\right) - \nabla^j\sigma^a(\vec{x})\langle O^a(\vec{x})\rangle = 0. \tag{B.14}$$

We can then obtain any desired WT identity by functionally differentiating this expression. In the main text, we do not consider correlators with both spin-1 and spin-2 conserved currents, so in the following we will set $A_i = 0$.

There is an important subtlety in deriving the WT identities for correlators with spin-2 currents that was absent in the spin-1 case. The covariant derivatives in (B.14) involve the metric, so they will contribute when we take functional derivatives. To make these additional contributions manifest, it is helpful to rewrite (B.14) as

$$\partial_i\langle T^{ij}(\vec{x})\rangle + \Gamma_{ik}^i(\vec{x})\langle T^{kj}(\vec{x})\rangle + \Gamma_{ik}^j(\vec{x})\langle T^{ik}(\vec{x})\rangle - \gamma^{ij}(\vec{x})\partial_i\sigma^a(\vec{x})\langle O^a(\vec{x})\rangle = 0. \tag{B.15}$$

To take the functional derivatives, we will use [141]

$$\frac{\delta\gamma_{ij}(\vec{x})}{\delta\gamma_{kl}(\vec{y})} = \mathbb{1}_{ij}^{kl}\,\delta(\vec{x}-\vec{y}), \tag{B.16}$$

$$\frac{\delta\gamma^{ij}(\vec{x})}{\delta\gamma_{kl}(\vec{y})} = -\gamma^{im}\gamma^{jn}\mathbb{1}_{mn}^{kl}\,\delta(\vec{x}-\vec{y}), \tag{B.17}$$

$$\frac{\delta\Gamma_{mn}^k(\vec{x})}{\delta\gamma_{ij}(\vec{y})} = \frac{\delta\gamma^{lk}(\vec{x})}{\delta\gamma_{ij}(\vec{y})}\Gamma_{l,mn}(\vec{x}) + \gamma^{lk}(\vec{x})\frac{\delta\Gamma_{l,mn}(\vec{x})}{\delta\gamma_{ij}(\vec{y})}, \tag{B.18}$$

$$\frac{\delta\Gamma_{l,mn}(\vec{x})}{\delta\gamma_{ij}(\vec{y})} = \frac{1}{2}\left(\mathbb{1}_{ln}^{ij}\frac{\partial}{\partial x^m}\delta(\vec{x}-\vec{y}) + \mathbb{1}_{lm}^{ij}\frac{\partial}{\partial x^n}\delta(\vec{x}-\vec{y}) - \mathbb{1}_{mn}^{ij}\frac{\partial}{\partial x^l}\delta(\vec{x}-\vec{y})\right), \tag{B.19}$$

where we have defined $\mathbb{1}_{kl}^{ij} \equiv \frac{1}{2}\left(\delta_k^i\delta_l^j + \delta_l^i\delta_k^j\right)$ and $\Gamma_{i,jk} \equiv \gamma_{il}\Gamma_{jk}^l$.

There are two further subtleties involved in the derivation of stress tensor WT identities, both related to the definition of the stress tensor. First, we are defining stress tensor insertions as functional derivatives of $\Psi$ *with* factors of $\sqrt{\gamma}$:

$$\langle T^{i_1j_1}(\vec{x}_1)\cdots T^{i_nj_n}(\vec{x}_n)\rangle \equiv \frac{2}{\sqrt{\gamma(\vec{x}_1)}}\frac{\delta}{\delta\gamma_{i_1j_1}(\vec{x}_1)}\cdots\frac{2}{\sqrt{\gamma(\vec{x}_n)}}\frac{\delta}{\delta\gamma_{i_nj_n}(\vec{x}_n)}\Psi, \tag{B.20}$$

where the functional derivatives act on everything to their right. There are then contributions where the functional derivatives act on the measure factors. Some authors define stress tensor insertions without the $\sqrt{\gamma}$ factors, which will cause the resulting WT identity to differ by contact terms (terms with delta functions). Additionally, there is a freedom to perform field redefinitions of the sources, and to define the stress tensor as a functional derivative of some

function of $\gamma_{ij}$. Consider $\gamma_{ij} = c^{-1}(e^{c\hat{\gamma}})_{ij}$ and define an alternative stress tensor as the functional derivative with respect to $\hat{\gamma}_{ij}$. The constant $c$ parametrizes this ambiguity. The relation between the two stress tensors is

$$\hat{T}^{ij} = \frac{\delta\gamma_{kl}}{\delta\hat{\gamma}_{ij}} T^{kl} = T^{ij} + \frac{c}{2}\gamma^{ik}T_k^j + \frac{c}{2}\gamma^{jk}T_k^i + \cdots, \tag{B.21}$$

where we have only kept the leading-order terms in $\gamma$. When we differentiate with respect to $\gamma$, we therefore see that this source ambiguity can contribute. In fact, this ambiguity is degenerate with the previously discussed one, and we will describe how to account for them both.

For correlation functions involving a single stress tensor, none of these subtleties matter, and the relevant identities can be obtained by differentiating (B.14) with respect to the sources. After Fourier transforming, this leads to the identities used in the main text, for example (165).

**Two stress tensors** We next consider identities with two stress tensor insertions. To obtain these, we differentiate (B.15) with respect to $\gamma_{ij}$ and then set $\gamma_{ij} = \delta_{ij}$ to obtain

$$\partial_{i_1}\langle T^{i_1 j_1}(\vec{x}_1)T^{i_2 j_2}(\vec{x}_2)\rangle + 2\,\delta^{lk}\frac{\delta\Gamma_{l,ki_1}(\vec{x}_1)}{\delta\gamma_{i_2 j_2}(\vec{x}_2)}\langle T^{i_1 j_1}(\vec{x}_1)\rangle + 2\delta^{j_1 k}\frac{\delta\Gamma_{k,i_1 l}(\vec{x}_1)}{\delta\gamma_{i_2 j_2}(\vec{x}_2)}\langle T^{i_1 l}(\vec{x}_1)\rangle$$
$$+ 2\times\mathbb{1}^{i_1 j_1, i_2 j_2}\delta(\vec{x}_1 - \vec{x}_2)\partial_{i_1}\sigma^a(\vec{x}_1)\langle O^a(\vec{x}_1)\rangle - \partial^{j_1}\sigma^a(\vec{x}_1)\langle O^a(\vec{x}_1)T^{i_2 j_2}(\vec{x}_2)\rangle = 0. \tag{B.22}$$

We can simplify this expression using the above identities (after contracting with auxiliary polarization vectors)

$$+ 2(\vec{\xi}_1 \cdot \vec{\xi}_2)\partial_{i_1}^{x_1}\delta(\vec{x}_1 - \vec{x}_2)\langle\xi_{2 j_1}T^{i_1 j_1}(\vec{x}_1)\rangle + 2\,\delta(\vec{x}_1 - \vec{x}_2)(\vec{\xi}_1 \cdot \vec{\xi}_2)\xi_2^l\partial_l\sigma^a(\vec{x}_1)\langle O^a(\vec{x}_1)\rangle$$
$$\partial_{i_1}\langle T^{i_1}(\vec{x}_1)T(\vec{x}_2)\rangle - \xi_1^l\partial_l^{x_1}\delta(\vec{x}_1 - \vec{x}_2)\langle\xi_{2 i_1}\xi_{2 j_1}T^{i_1 j_1}(\vec{x}_1)\rangle - \xi_{1l}\partial^l\sigma^a(\vec{x}_1)\langle O^a(\vec{x}_1)T(\vec{x}_2)\rangle = 0, \tag{B.23}$$

where we have left the contraction with external polarizations implicit when the contractions are associated to the insertion point; e.g. $T(\vec{x}_2) \equiv \xi_2^i\xi_2^j T_{ij}(\vec{x}_2)$. We can now differentiate this expression with respect to $\sigma$ to derive identities with two stress tensor insertions and arbitrarily many scalar operators. Note that the vanishing of one-point functions and of the two-point function $\langle OT\rangle$ implies that the right-hand side of $\langle TTO\rangle$ is trivial, as we found in the main text.

The next simplest example is $\langle TTOO\rangle$, for which we can derive the WT identity by taking two functional derivatives of (B.23):

$$0 = \partial_{i_1}\langle T^{i_1}(\vec{x}_1)T(\vec{x}_2)O^b(\vec{x}_3)O^c(\vec{x}_4)\rangle - \xi_1^l\partial_l^{x_1}\delta(\vec{x}_1 - \vec{x}_2)\langle\xi_{2 i_1}\xi_{2 j_1}T^{i_1 j_1}(\vec{x}_1)O^b(\vec{x}_3)O^c(\vec{x}_4)\rangle$$
$$+ 2(\vec{\xi}_1 \cdot \vec{\xi}_2)\partial_{i_1}^{x_1}\delta(\vec{x}_1 - \vec{x}_2)\langle\xi_{2 j_1}T^{i_1 j_1}(\vec{x}_1)O^b(\vec{x}_3)O^c(\vec{x}_4)\rangle$$
$$+ 2(\vec{\xi}_1 \cdot \vec{\xi}_2)\,\delta(\vec{x}_1 - \vec{x}_2)\big(\xi_2^l\partial_l^{x_1}\delta(\vec{x}_1 - \vec{x}_3)\langle O^b(\vec{x}_1)O^c(\vec{x}_4)\rangle$$
$$+ \xi_2^l\partial_l^{x_1}\delta(\vec{x}_1 - \vec{x}_4)\langle O^c(\vec{x}_1)O^b(\vec{x}_3)\rangle\big) - \xi_{1l}\partial_{x_1}^l\delta(\vec{x}_1 - \vec{x}_3)\langle O^b(\vec{x}_1)T(\vec{x}_2)O^c(\vec{x}_4)\rangle$$
$$- \xi_{1l}\partial_{x_1}^l\delta(\vec{x}_1 - \vec{x}_4)\langle O^c(\vec{x}_1)T(\vec{x}_2)O^b(\vec{x}_3)\rangle. \tag{B.24}$$

We now have to account for the subtleties related to the definition of the stress tensor. Notice that when evaluated for $\gamma_{ij} = \delta_{ij}$, (B.21) reduces to an unimportant rescaling of $T^{ij}$. For there to be a nontrivial contribution, we must differentiate the $\gamma_{ij}$ factors before setting them to the background, which leads to a contribution of the form

$$\left.\frac{\delta\hat{T}^{ij}(\vec{x})}{\delta\gamma_{kl}(\vec{y})}\right|_{\gamma_{ij}=\delta_{ij}} = -\frac{c}{2}\delta(\vec{x} - \vec{y})\big[\mathbb{1}^{kl,im}T_m^j + \mathbb{1}^{kl,jm}T_m^i\big]. \tag{B.25}$$

The only term affected by this consideration is the first term in (B.15), which adds a term in the final identity (B.24) of the form

$$-\frac{c}{2}\partial_i^{x_1}\delta(\vec{x}_1 - \vec{x}_2)\left[\xi_2^i\langle\xi_1^k\xi_2^l T_{kl}(\vec{x}_2)O^b(\vec{x}_3)O^c(\vec{x}_4)\rangle + (\vec{\xi}_1\cdot\vec{\xi}_2)\langle\xi_2^m T_m^i(\vec{x}_2)O^b(\vec{x}_3)O^c(\vec{x}_4)\rangle\right].$$
(B.26)

All we then have to do is to transform the sum of (B.24) and (B.26) to Fourier space:

$$
\begin{aligned}
k_1^i\langle T_{\vec{k}_1}^i T_{\vec{k}_2} O_{\vec{k}_3}^b O_{\vec{k}_4}^c\rangle = {}& -(\vec{\xi}_1\cdot\vec{k}_2)\xi_2^i\xi_2^j\langle T_{\vec{k}_2+\vec{k}_1}^{ij} O_{\vec{k}_3}^b O_{\vec{k}_4}^c\rangle + 2(\vec{\xi}_1\cdot\vec{\xi}_2)\xi_2^j k_2^i\langle T_{\vec{k}_2+\vec{k}_1}^{ij} O_{\vec{k}_3}^b O_{\vec{k}_4}^c\rangle \\
& -(\vec{\xi}_1\cdot\vec{k}_3)\langle O_{\vec{k}_3+\vec{k}_1}^b T_{\vec{k}_2} O_{\vec{k}_4}^c\rangle - (\vec{\xi}_1\cdot\vec{k}_4)\langle O_{\vec{k}_4+\vec{k}_1}^c T_{\vec{k}_2} O_{\vec{k}_3}^b\rangle \\
& +2(\vec{\xi}_1\cdot\vec{\xi}_2)(\vec{\xi}_2\cdot\vec{k}_3)\langle O_{\vec{k}_3+\vec{k}_2+\vec{k}_1}^b O_{\vec{k}_4}^c\rangle + 2(\vec{\xi}_1\cdot\vec{\xi}_2)(\vec{\xi}_2\cdot\vec{k}_4)\langle O_{\vec{k}_4+\vec{k}_2+\vec{k}_1}^c O_{\vec{k}_3}^b\rangle \\
& -\frac{c}{2}(\vec{\xi}_2\cdot\vec{k}_1)\xi_1^i\xi_2^j\langle T_{\vec{k}_2+\vec{k}_1}^{ij} O_{\vec{k}_3}^b O_{\vec{k}_4}^c\rangle - \frac{c}{2}(\vec{\xi}_1\cdot\vec{\xi}_2)\xi_2^i k_1^j\langle T_{\vec{k}_2+\vec{k}_1}^{ij} O_{\vec{k}_3}^b O_{\vec{k}_4}^c\rangle.
\end{aligned}
$$
(B.27)

The identity used in §5.3.2 is obtained from this one by permuting $2\leftrightarrow 3$.

**Three stress tensors**    As another example, we consider the WT identity for $\langle TTT\rangle$, which is relevant in §4.3.3. Setting the sources for $J$ and $O$ to zero, the fundamental identity (B.15) takes the form

$$\partial_i\langle T^{ij}(\vec{x}_1)\rangle + \Gamma_{ik}^i(\vec{x}_1)\langle T^{kj}(\vec{x}_1)\rangle + \Gamma_{ik}^j(\vec{x}_1)\langle T^{ik}(\vec{x}_1)\rangle = 0.$$
(B.28)

We have to differentiate this identity twice with respect to $\gamma_{ij}$. Due to the vanishing of the one-point function, second functional derivatives acting on the Christoffel symbols will not contribute to the final answer, which simplifies the algebra, so that we get

$$
\begin{aligned}
0 = {}& \partial_{i_1}\langle T^{i_1}(\vec{x}_1)T(\vec{x}_2)T(\vec{x}_3)\rangle - \xi_1^l\partial_l^{x_1}\delta(\vec{x}_1-\vec{x}_2)\langle\xi_{2i_1}\xi_{2j_1}T^{i_1j_1}(\vec{x}_1)T(\vec{x}_3)\rangle \\
& +2(\vec{\xi}_1\cdot\vec{\xi}_2)\partial_{i_1}^{x_1}\delta(\vec{x}_1-\vec{x}_2)\langle\xi_{2j_1}T^{i_1j_1}(\vec{x}_1)T(\vec{x}_3)\rangle \\
& -\xi_1^l\partial_l^{x_1}\delta(\vec{x}_1-\vec{x}_3)\langle\xi_{3i_1}\xi_{3j_1}T^{i_1j_1}(\vec{x}_1)T(\vec{x}_2)\rangle \\
& +2(\vec{\xi}_1\cdot\vec{\xi}_3)\partial_{i_1}^{x_1}\delta(\vec{x}_1-\vec{x}_3)\langle\xi_{3j_1}T^{i_1j_1}(\vec{x}_1)T(\vec{x}_2)\rangle.
\end{aligned}
$$
(B.29)

We again have to account for the field-redefinition ambiguity (B.21), which enters in essentially the same way, leading to an additional contribution of the form

$$
\begin{aligned}
& -\frac{c}{2}\partial_i^{x_1}\delta(\vec{x}_1-\vec{x}_2)\left[\xi_2^i\langle\xi_1^k\xi_2^l T_{kl}(\vec{x}_2)T(\vec{x}_3)\rangle + (\vec{\xi}_1\cdot\vec{\xi}_2)\langle\xi_2^m T_m^i(\vec{x}_2)T(\vec{x}_3)\rangle\right] \\
& -\frac{c}{2}\partial_i^{x_1}\delta(\vec{x}_1-\vec{x}_3)\left[\xi_3^i\langle\xi_1^k\xi_3^l T_{kl}(\vec{x}_3)T(\vec{x}_2)\rangle + (\vec{\xi}_1\cdot\vec{\xi}_3)\langle\xi_3^m T_m^i(\vec{x}_3)T(\vec{x}_2)\rangle\right].
\end{aligned}
$$
(B.30)

Putting (B.29) and (B.30) together and Fourier transforming, we obtain

$$
\begin{aligned}
k_i\langle T_{\vec{k}_1}^i T_{\vec{k}_2} T_{\vec{k}_3}\rangle = {}& -(\vec{\xi}_1\cdot\vec{k}_2)\xi_2^i\xi_2^j\langle T_{\vec{k}_2+\vec{k}_1}^{ij} T_{\vec{k}_3}\rangle + 2(\vec{\xi}_1\cdot\vec{\xi}_2)\xi_2^j k_2^i\langle T_{\vec{k}_2+\vec{k}_1}^{ij} T_{\vec{k}_3}\rangle \\
& -(\vec{\xi}_1\cdot\vec{k}_3)\xi_3^i\xi_3^j\langle T_{\vec{k}_2} T_{\vec{k}_3+\vec{k}_1}^{ij}\rangle + 2(\vec{\xi}_1\cdot\vec{\xi}_3)\xi_3^j k_3^i\langle T_{\vec{k}_2} T_{\vec{k}_3+\vec{k}_1}^{ij}\rangle \\
& -\frac{c}{2}(\vec{k}_1\cdot\vec{\xi}_2)\xi_1^i\xi_2^j\langle T_{\vec{k}_2+\vec{k}_1}^{ij} T_{\vec{k}_3}\rangle - \frac{c}{2}(\vec{\xi}_1\cdot\vec{\xi}_2)k_1^i\xi_2^j\langle T_{\vec{k}_2+\vec{k}_1}^{ij} T_{\vec{k}_3}\rangle \\
& -\frac{c}{2}(\vec{k}_1\cdot\vec{\xi}_3)\xi_1^i\xi_3^j\langle T_{\vec{k}_2} T_{\vec{k}_1+\vec{k}_3}^{ij}\rangle - \frac{c}{2}(\vec{\xi}_1\cdot\vec{\xi}_3)k_1^i\xi_3^j\langle T_{\vec{k}_2} T_{\vec{k}_1+\vec{k}_3}^{ij}\rangle.
\end{aligned}
$$
(B.31)

This is precisely the identity we solved in §4.3.3 with $c = -2$, which corresponds to the natural bulk inflationary choice of graviton field variables.

**Trace identity**

We next consider the Ward–Takahashi identity coming from the fact that the stress tensor is traceless. From the bulk perspective, this identity follows from the Hamiltonian constraint

$$H\Psi[\gamma_{ij}, A_i^B, \sigma^a] = 0, \tag{B.32}$$

where $H$ is the Hamiltonian associated to the bulk fields. This equation is often also called the Wheeler–DeWitt equation. Given an explicit form of the bulk Hamiltonian, the late-time limit of this equation can be approximated as [141, 143]

$$\left[2\gamma_{ij}\frac{\delta}{\delta\gamma_{ij}} - \Delta_-\sigma^a\frac{\delta}{\delta\sigma^a}\right]\Psi = 0, \tag{B.33}$$

where we have written the coefficient of the scalar functional derivative as $\Delta_-$ to emphasize that it is the weight of the late-time bulk field profile. The equation (B.33) can also be derived from a purely boundary perspective by demanding invariance of the generating function under the following Weyl transformation [89]

$$\delta\gamma_{ij} = 2\Omega(\vec{x})\gamma_{ij}, \tag{B.34}$$

$$\delta\sigma^a = -\Delta_-\Omega(\vec{x})\sigma^a, \tag{B.35}$$

where the vector current source does not transform.

Recalling the definitions (B.1)–(B.3), we can write (B.33) as

$$\langle T(\vec{x})\rangle - (3 - \Delta_a)\sigma^a\langle O^a(\vec{x})\rangle = 0, \tag{B.36}$$

where $T$ is the trace of the stress tensor (not to be confused with index-free notation) and we have rewritten $\Delta_- = 3 - \Delta_a$ in terms of the weight of the boundary operator $O^a$. We can now use this master equation in the same way as (B.15) by taking functional derivatives and then setting the sources to zero. Much like for the current conservation identity, additional stress tensor insertions must be treated with care, but for our purposes we will not need these identities.

The only trace WT identity that we need in the main text is for the correlator $\langle TOO\rangle$. This can be obtained by differentiating (B.36) twice with respect to $\sigma$:

$$\langle T(\vec{x}_1)O^a(\vec{x}_2)O^b(\vec{x}_3)\rangle = (3 - \Delta)\left[\delta(\vec{x}_1 - \vec{x}_2)\langle O^a(\vec{x}_1)O^b(\vec{x}_3)\rangle + \delta(\vec{x}_1 - \vec{x}_3)\langle O^b(\vec{x}_1)O^a(\vec{x}_2)\rangle\right], \tag{B.37}$$

where we have set the sources to zero and used the fact that the weights have to be equal for the two-point function to be non-vanishing. Transforming to Fourier space, we obtain

$$\langle T_{\vec{k}}O_{\vec{k}_2}^a O_{\vec{k}_3}^b\rangle = (3 - \Delta)\left[\langle O_{\vec{k}_2+\vec{k}_1}^a O_{\vec{k}_3}^b\rangle + \langle O_{\vec{k}_2}^a O_{\vec{k}_3+\vec{k}_1}^b\rangle\right]. \tag{B.38}$$

This is the trace identity required to evaluate the longitudinal parts of the $\langle TOTO\rangle$ Ward–Takahashi identity (175).

# C  Spinor Helicity Formalism

The complexity of the scattering amplitudes of massless particles decreases dramatically in spinor helicity variables [144–147]. Similar simplifications occur for massless spinning particles in de Sitter space [42]. In this appendix, we review the basics of the spinor helicity formalism both in flat space and in de Sitter space.[60]

---

[60]For other approaches to spinor helicity variables in de Sitter space, see [148–151].

## C.1 Flat Space

We will begin with a discussion of the spinor helicity formalism in flat space. Although this is textbook material, we have included it for the benefit of the non-expert reader and to provide an easy comparison with the more non-standard treatment in de Sitter space. We will follow closely the excellent treatment in [45].

**Spinor helicity variables**

Given the momentum four-vector $p^\mu$, we can define the following two-by-two matrix

$$p_{\alpha\dot\alpha} = p_\mu \sigma^\mu_{\alpha\dot\alpha} = \begin{pmatrix} p_0 + p_3 & p_1 - ip_2 \\ p_1 + ip_2 & p_0 - p_3 \end{pmatrix}, \tag{C.1}$$

where $\sigma^\mu = (1, \vec\sigma)$ is a four-vector of Pauli matrices. For massless on-shell particles, the determinant of this matrix vanishes

$$\det(p) = p^\mu p_\mu = 0. \tag{C.2}$$

The matrix $p_{\alpha\dot\alpha}$ is therefore rank one and can be decomposed as the outer product of two commuting spinors

$$p_{\alpha\dot\alpha} = \lambda_\alpha \tilde\lambda_{\dot\alpha}, \tag{C.3}$$

where $\lambda_\alpha$ and $\tilde\lambda_{\dot\alpha}$ are called holomorphic and anti-holomorphic, respectively. We typically consider complex momenta, corresponding to complexifying the Lorentz group to two copies of $SL(2, \mathbb{C})$. In this case, the spinors $\lambda_\alpha$ and $\tilde\lambda_{\dot\alpha}$ are independent. For real momenta, $p_{\alpha\dot\alpha}$ is Hermitian and we must have $\tilde\lambda_{\dot\alpha} = \pm(\lambda^*)_{\dot\alpha}$, where the sign corresponds to the sign of the energy of the associated four-momentum. Parity exchanges $\lambda_\alpha$ and $\tilde\lambda_{\dot\alpha}$.

Lorentz-invariant building blocks are constructed by contracting the spinor indices using the Levi-Civita tensors $\epsilon^{\alpha\beta}$ and $\epsilon^{\dot\alpha\dot\beta}$.[61] Denoting two particles by $a$ and $b$, we define the following "angle" and "square" brackets

$$\langle ab \rangle \equiv \lambda_\alpha^a \lambda_\beta^b \epsilon^{\alpha\beta}, \tag{C.4}$$

$$[ab] \equiv \tilde\lambda_{\dot\alpha}^a \tilde\lambda_{\dot\beta}^b \epsilon^{\dot\alpha\dot\beta}. \tag{C.5}$$

When we need to raise and lower spinor indices, we do so with the convention of contracting with the first index of the epsilon symbol, for example $\lambda_\alpha = \epsilon_{\beta\alpha}\lambda^\beta$. For massless particles in four dimensions, all kinematical information can be expressed in these angle and square brackets. For example, the Mandelstam invariants are

$$s_{ab} \equiv -(p_a + p_b)^2 = -2p_a \cdot p_b = \langle ab \rangle [ab]. \tag{C.6}$$

The spinor brackets satisfy a number of important identities. First, by definition, they obey $\langle ab \rangle = -\langle ba \rangle$ and $[ab] = -[ba]$, which implies $\langle aa \rangle = [aa] = 0$. Second, any spinor can be written as a linear combination of two linearly independent spinors:[62]

$$\langle ab \rangle \lambda_c + \langle ca \rangle \lambda_b + \langle bc \rangle \lambda_a = 0, \tag{C.7}$$

which is called the *Schouten identity*. Finally, momentum conservation implies

$$\sum_{a=1}^n p_a^\mu = 0 \quad \Leftrightarrow \quad \sum_{a=1}^n \lambda_a^\alpha \tilde\lambda_a^{\dot\alpha} = 0 \quad \Leftrightarrow \quad \sum_{a=1}^n \langle ca \rangle [ad] = 0, \tag{C.8}$$

for arbitrary $\lambda_c$ and $\tilde\lambda_d$.

---

[61]We adopt the convention that $\epsilon^{12} = -\epsilon^{21} = \epsilon_{21} = -\epsilon_{12} = 1$.

[62]This follows simply from the fact that each of the spinors defines a two-dimensional vector, and it is not possible for three 2-vectors to be linearly independent.

**Little group scaling**

The subset of Lorentz transformations that leave the momentum of a particle invariant is the little group. Under the little group, the spinor helicity variables transform as

$$\lambda_a \to r_a \lambda_a \,,$$
$$\tilde{\lambda}_a \to r_a^{-1} \tilde{\lambda}_a \,. \tag{C.9}$$

For real momenta, this transformation is constrained to be a pure phase. Particles with spin correspond to nontrivial representations of the little group which carry helicity quantum numbers. In spinor helicity variables, we write polarization vectors as

$$\xi^+_{\alpha\dot{\alpha}} = \frac{\eta_\alpha \tilde{\lambda}_{\dot{\alpha}}}{\langle \eta \lambda \rangle} \quad \text{and} \quad \xi^-_{\alpha\dot{\alpha}} = \frac{\lambda_\alpha \tilde{\eta}_{\dot{\alpha}}}{[\lambda \eta]} \,, \tag{C.10}$$

where the subscripts $\pm$ label the helicity. The reference spinors $\eta$ and $\tilde{\eta}$ must be linearly independent of $\lambda$ and $\tilde{\lambda}$, but are otherwise arbitrary. Any change of the reference spinors corresponds to a gauge transformation. Under the little group, the polarization vectors transform as

$$\xi^+_{\alpha\dot{\alpha}} \to r^{-2} \xi^+_{\alpha\dot{\alpha}} \quad \text{and} \quad \xi^-_{\alpha\dot{\alpha}} \to r^2 \xi^-_{\alpha\dot{\alpha}} \,. \tag{C.11}$$

Contracting the output of Feynman diagrams with polarization vectors gives amplitudes with the correct helicity weights.

Amplitudes are Lorentz-invariant, but little group covariant. This means that they must be built out of the scalar quantities $\langle ab \rangle$ and $[ab]$, and have the correct scaling weight under the rescaling (C.9):

$$A(1^{h_1} \cdots n^{h_n}) \to \prod_a r_a^{-2h_a} A(1^{h_1} \cdots n^{h_n}) \,, \tag{C.12}$$

where $h_a$ denotes the helicity quantum number of the $a^{\text{th}}$ particle. Notice that for real momenta the rescaling is a pure phase and doesn't affect observables (which only depend on squares of amplitudes). Despite not being observable, little group covariance is an important constraint dictating the structure of scattering amplitudes for massless particles. In the following, we provide a few examples of the power of the spinor helicity formalism.

**Three-particle amplitudes**

Using momentum conservation, $p_1 + p_2 + p_3 = 0$, it is easy to show that all three-particle amplitudes vanish on-shell for real momenta. Nonzero amplitudes therefore require complex momenta in only two possible kinematic configurations: *i*) all square brackets vanish, or *ii*) all angle brackets vanish. Little group covariance then dictates that the most general three-particle amplitude of massless particles in four dimensions is

$$A(1^{h_1} 2^{h_2} 3^{h_3}) = \langle 12 \rangle^{h_3 - h_1 - h_2} \langle 23 \rangle^{h_1 - h_2 - h_3} \langle 31 \rangle^{h_2 - h_3 - h_1} \,, \quad h \le 0 \,, \tag{C.13}$$

where $h \equiv \sum_a h_a$. Flipping the signs of all helicities gives the same result with angle and square brackets interchanged. Important special cases are:

- **Scalars** The three-particle amplitude of scalars (with $h_a = 0$) is simply a constant

$$A(1_A 2_B 3_C) = \omega_{ABC} \,, \tag{C.14}$$

  where we have allowed for the possibility of multiple distinct scalars, as indicated by the subscripts. This result corresponds to a cubic potential term in the Lagrangian, while all derivative interactions vanish on-shell for massless particles.

- **Vectors** The three-particle amplitude of identical vectors (with $h_a = \pm 1$) vanishes by Bose symmetry. Allowing for multiple vector species, we get

$$A(1_A^- 2_B^- 3_C^+) = f_{ABC} \frac{\langle 12 \rangle^3}{\langle 13 \rangle \langle 32 \rangle}, \qquad A(1_A^- 2_B^- 3_C^-) = f_{ABC} \langle 12 \rangle \langle 23 \rangle \langle 31 \rangle, \qquad (C.15)$$

where $f_{ABC}$ must be antisymmetric in its indices.

- **Tensors** The three-particle amplitude of identical gravitons (with $h_a = \pm 2$) is

$$A(1^{-2} 2^{-2} 3^{+2}) = \frac{\langle 12 \rangle^6}{\langle 13 \rangle^2 \langle 32 \rangle^2}, \qquad A(1^{-2} 2^{-2} 3^{-2}) = \langle 12 \rangle^2 \langle 23 \rangle^2 \langle 31 \rangle^2. \qquad (C.16)$$

Note that after stripping the color factor, the graviton amplitudes (C.16) are the squares of the corresponding Yang–Mills amplitudes (C.15).

**Four-particle amplitudes**

While three-particle amplitudes are completely fixed by kinematics, four-particle amplitudes must satisfy additional constraints. One important extra requirement is *locality*, which is encoded in the singularity structure of the amplitude. For example, in the $s$-channel, amplitudes can have at most simple poles, $1/S$. Moreover, on the pole, the four-particle amplitude must factorize into a product of on-shell three-particle amplitudes. At tree-level, this means

$$\lim_{S \to 0} S A_4 = A_3 A_3. \qquad (C.17)$$

Consistent factorization is a remarkably efficient way to bootstrap four-particle amplitudes from three-particles amplitudes alone. In the following, we illustrate this in a number of important examples:

- **Scalars** For $\phi^3$ theory, the four-particle amplitude is

$$A_4 = S^{-1} + T^{-1} + U^{-1}. \qquad (C.18)$$

For derivative interactions, the four-particle amplitudes are all regular (corresponding to contact interactions), consistent with the fact that associated three-particle amplitudes are zero.

- **Vectors** The most general four-particle amplitude of massless vectors consistent with little group covariance, permutation symmetry and dimensional analysis is

$$A(1_A^- 2_B^- 3_C^+ 4_D^+) = \langle 12 \rangle^2 [34]^2 \left( \frac{c_1}{ST} + \frac{c_2}{TU} + \frac{c_3}{US} \right). \qquad (C.19)$$

The correct factorization in all channels implies

$$c_1 - c_3 = \sum_E f_{ABE} f_{CDE}, \qquad (C.20)$$

$$c_2 - c_1 = \sum_E f_{BCE} f_{ADE}, \qquad (C.21)$$

$$c_3 - c_2 = \sum_E f_{CAE} f_{BDE}. \qquad (C.22)$$

The sum of these relations is

$$\sum_E f_{ABE} f_{CDE} + f_{BCE} f_{ADE} + f_{CAE} f_{BDE} = 0, \qquad (C.23)$$

which we recognize as the *Jacobi identity*.

- **Tensors** The most general four-particle amplitude of massless tensors consistent with little group covariance, permutation symmetry and dimensional analysis is

$$A(1^{-2}2^{-2}3^{+2}4^{+2}) = \langle 12 \rangle^4 [34]^4 \frac{1}{STU}. \tag{C.24}$$

It is straightforward to check that this answer factorizes correctly on all poles, although this was not used in its derivation.

## C.2 De Sitter Space

In slightly modified form the spinor helicity formalism also applies in de Sitter space, where it leads to similarly dramatic simplifications for the correlators of massless fields. In this case, the relevant variables are spinors of the complexified three-dimensional group of rotations, $SL(2, \mathbb{C})$. In §3.4, we introduced a set of spinor variables adapted to this group of rotations directly. However, in order to match back to bulk physics, it is useful to understand the sense in which boundary spinor helicity variables are induced from the flat-space construction. Our review will follow closely the treatment in [42, 98].

**Spinor helicity variables**

The primary difference between the flat space and cosmology is that there is a preferred foliation in cosmology. Moreover, energy is no longer conserved. To apply the spinor helicity formalism, we embed the three-momentum vector $\vec{k}$ into a null four-momentum vector

$$k_\mu = (k, \vec{k}), \tag{C.25}$$

where $k \equiv |\vec{k}|$. Using this four-momentum, we can define spinors as we did in flat space:

$$k_{\alpha\dot\alpha} = k_\mu \sigma^\mu_{\alpha\dot\alpha} = \lambda_\alpha \tilde\lambda_{\dot\alpha}. \tag{C.26}$$

Like in flat space, contractions with the Levi–Civita tensors give Lorentz-invariant spinor brackets, $\langle \lambda\lambda \rangle$ and $[\tilde\lambda\tilde\lambda]$. Unlike in flat space, there is only one physically meaningful $SL(2, \mathbb{C})$, so it is possible to define a contraction between $\lambda$ and $\tilde\lambda$. To see this, we introduce a time-like vector normal to the foliation of the spacetime [152], $\tau^\mu = (1, \vec{0})$, which in spinor variables reads $\tau_{\alpha\dot\alpha} = \tau_\mu \sigma^\mu_{\alpha\dot\alpha} = -\mathbb{1}_{\alpha\dot\alpha}$. This tensor provides an identification between the two $SL(2, \mathbb{C})$ groups, allowing us to convert dotted indices into un-dotted indices, and vice versa.

In order to convert indices, it is useful to introduce the tensor

$$\tau^{\dot\alpha}_{\ \alpha} = -\epsilon^{\dot\alpha\dot\beta} \mathbb{1}_{\dot\beta\alpha}. \tag{C.27}$$

In order to avoid confusion related to the fact that $\epsilon^{\alpha\beta} = -\epsilon^{\alpha\rho}\epsilon^{\beta\sigma}\epsilon_{\rho\sigma}$, it is convenient to use the tensor (C.27) to convert dotted to undotted indices, and then never again consider dotted indices. Using this tensor, we define a new set of *barred* spinors related to the tilde spinors:

$$\bar\lambda_\alpha \equiv \tilde\lambda_{\dot\alpha} \tau^{\dot\alpha}_{\ \alpha}. \tag{C.28}$$

Using this convention, we see that to a three-momentum, we associate the spinors

$$\lambda_\alpha \bar\lambda_\beta = k_i (\hat\sigma^i)_{\alpha\beta} + k\epsilon_{\alpha\beta}, \tag{C.29}$$

where $\hat\sigma^i_{\alpha\beta} \equiv \sigma^i_{\alpha\dot\alpha} \tau^{\dot\alpha}_{\ \beta} = (\sigma_z, -i\mathbb{1}, -\sigma_x)_{\alpha\beta}$. Comparing with (41), we see that this is precisely the set of natural $SL(2, \mathbb{C})$ spinors introduced there, with $\bar\lambda_\beta = \epsilon_{\alpha\beta}\bar\lambda^\alpha$.

Since there is now only one set of indices for all spinors, in addition to the usual brackets

$$\langle ab \rangle \equiv \epsilon^{\alpha\beta} \lambda^a_\alpha \lambda^b_\beta \,, \langle \bar{a}\bar{b} \rangle \equiv \epsilon^{\alpha\beta} \bar{\lambda}^a_\alpha \bar{\lambda}^b_\beta \,, \tag{C.30}$$

there exists a pairing between barred and un-barred spinors

$$\langle a\bar{b} \rangle \equiv \epsilon^{\alpha\beta} \lambda^a_\alpha \bar{\lambda}^b_\beta \,. \tag{C.31}$$

If we consider this bracket between the two spinors associated to a given momentum, we isolate the energy component

$$\langle \lambda\bar{\lambda} \rangle = -2k \,. \tag{C.32}$$

The fact that is is possible to isolate the energy component is a reflection of the fact that the setup is not Lorentz invariant any longer. We now turn to deriving some identities which are useful in simplifying spinor expressions.

**Momentum conservation**

In the cosmological background, we no longer have energy conservation, but momentum is still conserved. This implies that the sum of spinors is proportional to the total energy

$$\sum_{a=1}^{n} \lambda^a_\alpha \bar{\lambda}^a_\beta = E\,\epsilon_{\alpha\beta} \,, \tag{C.33}$$

where $E \equiv \sum_a k_a$. Contracting this identity with other spinors, we obtain the following identities:

$n = 3$:

$$\begin{aligned}
\langle ba \rangle \langle \bar{a}c \rangle &= E\langle b\bar{c} \rangle \,, \\
\langle ba \rangle \langle \bar{a}c \rangle &= (E - 2k_c)\langle bc \rangle \,, \\
\langle \bar{b}a \rangle \langle \bar{a}c \rangle &= (E - 2k_b)\langle \bar{b}c \rangle \,, \\
\langle \bar{b}a \rangle \langle \bar{a}c \rangle &= (E - 2k_b - 2k_c)\langle \bar{b}c \rangle \,, \\
\langle ba \rangle \langle \bar{a}b \rangle + \langle bc \rangle \langle \bar{c}b \rangle &= 0 \,, \\
\langle ab \rangle \langle a\bar{b} \rangle &= E(E - 2k_c) \,, \\
\langle \bar{b}a \rangle \langle \bar{a}b \rangle &= (E - 2k_a)(E - 2k_b) = k_c^2 - (k_a - k_b)^2 \,,
\end{aligned} \tag{C.34}$$

where $a \neq b \neq c$ and $E = k_1 + k_2 + k_3$.

$n = 4$:

$$\begin{aligned}
\langle ba \rangle \langle \bar{a}c \rangle + \langle bd \rangle \langle \bar{d}c \rangle &= E\langle b\bar{c} \rangle \,, \\
\langle ba \rangle \langle \bar{a}c \rangle + \langle bd \rangle \langle \bar{d}c \rangle &= (E - 2k_c)\langle bc \rangle \,, \\
\langle \bar{b}a \rangle \langle \bar{a}c \rangle + \langle \bar{b}d \rangle \langle \bar{d}c \rangle &= (E - 2k_b)\langle \bar{b}c \rangle \,, \\
\langle \bar{b}a \rangle \langle \bar{a}c \rangle + \langle \bar{b}d \rangle \langle \bar{d}c \rangle &= (E - 2k_b - 2k_c)\langle \bar{b}c \rangle \,, \\
\langle ba \rangle \langle \bar{a}b \rangle + \langle bc \rangle \langle \bar{c}b \rangle + \langle bd \rangle \langle \bar{d}b \rangle &= 0 \,, \\
\langle ba \rangle \langle a\bar{b} \rangle - \langle cd \rangle \langle \bar{d}c \rangle &= E(2k_c + 2k_d - E) \,, \\
\langle \bar{b}a \rangle \langle \bar{a}b \rangle - \langle \bar{c}d \rangle \langle \bar{d}c \rangle &= (k_c - k_b)^2 - (k_a - k_b)^2 \,,
\end{aligned} \tag{C.35}$$

where $a \neq b \neq c \neq d$ and $E = k_1 + k_2 + k_3 + k_4$.

**Polarization vectors**

When defining polarization vectors in the cosmological context, it is convenient to chose a gauge where the zero component vanishes. This amounts to choosing $\bar{\lambda}$ for the reference spinor $\eta$ in (C.10). Converting everything to undotted indices, the polarization vectors become

$$\xi^+_{\alpha\beta} = \frac{\bar{\lambda}_\alpha \bar{\lambda}_\beta}{2k} \quad \text{and} \quad \xi^-_{\alpha\beta} = \frac{\lambda_\alpha \lambda_\beta}{2k}. \tag{C.36}$$

Dot products between momenta and polarization vectors are

$$2k_a \cdot k_b = -\langle ab \rangle \langle \bar{a}\bar{b} \rangle, \qquad 2k_a \cdot \xi^+_b = \frac{\langle a\bar{b} \rangle \langle \bar{b}\bar{a} \rangle}{2k_b}, \qquad 2k_a \cdot \xi^-_b = \frac{\langle \bar{a}b \rangle \langle ba \rangle}{2k_b},$$

$$2\xi^+_a \cdot \xi^+_b = -\frac{\langle \bar{a}\bar{b} \rangle^2}{4k_a k_b}, \qquad 2\xi^-_a \cdot \xi^-_b = -\frac{\langle ab \rangle^2}{4k_a k_b}, \qquad 2\xi^+_a \cdot \xi^-_b = -\frac{\langle \bar{a}b \rangle^2}{4k_a k_b}. \tag{C.37}$$

In deriving these identities, we used the identity

$$v^\mu u_\mu = -\frac{1}{2} \epsilon^{\alpha_1 \alpha_2} \epsilon^{\beta_1 \beta_2} v_{\alpha_1 \beta_1} u_{\alpha_2 \beta_2}. \tag{C.38}$$

# D  Spinor Conformal Generator

In this appendix, we derive the action of the special conformal generator written in spinor helicity variables,

$$\widetilde{K}^i = 2(\sigma^i)_\alpha{}^\beta \frac{\partial^2}{\partial \lambda_\alpha \partial \bar{\lambda}^\beta}, \tag{D.1}$$

on conformally coupled scalars, spin-1 currents and the stress tensor.

## D.1  Action on $\varphi$

First, we consider the action of $\widetilde{K}$ on the dual to conformally coupled scalars $\varphi$. The relevant spinor derivatives are

$$\frac{\partial \varphi}{\partial \bar{\lambda}^\beta} = \frac{\partial k^j}{\partial \bar{\lambda}^\beta} \partial_{k^j} \varphi = \frac{1}{2}(\sigma^j)_\beta{}^\gamma \lambda_\gamma \partial_{k^j} \varphi, \tag{D.2}$$

$$\frac{\partial^2 \varphi}{\partial \lambda_\alpha \partial \bar{\lambda}^\beta} = \frac{1}{2}(\sigma^j)_\beta{}^\alpha \partial_{k^j} \varphi + \frac{1}{4} \lambda_\delta \bar{\lambda}^\gamma (\sigma^j)_\beta{}^\delta (\sigma^l)_\gamma{}^\alpha \partial_{k^j} \partial_{k^l} \varphi. \tag{D.3}$$

Contracting the two terms in (D.3) with $2(\sigma^i)_\alpha{}^\beta$, we get

$$(\sigma^i)_\alpha{}^\beta (\sigma^j)_\beta{}^\alpha \partial_{k^j} = 2\partial_{k_i}, \tag{D.4}$$

$$\frac{1}{2} \lambda_\delta \bar{\lambda}^\gamma (\sigma^l)_\gamma{}^\alpha (\sigma^i)_\alpha{}^\beta (\sigma^j)_\beta{}^\delta \partial_{k^i} \partial_{k^j} = 2k^j \partial_{k^j} \partial_{k_i} - k^i \partial^2_{k_j}, \tag{D.5}$$

where we have repeatedly applied the identity

$$(\sigma^i)_\alpha{}^\gamma (\sigma^j)_\gamma{}^\beta = \delta^{ij} \mathbb{1}_\alpha{}^\beta + i\epsilon^{ij}{}_k (\sigma^k)_\alpha{}^\beta. \tag{D.6}$$

We also used the definition of the spinor variables $\lambda_\alpha \bar{\lambda}^\beta = k_i (\sigma^i)_\alpha{}^\beta + k \mathbb{1}_\alpha{}^\beta$. Combining (D.4) and (D.5), we find

$$\widetilde{K}^i \varphi = -K^i \varphi, \tag{D.7}$$

where $K^i = -2\partial_{k_i} - 2k^j \partial_{k^j} \partial_{k_i} + k^i \partial_{k^j} \partial_{k_j}$ is the special conformal generator when it acts on a scalar with $\Delta = 2$, cf. (22).

## D.2 Action on $J$

Next, we derive the action of $\widetilde{K}$ on a spin-1 current $J_i$. When the current is contracted with a polarization vector, $J \equiv \xi^i J_i$, the operator $\widetilde{K}$ will act on the current and also on the polarization vector. This introduces some new features. In particular, we obtain two pieces—one proportional to the special conformal generator and a second proportional to the divergence of the current.

If the current operator has negative helicity, we write $J^- \equiv (\xi^-)^i J_i$, where the polarization vector is

$$(\xi^-)^i = \frac{(\sigma^i)_\beta{}^\alpha \lambda_\alpha \lambda^\beta}{4k} \, . \tag{D.8}$$

The spinor derivatives of this polarization vector are

$$\frac{\partial (\xi^-)^i}{\partial \bar{\lambda}^\beta} = -\frac{\lambda_\beta}{2k}(\xi^-)^i \, , \tag{D.9}$$

$$\frac{\partial (\xi^-)^i}{\partial \lambda_\alpha} = -\frac{\bar{\lambda}^\alpha}{2k}(\xi^-)^i + \frac{(\sigma^i)_\gamma{}^\delta}{4k}\big(\delta^\alpha{}_\delta \lambda^\gamma + \epsilon^{\alpha\gamma}\lambda_\delta\big) \, . \tag{D.10}$$

Using these expressions, we can can find the spinor derivatives of $J^-$. The second derivative has several contributions

$$\frac{\partial^2 J^-}{\partial \lambda_\alpha \partial \bar{\lambda}^\beta} = -\frac{\delta_\beta{}^\alpha}{2k}J^- + (\xi^-)^j \frac{\partial^2 J_j}{\partial \lambda_\alpha \partial \bar{\lambda}^\beta}$$
$$+ \frac{\lambda_\beta}{2k^2}\frac{\partial k}{\partial \lambda_\alpha}J^- - \frac{\lambda_\beta}{2k}(\xi^-)^j \frac{\partial J_j}{\partial \lambda_\alpha} - \frac{\lambda_\beta}{2k}\frac{\partial (\xi^-)^j}{\partial \lambda_\alpha}J_j + \frac{\partial (\xi^-)^j}{\partial \lambda_\alpha}\frac{\partial J_j}{\partial \bar{\lambda}^\beta} \, . \tag{D.11}$$

The first term vanishes when contracted with Pauli matrices. The second term is proportional to the action on a conformally coupled scalar, which we have already determined. Evaluating the terms in the second line of (D.11) requires a bit more work. Contracting the first two terms with $2(\sigma^i)_\alpha{}^\beta$, we obtain

$$(\sigma^i)_\alpha{}^\beta \frac{\lambda_\beta}{k^2}\frac{\partial k}{\partial \lambda_\alpha}J^- = \frac{k^i}{k^2}J^- \, , \tag{D.12}$$

$$-(\sigma^i)_\alpha{}^\beta \frac{\lambda_\beta}{k}(\xi^-)^j \frac{\partial J_j}{\partial \lambda_\alpha} = -(\xi^-)^j \frac{\partial J_j}{\partial k_i} - i\epsilon^{lim}\frac{k_m}{k}(\xi^-)^j \frac{\partial J_j}{\partial k^l} \, . \tag{D.13}$$

The last two terms in (D.11) lead to

$$2(\sigma^i)_\alpha{}^\beta \left( -\frac{\lambda_\beta}{2k}\frac{\partial (\xi^-)^j}{\partial \lambda_\alpha}J_j + \frac{\partial (\xi^-)^j}{\partial \lambda_\alpha}\frac{\partial J_j}{\partial \bar{\lambda}^\beta} \right) = (\sigma^i)_\alpha{}^\beta \left( -\frac{\lambda_\beta}{k}J_j + (\sigma^l)_\beta{}^\gamma \lambda_\gamma \frac{\partial J_j}{\partial k^l} \right)\frac{\partial (\xi^-)^j}{\partial \lambda_\alpha}$$
$$= \frac{k^i}{k^2}J^- - 2i\frac{\epsilon^{ji}{}_l}{k}(\xi^-)^l J_j - 2(\xi^-)^i \frac{\partial J_j}{\partial k_j} + (\xi^-)^j \frac{\partial J_j}{\partial k_i} + 2(\xi^-)^j \frac{\partial J^i}{\partial k^j} - i\epsilon^{im}{}_l \frac{k^l}{k}(\xi^-)^j \frac{\partial J_j}{\partial k^m} \, . \tag{D.14}$$

To arrive at the expression in the second line, we have used (D.10) and (D.6), as well as the identity

$$\epsilon^{\alpha\gamma}(\sigma^j)_\gamma{}^\delta (\sigma^i)_\alpha{}^\beta = \delta^{ji}\epsilon^{\delta\beta} + i\epsilon^{ji}{}_l(\sigma^l)_\gamma{}^\beta \epsilon^{\delta\gamma} \, . \tag{D.15}$$

Combining (D.12) – (D.14), we find that the second line of (D.11) leads to

$$2\frac{k^i}{k^2}J^- - 2i\frac{\epsilon^{ji}{}_l}{k}(\xi^-)^l J_j - 2(\xi^-)^i \frac{\partial J_j}{\partial k_j} + 2(\xi^-)^j \frac{\partial J^i}{\partial k^j} \, . \tag{D.16}$$

Adding the contribution from the first line of (D.11), we then get

$$\widetilde{K}^i J^- = 2\frac{k^i}{k^2}(\xi^-)^j J_j - 2i\frac{\epsilon^{ji}{}_l}{k}(\xi^-)^l J_j - 2(\xi^-)^i\frac{\partial J_j}{\partial k_j} + 2(\xi^-)^j\frac{\partial J^i}{\partial k^j} + (\xi^-)^j \widetilde{K}^i J_j. \qquad \text{(D.17)}$$

This expression can be further simplified. In particular, the second term in (D.17) can be rewritten by replacing the current with

$$J_j = \left[\left(\delta_j{}^m - \frac{k_j k^m}{k^2}\right) + \frac{k_j k^m}{k^2}\right] J_m = \left[-2\left((\xi^-)_j(\xi^+)^m + (\xi^+)_j(\xi^-)^m\right) + \frac{k_j k^m}{k^2}\right] J_m \qquad \text{(D.18)}$$

$$= -2\left(\xi_j^-(\xi^+)^m J_m + \xi_j^+(\xi^-)^m J_m\right) + \frac{k_j}{k^2} k^m J_m. \qquad \text{(D.19)}$$

We then have

$$-2i\frac{\epsilon^{jil}}{k}\xi_l^- J_j = \frac{4i}{k}\epsilon^{jil}\xi_l^-\xi_j^+(\xi^-)^m J_m - \frac{2i}{k^3}\epsilon^{jil}\xi_l^- k_j k^m J_m \qquad \text{(D.20)}$$

$$= -\frac{2}{k^2}k^i(\xi^-)^m J_m + \frac{2}{k^2}(\xi^-)^i k^m J_m, \qquad \text{(D.21)}$$

where we have used that[63]

$$\epsilon^{jil}\xi_l^- k_j = ik(\xi^-)^i, \qquad \text{(D.22)}$$

$$\epsilon^{jil}\xi_l^-\xi_j^+ = \frac{i}{2k}k^i. \qquad \text{(D.23)}$$

The action of $\widetilde{K}^i$ on $J^-$ then is

$$\widetilde{K}^i J^- = \frac{2}{k^2}(\xi^-)^i k^m J_m - 2(\xi^-)^i\frac{\partial J_j}{\partial k_j} + 2(\xi^-)^j\frac{\partial J^i}{\partial k^j} + (\xi^-)^j \widetilde{K}^i J_j. \qquad \text{(D.24)}$$

Since the vector current $J_j$ only depends on the momentum, we can use (D.7) to write the last term as $(\xi^-)^j\widetilde{K}^i J_j = (\xi^-)^j(2\partial_{k_i} + 2k^j\partial_{k_j}\partial_{k_i} - k^i\partial_{k_l}\partial_{k_l})J_j$. However, in contrast with the scalar case, this is not proportional to the action of the special conformal generator $K^i$ on the field. The reason is that $K^i$ contains derivatives of the polarization vectors, cf. (22) and (51). Taking the action of the special conformal generator on a vector operator into account and substituting this into (D.24), we finally get

$$\widetilde{K}^i J^- = \left(-\xi_-^j K^i + 2\xi_-^i\frac{k^j}{k^2}\right) J_j, \qquad \text{(D.25)}$$

which is the result (53) used in the main text.

## D.3 Action on $T$

The analysis for the stress tensor is conceptually similar, but algebraically slightly more involved. Since the stress tensor has dimension $\Delta = 3$, the operator $\widetilde{K}$ acts more naturally on

$$\frac{1}{k}T^- \equiv \frac{1}{k}(\xi^-)^{jl} T_{jl} = \frac{(\sigma^j)_\delta{}^\gamma \lambda_\gamma \lambda^\delta}{4k^2}(\xi^-)^l T_{jl}, \qquad \text{(D.26)}$$

---

[63]These identities can be derived by writing each vector as a linear combination of $\vec{k}$, $\vec{\xi}^-$ and $\vec{\xi}^+$. Contracting the free index with those same vectors then shows that $\epsilon^{jil}\xi_l^- k_j \propto (\xi^-)^i$ and $\epsilon^{jil}\xi_l^-\xi_j^+ \propto k^i$. Finally, the proportionality constants are obtained by plugging in an explicit example.

where the rewriting in the last equality is for later convenience. The relevant spinor derivatives then are

$$\frac{\partial}{\partial \bar{\lambda}^{\beta}} \left( \frac{T^-}{k} \right) = \frac{(\sigma^j)_{\delta}{}^{\gamma} \lambda_{\gamma} \lambda^{\delta}}{4} \frac{\partial k^{-2}}{\partial \bar{\lambda}^{\beta}} (\xi^-)^l T_{jl} + \frac{(\sigma^j)_{\delta}{}^{\gamma} \lambda_{\gamma} \lambda^{\delta}}{4k^2} \frac{\partial}{\partial \bar{\lambda}^{\beta}} \left( (\xi^-)^l T_{jl} \right), \qquad \text{(D.27)}$$

$$\frac{\partial^2}{\partial \lambda_{\alpha} \partial \bar{\lambda}^{\beta}} \left( \frac{T^-}{k} \right) = k \frac{\partial^2 k^{-2}}{\partial \lambda_{\alpha} \partial \bar{\lambda}^{\beta}} T^- + \frac{(\sigma^j)_{\eta}{}^{\gamma}}{4} \left( \delta^{\alpha}{}_{\gamma} \lambda^{\eta} + \epsilon^{\eta \alpha} \lambda_{\gamma} \right) \left[ \frac{\partial k^{-2}}{\partial \bar{\lambda}^{\beta}} (\xi^-)^l T_{jl} \right.$$
$$\left. + \frac{1}{k^2} \frac{\partial}{\partial \bar{\lambda}^{\beta}} \left( (\xi^-)^l T_{jl} \right) \right] + k \frac{\partial k^{-2}}{\partial \bar{\lambda}^{\beta}} (\xi^-)^j \frac{\partial}{\partial \lambda_{\alpha}} \left( (\xi^-)^l T_{jl} \right) \qquad \text{(D.28)}$$
$$+ k \frac{\partial k^{-2}}{\partial \lambda_{\alpha}} (\xi^-)^j \frac{\partial}{\partial \bar{\lambda}^{\beta}} \left( (\xi^-)^l T_{jl} \right) + \frac{1}{k} (\xi^-)^j \frac{\partial^2}{\partial \lambda_{\alpha} \partial \bar{\lambda}^{\beta}} \left( (\xi^-)^l T_{jl} \right).$$

The single derivatives on $k^{-2}$ are

$$\frac{\partial k^{-2}}{\partial \bar{\lambda}^{\beta}} = -\frac{\lambda_{\beta}}{k^3}, \qquad \text{(D.29)}$$

$$\frac{\partial k^{-2}}{\partial \lambda_{\alpha}} = -\frac{\bar{\lambda}^{\alpha}}{k^3}, \qquad \text{(D.30)}$$

and the double derivative, after contracting with $2(\sigma^i)_{\alpha}{}^{\beta}$, gives

$$2(\sigma^i)_{\alpha}{}^{\beta} \frac{\partial^2 k^{-2}}{\partial \lambda_{\alpha} \partial \bar{\lambda}^{\beta}} = \frac{6}{k^4} k^i. \qquad \text{(D.31)}$$

The action of $\partial / \partial \bar{\lambda}^{\beta}$ on $(\xi^-)^l T_{jl}$ is easy to compute using (D.9), and the double derivative, after contracting with $2(\sigma^i)_{\alpha}{}^{\beta}$, is given by (D.25). What remains therefore is to compute the derivative with respect to $\lambda_{\alpha}$, using (D.10):

$$\frac{\partial}{\partial \lambda_{\alpha}} \left( (\xi^-)^l T_{jl} \right) = \frac{1}{4k} \left( -2\bar{\lambda}^{\alpha} (\xi^-)^l + \lambda^{\gamma} (\sigma^l)_{\gamma}{}^{\alpha} + \epsilon^{\alpha \gamma} (\sigma^l)_{\gamma}{}^{\delta} \lambda_{\delta} \right) T_{jl} + (\xi^-)^l \frac{\partial T_{jl}}{\partial \lambda_{\alpha}}. \qquad \text{(D.32)}$$

After some algebra, we find

$$\widetilde{K}^i \left( \frac{T^-}{k} \right) = \frac{10}{k^3} k^i T^- - 2(\sigma^i)_{\alpha}{}^{\beta} \frac{\lambda_{\beta}}{k^2} (\xi^-)^{jl} \frac{\partial T_{jl}}{\partial \lambda_{\alpha}}$$
$$+ \frac{(\sigma^i)_{\alpha}{}^{\beta} (\sigma^j)_{\delta}{}^{\gamma}}{2k^2} \left( \delta^{\alpha}{}_{\gamma} \lambda^{\delta} + \epsilon^{\delta \alpha} \lambda_{\gamma} \right) (\xi^-)^l \left( -\frac{5\lambda_{\beta}}{k} T_{jl} + \frac{\partial T_{jl}}{\partial \bar{\lambda}^{\beta}} \right)$$
$$- 2(\sigma^i)_{\alpha}{}^{\beta} \frac{\bar{\lambda}^{\alpha}}{k^2} (\xi^-)^{jl} \frac{\partial T_{jl}}{\partial \bar{\lambda}^{\beta}} + \frac{1}{k} (\xi^-)^j \widetilde{K}^i \left( (\xi^-)^l T_{jl} \right). \qquad \text{(D.33)}$$

The last term is given by (D.25), but again we need to take into account that the action of the special conformal generator $K^i$ on $T_{jl}$ is different from its action on $J_j$; cf. (51). The result is

$$\widetilde{K}^i \left( (\xi^-)^l T_{jl} \right) = -(\xi^-)^l K^i T_{jl} + \frac{2}{k^2} (\xi^-)^i k^l T_{jl} + 2(\xi^-)^l \frac{\partial T_{jl}}{\partial k_i}$$
$$- 2(\xi^-)^m \frac{\partial T_j{}^i}{\partial k^m} + 2(\xi^-)^i \frac{\partial T_{jl}}{\partial k_l}. \qquad \text{(D.34)}$$

The spinor derivatives in (D.33) are

$$\frac{\partial T_{jl}}{\partial \bar{\lambda}^{\beta}} = \frac{\lambda_{\gamma}}{2} (\sigma^m)_{\beta}{}^{\gamma} \frac{\partial T_{jl}}{\partial k^m}, \qquad \text{(D.35)}$$

$$\frac{\partial T_{jl}}{\partial \lambda_{\alpha}} = \frac{\bar{\lambda}^{\gamma}}{2} (\sigma^m)_{\gamma}{}^{\alpha} \frac{\partial T_{jl}}{\partial k^m}. \qquad \text{(D.36)}$$

Substituting the previous expressions in (D.33), and performing some Pauli matrix algebra, we get

$$
\begin{aligned}
\widetilde{K}^i\left(\frac{T^-}{k}\right) = &-\frac{1}{k}(\xi^-)^{jl}K^i T_{jl} + \frac{2}{k^3}(\xi^-)^{ji}k^l T_{jl} \\
&+ \frac{10}{k^3}k^i T^- - i\frac{10}{k^2}\epsilon^{ji}{}_k(\xi^-)^{kl}T_{jl}.
\end{aligned}
\tag{D.37}
$$

To simplify the last term, we use the same trick as before and write the stress tensor as

$$
T_{jl} = \left[\left(\delta_j{}^m - \frac{k_j k^m}{k^2}\right) + \frac{k_j k^m}{k^2}\right]T_{ml} = \left[-2\left((\xi^-)_j(\xi^+)^m + (\xi^+)_j(\xi^-)^m\right) + \frac{k_j k^m}{k^2}\right]T_{ml} \quad \text{(D.38)}
$$

$$
= -2\left[\xi_j^-(\xi^+)^m T_{ml} + \xi_j^+(\xi^-)^m T_{ml}\right] + \frac{k_j}{k^2}k^m T_{ml}. \quad \text{(D.39)}
$$

This leads to

$$
\begin{aligned}
-i\frac{10}{k^2}\epsilon^{ji}{}_k(\xi^-)^{kl}T_{jl} &= i\frac{20}{k^2}\epsilon^{ji}{}_k(\xi^-)^k\xi_j^+ T^- - i\frac{10}{k^4}\epsilon^{ji}{}_k k_j(\xi^-)^{kl}k^m T_{ml} \\
&= -\frac{10}{k^3}k^i T^- + \frac{10}{k^3}(\xi^-)^{il}k^m T_{ml},
\end{aligned}
\tag{D.40}
$$

where in the last step we used (D.22) and (D.23). Substituting (D.40) into (D.37), we finally get

$$
\widetilde{K}^i\left(\frac{T^-}{k}\right) = \left(-\frac{1}{k}\xi_-^{(j}\xi_-^{l)}K^i + 12\,\xi_-^i\frac{\xi_-^{(j}k^{l)}}{k^3}\right)T_{jl},
\tag{D.41}
$$

which confirms the result (54) used in the main text.

# E  Polarization Tensors and Sums

In order to deal with correlation functions involving the exchange of operators with spin, we require explicit expressions for the polarization tensors for spinning operators. These are the numerators that appear in the two-point functions of spinning operators:

$$
\langle O^{i_1\cdots i_\ell}O_{j_1\cdots j_\ell}\rangle = \frac{(\Pi_\ell)^{i_1\cdots i_\ell}_{j_1\cdots j_\ell}}{k^{3-2\Delta}}.
\tag{E.1}
$$

Since we only consider the exchange of spin-1 and spin-2 operators, we will restrict our attention to these cases. For a more general discussion and derivation of the polarization structures we consider here, see Appendix C of [48].

## E.1  Polarization Tensors

Let us record the polarization tensors that appear in the two-point function numerator for fields of spin-1 and spin-2. These objects depend on the spatial momentum of the relevant spinning operator, $k^i$, so it is helpful to introduce the transverse projector

$$
\pi^i_j \equiv \delta^i_j - \hat{k}^i\hat{k}_j,
\tag{E.2}
$$

where $\hat{k}_i \equiv k_i/k$. It will also prove to be convenient to decompose the polarization tensors $\Pi_\ell$ into a helicity-like basis that is orthonormal, complete, and transverse in a sense that we will make precise.

- **Spin 1:** For spin 1, we split the polarization tensor into the following transverse and longitudinal components

$$(\Pi_1)^i_j = \pi^i_j + \frac{(1-\Delta)}{(\Delta-2)}\hat{k}^i\hat{k}_j. \tag{E.3}$$

The components $\pi_{ij}$ and $\hat{k}_i\hat{k}_j$ are orthonormal and form a complete basis of projectors for a vector. Further note that $k^i\pi_{ij} = 0$. For $\Delta = 2$, the coefficient of the longitudinal term diverges, which is a reflection of the fact that the corresponding operator is conserved at this point. The bulk dual of this operator is a massless vector field.

- **Spin 2:** For spin 2, we write the relevant polarization tensor as

$$(\Pi_2)^{i_1 i_2}_{j_1 j_2} = (\Pi_{2,2})^{i_1 i_2}_{j_1 j_2} + \frac{\Delta}{3-\Delta}(\Pi_{2,1})^{i_1 i_2}_{j_1 j_2} + \frac{\Delta(\Delta-1)}{(3-\Delta)(2-\Delta)}(\Pi_{2,0})^{i_1 i_2}_{j_1 j_2}, \tag{E.4}$$

where we have introduced the orthonormal basis of projectors for traceless two-index tensors

$$(\Pi_{2,2})^{i_1 i_2}_{j_1 j_2} = \pi^{(i_1}_{(j_1}\pi^{i_2)}_{j_2)} - \frac{1}{2}\pi^{i_1 i_2}\pi_{j_1 j_2}, \tag{E.5}$$

$$(\Pi_{2,1})^{i_1 i_2}_{j_1 j_2} = 2\hat{k}^{(i_1}\hat{k}_{(j_1}\pi^{i_2)}_{j_2)}, \tag{E.6}$$

$$(\Pi_{2,0})^{i_1 i_2}_{j_1 j_2} = \frac{3}{2}\left(\hat{k}^{i_1}\hat{k}^{i_2} - \frac{1}{3}\delta^{i_1 i_2}\right)\left(\hat{k}_{j_1}\hat{k}_{j_2} - \frac{1}{3}\delta_{j_1 j_2}\right). \tag{E.7}$$

These projectors have the following properties

$$\text{orthonormality}: \quad (\Pi_{2,m})^{i_1 i_2}_{j_1 j_2}(\Pi_{2,m'})^{j_1 j_2}_{l_1 l_2} = \delta_{mm'}(\Pi_{2,m})^{i_1 i_2}_{l_1 l_2}, \tag{E.8}$$

$$\text{completeness}: \quad (\Pi_{2,2})^{i_1 i_2}_{j_1 j_2} + (\Pi_{2,1})^{i_1 i_2}_{j_1 j_2} + (\Pi_{2,0})^{i_1 i_2}_{j_1 j_2} = \delta^{(i_1}_{(j_1}\delta^{i_2)_T}_{j_2)_T}, \tag{E.9}$$

$$\text{transversality}: \quad k^{j_1}(\Pi_{2,2})^{i_1 i_2}_{j_1 j_2} = k^{j_1}k^{j_2}(\Pi_{2,1})^{i_1 i_2}_{j_1 j_2} = 0, \tag{E.10}$$

where $(\cdots)_T$ denotes the traceless symmetrization of the enclosed indices. From (E.4), we see that there are two distinguished values of $\Delta$ for spin-2 operators. When $\Delta = 3$, the coefficients of the helicity-1 and helicity-0 components diverge. This signals that these states are becoming null and decouple from the theory. This corresponds to the operator being conserved if we take a single divergence. The bulk field dual to such a singly conserved spin-2 operator is the graviton. The second interesting value is $\Delta = 2$. At this point, the spin-2 operator is conserved if we take two divergences, and correspondingly we see that the coefficient of the helicity-0 mode diverges, indicating it decouples. A spin-2 operator with this weight is dual to a partially massless spin-2 field.

Although we do not require them here, these formulas have generalizations to higher spin, which can be found in [48].

## E.2 Polarization Sums

The polarization tensors (E.3) and (E.4) typically arise contracted with external vectors to form scalar polarization sums. These provide useful building blocks in the exchange of spinning operators. In this section, we collect some useful formulas involving these polarization sums. For concreteness, we present these expressions in the $s$-channel, but they can be permuted to any other channel desired.

In the $s$-channel, the polarization tensors typically occur contracted with the combinations of external momenta $\vec{\alpha} = \vec{k}_1 - \vec{k}_2$ and $\vec{\beta} = \vec{k}_3 - \vec{k}_4$, and the momentum on which they depend is $\vec{s} = \vec{k}_1 + \vec{k}_2$.

- **Spin 1:** In the spin-1 case, the two scalar polarization sums of interest are

$$\Pi_{1,1} \equiv \frac{\alpha^i \pi_{ij} \beta^j}{s^2} = \frac{k_{12} k_{34} \hat{\alpha}\hat{\beta} - (t^2 - u^2)}{s^2}, \tag{E.11}$$

$$\Pi_{1,0} \equiv -\frac{s^2}{k_{12} k_{34}} \frac{\hat{s}\cdot\vec{\alpha}\,\hat{s}\cdot\vec{\beta}}{s^2} = \hat{\alpha}\hat{\beta}, \tag{E.12}$$

where we have defined these polarization sums in relation to the orthonormal spin-1 projectors for the exchanged operator.

- **Spin 2:** The relevant spin-2 polarization sums are defined by

$$\Pi_{2,2} \equiv \frac{3}{2s^4} \alpha_i \alpha_j (\Pi_{2,2})^{ij}_{lm} \beta^l \beta^m, \tag{E.13}$$

$$\Pi_{2,1} \equiv -\frac{s^2}{k_{12} k_{34}} \frac{3}{2s^4} \alpha_i \alpha_j (\Pi_{2,1})^{ij}_{lm} \beta^l \beta^m = 3\hat{\alpha}\hat{\beta}\frac{\alpha^i \pi_{ij} \beta^j}{s^2}, \tag{E.14}$$

$$\Pi_{2,0} \equiv \frac{1}{4}(1 - 3\hat{\alpha}^2)(1 - 3\hat{\beta}^2), \tag{E.15}$$

where the polarization sum $\Pi_{2,0}$ is defined in such a way that it depends only on $\hat{\alpha} = (k_1 - k_2)/s$ and $\hat{\beta} = (k_3 - k_4)/s$.

These polarization sums are the objects that naturally arise when we contract external momenta with (E.3) and (E.4). The organizing principle in their definition is to remove the overall dependence on $w$ and $v$, so that they are purely functions of the angular variables. A further remarkable quality of these scalar sums is that the combination $s^{-1}\Pi_{\ell,m}$ solves the conformal Ward identities with $\Delta = 2$.

**Flat-space limits**

In various places in the text, we are interested in the $E \to 0$ singularities displayed by correlation functions. This limit is essentially probing the flat space limit of the bulk physics. The individual polarization sums do not have any particularly transparent interpretation in this limit, but the following combinations simplify greatly:

$$\mathcal{P}_1 \equiv s^2 \Pi_{1,1} - E_L E_R \Pi_{1,0} \xrightarrow{E\to 0} -S\,P_1\left(1 + \frac{2U}{S}\right), \tag{E.16}$$

$$\mathcal{P}_2 \equiv s^4 \Pi_{2,2} - E_L E_R s^2 \Pi_{2,1} + E_L^2 E_R^2 \Pi_{2,0} \xrightarrow{E\to 0} S^2 P_2\left(1 + \frac{2U}{S}\right), \tag{E.17}$$

where $P_\ell$ is a Legendre polynomial and recall that $E_L = k_{12} + s$ and $E_R = k_{34} + s$, while $S$ and $U$ are flat-space Mandelstam variables. Hence, we see that these combinations of polarization sums reproduce the expected flat-space angular structure given by Legendre polynomials.

# F  Compton Scattering Amplitudes

In this appendix, we present the amplitudes for Compton scattering of photons, gluons and gravitons (see Fig. 14). These results are needed to check the flat-space limits of the corresponding correlators in Sections 5 and 6.

### F.1 Spin-1 Compton Scattering

In (scalar) Compton scattering, $\gamma\phi \to \gamma\phi$, a charged scalar and a photon scatter off each other. The $s$- and $t$-channel contributions to the amplitude for this process are

$$A_s = e_2 e_4 \frac{(\epsilon_1 \cdot p_2)(\epsilon_3 \cdot p_4)}{S}, \tag{F.1}$$

$$A_t = e_2 e_4 \frac{(\epsilon_1 \cdot p_4)(\epsilon_3 \cdot p_2)}{T}. \tag{F.2}$$

These answers are not gauge invariant. They don't vanish when either $\epsilon_1$ or $\epsilon_3$ are replaced by the corresponding momenta. Even the sum of the two channels is not gauge invariant:

$$A_s + A_t \xrightarrow{\epsilon_1 \mapsto p_1} -\frac{e_2 e_4}{2}\, \epsilon_3 \cdot (p_2 + p_4) \neq 0. \tag{F.3}$$

To obtain a gauge-invariant result we must add the contact interaction $A_c = -\frac{1}{2} e_2 e_4 (\epsilon_1 \cdot \epsilon_3)$, so that the total amplitude becomes

$$A_{s+t+c} = e_2 e_4 \left( \frac{(\epsilon_1 \cdot p_2)(\epsilon_3 \cdot p_4)}{S} + \frac{(\epsilon_1 \cdot p_4)(\epsilon_3 \cdot p_2)}{T} - \frac{1}{2}(\epsilon_1 \cdot \epsilon_3) \right). \tag{F.4}$$

This is indeed gauge invariant, after using momentum conservation and transversality of the polarization vectors ($p_a \cdot \epsilon_a = 0$). From a field theory perspective, this feature of Compton scattering is, of course, expected. Minimally coupling a scalar field to a photon, the covariant derivative gives rise to both a cubic coupling $2eA^\mu \phi^* \partial_\mu \phi$ and a contact interaction $e^2 A^2 |\phi|^2$ with a precise relative coefficient.

If we have multiple gauge fields, then Compton scattering can also proceed by the exchange of the gauge field in the $u$-channel (see Fig. 14). In this case, we associate to each interaction between the gauge fields and two scalars a coupling matrix $T^A_{ab}$. The $s$- and $t$-channel amplitudes are basically the same as before, just dressed by the coupling matrices

$$A_s = T^A_{ac} T^B_{cb} \frac{(\epsilon_1 \cdot p_2)(\epsilon_3 \cdot p_4)}{S}, \tag{F.5}$$

$$A_t = T^B_{ac} T^A_{cb} \frac{(\epsilon_1 \cdot p_4)(\epsilon_3 \cdot p_2)}{T}. \tag{F.6}$$

The extra $u$-channel amplitude is

$$A_u = -\frac{1}{4} f^{ABC} T^C_{ab} \left( \frac{T-S}{U}(\epsilon_1 \cdot \epsilon_3) + \frac{4}{U}\Big[(\epsilon_1 \cdot p_3)(\epsilon_3 \cdot p_2) - (\epsilon_1 \cdot p_2)(\epsilon_3 \cdot p_1)\Big] \right). \tag{F.7}$$

As in the Abelian case, it is necessary to add a contact interaction for the final answer to be gauge invariant

$$A_c = -\frac{1}{2} T^{(A}_{ac} T^{B)}_{cb}(\epsilon_1 \cdot \epsilon_3). \tag{F.8}$$

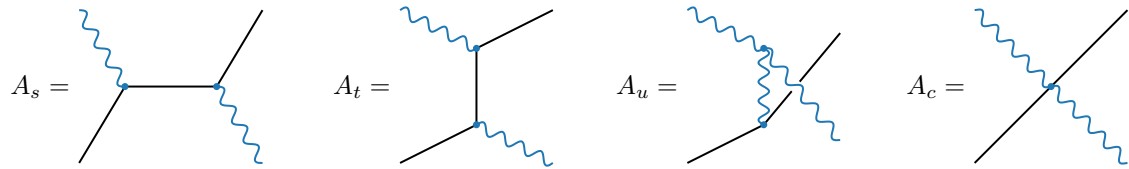

Figure 14: Feynman diagrams of the different tree-level contributions to Compton scattering.

Putting everything together, we get

$$A_{s+t+u+c} \xrightarrow{\epsilon_1 \mapsto p_1} -\frac{1}{2}[T^A, T^B]_{ab}\, \epsilon_3 \cdot \left(p_4 + \frac{p_1}{2}\right) - \frac{1}{2} f^{ABC} T^C_{ab}\, \epsilon_3 \cdot \left(p_2 + \frac{p_1}{2}\right). \tag{F.9}$$

We see that the amplitude is only gauge invariant if the coupling matrices satisfy

$$[T^A, T^B]_{ab} = f^{ABC} T^C_{ab}, \tag{F.10}$$

i.e. we require that the couplings transform in a linear representation of the gauge group [3]. In addition to this requirement, we have had to fix the relative couplings of the contributions to the amplitude rather delicately. Of course, this is expected from the Lagrangian viewpoint, where minimally coupling a scalar to gluons fixes both the three- and four-point couplings.

### F.2 Graviton Compton Scattering

Next, we record the amplitude for Compton scattering of gravitons off of scalar particles. This computation was first done in [44], but is recorded in a more convenient form in [153]. Working in de Donder gauge, the answers for the different exchange contributions and the contact interaction were found to be

$$A_s = \kappa^2 \frac{(\epsilon_1 \cdot p_2)^2 (\epsilon_3 \cdot p_4)^2}{S}, \tag{F.11}$$

$$A_t = \kappa^2 \frac{(\epsilon_1 \cdot p_4)^2 (\epsilon_3 \cdot p_2)^2}{T}, \tag{F.12}$$

$$A_u = -\frac{\kappa \kappa_g}{U} \left[ \frac{(\epsilon_1 \cdot \epsilon_3)^2}{4} (ST - U^2) + \epsilon_1 \cdot \epsilon_3 \left( \epsilon_1 \cdot p_2\, \epsilon_3 \cdot p_4\, S + \epsilon_1 \cdot p_4\, \epsilon_3 \cdot p_2\, T \right) - (\epsilon_1 \circ \epsilon_3)^2 \right], \tag{F.13}$$

$$A_c = -\kappa_c^2 \left[ \frac{(\epsilon_1 \cdot \epsilon_3)^2}{4} U + \epsilon_1 \cdot \epsilon_3 \left( \epsilon_1 \cdot p_2\, \epsilon_3 \cdot p_4 + \epsilon_1 \cdot p_4\, \epsilon_3 \cdot p_2 \right) \right], \tag{F.14}$$

where we have defined $\epsilon_1 \circ \epsilon_3 \equiv \epsilon_1 \cdot p_2\, \epsilon_3 \cdot p_4 - \epsilon_1 \cdot p_4\, \epsilon_3 \cdot p_2$, and the couplings are the same as those in Fig. 9. Note that computing this scattering amplitude in a different gauge for the graviton propagator will shift the terms appearing in $A_u$ and $A_c$ around, which is why our check of the flat-space limit in the main text only matches the sum of these two terms.

As in the vector case, we can check that gauge invariance requires adding together multiple channels with fixed couplings. Indeed, the gauge variation of the sum of channels plus the contact term takes the form (after using momentum conservation)

$$A_{s+t+u+c} \rightarrow (\kappa_c^2 - \kappa \kappa_g) \left[ \frac{T}{2}(\epsilon_3 \cdot p_1)(\epsilon_1 \cdot \epsilon_3) + \frac{T - S}{2}(\epsilon_3 \cdot p_2)(\epsilon_1 \cdot \epsilon_3) + (\epsilon_3 \cdot p_1)(\epsilon_3 \cdot p_2)(\epsilon_1 \cdot p_3) \right]$$
$$+ (\kappa_c^2 - \kappa^2) \left[ (\epsilon_3 \cdot p_1)^2 (\epsilon_1 \cdot p_2) + 2(\epsilon_3 \cdot p_1)(\epsilon_3 \cdot p_2)(\epsilon_1 \cdot p_2) \right]$$
$$+ (\kappa^2 - \kappa \kappa_g)(\epsilon_3 \cdot p_2)^2 (\epsilon_1 \cdot p_3). \tag{F.15}$$

The only way to make this vanish is to set the couplings of all particles to the graviton (including its self-coupling) equal

$$\kappa_c = \kappa_g \equiv \kappa, \tag{F.16}$$

which is of course a manifestation of the equivalence principle. Remarkably, once this has been done the sum of all contributions greatly simplifies

$$A_{T\varphi T\varphi} = -\frac{\kappa^2}{e^4} \frac{ST}{U} (A_{J\varphi J\varphi})^2, \tag{F.17}$$

where $A_{J\varphi J\varphi}$ is given by (F.4), with $e_2 e_4 = -e^2$. We see that the amplitude for gravitational Compton scattering is related to the square of the amplitude for photon Compton scattering.

# G  Derivation of Compton Correlators

In this appendix, we present alternative derivations of $\langle J\varphi J\varphi\rangle_u$ and $\langle T\varphi T\varphi\rangle_u$ that are somewhat simpler than the weight-shifting approach presented in Section 5. We also give explicit results for $\langle TT\phi\phi\rangle$, which were used in Section 7.

## G.1  $\langle J\varphi J\varphi\rangle$

Consider the four-point function of conformally coupled scalars exchanging a spin-1 current in the $u$-channel:

$$\langle\varphi\varphi\varphi\varphi\rangle_{u,J} = \left(\Pi_{1,1}^{(u)}D_{wv} + \Pi_{1,0}^{(u)}\Delta_w\right)F_{\Delta_\sigma=2}\,, \tag{G.1}$$

where $F_{\Delta_\sigma=2} = u^{-1}\hat{F}_{\Delta_\sigma=2}$. The correlator we wish to compute, $\langle J\varphi J\varphi\rangle_u$, differs from $\langle\varphi\varphi\varphi\varphi\rangle_{u,J}$ in one of the vertices, namely the Yang–Mills self-coupling of the vector field. Notice that the three-point correlators corresponding to these two different vertices, $\langle\varphi J\varphi\rangle$ and $\langle JJJ\rangle$, only have different polarization structures:

$$\langle\varphi_{\vec{k}_1}J_{\vec{k}_I}\varphi_{\vec{k}_3}\rangle = (\vec{\alpha}_u\cdot\vec{\xi}_I)f(k_1,k_I,k_3)\,, \tag{G.2}$$

$$\langle J_{\vec{k}_1}J_{\vec{k}_I}J_{\vec{k}_3}\rangle = \left[(\vec{\xi}_1\cdot\vec{\xi}_3)(\vec{\alpha}_u\cdot\vec{\xi}_I) + 2(\vec{k}_3\cdot\vec{\xi}_1)(\vec{\xi}_3\cdot\vec{\xi}_I) - 2(\vec{k}_1\cdot\vec{\xi}_3)(\vec{\xi}_1\cdot\vec{\xi}_I)\right]f(k_1,k_I,k_3)\,, \tag{G.3}$$

where $f(k_1,k_I,k_3) = K^{-1}$, with $K \equiv k_1 + k_I + k_3$. We can therefore obtain $\langle J\varphi J\varphi\rangle_u$ by modifying (G.1) in accordance with this difference in the polarization factors.

The first term in (G.1) contains the polarization sum $\Pi_{1,1}^{(u)}$, which comes from contracting two copies of the polarization structure in (G.2) and summing over helicities. It will now have to be replaced by

$$\Pi_{1,1}^{(u)} = \frac{\alpha_u^i\pi_{ij}\beta_u^j}{u^2} \rightarrow \left[(\vec{\xi}_1\cdot\vec{\xi}_3)\alpha_u^i + 2(\vec{k}_3\cdot\vec{\xi}_1)\xi_3^i - 2(\vec{k}_1\cdot\vec{\xi}_3)\xi_1^i\right]\frac{\pi_{ij}\beta_u^j}{u^2}$$
$$= (\vec{\xi}_1\cdot\vec{\xi}_3)\Pi_{1,1}^{(u)} + 2(\vec{\xi}_1\circ\vec{\xi}_3)\,. \tag{G.4}$$

The second term in (G.1), proportional to $\Pi_{1,0}^{(u)}$, comes from the longitudinal piece of $\langle\varphi J\varphi\rangle$. This piece is absent in (G.2), which is written in terms of transverse polarization vectors, but can be reconstructed from the Ward–Takahashi identity. First, notice that taking $\vec{\xi}_I \rightarrow \vec{k}_I$ in (G.2) gives $(k_3^2 - k_1^2)f(k_1,k_I,k_3)$. Comparing this to the WT identity, $k_I^i\langle\varphi_{\vec{k}_1}J_{\vec{k}_I}^i\varphi_{\vec{k}_3}\rangle = k_3 - k_1$, we see that the longitudinal part of the correlator must be

$$\langle\varphi_{\vec{k}_1}J_{\vec{k}_I}\varphi_{\vec{k}_3}\rangle_L = \frac{(\vec{z}_I\cdot\vec{k}_I)(k_3 - k_1)}{k_I K}\,. \tag{G.5}$$

A similar analysis for $\langle JJJ\rangle$ shows that its longitudinal piece is $\langle JJJ\rangle_L = (\vec{\xi}_1\cdot\vec{\xi}_3)\langle\varphi J\varphi\rangle_L$. Hence, the contribution proportional to $\Pi_{1,0}^{(u)}$ in the four-point correlator will just differ by a factor of $(\vec{\xi}_1\cdot\vec{\xi}_3)$. Putting everything together, we get

$$\langle J\varphi J\varphi\rangle_u = \left[(\vec{\xi}_1\cdot\vec{\xi}_3)\left(\Pi_{1,1}^{(u)}D_{wv} + \Pi_{1,0}^{(u)}\Delta_w\right) + 2(\vec{\xi}_1\circ\vec{\xi}_3)D_{wv}\right]F_{\Delta_\sigma=2}\,, \tag{G.6}$$

which is the same as the result (154) obtained through weight-shifting.

## G.2 $\langle T\varphi T\varphi\rangle$

The strategy to obtain $\langle T\varphi T\varphi\rangle_u$ is very similar. In this case, we start from the correlator of two massless scalars, $\phi$, and two conformally coupled scalars, $\varphi$, exchanging a spin-2 current in the $u$-channel, which can be written as

$$
\langle\phi\varphi\phi\varphi\rangle_{u,T} = u^2\Big[\Pi_{2,2}^{(u)}U_{13}^{(2,2)}D_{wv}^2 + \Pi_{2,1}^{(u)}U_{13}^{(2,1)}D_{wv}(\Delta_w - 2) \\
+ \Pi_{2,0}^{(u)}U_{13}^{(2,0)}\Delta_w(\Delta_w - 2)\Big]F_{\Delta_\sigma = 3}\,,
\tag{G.7}
$$

where explicit expressions for the weight-shifting operators $U_{ab}^{(2,m)}$ can be found in [47, 48]. The difference between (G.7) and the desired correlator is again in one of the vertices, and the three-point functions corresponding to these vertices differ only in their polarization structure:

$$
\langle\phi_{\vec{k}_1}T_{\vec{k}_I}\phi_{\vec{k}_3}\rangle = \big(\vec{a}_u\cdot\vec{\xi}_I\big)^2 g(k_1,k_I,k_3)\,,
\tag{G.8}
$$

$$
\langle T_{\vec{k}_1}T_{\vec{k}_I}T_{\vec{k}_3}\rangle = \Big[(\vec{\xi}_1\cdot\vec{\xi}_3)(\vec{a}_u\cdot\vec{\xi}_I) + 2(\vec{k}_3\cdot\vec{\xi}_1)(\vec{\xi}_3\cdot\vec{\xi}_I) - 2(\vec{k}_1\cdot\vec{\xi}_3)(\vec{\xi}_1\cdot\vec{\xi}_I)\Big]^2 g(k_1,k_I,k_3)\,,
\tag{G.9}
$$

where the common function is now

$$
g(k_1,k_I,k_3) = \frac{K^3 - K(k_1 k_I + k_1 k_3 + k_I k_3) - k_1 k_I k_3}{K^2}\,.
\tag{G.10}
$$

The WT identities allow to reconstruct the longitudinal parts of these correlators. They are also the same up to the polarization structure:

$$
\langle\phi_{\vec{k}_1}T_{\vec{k}_I}\phi_{\vec{k}_3}\rangle_L = \big(\vec{a}_u\cdot\vec{z}_I\big)\big(\vec{k}_I\cdot\vec{z}_I\big)h_1(k_1,k_I,k_3) + \big(\vec{k}_I\cdot\vec{z}_I\big)^2 h_0(k_1,k_I,k_3)\,,
\tag{G.11}
$$

$$
\langle T_{\vec{k}_1}T_{\vec{k}_I}T_{\vec{k}_3}\rangle_L = \big(\vec{\xi}_1\cdot\vec{\xi}_3\big)\Big[\big(\vec{\xi}_1\cdot\vec{\xi}_3\big)\big(\vec{a}_u\cdot\vec{z}_I\big) + 2\big(\vec{k}_3\cdot\vec{\xi}_1\big)\big(\vec{\xi}_3\cdot\vec{z}_I\big)
\tag{G.12}
$$

$$
-2\big(\vec{k}_1\cdot\vec{\xi}_3\big)\big(\vec{\xi}_1\cdot\vec{z}_I\big)\Big]\big(\vec{k}_I\cdot\vec{z}_I\big)h_1(k_1,k_I,k_3) + \big(\vec{\xi}_1\cdot\vec{\xi}_3\big)^2\big(\vec{k}_I\cdot\vec{z}_I\big)^2 h_0(k_1,k_I,k_3)\,.
$$

We see that the part proportional to $h_0$ only differs by a factor of $(\vec{\xi}_1\cdot\vec{\xi}_3)^2$, so the term proportional to $\Pi_{2,0}^{(u)}$ in (G.7) will just pick up this factor. In the $\Pi_{2,2}^{(u)}$ and $\Pi_{2,1}^{(u)}$ terms, the substitution is more involved and will produce more complex polarization structures in a similar fashion as in (G.4):

$$
\Pi_{2,2}^{(u)} \to (\vec{\xi}_1\cdot\vec{\xi}_3)^2\Pi_{2,2}^{(u)} + 6(\vec{\xi}_1\cdot\vec{\xi}_3)(\vec{\xi}_1\circ\vec{\xi}_3)\Pi_{1,1}^{(u)} + 6(\vec{\xi}_1\circ\vec{\xi}_3)^2\,,
\tag{G.13}
$$

$$
\Pi_{2,1}^{(u)} \to (\vec{\xi}_1\cdot\vec{\xi}_3)^2\Pi_{2,1}^{(u)} + 6(\vec{\xi}_1\cdot\vec{\xi}_3)(\vec{\xi}_1\circ\vec{\xi}_3)\Pi_{1,0}^{(u)}\,.
\tag{G.14}
$$

After introducing all of these modifications in (G.7), we get

$$
\langle T\varphi T\varphi\rangle_u \stackrel{?}{=} u^2\Big[\big((\vec{\xi}_1\cdot\vec{\xi}_3)^2\Pi_{2,2}^{(u)} + 6(\vec{\xi}_1\cdot\vec{\xi}_3)(\vec{\xi}_1\circ\vec{\xi}_3)\Pi_{1,1}^{(u)} + 6(\vec{\xi}_1\circ\vec{\xi}_3)^2\big)U_{13}^{(2,2)}D_{wv}^2 +
$$

$$
+\big((\vec{\xi}_1\cdot\vec{\xi}_3)\Pi_{2,1}^{(u)} + 6(\vec{\xi}_1\circ\vec{\xi}_3)\Pi_{1,0}^{(u)}\big)(\vec{\xi}_1\cdot\vec{\xi}_3)U_{13}^{(2,1)}D_{wv}(\Delta_w - 2) +
\tag{G.15}
$$

$$
+(\vec{\xi}_1\cdot\vec{\xi}_3)^2\Pi_{2,0}^{(u)}U_{13}^{(2,0)}(\Delta_w - 2)\Delta_w\Big]F_{\Delta_\sigma = 3}\,.
$$

This solution is not yet the pure exchange solution, but still contains within it a contact solution diverging as $E^{-5}$. Using the explicit expressions for $U_{13}^{(2,m)}$ in [47, 48], we find

$$
U_{13}^{(2,2)}w^2\partial_w\big(w^2\partial_w F_{\Delta_\sigma = 3}\big) = O_{13}\partial_w\big(w\Delta_w F_{\Delta_\sigma = 3}\big)\,,
\tag{G.16}
$$

$$
U_{13}^{(2,1)}w^2\partial_w(\Delta_w - 2)F_{\Delta_\sigma = 3} = \frac{1}{w}O_{13}\Delta_w(\Delta_w - 2)F_{\Delta_\sigma = 3}\,,
\tag{G.17}
$$

where we have introduced the operator

$$O_{13} = 1 - \frac{k_1 k_3}{k_{13}} \partial_{k_{13}}. \tag{G.18}$$

This implies that (G.15) can be written as

$$\langle T\varphi T\varphi \rangle_u \overset{?}{=} \mathcal{D}_{T\varphi T\varphi} \Delta_w F_{\Delta_\sigma = 3}. \tag{G.19}$$

We see that the seed is acted on by $\Delta_w$, which yields a sum of the seed itself and a contact solution, cf. (120). The contact part can be removed by just replacing $\Delta_w F_{\Delta_\sigma = 3}$ with $2 F_{\Delta_\sigma = 3}$, and the outcome is then precisely the weight-shifting result (183).

## G.3 $\langle TT\phi\phi \rangle$

In this subsection, we compute the correlator $\langle TT\phi\phi \rangle$ of two spin-2 currents and two massless scalars. The longitudinal part of this correlator is used in §7.2 to compute the inflationary bispectrum $\langle \gamma\gamma\zeta \rangle$. The final result for each channel will be normalized in such a way that a simple sum (without additional factors) gives the correct answer for the full correlator $\langle TT\phi\phi \rangle$.
*Scalar exchange.*—We will first find the $u$-channel contribution coming from the exchange of a massless scalar field. We start by noticing that

$$\langle T\phi\phi \rangle = D_{11} D_{12} \mathcal{W}_{12}^{++} \langle \phi\phi\phi \rangle, \tag{G.20}$$

where the differential operators only act on legs 1 and 2. This suggests that the desired four-point function can we written as

$$\begin{aligned}
\langle TT\phi\phi \rangle_u &\overset{?}{\propto} (D_{22} D_{24} \mathcal{W}_{24}^{++})(D_{11} D_{13} \mathcal{W}_{13}^{++}) \langle \phi\phi\phi\phi \rangle_{u,\phi} \\
&= D_{22} D_{24} D_{11} D_{13} \left( \mathcal{W}_{24}^{++} \mathcal{W}_{13}^{++} \right)^2 F_{\Delta_\sigma = 3} \\
&= (\vec{\xi}_1 \cdot \vec{k}_3)^2 (\vec{\xi}_2 \cdot \vec{k}_4)^2 O_{13} O_{24} \partial_w \partial_v \left[ wv(\Delta_w - 12)^2 (\Delta_w - 2)^2 F_{\Delta_\sigma = 3} \right]. \tag{G.21}
\end{aligned}$$

As in §5.2.2 and §5.3.2, however, the result in (G.21) is not yet the correct answer because the differential operators acting on the seed $F_{\Delta_\sigma = 3}$ eliminate the poles corresponding to particle exchange. As before, the correct result is obtained by removing these operators

$$\langle TT\phi\phi \rangle_u = 4(\vec{\xi}_1 \cdot \vec{k}_3)^2 (\vec{\xi}_2 \cdot \vec{k}_4)^2 O_{13} O_{24} \partial_w \partial_v \left( wv F_{\Delta_\sigma = 3} \right), \tag{G.22}$$

where we have introduced the correct normalization. The $t$-channel contribution, $\langle TT\phi\phi \rangle_t$, follows from this solution simply by permuting the legs 3 and 4. It is easy to see that both $u$- and $t$-channel contributions vanish as we take the soft limit in one of the scalar legs, so they do not contribute to the bispectrum $\langle \gamma\gamma\zeta \rangle$ computed in §7.2.
*Graviton exchange.*—Next, we compute the $s$-channel contribution $\langle TT\phi\phi \rangle_s$ arising from graviton exchange. The derivation is similar to that of $\langle T\varphi T\varphi \rangle_u$ in §G.2. The only difference is that now the external scalar fields are massless, so we first act with the weight-raising operator $\mathcal{W}_{34}^{++}$ on the $s$-channel equivalent of (G.7) to get

$$\begin{aligned}
\langle \phi\phi\phi\phi \rangle_{s,T} \overset{?}{\propto} s^4 \Big[ &\Pi_{2,2} U_{34}^{(2,2)} U_{12}^{(2,2)} D_{wv}^2 + \Pi_{2,1} U_{34}^{(2,1)} U_{12}^{(2,1)} D_{wv}(\Delta_w - 2) \\
&+ \Pi_{2,0} U_{34}^{(2,0)} U_{12}^{(2,0)} \Delta_w(\Delta_w - 2) \Big] F_{\Delta_\sigma = 3}. \tag{G.23}
\end{aligned}$$

The result has a leading divergence scaling as $1/E^7$, while the expected divergence is $1/E^3$. Using (G.16) and (G.17), and replacing $\Delta_w F_{\Delta_\sigma = 3}$ with $2 F_{\Delta_\sigma = 3}$, reduces the scaling to $1/E^5$. To obtain the correct $1/E^3$ scaling, we must subtract an additional contact solution. We find

such a contact solution by weight-shifting the seed $C_0$. The final result for the contribution associated to particle exchange then is

$$
\begin{aligned}
\langle \phi\phi\phi\phi \rangle_{s,T} &\propto s^4 \bigg[ \Pi_{2,2} O_{34} O_{12} \partial_w \partial_v \big( wv\Delta_w F_{\Delta_\sigma=3} \big) + \Pi_{2,1} \frac{1}{wv} O_{34} O_{12} \Delta_w (\Delta_w - 2) F_{\Delta_\sigma=3} \\
&\quad + \Pi_{2,0} U_{34}^{(2,0)} U_{12}^{(2,0)} (\Delta_w - 2) F_{\Delta_\sigma=3} \bigg] \\
&\quad + \Big( 2s^4 U_{34}^{(0,0)} U_{12}^{(0,0)} - 3u^4 U_{24}^{(0,0)} U_{13}^{(0,0)} - 3t^4 U_{23}^{(0,0)} U_{14}^{(0,0)} \Big) C_0 \, .
\end{aligned}
$$

(G.24)

To relate this to $\langle TT\phi\phi \rangle_s$, we follow exactly the same procedure as in §G.2—i.e. we change the polarization structure of (G.24) to account for the difference in the three-point vertices. This gives

$$
\begin{aligned}
\langle TT\phi\phi \rangle_s &\overset{?}{\propto} 6s^4 \Big[ (\vec{\xi}_1 \cdot \vec{\xi}_2)(\vec{\xi}_1 \circ \vec{\xi}_2) \Pi_{1,1} + (\vec{\xi}_1 \circ \vec{\xi}_2)^2 \Big] O_{34} O_{12} \partial_w \partial_v \big( wv\Delta_w F_{\Delta_\sigma=3} \big) \\
&\quad + 6s^4 (\vec{\xi}_1 \cdot \vec{\xi}_2)(\vec{\xi}_1 \circ \vec{\xi}_2) \Pi_{1,0} \frac{1}{wv} O_{34} O_{12} (\Delta_w - 2) \Delta_w F_{\Delta_\sigma=3} \\
&\quad + (\vec{\xi}_1 \cdot \vec{\xi}_2)^2 \langle \phi\phi\phi\phi \rangle_{s,T} \, .
\end{aligned}
$$

(G.25)

However, this procedure has re-introduced the $E^{-5}$ singularity that we cancelled in (G.24). In particular, the first two lines do not contain the contact solution and therefore diverge again as $E^{-5}$. To reduce the order of the divergence to $E^{-3}$, we replace $\Delta_w F_{\Delta_\sigma=3}$ by $2F_{\Delta_\sigma=3}$ in these terms. The final solution then is

$$
\begin{aligned}
\langle TT\phi\phi \rangle_s &= -s^4 \Big[ (\vec{\xi}_1 \cdot \vec{\xi}_2)(\vec{\xi}_1 \circ \vec{\xi}_2) \Pi_{1,1} + (\vec{\xi}_1 \circ \vec{\xi}_2)^2 \Big] O_{34} O_{12} \partial_w \partial_v \big( wv F_{\Delta_\sigma=3} \big) \\
&\quad - s^4 (\vec{\xi}_1 \cdot \vec{\xi}_2)(\vec{\xi}_1 \circ \vec{\xi}_2) \Pi_{1,0} \frac{1}{wv} O_{34} O_{12} (\Delta_w - 2) F_{\Delta_\sigma=3} \\
&\quad - \frac{1}{12} (\vec{\xi}_1 \cdot \vec{\xi}_2)^2 \langle \phi\phi\phi\phi \rangle_{s,T} \, ,
\end{aligned}
$$

(G.26)

where we have fixed the overall normalization. It is easy to see that $\vec{\xi}_1 \circ \vec{\xi}_2$ vanishes in the soft limit of either scalar leg, so only the term proportional to $(\vec{\xi}_1 \cdot \vec{\xi}_2)^2$ is relevant for the bispectrum $\langle \gamma\gamma\zeta \rangle$.

*Contact solution.*—Finally, the contact contribution $\langle TT\phi\phi \rangle_c$ can be written in terms of weight-shifting operators as

$$
\begin{aligned}
\langle TT\phi\phi \rangle_c &= \frac{1}{18} \mathcal{S}_{12}^{++} \bigg[ \mathcal{W}_{34}^{++} (D_{24} D_{13} + D_{23} D_{14}) C_0 \\
&\quad - \frac{1}{2} (k_3 k_4)^3 (D_{24} D_{13} + D_{23} D_{14}) \mathcal{W}_{12}^{++} \left( \frac{C_0}{k_3 k_4} \right) \bigg] \, ,
\end{aligned}
$$

(G.27)

where $C_0 = s^{-1}\hat{C}_0$ is given in (119) and $\mathcal{C}_0$ is the $\phi^4$ contact solution (168). The result vanishes in the soft limit of either scalar leg, so it does not contribute to the inflationary bispectrum in Section 7.

# H   Notation and Conventions

| Symbol | Meaning | Reference |
|---|---|---|
| $\Psi$ | Wavefunction of the universe | (5) |
| $\Psi_n$ | $n$-point wavefunction coefficient | (6) |
| $\widetilde{\Psi}_n$ | Shifted wavefunction coefficient | (221) |
| $\Psi_4^{(s,t,u)}$ | $s$, $t$, $u$-channel correlator | §5+6 |
| $\Psi_4^{(c)}$ | Contact correlator | §5+6 |
| $P_i$ | Translation generator | (14) |
| $J_{ij}$ | Rotation generator | (14) |
| $D$ | Dilatation generator | (14) |
| $K_i$ | Special conformal generator | (14) |
| $\widetilde{K}_i$ | Conformal generator in spinor variables | (49) |
| $\eta$ | Conformal time | (4) |
| $H$ | Hubble parameter | (4) |
| $\vec{x}$ | Spatial three-vector | §1 |
| $x^i$ | Component of $\vec{x}$ | §1 |
| $\vec{k}$ | Three-momentum vector | §1 |
| $k^i$ | Component of $\vec{k}$ | §1 |
| $\vec{k}_a$ | Momentum of the $a$-th leg | §1 |
| $k_a$ | Magnitude of $\vec{k}_a$, $k_a \equiv |\vec{k}_a|$ | §1 |
| $k_{ab}$ | Sum of $k_a$ and $k_b$, $k_{ab} \equiv k_a + k_b$ | §1 |
| $E$ | Total energy, $E \equiv \sum_a k_a$ | §2.3 |
| $K$ | Total energy (3pt-function), $K \equiv k_1 + k_2 + k_3$ | §4.1 |
| $s$ | Exchange momentum ($s$-channel), $s \equiv |\vec{k}_1 + \vec{k}_2|$ | §1 |
| $t$ | Exchange momentum ($t$-channel), $t \equiv |\vec{k}_1 + \vec{k}_4|$ | §1 |
| $u$ | Exchange momentum ($u$-channel), $u \equiv |\vec{k}_1 + \vec{k}_3|$ | §1 |
| $w$ | Ratio of $s$ and $k_{12}$, $w \equiv s/k_{12}$ | (117) |
| $v$ | Ratio of $s$ and $k_{34}$, $v \equiv s/k_{34}$ | (117) |
| $\alpha$ | Ratio of $k_1 - k_2$ and $s$, $\alpha \equiv (k_1 - k_2)/s$ | (123) |
| $\beta$ | Ratio of $k_3 - k_4$ and $s$, $\beta \equiv (k_3 - k_4)/s$ | (123) |
| $\vec{\alpha}$ | Difference of $\vec{k}_1$ and $\vec{k}_2$, $\vec{\alpha} \equiv \vec{k}_1 - \vec{k}_2$ | (123) |
| $\vec{\beta}$ | Difference of $\vec{k}_3$ and $\vec{k}_4$, $\vec{\beta} \equiv \vec{k}_3 - \vec{k}_4$ | (123) |
| $\tau$ | Angular variable, $\tau \equiv \vec{\alpha} \cdot \vec{\beta}$ | (123) |
| $\hat{\alpha}_t$ | Ratio of $k_1 - k_4$ and $t$, $\alpha_t \equiv (k_1 - k_4)/t$ | (127) |
| $\hat{\beta}_t$ | Ratio of $k_2 - k_3$ and $t$, $\beta_t \equiv (k_2 - k_3)/t$ | (127) |
| $\vec{\alpha}_t$ | Difference of $\vec{k}_1$ and $\vec{k}_4$, $\vec{\alpha}_t \equiv \vec{k}_1 - \vec{k}_4$ | (127) |
| $\vec{\beta}_t$ | Difference of $\vec{k}_2$ and $\vec{k}_3$, $\vec{\beta}_t \equiv \vec{k}_2 - \vec{k}_3$ | (127) |
| $\tau_t$ | Angular variable, $\tau \equiv \vec{\alpha}_t \cdot \vec{\beta}_t$ | (127) |
| $\hat{\alpha}_u$ | Ratio of $k_1 - k_3$ and $u$, $\alpha_u \equiv (k_1 - k_3)/u$ | (128) |
| $\hat{\beta}_u$ | Ratio of $k_2 - k_4$ and $u$, $\beta_u \equiv (k_2 - k_4)/u$ | (128) |
| $\vec{\alpha}_u$ | Difference of $\vec{k}_1$ and $\vec{k}_3$, $\vec{\alpha}_u \equiv \vec{k}_1 - \vec{k}_3$ | (128) |
| $\vec{\beta}_u$ | Difference of $\vec{k}_2$ and $\vec{k}_4$, $\vec{\beta}_u \equiv \vec{k}_2 - \vec{k}_4$ | (128) |

| Symbol | Meaning | Reference |
|---|---|---|
| $\tau_u$ | Angular variable, $\tau_u \equiv \vec{\alpha}_u \cdot \vec{\beta}_u$ | (128) |
| $E_L$ | Partial energy ($s$-channel), $E_L \equiv k_{12} + s$ | §2.3 |
| $E_R$ | Partial energy ($s$-channel), $E_R \equiv k_{34} + s$ | §2.3 |
| $E_L^{(t)}$ | Partial energy ($t$-channel), $E_L^{(t)} \equiv k_{14} + t$ | (223) |
| $E_R￼^{(t)}$ | Partial energy ($t$-channel), $E_R^{(t)} \equiv k_{23} + t$ | (223) |
| $E_L^{(u)}$ | Partial energy ($u$-channel), $E_L^{(u)} \equiv k_{13} + u$ | (224) |
| $E_R^{(u)}$ | Partial energy ($u$-channel), $E_R^{(u)} \equiv k_{24} + u$ | (224) |
| $y_{nm}$ | Energy running between vertices $n$ and $m$ | §6.1 |
| $E_n$ | Energy entering vertex $n$ | §6.1 |
| $E_{\text{tot}}$ | Total energy entering a graph | §6.1 |
| $\vec{z}$ | Auxiliary null vector, $z^2 = 0$ | §1 |
| $\vec{\xi}_a$ | Transverse polarization vector, $\vec{k}_a \cdot \vec{\xi}_a = 0$ | §1 |
| $\vec{\xi}_a \circ \vec{\xi}_b$ | Polarization structure | (152) |
| $\lambda_\alpha$ | Momentum spinor variable (holomorphic) | (41) |
| $\bar{\lambda}_\alpha$ | Momentum spinor variable (anti-holomorphic) | (41) |
| $\sigma^i_{\alpha\beta}$ | Pauli matrices | [154] |
| $\epsilon_{\alpha\beta}$ | Totally antisymmetric 2-index tensor | [154] |
| $\langle ab \rangle$ | Inner product between momentum spinors, $\langle ab \rangle \equiv \epsilon^{\alpha\beta} \lambda^a_\alpha \lambda^b_\beta$ | (44) |
| $\langle a\bar{b} \rangle$ | Inner product between momentum spinors, $\langle a\bar{b} \rangle \equiv \epsilon^{\alpha\beta} \lambda^a_\alpha \bar{\lambda}^b_\beta$ | (45) |
| $m$ | Helicity (3d) | (124) |
| $h$ | Helicity (4d) | §C.1 |
| $\Delta$ | Scaling dimension (conformal weight) | (16) |
| $\ell$ | Spin | §2.2 |
| $O_\Delta^{(\ell)}$ | Operator with weight $\Delta$ and spin $\ell$ | §2.1 |
| $O$ | Operator dual to the field $\bar{\sigma}$ | (7) |
| $\sigma$ | Generic bulk field | §2.1 |
| $\sigma_\pm$ | Boundary values of the field $\sigma$ | (15) |
| $m_\sigma$ | Mass of the field $\sigma$ | §2.2 |
| $\varphi$ | Scalar operator with $\Delta = 2$ (dual to conformal scalar $\varphi$) | §2.2 |
| $\varphi$ | Conformal bulk scalar | §2.2 |
| $\phi$ | Scalar operator with $\Delta = 3$ (dual to massless scalar $\phi$) | §2.2 |
| $\phi$ | Massless bulk scalar | §2.2 |
| $J_i$ | Conserved spin-1 current (dual to photon $A_\mu$) | §2.2 |
| $J^\pm$ | Helicity components of $J^i$, $J^\pm \equiv \xi^\pm_i J^i$ | §3.4 |
| $A_\mu$ | Bulk vector field | §2.2 |
| $F_{\mu\nu}$ | Field strength | §4.2 |
| $T_{ij}$ | Conserved spin-2 current (dual to graviton $\gamma_{\mu\nu}$) | §2.2 |
| $T^\pm$ | Helicity components of $T^{ij}$, $T^\pm \equiv \xi^\pm_i \xi^\pm_j T^{ij}$ | §3.4 |
| $\gamma_{\mu\nu}$ | Bulk graviton | §2.2 |
| $W_{\mu\nu\rho\sigma}$ | Weyl tensor | §4.3 |
| $\sigma_{\text{cl}}$ | Classical solution | (193) |
| $\bar{\sigma}$ | Boundary value of $\sigma$ | §6.1 |

| Symbol | Meaning | Reference |
|---|---|---|
| $\mathcal{K}$ | Bulk-to-boundary propagator | (194) |
| $\mathcal{G}$ | Bulk-to-bulk propagator | (195) |
| $iV$ | Vertex factor | §6.1 |
| $A_n$ | $n$-point amplitude | §2.3 |
| $\widetilde{A}_n$ | Rescaled amplitude | (220) |
| $A_{s,t,u}$ | $s$, $t$, $u$-channel amplitude | §5+6 |
| $A_c$ | Contact amplitude | §5+6 |
| $p_a^\mu$ | Four-momentum of the $a$-th leg | §1 |
| $\epsilon_a^\mu$ | Polarization vector of particle $a$, $p_a \cdot \epsilon_a = 0$ | §1 |
| $S$ | Flat-space Mandelstam variable, $S \equiv -(p_1 + p_2)^2$ | §1 |
| $T$ | Flat-space Mandelstam variable, $T \equiv -(p_1 + p_4)^2$ | §1 |
| $U$ | Flat-space Mandelstam variable, $U \equiv -(p_1 + p_3)^2$ | §1 |
| $D_z^i$ | Operator removing auxiliary null vectors | (3) |
| $\Delta_w$ | Hypergeometric differential operator | (118) |
| $\mathcal{W}_{12}^{--}$ | Weight-lowering operator, $\Delta \mapsto \Delta - 1$ at 1 and 2 | (24) |
| $\mathcal{W}_{12}^{++}$ | Weight-raising operator, $\Delta \mapsto \Delta + 1$ at 1 and 2 | (25) |
| $\mathcal{S}_{12}^{++}$ | Spin-raising operator, $\ell \mapsto \ell + 1$ at 1 and 2 | (26) |
| $H_{12}$ | Operator that maps $[\Delta, \ell] \mapsto [\Delta - 1, \ell + 1]$ at 1 and 2 | (27) |
| $D_{12}$ | Operator that raises spin at 1 and lowers weight at 2 | (28) |
| $D_{11}$ | Operator that maps $[\Delta, \ell] \mapsto [\Delta - 1, \ell + 1]$ at point 1 | (29) |
| $Q_{ab}$ | Auxiliary weight-shifting operator | (178) |
| $O_{ab}$ | Auxiliary weight-shifting operator | (G.18) |
| $U_{ab}^{(\ell,m)}$ | Weight-shifting operator | [47, 48] |
| $\mathcal{D}_{wv}^{(\ell,m)}$ | Differential operator in the spin-exchange solution | (124) |
| $(\mathsf{P}_1)_{ij}$ | Spin-1 projector | (34) |
| $(\mathsf{P}_2)_{ij,lm}$ | Spin-2 projector | (37) |
| $\mathsf{P}_a^{(1)}$ | Scalar spin-1 projector, $\mathsf{P}_a^{(1)} \equiv z_a^i (\mathsf{P}_1)_{ij} D_{z_a}^j$ | (35) |
| $\mathsf{P}_a^{(2)}$ | Scalar spin-2 projector, $\mathsf{P}_a^{(2)} \equiv z_a^i z_a^j (\mathsf{P}_2)_{ij,lm} D_{z_a}^l D_{z_a}^m$ | (39) |
| $e_a$ | Charge of particle $a$ | Fig. 4 |
| $f^{ABC}$ | Self-coupling of non-Abelian vector | Fig. 7 |
| $T_{ab}^A$ | Vector-scalar coupling | Fig. 7 |
| $\kappa_a$ | Gravitational coupling of particle $a$ | Fig. 9 |
| $\kappa_g$ | Gravitational self-coupling | Fig. 9 |
| $\kappa_c$ | Nonlinear gravitational coupling to matter | Fig. 9 |
| $\kappa$ | Universal gravitational coupling | Fig. 11 |
| $\tilde{\kappa}_c$ | Quartic graviton self-coupling | Fig. 11 |
| $\pi_{ij}$ | Transverse projector, $\pi_{ij} \equiv \delta_{ij} - \hat{k}_i \hat{k}_j$ | (38) |
| $\Pi_{\ell,m}$ | Polarization sum ($s$-channel) | §E.2 |
| $\Pi_{\ell,m}^{(t)}$ | Polarization sum ($t$-channel) | §E.2 |
| $\Pi_{\ell,m}^{(u)}$ | Polarization sum ($u$-channel) | §E.2 |
| $\mathcal{Q}_1$ | Spin-1 angular function (top-helicity) | (185) |
| $\mathcal{Q}_2$ | Spin-2 angular function (top-helicity) | (185) |

| Symbol | Meaning | Reference |
|---|---|---|
| $\mathcal{P}_1$ | Spin-1 angular function (exchange) | (185) |
| $\mathcal{P}_2$ | Spin-2 angular function (exchange) | (185) |
| $\mathcal{M}$ | Polarization structure of $\langle T\varphi T\varphi\rangle_u$ | (184) |
| $\mathcal{N}$ | Polarization structure of $\langle T\varphi T\varphi\rangle_u$ | (184) |
| $\mathcal{L}$ | Polarization structure of $\langle T\varphi T\varphi\rangle_u$ | (184) |
| $F$ | Four-point function of $\Delta=2$ scalars | §5.1 |
| $\hat{F}$ | Dimensionless four-point function, $\hat{F}=sF$ ($s$-channel) | (117) |
| $\hat{F}_{\Delta_\sigma=2}$ | Exchange solution ($\Delta_\sigma=2$) | (121) |
| $\hat{F}_{\Delta_\sigma=3}$ | Exchange solution ($\Delta_\sigma=3$) | (122) |
| $\hat{F}^{(\ell)}$ | Spin-$\ell$ exchange solution | (124) |
| $\hat{C}_0$ | Lowest-order contact solution, $\hat{C}_0 \equiv wv/(w+v)$ | (119) |
| $\hat{C}_n$ | Higher-order contact solutions, $\hat{C}_n = \Delta_w^n \hat{C}_0$ | (119) |
| $P_\ell$ | Legendre polynomial | [154] |
| $K_\nu$ | Bessel function | [154] |
| $H_\nu^{(1,2)}$ | Hankel functions | [154] |
| $_2F_1$ | Hypergeometric function | [154] |
| $\mathrm{Li}_2$ | Dilogarithm | [154] |

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
