# Peer review of "The Cosmological Bootstrap: Spinning Correlators from Symmetries and Factorization"

_SciPost Physics, doi:SciPost Phys. 11, 071 (2021)_

## Round 2 · Referee Report · Anonymous (Referee 1) · 2020-10-16

Report

The work under consideration is the third in a series of papers centered on a bootstrap approach to de Sitter correlators, initiating the study of spinning correlators focusing on external massless spin-1 and spin-2 legs. The results of this work are interesting and well motivated. The paper is rather long but contains many useful/pedagogical details for readers looking to learn something/reproduce. I can strongly recommend publication. I would like to list a few minor comments which I feel would be beneficial to the reader.

Technical points:

  1. At the beginning of section 3 it is claimed that the reason why weight-shifting operators have been introduced is that it is difficult to solve conformal Ward identities. To me this statement is confusing. Solving conformal Ward identities is precisely the way weight-shifting operators were first constructed (see e.g. (2.50) of 1706.07813). It might actually be even simpler to directly solve the Ward identities in momentum space following the analogous logic to that of 1706.07813 in position space. Did the authors try to address this question? In any case, it would be useful to have some clarification.

  2. When the authors discuss 3pt correlators they are very careful in including all longitudinal terms which are crucial to solve the Ward identities (see e.g. eq 4.8). On the other hand when they focus on 4pt correlator they seem to drop longitudinal terms (see e.g. eq. 5.15). I was not able to find in the text why these longitudinal terms have been dropped. Are the authors saying that longitudinal terms should not be there? This seems strange. It would be important to write down all longitudinal terms explicitly or comment about them. A similar discussion should be made in the second part of the paper where the authors study factorization of 4pt correlators. In eq. 6.65 the authors seem to drop again longitudinal contributions to the 3pt function. While such longitudinal terms might not be needed to impose the constraints of Ward Takahashi identities, in the introductory text and abstract (and therefore to the general reader) it looks like the goal of the paper is to compute the three- and four-point functions — which contain such longitudinal terms (unless they can be ignored for some reason?).

Presentational points:

  1. The paper is focused on applying existing weight-shifting technology to certain correlators with massless spin-1 and spin-2 external legs. However in the title, the first sentence of the abstract and the introduction the emphasis is on spinning fields in general, which could give the impression to the wider community that the results obtained are more general than they are (logic: the technology was already there for spin in general, so such emphasis on spin in general implies the new results are for spin in general….). A suggestion could be to make the first sentence, which reads:

“We extend the cosmological bootstrap to correlators involving massless particles with spin.” ,

a little more precise because, since this work is a first (but important) step focussing on certain correlators of massless spin-1 and spin-2 (and not including the longitudinal modes it seems), it is not true that “extend the cosmological bootstrap to correlators involving massless particles with spin.” has been reached.

  1. The authors give a lot of emphasis to the fact that non-trivial Ward identities require the simultaneous presence of multiple channels. Wasn’t this known beforehand in flat space? Is there some new observation on top of the flat space statement?

  2. Some papers seem to be referenced for original results that they do not contain. Most striking is footnote 18: There are plenty of important earlier works on correlation functions of conserved currents, e.g.

https://arxiv.org/abs/hep-th/9605009 https://arxiv.org/abs/1104.4317 https://arxiv.org/abs/1206.5639 https://arxiv.org/abs/1206.6370

And since [90] does not contain new results on top of the above, and only applies when all operators are conserved, it does not seem correct that it is given (what appears to be) all credit for three point functions involving conserved currents. Similarly, [91] is purely about flat space and does not even contain the word Ward identity. On top of this, all potentially relevant original results of [91] appear in the seminal works of Ruslan Metsaev dating back to the early 90s on this light front approach, including Yang Mills and universality of gravity.

  • validity: top
  • significance: top
  • originality: top
  • clarity: top
  • formatting: perfect
  • grammar: perfect

Author:  Guilherme L. Pimentel  on 2021-07-23  [id 1609]

(in reply to Report 1 on 2020-10-16)

We thank the referee for their careful reading and remarks on the paper.

About the technical points:

  1. The fact that the weight-shifting operators were constructed solving the conformal Ward identities implies that we can skip that step by just Fourier transforming them to momentum space. Once this is done, the problem reduces to applying differential operators on a seed function, and the result is guaranteed to solve the Ward identities. That is a foreseeably easier method than solving a set of PDEs for each spinning correlator. In fact, directly solving the PDEs was already a challenge in 1811.00024 for the conformally coupled scalar four-point function, which has much simpler symmetry properties than any of the spinning cases.

  2. Three- and four-point correlators contain longitudinal terms, but they do not contribute to the observables—physical polarization states are transverse and traceless—so we decided to omit them. This way the correlators become much simpler and easier to compute, particularly at four points. We only included the longitudinal part in the simplest cases— (4.8), (4.18) and (4.32)—in which it is easier to check the Ward Takahashi identities directly rather than using the operator (3.26). Another reason to include the longitudinal terms in (4.8) and (4.32) was that they are necessary in evaluating the right hand side of the Ward identities (5.31) and (5.59), respectively. We have added a sentence reminding this at the beginning of section 5.

About the presentational points:

  1. We have made the abstract a bit more precise to avoid misinterpretation.

  2. Although this is a well-known fact in flat space, its analogy in de Sitter had not been explored before. But it is not just a technical curiosity, the bootstrap was lacking an argument for the presence of multiple channels and the value of their relative couplings that did not refer to the bulk. Our work shows that the Ward Takahashi identity provides such a pure boundary argument.

  3. We have added the suggested references in chronological order.

---

## Round 2 · Referee Report · Anonymous (Referee 2) · 2020-12-17

Report

The paper initiated the study of scattering amplitudes of massless spin-1 and spin-2 particles in 4D dS space. This is a much uncharted area which is both theoretically interesting and phenomenologically relevant. The direct computation of amplitudes of nonzero-spin states could be quite tedious. The paper tackled the problem using the previously developed weight-shifting techniques. In addition, the paper showed that the Ward identities of conformal and gauge symmetries, together with the pole structures, can sometimes fix the correlators at tree level. The presentation of the paper is clear and pedagogical which could be very useful for future studies. I highly recommend the paper for publication. I also have several minor questions and comments. Addressing these points is optional to the authors but would be helpful to the readers.

  1. The paper considered correlators made of massless spin-1 and spin-2 particles, as well as conformally coupled scalars. On the other hand, when the authors considered the scalar correlators in their previous two papers, they did study the exchange of massive particles with or without spin. Given all the techniques at hand, I'm wondering if it is straightforward to get the result of spinning correlators with massive particle exchange. This might be beyond the scope of the paper, but I think it would be nice at least to comment on this more general case, in particular because this might also lead to richer phenomenology.

  2. The scalar seed correlators in Sec. 4.1 for three $\Delta=2$ opeartors and for three $\Delta=3$ operators contain an arbitrary scale $\mu$. (Eqs. 4.2 and 4.3) From bulk viewpoint this could be related to the secular growth of in-in integral. I'm wondering if there is any good boundary explanation for this logarithmic dependence on $\mu$. Can the authors possibly comment on this point?

  3. When talking about the future directions in Sec. 8, can the authors comment on the loop diagrams? In particular, what analytical structure input would be necessary/sufficient to construct loop correlators? Can the spin helicity formalism shed any light on this?

  4. In Appendix A when introducing the dS UIRs, the authors mentioned that the complementary series of the scalar include $0 < \Delta < 3$. Clearly $\Delta = 0$ or 3 are excluded. But the authors also mentioned that massless scalar corresponds to $\Delta = 3$ (and this $\Delta=3$ case was also considered in the main text). Can the authors provide a bit more explanation of this? Specifically, is a massless scalar not allowed in dS field content? And if so, why is it valid to include such operators in correlators? Are such correlators respect all dS symmetry?

  • validity: top
  • significance: top
  • originality: top
  • clarity: top
  • formatting: perfect
  • grammar: perfect

Author:  Guilherme L. Pimentel  on 2021-07-23  [id 1608]

(in reply to Report 2 on 2020-12-17)

We thank the referee for their remarks, questions and suggestions, which we address below:

  1. With the weight-shifting technique presented in the paper it is straightforward—though sometimes computationally heavy—to obtain spinning correlators with exchange of a massive particle of any integer spin. The strategy is simply to use as a seed the correlator of conformally coupled scalars obtained in the previous two papers, for the desired values of mass and spin of the exchanged field. Throughout this paper we use different scalar seeds depending on the aimed correlator, so it is somewhat implicit that the method works for generic seeds. The study of correlators from massive spinning particle exchange and their implications for phenomenology is the topic of an ongoing project.

  2. As mentioned in section 4.1, from the boundary point of view the arbitrary scale $\mu$ is related to the freedom of adding an arbitrary contribution from a local term. This local term solves the Ward identities by itself and it is a constant in the case of $\langle \varphi \varphi \varphi\rangle$, $\sum_n 􏰒k_n^3$ in the case of $\langle \phi \phi \phi \rangle$ and $k_1 + k_2$ in the case of $\langle \varphi \varphi \phi\rangle$. The presence of this arbitrary scale is related to the fact that the correlator solves the conformal Ward identities only at separated points, i.e. anomalously. This was carefully studied in 1510.08442.

  3. In the context of the paper, the most important observation regarding loops is made at the end of section 3.3—because the two- and three-point functions are completely fixed by symmetries, adding loops will not change the right hand side of four-point Ward-Takahashi identities and so they can only modify the four-point correlators by a rescaling of the tree-level answer and/or an addition of identically conserved pieces. At higher points, this observation does not apply and the effect of loop corrections is not so constrained. Besides this, not much is known so far about the analytic structure of loop diagrams in de Sitter space, but it is a topic of active investigation.

  4. Strictly speaking, the complementary series covers the range $0 < \Delta < 3$ and the massless scalar, which corresponds to $\Delta = 3$, is in the discrete series. We added a footnote with a reference to explain this point.

---

## Round 3 · Referee Report · Anonymous (Referee 3) · 2021-7-27

Report

The authors have properly addressed my previous questions. I am happy to recommend the paper for publication at SciPost. Congratulations to the authors for a very nice piece of work.

---

## Round 3 · Referee Report · Anonymous (Referee 4) · 2021-8-5

Report

Thanks to the authors for their clarifications and modifications. I am happy to recommend their paper for publication.

---

## Editorial Decision

published